# MSC-180: A Benchmark for Automated Formal Theorem Proving from Mathematical Subject Classification

## Abstract

Automated Theorem Proving (ATP) represents a core research direction in artificial intelligence for achieving formal reasoning and verification, playing a significant role in advancing machine intelligence. However, current large language model (LLM)-based theorem provers suffer from limitations such as restricted domain coverage and weak generalization in mathematical reasoning. To address these issues, we propose MSC-180, a benchmark for evaluation based on the MSC2020 mathematical subject classification. It comprises 180 formal verification problems—3 advanced problems from each of 60 mathematical branches—spanning from undergraduate to graduate levels. Each problem has undergone multiple rounds of verification and refinement by domain experts to ensure formal accuracy. Evaluations of state-of-the-art LLM-based theorem provers under the pass@32 setting reveal that the best model achieves only an 18.89% overall pass rate, with prominent issues including significant domain bias (maximum domain coverage 41.7%) and a difficulty gap (significantly lower pass rates on graduate-level problems). To further quantify performance variability across mathematical domains, we introduce the coefficient of variation (CV) as an evaluation metric. The observed CV values are 4–6 times higher than the statistical high-variability threshold, indicating that the models still rely on pattern matching from training corpora rather than possessing transferable reasoning mechanisms and systematic generalization capabilities. MSC-180, together with its multi-dimensional evaluation framework, provides a discriminative and systematic benchmark for driving the development of next-generation AI systems with genuine mathematical reasoning abilities.

## 1 Introduction

Automated Theorem Proving (ATP) aims to develop systems capable of performing mathematical reasoning and generating proofs automatically. Such systems can provide rigorous formal guarantees, which are crucial for high-reliability fields like mathematics, hardware verification, and software security (Robinson, 1965; Harrison, 2009). In recent years, large language models (LLMs) have been successfully integrated into this domain, leveraging their powerful pattern recognition and strategy generation capabilities. This integration has, to some extent, mitigated the limitations of traditional purely symbolic methods in intuitive reasoning and generalization (Polu & Sutskever, 2020). The integration of LLMs with Interactive Theorem Provers (ITPs) has formed a "generate-and-verify" paradigm, further enhancing the efficiency and scalability of ATP For instance, Seed-Prover (Chen et al., 2025), utilizing a lemma-based proof approach, successfully solved four problems from the International Mathematical Olympiad (IMO) 2025. This demonstrates its potential in addressing certain types of complex mathematical problems and marks a notable advancement in the application of machine reasoning within mathematics and artificial intelligence.

State-of-the-art LLM-based automated theorem provers have reported high performance metrics on certain benchmarks. For example, Goedel-Prover-V2-32B (Lin et al., 2025) achieved a success rate exceeding 90.4% on miniF2F(Zheng et al., 2022). However, these results may partly reflect the models' adaptation to particular data distributions rather than indicating universal mathematical reasoning capabilities. Studies suggest that high performance might stem from over-optimization

towards specific problem patterns and solving strategies, which poses challenges for cross-domain generalization. Experimental results from Goedel-Prover (Lin et al., 2025) indicate a negative correlation trend between its performance on ProofNet (Azerbayev et al., 2023) (undergraduate pure mathematics) and miniF2F (high school competitions). Furthermore, after training with the Mathlib4 (Mathlib Community, 2020) library, the model's performance on ProofNet improved, while its performance on miniF2F declined. This suggests potential trade-offs in performance across different domains and highlights the significant influence of training data distribution on current model capabilities. These observations collectively suggest that high performance on specific tasks may rely considerably on learning dataset-specific patterns. The ability of these models to generalize to unseen domains or problem types requires further investigation.

Challenges in the generalization capabilities of existing theorem provers are partly attributable to limitations in current benchmarking practices. For instance, FormalMATH (Yu et al., 2025) has a higher proportion of problems related to integrals and elementary calculus, while MiniF2F (Zheng et al., 2022) focuses more on high school algebra and number theory. Such imbalances in domain coverage may affect the comprehensive evaluation of models' reasoning abilities. Constructing formal mathematical benchmarks with broad coverage across mathematical domains and a balanced difficulty distribution faces several challenges. Firstly, resources containing high-quality formal representations of mathematical problems remain limited. Secondly, establishing a systematic disciplinary classification framework is necessary to ensure comprehensive coverage. Additionally, the benchmark construction process demands expertise in specialized domains, resulting in high labor costs that constrain the scale and diversity of benchmarks.

To address these challenges, we propose the MSC-180 benchmark. This benchmark is constructed with reference to the MSC2020 mathematical subject classification system (AMS, 2020). It covers 60 mathematical branches, each containing three representative problems, totaling 180 problems. The selected problems are of high difficulty, primarily sourced from classic graduate and undergraduate textbooks. A team of researchers with backgrounds in formal mathematics systematically conducted the formalization and cross-validation of these problems. This process aimed to maintain linguistic consistency and reasoning complexity while ensuring disciplinary representativeness.

We evaluated several automated theorem provers using the MSC-180 benchmark. Experimental results indicate that existing models face challenges when dealing with problems spanning a wide range of mathematical domains. Under the pass@32 metric, DeepSeek-Prover-V2 (Ren et al., 2025) achieved a success rate of 18.89%, while BFS-Prover 7B (Xin et al., 2025) attained a rate of 4.44%. Further analysis revealed considerable variation in model performance across different mathematical domains, with the highest domain coverage reaching 41.7%. Moreover, the pass rates for graduate-level problems were lower than those for undergraduate-level problems. These observations suggest potential limitations in the abstract reasoning and systematic generalization capabilities of current approaches. Furthermore, we introduced the coefficient of variation (CV) as an evaluation metric. The relatively high CV values observed indicate substantial fluctuation in model performance across domains. This may imply a tendency for models to rely on specific patterns present in the training data rather than employing general-purpose reasoning mechanisms.

The main contributions of this study are as follows:

**A Systematic,Advanced, and Domain-Balanced Benchmark (MSC-180)**: To address the limited disciplinary coverage and uneven difficulty distribution of existing benchmarks, we introduce MSC-180. Constructed upon the MSC2020 classification system (AMS, 2020), it comprises 180 problems spanning 60 mathematical domains, providing a more comprehensive and equitable evaluation framework for automated theorem proving.

**Systematic Evaluation of Reasoning and Generalization**: The evaluation based on MSC-180 reveals two notable phenomena regarding current models: first, there exists significant variation in model performance across different mathematical domains; second, model performance shows correlation with problem difficulty. These findings provide reference points indicating certain limitations in current models' capabilities for abstract reasoning and cross-domain generalization.

**A Novel Metric for Assessing Disciplinary Generalization Capability (Coefficient of Variation@k)**: This paper proposes the Coefficient of Variation@k (CV@$k$) as a metric for evaluating consistency of model performance across domains. CV@$k$, defined as the ratio of the standard deviation to the mean of domain-level Pass@$k$ values, is a dimensionless measure that eliminates

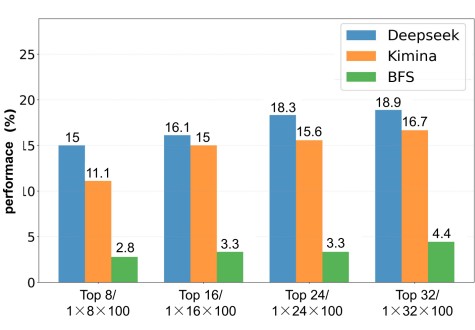
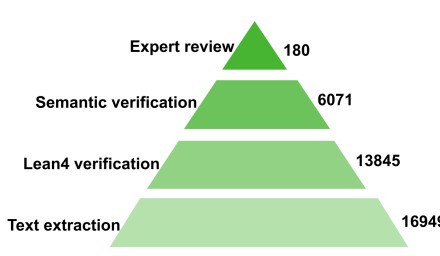

**(a)** Performance of current provers on MSC-180.

**(b)** Quality refinement pyramid.

Figure 1: The proving performance on the MSC-180 benchmark and the refinement process. (a) The performance of automated theorem provers, where the x-axis represents Top k for Deepseek and Kimina, and 1×k×100 for BFS. (b) The multi-stage quality refinement pyramid illustrating the dataset construction process, with the left side showing the dataset filtering steps and the right side showing the corresponding number of remaining problems.

absolute performance discrepancies, prevents dominance by high-scoring domains, and better reflects model stability. By shifting the focus from pure accuracy to performance balance, CV@$k$ provides an objective basis for distinguishing between generalist models with consistent generalization abilities and domain-specialized models.

## 2 RELATED WORK

**Autoformalization** refers to the task of automatically converting informal mathematical statements into formal code (e.g., in Lean (de Moura & Ullrich, 2021) or Isabelle (Nipkow et al., 2002)) using LLMs. Common technical approaches include: (1) Supervised fine-tuning (SFT), which uses methods such as back-translation and expert iteration to create high-quality aligned data for fine-tuning LLMs, with the aim of improving autoformalization performance (Azerbayev et al., 2023; Yu et al., 2024); (2) Retrieval-augmented generation (RAG), where relevant theorems and definitions retrieved from formal libraries (e.g., Mathlib (Mathlib Community, 2020)) provide contextual cues to help generate syntactically correct and semantically consistent formal code (Liu et al., 2025); and (3) Iterative refinement, which leverages compiler error feedback in a generate-verify-revise loop—using error messages from proof assistants to guide iterative corrections—thereby combining the rigor of formal verification with the generative capacity of LLMs (Yang et al., 2024; Li et al., 2024). Having evolved from early rule-based systems, modern autoformalization often integrates SFT, retrieval augmentation, and iterative feedback to handle the inherent ambiguity of natural language and meet the precision requirements of formal systems, with the goal of supporting more reliable and scalable automated reasoning.

**Automated Theorem Proving (ATP)** focuses on generating machine-verifiable proofs for mathematical statements. Recent LLM-based approaches can be broadly categorized into three paradigms: (1) End-to-end proof generation, in which models produce complete formal proof scripts without interacting with proof assistants, often relying on large-scale pre-training and high-quality synthetic data (e.g., Goedel-Prover (Lin et al., 2025)). Inference frequently involves sampling multiple candidates (e.g., Pass@N) to account for output variability. Some models, such as Kimina-Prover (Wang et al., 2025), also incorporate explicit internal reasoning chains to enhance proof validity. (2) Interactive tactic generation, where models collaborate with proof assistants (e.g., Lean) to generate tactics step by step, often using search algorithms (e.g., breadth-first search in BFS-Prover (Xin et al., 2025)) to explore the large space of possible tactics efficiently. (3) Hierarchical and lemma-based proof generation, a divide-and-conquer strategy that breaks complex theorems into simpler subgoals. Hierarchical methods (e.g., DeepSeek-Prover-V2 (Ren et al., 2025)) may use a planner model to decompose the problem and subsidiary models to solve subgoals, while lemma-based approaches (e.g., Seed-Prover (Chen et al., 2025)) automatically propose and prove auxiliary lemmas, accumulating a reusable "lemma pool" to facilitate the main proof. These paradigms, particularly

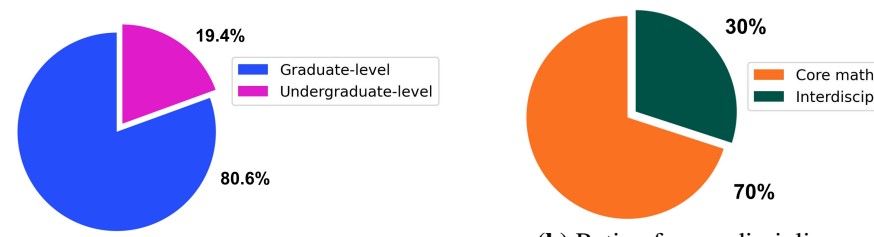

**(a)** Distribution of problem difficulty levels.

**(b)** Ratio of cross-disciplinary and core math problems.

Figure 2: Analysis of mathematical problem characteristics. (a) Quantity distribution across difficulty levels, showing comparisons such as graduate-level versus undergraduate-level problem counts. (b) Quantitative ratio between cross-disciplinary and core math problems.

through interactive collaboration and hierarchical planning, have contributed to improvements in the performance of LLM-based ATP systems on benchmarks such as MiniF2F.

**Formal Mathematics Benchmarks** consist of mathematical problems expressed in formal languages (e.g., Lean (de Moura & Ullrich, 2021)) and provide objective, reproducible environments for evaluating the reasoning capabilities of AI systems. Several key benchmarks have been developed: MiniF2F (Zheng et al., 2022) and ProofNet (Azerbayev et al., 2023) serve as standard test sets for contest-style and undergraduate-level problems, assessing creative problem-solving and systematic knowledge application, respectively. For more advanced challenges, FIMO (Liu et al., 2023) offers problems from IMO preliminary contests, while PutnamBench (Tsoukalas et al., 2024) aims to enhance generalization through a large-scale collection of competition problems. Additionally, FormalMATH (Yu et al., 2025) provides a comprehensive, mixed-difficulty dataset. By offering machine-verifiable, unambiguous, and standardized evaluation environments, these benchmarks help establish a common ground for measuring progress in AI-driven mathematical reasoning and encourage the community to address increasingly complex and profound challenges.

## 3 MSC-180: A SYSTEMATIC BENCHMARK FOR EVALUATING DOMAIN GENERALIZATION IN AUTOMATED THEOREM PROVING

### 3.1 DATASET OVERVIEW

MSC-180 is a rigorously validated benchmark dataset for Lean 4, constructed based on the Mathematics Subject Classification (MSC2020) framework. It contains 180 high-quality formalized propositions, each independently verified through a hybrid process that combines semantic checks by LLMs and manual review by domain experts. The propositions span 60 major branches of mathematics and cover topics ranging from undergraduate foundations to graduate-level specialties.

A comparison with existing benchmarks highlighting MSC-180's design focus is provided in Table 1 and the complete list of covered MSC codes is included in the Appendix A. The distribution of difficulty levels across these topics is shown in Figure 3. The dataset is composed of two main categories:

**Core Mathematical Disciplines (70%)**: This portion encompasses both fundamental subjects (e.g., calculus, abstract algebra) and advanced pure mathematical areas (e.g., real and functional analysis, topology, Lie groups, and representation theory), representing the central body of mathematical knowledge.

**Interdisciplinary Fields (30%)**: This portion extends to fields at the intersection of mathematics and other disciplines, such as mathematical physics, computational mathematics, information science, and mathematical biology.

This comprehensive coverage, balancing depth in core mathematics with breadth in interdisciplinary applications, enhances the dataset's representativeness and practical utility. It establishes a solid foundation for applying formal mathematics in broader scientific and engineering contexts. For

detailed dataset, please refer to the following link: `https://anonymous.4open.science/r/MSC-180-34F3`.

Table 1: A Comparison of Formal Mathematical Reasoning Benchmarks

| Benchmark | Field Coverage | Problems | Difficulty Distribution |
|-----------|----------------|----------|--------------------------|
| MSC-180 | 60 fields (MSC2020) | 180 | Graduate-dominant |
| ProofNet | Core undergrad. pure math | 371 | Undergraduate |
| miniF2F | Olympiad (e.g., IMO) problems | 488 | High School to Olympiad |
| FormalMATH | 12 subfields | 5,560 | Olympiad to Undergraduate |

## 3.2 DATASET CONSTRUCTION PIPELINE

The dataset was built through a three-stage pipeline, as illustrated in Figure 2.

**Data Sources and Preliminary Processing** To ensure comprehensive coverage across all 60 major mathematical domains, we selected undergraduate and graduate textbooks from authoritative publishers (e.g., Springer) based on the MSC2020 classification system. Our selection prioritized recent editions (post-2010) for contemporary content depth. The selected textbooks were digitized using optical character recognition (OCR) with high reported accuracy. A large language model (DeepSeek) was then employed to automatically identify and extract mathematical propositions along with their complete statements and proofs.

**Automated Formalization and Preliminary Filtering** The automated formalization process uses the kimina-Autoformalizer-7B to generate 20 candidate Lean 4 code snippets for each natural-language mathematical proposition. This step initially produces a candidate pool of 16,949 raw formal expressions. These candidate codes then undergo rigorous syntactic validation using the Lean 4 compiler. Only compilable snippets are retained, resulting in 13,845 syntactically valid statements, each paired with its original problem. This represents a syntactic conversion rate of 81.7% from the initial candidate set. To ensure semantic consistency, we further employ a semantic verification step using the DeepSeek API (model: DeepSeek-V2). For each problem, the Lean 4 code and its original proposition are submitted to the API with the specific instruction: "Determine whether the Lean 4 code is semantically equivalent to the original proposition (Yes/No binary classification)." The API call uses parameters 'temperature=0.7' and 'top-p=0.9'. Only the snippets classified as semantically equivalent (Yes) are retained. The final step of this automated phase yields a high-quality dataset of 6,071 semantically consistent proposition–code pairs, corresponding to a retention rate of 43.8% from the syntactically valid set.

**Human Verification and Reconstruction** From each major MSC domain, we purposefully selected three propositions that best represented the core subfields (180 propositions in total). This approach aimed to ensure disciplinary completeness and conceptual representativeness. We observed that over 90% of the auto-generated code exhibited semantic incompleteness. As a result, all propositions were manually reconstructed and rewritten. The manual verification followed strict criteria to guarantee the quality of formalization. Each proposition was assessed based on:

Logical Equivalence: The formal statement must be logically equivalent to the original natural language proposition.

Precise Formalization of Premises and Conclusions: All assumptions must be explicitly declared as premises, and the conclusion must be precisely formulated.

Independent Compilation: The Lean 4 code must be syntactically correct and compilable by the Lean 4 compiler in isolation.

These criteria collectively define the **completeness of a formalized problem**, which requires: (1) explicit declaration of all premises, (2) precise formalization of the conclusion, and (3) the code being independently checkable by the Lean 4 compiler. The final dataset was constructed through multiple rounds of cross-validation. Each theorem underwent three rounds of review by at least two independent evaluators against the above standards to ensure the correctness and completeness of the formalized statements. This pipeline improved the overall reliability and formalization quality,

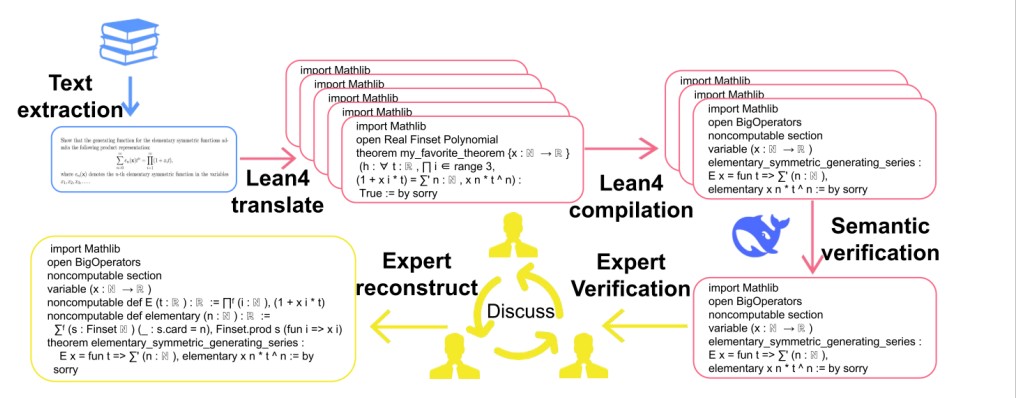

Figure 3: The three-stage dataset construction pipeline: (1) data sourcing and preliminary processing from MSC2020-based textbook selection; (2) automated formalization and initial filtering using the Kimina tool and multi-model semantic alignment; (3) manual verification and reconstruction with expert review.

resulting in a systematic, challenging, and domain-balanced mathematical reasoning benchmark for Lean 4.

## 4 EXPERIMENTS AND ANALYSIS

### 4.1 EVALUATION OF EXISTING ATP MODELS ON MSC-180

#### 4.1.1 MODEL SELECTION

We evaluated several existing automated theorem proving (ATP) models based on three canonical proof generation paradigms:

**Interactive Tactics Generation.** We selected the BFS-Prover 7B (Xin et al., 2025), which formulates proof construction as an interactive tactic generation task within a best-first search (BFS) framework. At each step, the model generates multiple candidate tactics from the current proof state. A verifier then guides the search to select the most promising tactic, incrementally building a complete proof.

**End-to-End Full-Proof Generation.** We selected the Kimina-Prover-Preview-Distill-7B (Wang et al., 2025), which adheres to the single-pass generation (SPG) paradigm. This model directly generates complete Lean 4 proof code in an end-to-end manner, without relying on explicit intermediate states or search. It is trained by distilling long proof trajectories from a more powerful teacher model to internalize holistic proof patterns.

**Hierarchical and Lemma-based Proof Generation.** We selected the DeepSeek-Prover-V2 7B (Ren et al., 2025), which emphasizes structured proof planning. This model generates proofs by incorporating intermediate lemmas and hierarchical reasoning steps, mimicking the human approach of decomposing complex problems into manageable subgoals. Trained to construct such hierarchical structures, it demonstrates enhanced capabilities in complex mathematical reasoning. We evaluated the theorem provers using the Pass@$k$ metric, alongside our custom-defined Domain@$k$ and Coefficient of Variation@$k$ (CV@$k$) metrics. All Lean 4 code compilation and testing were performed using the kimina-lean-server(Wang et al., 2025) to ensure efficient validation.

#### 4.1.2 METRIC

We evaluate the theorem provers using the Pass@$k$ metric, along with two custom metrics: Domain@$k$ (domain coverage) and the Coefficient of Variation@$k$ (CV@$k$). The correctness of the generated Lean code is verified through rapid compilation and testing using the kimina-lean-server.

**Pass@$k$**: This metric measures the proportion of problems solved within a limited number of attempts. The definition of an attempt depends on the proving paradigm: For end-to-end and hierarchical/lemma-based models, $k$ refers to the number of independently sampled proof scripts. For the interactive tactic-generation model (BFS-Prover), the effective attempt count $K$ is derived from the search space scale: $K = N \times S \times T$, where $N$ is the number of parallel searches, $S$ is the number of candidate tactics per state expansion, and $T$ is the maximum number of iterations. This approach is designed to support consistent comparison across different proving paradigms.

**Domain@$k$**: This metric evaluates the breadth of a model's capabilities across mathematical domains. It is defined as the number of distinct domains in which the model solves at least one problem under a given Pass@$k$ setting, divided by the total number of domains in the benchmark (60). A higher Domain@$k$ value indicates wider applicability across different mathematical fields.

**Coefficient of Variation@$k$ (CV@$k$)**: This metric assesses the dispersion of a model's performance across domains. It is defined as $CV@k = \sigma/\mu$, where $\sigma$ is the standard deviation and $\mu$ is the mean of the per-domain Pass@$k$ rates. A low value indicates uniform performance across domains, while a high value points to a high concentration of success in a limited subset of domains.

The calculation proceeds as follows:

1. Calculate the average pass rate ($\mu$):

$$\mu = \frac{1}{D}\sum_{d=1}^{D} \text{Pass}@k_d$$

2. Calculate the standard deviation of cross-domain performance ($\sigma$):

$$\sigma = \sqrt{\frac{1}{D}\sum_{d=1}^{D}(\text{Pass}@k_d - \mu)^2}$$

3. Calculate the coefficient of variation (CV):

$$CV = \frac{\sigma}{\mu}$$

### 4.2 THE PERFORMANCE ON THE MSC-180 DATASET

### 4.3 OVERALL PERFORMANCE ON THE MSC-180 DATASET

The overall performance data presented in Table 2 shows that even the most advanced theorem-proving models achieve a top pass rate of only 19.8% when faced with graduate-level mathematical propositions. This result clearly reveals that current models still face fundamental challenges in automatically generating complete and rigorous formalized proofs. Further analysis of the Domain@k and CV@k metrics delineates the specific patterns of their capability gaps.

**the models exhibit a characteristic of "breadth over depth of understanding"**. For example, DeepSeek-Prover-V2 and Kimina-Prover cover 25 and 23 mathematical subdomains respectively, indicating broad coverage in their training data. However, their pass rates remain low in the vast majority of domains. This pattern of "high coverage yet low pass rates" suggests that the models may be better at memorizing and matching common proof syntax and strategy fragments, rather than deeply understanding the intrinsic logic of mathematical concepts and their precise formalization in Lean 4. Consequently, their performance drops significantly when a proof requires flexibly combining multiple definitions or performing non-standard reasoning.

**different provers show notable differences in stability.** DeepSeek-Prover-V2's lower coefficient of variation (CV=1.27) may stem from its ability to robustly invoke strategies for structurally standard proofs, demonstrating stronger capability in template-based reasoning. Yet, this "stability" tends to break down when faced with problems requiring creative construction, complex induction, or subtle type conversions, often leaving proofs at a skeletal stage. In sharp contrast, the extremely high coefficient of variation of BFS-Prover (CV=1.72) directly exposes the inherent fragility of pure search-based strategies—its success heavily relies on randomly "hitting" effective sequences in the vast strategy space, rather than being guided by deep semantic understanding.

Table 2: Performance comparison of 7B parameter theorem provers on MSC-180 benchmark

| Model | Total Passed | Average Pass Rate | Std Dev | CV |
|---|---|---|---|---|
| DeepSeek-prover 7B | 34 | 0.189 | 0.24 | 1.27 |
| Kimina-Prover-Preview-Distill 7B | 30 | 0.167 | 0.239 | 1.43 |
| BFS-Prover 7B | 8 | 0.044 | 0.077 | 1.72 |

Table 3: Model Performance on Core and Interdisciplinary Mathematical Problems under Different Sampling Scales

| Model | Budget | Total rate | Core rate | Interdisciplinary rate |
|---|---|---|---|---|
| DeepSeek-prover 7B | 8 | 0.150 | 0.111 | 0.241 |
| DeepSeek-prover 7B | 16 | 0.161 | 0.127 | 0.241 |
| DeepSeek-prover 7B | 24 | 0.183 | 0.151 | 0.259 |
| DeepSeek-prover 7B | 32 | 0.189 | 0.159 | 0.259 |
| Kimina-Prover-Preview-Distill 7B | 8 | 0.111 | 0.056 | 0.241 |
| Kimina-Prover-Preview-Distill 7B | 16 | 0.150 | 0.087 | 0.296 |
| Kimina-Prover-Preview-Distill 7B | 24 | 0.156 | 0.095 | 0.296 |
| Kimina-Prover-Preview-Distill 7B | 32 | 0.167 | 0.111 | 0.296 |
| BFS-Prover 7B | 1×8×100 | 0.028 | 0.008 | 0.074 |
| BFS-Prover 7B | 1×16×100 | 0.033 | 0.008 | 0.093 |
| BFS-Prover 7B | 1×24×100 | 0.033 | 0.008 | 0.093 |
| BFS-Prover 7B | 1×32×100 | 0.044 | 0.024 | 0.093 |

## 4.4 RESULTS AND ANALYSIS BY DOMAIN AND DIFFICULTY LEVEL

We categorized the problems into two major classes based on the MSC classification standards: Core Mathematical Disciplines (Pure & Core Mathematics) and Interdisciplinary Applied Disciplines (i.e., fields arising from the intersection of mathematics with other scientific disciplines). We conducted an in-depth analysis of the three theorem provers' performance across different disciplinary areas and difficulty levels (Undergraduate vs. Graduate).

## 4.5 ANALYSIS OF RESULTS BY FIELD AND DIFFICULTY LEVEL

The data in Table 2 show that the performance of all models in core mathematical fields is significantly weaker than in cross-disciplinary applied fields. This phenomenon reveals an important limitation of current models: their capabilities are more inclined towards handling problems with clear computational patterns or structured algorithmic features, rather than dealing with highly abstract conceptual proofs.

A distinct pattern regarding sampling efficiency emerges from the data: its impact differs fundamentally between domains. As shown in Table 2, increasing the sampling budget leads to a much higher performance improvement in core mathematical fields (e.g., DeepSeek-Prover from 11.1% to 15.9%) compared to cross-disciplinary applied fields (from 24.1% to 25.9%). This suggests that core mathematical problems often have more diverse yet equally difficult proof paths; increasing sampling can, to some extent, improve the probability of "hitting" the correct path. In contrast, once a cross-disciplinary applied problem is "understood" by the model, its proof path is likely more standard, resulting in lower marginal benefits from increased sampling.

### 4.5.1 ATTRIBUTION OF PERFORMANCE DIFFERENCES ACROSS DISCIPLINARY FIELDS

**The Challenge of Abstraction in Core Mathematics**: In fields such as Algebra (MSC 13), Number Theory (MSC 11), and Topology (MSC 54), proof construction heavily relies on the rigorous manip-

ulation of abstract axioms and definitions. Models often require dozens of generation attempts yet still struggle to construct complete reasoning chains. This reflects a lack of deep modeling capability for the intrinsic structural relationships between mathematical objects, especially evident when handling multi-step proofs that require recursive or inductive application of definitions and lemmas. For instance, in proofs involving ideal powers, valuation maps, or localization constructions, models frequently fail to correctly concatenate the fundamental properties of ring theory and field theory, leading to broken proof logic or circular arguments.

**The Relative Advantage in Cross-Disciplinary Applied Fields**: In fields such as Mathematical Physics (MSC 70) and Computer Science-related Mathematics (MSC 68), problems are often modeled as equations, algorithms, or discrete structures. Their proof processes typically involve more computational steps, determinate inductive patterns, or structures similar to programming languages. For example, DeepSeek-Prover successfully handled complex algebraic calculations and physical parameter transformations in the MSC 70 problem "relativistic scattering energy ratio." This success may be attributed to the richness of such "procedural" content in its pre-training data, as its proof strategy shares certain structural similarities with program code generation.

### 4.5.2 PERFORMANCE BY DIFFICULTY LEVEL

The pass rates of all models on graduate-level problems ($\leq 20\%$) are significantly lower than those on undergraduate-level problems ($\leq 40\%$). This gap arises not only from the increased complexity of the problems themselves but, more importantly, from the higher demands graduate-level problems place on the completeness and rigor of formalized proofs. Specifically, such problems typically require finer-grained assumption management, more complex nested quantifier logic, and deeper invocation of advanced theorems in the Mathlib library. Models often struggle to maintain the overall logical coherence of longer proofs and fail to adequately account for all implicit premises. Consequently, their performance declines markedly when confronted with such challenges requiring high levels of abstraction and rigorous reasoning.

### 4.6 INVESTIGATION INTO BFS-PROVER 7B

Based on detailed performance data of BFS-Prover 7B under the settings of 1×32×100 and 1×100×100, the model demonstrated relatively apparent instability and strategic limitations.

Under the 1×32×100 setting, the model solved only 8 problems. When the search breadth was expanded to 100, the number of solved problems increased by merely 3 (totaling 11). Concurrently, the model's overall pass rate on the benchmark only slightly increased from 4.44% to 6.11%. This indicates that the returns from simply expanding the brute-force search scope are quite limited

Although the model achieved a pass rate of approximately 50% on a few specific problems, its success on many other problems often depended on a single correct proof discovered by chance. This instability suggests that, due to the lack of deep reasoning capabilities, the BFS method is relatively sensitive to search randomness. Collectively, these performances demonstrate that pure search strategies struggle to systematically solve higher-level abstract mathematical reasoning problems

## 5 DISCUSSION AND OUTLOOK

The findings of this study establish a valuable benchmark for comprehensively assessing the capabilities of current models in formal mathematical reasoning. The results indicate that automated theorem provers are at a critical juncture—filled with potential yet requiring fundamental breakthroughs.

**Demonstrated Potential and Strengths**

The initial success of diverse methods is promising: three distinct proving paradigms (interactive search, end-to-end generation, and hierarchical planning) each succeeded on certain problems, demonstrating the potential of LLM to form a foundation for automated theorem proving. Models exhibited robust performance on well-structured problems. DeepSeek-Prover-V2, in particular, showed strong reasoning in applied mathematics domains like physics and computer science, highlighting the applicability of LLMs to problems with clear constraints. A preliminary generalization

ability was also observed, with Kimina-Prover and DeepSeek-Prover-V2 covering 23 and 25 different mathematical domains, respectively, suggesting a capacity for transferring learned reasoning patterns.

**Challenges and Opportunities for Breakthrough**

A significant challenge is that existing models still lack sufficient abstract reasoning capabilities and mathematical intuition. The overall low performance (Pass@$k$ below 20%) on the MSC-180 benchmark indicates that models rely, to some extent, on pattern matching and local reasoning, lacking the deep abstract thinking and creative intuition characteristic of human mathematicians. This inability to systematically transfer knowledge to novel or highly abstract problems remains a major obstacle Furthermore, the models' generalization ability is insufficient; with Domain@k not exceeding 50%, most models effectively function as 'specialists' in limited domains. They may perform well within areas covered by their training data but fail to generalize reliably across broader mathematical fields.

https://anonymous.4open.science/r/MSC-180-34F3

## 5.1 REPRODUCIBILITY STATEMENT

To ensure the reproducibility of our work, we have provided the complete source code in the supplementary materials. An anonymized version of the code is also available at the following link: `https://anonymous.4open.science/r/MSC-180-34F3`. Detailed experimental setups, hyperparameters, and prompts used for LLM are provided in the appendix.

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

# A  COMPLETE LIST OF 60 MSC CODES COVERED BY MSC-180

03 Mathematical logic and foundations
05 Combinatorics
06 Order, lattices, ordered algebraic structures
08 General algebraic systems
11 Number theory
12 Field theory and polynomials
13 Commutative algebra
14 Algebraic geometry
15 Linear and multilinear algebra; matrix theory
16 Associative rings and algebras
17 Nonassociative rings and algebras
18 Category theory; homological algebra
19 K-theory
20 Group theory and generalizations
22 Topological groups, Lie groups
26 Real functions
28 Measure and integration
30 Functions of a complex variable
31 Potential theory
32 Several complex variables and analytic spaces
33 Special functions
34 Ordinary differential equations
35 Partial differential equations
37 Dynamical systems and ergodic theory
39 Difference and functional equations
40 Sequences, series, summability
41 Approximations and expansions
42 Harmonic analysis on Euclidean spaces
43 Abstract harmonic analysis
44 Integral transforms, operational calculus
45 Integral equations
46 Functional analysis
47 Operator theory
49 Calculus of variations and optimal control; optimization
51 Geometry
52 Convex and discrete geometry
53 Differential geometry
54 General topology
55 Algebraic topology
57 Manifolds and cell complexes
58 Global analysis, analysis on manifolds
60 Probability theory and stochastic processes
62 Statistics
65 Numerical analysis
68 Computer science
70 Mechanics of particles and systems
74 Mechanics of deformable solids
76 Fluid mechanics
78 Optics, electromagnetic theory
80 Classical thermodynamics, heat transfer
81 Quantum theory
82 Statistical mechanics, structure of matter
83 Relativity and gravitational theory
85 Astronomy and astrophysics
86 Geophysics
90 Operations research, mathematical programming
91 Game theory, economics, social and behavioral sciences

92 Biology and other natural sciences
93 Systems theory; control
94 Information and communication, circuits

## B  Example of Lean 4 Code Generated by Multiple Models

**Lean-statement：**

```
import Mathlib
theorem curve_union : {(x, y, z) : ℝ × ℝ × ℝ | (x + y + z) * (x ^ 2 + y ^ 2 - z ^ 2) = 0} =
    {(x, y, z) | x + y + z = 0} ∪ {(x, y, z) | x ^ 2 + y ^ 2 - z ^ 2 = 0} := by
  have h_main : {(x, y, z) : ℝ × ℝ × ℝ | (x + y + z) * (x ^ 2 + y ^ 2 - z ^ 2) = 0} =
          {(x, y, z) | x + y + z = 0} ∪ {(x, y, z) | x ^ 2 + y ^ 2 - z ^ 2 = 0} := by
    sorry
```

**Lean-statement-solution：(BFS)**

```
import Mathlib
theorem curve_union : {(x, y, z) : ℝ × ℝ × ℝ | (x + y + z) * (x ^ 2 + y ^ 2 - z ^ 2) = 0} =
    {(x, y, z) | x + y + z = 0} ∪ {(x, y, z) | x ^ 2 + y ^ 2 - z ^ 2 = 0} := by
have h_main : {(x, y, z) : ℝ × ℝ × ℝ | (x + y + z) * (x ^ 2 + y ^ 2 - z ^ 2) = 0} =
        {(x, y, z) | x + y + z = 0} ∪ {(x, y, z) | x ^ 2 + y ^ 2 - z ^ 2 = 0} := by
 ext ⟨ x, y, z⟩
 simp [curve_union]
```

**Lean-statement-solution(kimina)：**

```
import Mathlib
theorem curve_union : {(x, y, z) : ℝ × ℝ × ℝ | (x + y + z) * (x ^ 2 + y ^ 2 - z ^ 2) = 0} =
    {(x, y, z) | x + y + z = 0} ∪ {(x, y, z) | x ^ 2 + y ^ 2 - z ^ 2 = 0} := by
have h_main : {(x, y, z) : ℝ × ℝ × ℝ | (x + y + z) * (x ^ 2 + y ^ 2 - z ^ 2) = 0} =
        {(x, y, z) | x + y + z = 0} ∪ {(x, y, z) | x ^ 2 + y ^ 2 - z ^ 2 = 0} := by
 ext ⟨ x, y, z⟩
 simp [mul_eq_zero]
 tauto
```

Figure 4: First part of Lean 4 code comparison generated by different models.

Figures 4 and 5 provide concrete examples illustrating the Lean 4 code generated by three distinct models (BFS, Kimina, and Deepseek) for the same theorem. Our analysis proceeds as follows:

The proof generated by the BFS model exhibits an extremely concise approach in terms of tactic usage, primarily relying on `ext` and `simp [curve_union]`. However, a fundamental issue arises in its proof structure: the `simp` tactic references the very theorem being proved, `curve_union`, thereby creating a circular dependency. This indicates that, in this instance, the model's proof generation mechanism may prioritize code brevity excessively while failing to avoid the logical risk of circular reasoning, ultimately resulting in an invalid proof.

In contrast, the proof generated by the Kimina model demonstrates a better sense of balance. It also begins with the `ext` tactic, but the key distinction lies in its `simp` step correctly referencing the existing theorem `mul_eq_zero` from the standard library to simplify the goal. Subsequently, it employs the `tauto` tactic to automatically handle the resulting logical structure. This combination of strategies not only ensures the correctness of the proof but also maintains code conciseness, reflecting the model's appropriate selection of mathematical theorems and effective use of automation tools.

The Deepseek model, on the other hand, adopts a more detailed, step-by-step proof style. The proof process starts by explicitly applying the `Set.ext` rule, followed by gradually introducing variables and simplifying via `simp only`. Using the `constructor` tactic, the proof goal is clearly divided into two directions, each rigorously addressed through case analysis based on fundamental theorems such as `eq_zero_or_eq_zero_of_mul_eq_zero`. Although this approach increases the length of the proof, each step of reasoning remains highly transparent, enhancing the overall verifiability of the proof process, making it more suitable for scenarios where extreme rigor is required.

## Lean-statement-solution：(deepseek)

```
import Mathlib
theorem curve_union : {(x, y, z) : ℝ × ℝ × ℝ | (x + y + z) * (x ^ 2 + y ^ 2 - z ^ 2) = 0} =
    {(x, y, z) | x + y + z = 0} ∪ {(x, y, z) | x ^ 2 + y ^ 2 - z ^ 2 = 0} := by
  have h_main : {(x, y, z) : ℝ × ℝ × ℝ | (x + y + z) * (x ^ 2 + y ^ 2 - z ^ 2) = 0} =
        {(x, y, z) | x + y + z = 0} ∪ {(x, y, z) | x ^ 2 + y ^ 2 - z ^ 2 = 0} := by
    apply Set.ext
    intro ⟨ x, y, z⟩
    simp only [Set.mem_setOf_eq, Set.mem_union, Set.mem_setOf_eq]
    constructor
    ·
      intro h
      have h₁  : (x + y + z) * (x ^ 2 + y ^ 2 - z ^ 2) = 0 := by simpa using h
      have h₂  : x + y + z = 0 ∨ x ^ 2 + y ^ 2 - z ^ 2 = 0 := by
        apply eq_zero_or_eq_zero_of_mul_eq_zero h₁
      cases h₂  with
      | inl h₂  => exact Or.inl h₂
      | inr h₂  => exact Or.inr h₂
    ·
      intro h
      cases h with
      | inl h =>
        have h₁  : (x + y + z) * (x ^ 2 + y ^ 2 - z ^ 2) = 0 := by rw [h]; ring
        exact h₁
      | inr h =>
        have h₁  : (x + y + z) * (x ^ 2 + y ^ 2 - z ^ 2) = 0 := by rw [h]; ring
        exact h₁
  exact h_main
```

Figure 5: Second part of Lean 4 code comparison showing detailed proof structures.

## C  PERFORMANCE ACROSS DIFFICULTY LEVEL

Figure 6 illustrates a comparative analysis of the performance of three theorem proving models—BFS, Kimina, and DeepSeek—on undergraduate-level (n=35) and graduate-level (n=145) mathematical problem sets.

**Deepseek:** When the value of k increased from 8 to 32, the total number of problems solved by the DeepSeek model rose from 27 to 34 (an increase of 7 problems). Among these, the number of undergraduate-level problems solved increased slightly from 10 to 11 (a gain of 1 problem), while the number of graduate-level problems solved grew from 17 to 23 (an increase of 6 problems). The larger growth in graduate-level problems suggests that increasing k may be more effective for addressing graduate-level questions. For undergraduate problems, the model's performance remained relatively stable, with a maximum of 11 problems solved and a normalized success rate of 31.4% (11/35). For graduate problems, the maximum number solved was 23, corresponding to a normalized success rate of 15.9% (23/145). Overall, DeepSeek demonstrated strong performance on undergraduate problems (with a success rate consistently above 30%), and while its performance on graduate problems improved with higher k values, the success rate remained lower than that for undergraduate problems. In comprehensive comparison, DeepSeek exhibited the best overall performance among the three models.

**Kimina:** As k increased from 8 to 32, the total number of problems solved by the Kimina model increased from 20 to 30 (a gain of 10 problems). The number of graduate-level problems solved showed significant growth, rising from 14 to 23 (an increase of 9 problems), while the number of undergraduate-level problems solved only increased marginally from 6 to 7 (a gain of 1 problem). This pronounced improvement in graduate-level problems may indicate that the Kimina model is more sensitive to higher k values; meanwhile, undergraduate-level performance changed very little, with a maximum of 7 problems solved and a normalized success rate of 20.0% (7/35). On graduate-level problems, when k=32, Kimina's number of solved problems (23) matched that of DeepSeek, with a normalized success rate of 15.9% (23/145). However, on undergraduate problems, Kim-

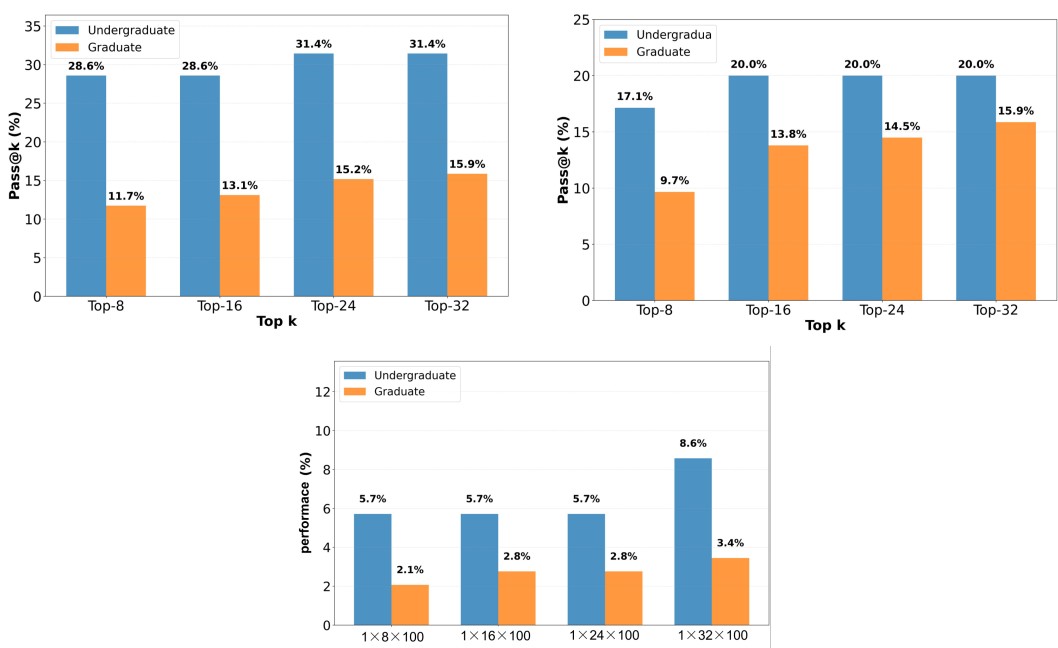

Figure 6: Performance comparison of three theorem proving models across educational backgrounds. Success rates are normalized by population size (undergraduate: 35, graduate: 145) and shown for increasing pass rate thresholds. Deepseek (left) and Kimina (right) show similar trends with Deepseek outperforming, while BFS (bottom) exhibits significantly lower performance across all evaluation levels.

ina's success rate (20.0%) was noticeably lower than DeepSeek's (31.4%), indicating that Kimina's overall performance is relatively weaker, particularly on undergraduate-level questions.

**BFS:** Under different search configurations (such as Pass@1×32×100), the BFS model's number of solved problems remained consistently very low and showed no significant improvement as k increased: the total number of solved problems rose slightly from 5 to 8, undergraduate-level problems solved increased from 2 to 3, and graduate-level problems solved increased from 3 to 5. The normalized success rates reached a maximum of 8.6% (3/35) for undergraduate problems and 3.4% (5/145) for graduate problems. Even with higher k values, there was no substantial increase in the number of problems solved, suggesting that a pure search-based strategy may face significant challenges in current theorem-proving tasks. Overall, the BFS model performed poorly across all configurations, with success rates below 10%, far inferior to the other models. This is likely due to its inefficiency in navigating the vast search spaces of complex mathematical problems.

## D EXPERIMENTAL SETUP

**Experimental Environment:**

Our experimental setup utilizes a single NVIDIA A100 80GB GPU as the hardware foundation. The key software stack includes PyTorch 2.0+ (requiring CUDA 12.1 or higher), Transformers 4.37.2, and vLLM 0.7.2, among other essential libraries, all deployed and running on a Linux operating system.

**Hyperparameter Settings:**

Kimina-Prover: the temperature is set to 0.7, Top-p is set to 0.95, the maximum token length for a single generation is limited to 8096 tokens, and the batch size is set to 10.

DeepSeek-V2: Its hyperparameter settings include a temperature of 0.7, Top-p of 0.9, a maximum generation length of 2048 tokens, and a batch size of 8.

BFS-Prover: This proving system generates k candidate strategy samples for each problem. Its hyperparameters include a temperature of 0.7 and a maximum generation length of 512 tokens.

# E STATEMENT ON THE USE OF LARGE LANGUAGE MODELS

During the preparation of this research, we utilized large language models (LLMs), specifically DeepSeek, as an auxiliary tool. More precisely, the LLM was employed for:

Text Polishing: Grammar checking, wording improvement, and fluency enhancement of certain paragraphs in the initial draft. However, all core academic content, arguments, and analyses were independently developed by the authors.

The LLM played only a supporting role in this work and was not involved in research conception, theoretical derivation, experimental design, result analysis, or the formation of scientific conclusions. The authors assume full responsibility for all content presented in this paper.

