# MSC2020-Mathematics Subject Classification System

## Associate Editors of Mathematical Reviews and zbMATH

**00** General and overarching topics; collections

**01** History and biography

**03** Mathematical logic and foundations

**05** Combinatorics

**06** Order, lattices, ordered algebraic structures

**08** General algebraic systems

**11** Number theory

**12** Field theory and polynomials

**13** Commutative algebra

**14** Algebraic geometry

**15** Linear and multilinear algebra; matrix theory

**16** Associative rings and algebras

**17** Nonassociative rings and algebras

**18** Category theory; homological algebra

**19** $K$-theory

**20** Group theory and generalizations

**22** Topological groups, Lie groups

**26** Real functions

**28** Measure and integration

**30** Functions of a complex variable

**31** Potential theory

**32** Several complex variables and analytic spaces

**33** Special functions

**34** Ordinary differential equations

**35** Partial differential equations

**37** Dynamical systems and ergodic theory

**39** Difference and functional equations

**40** Sequences, series, summability

**41** Approximations and expansions

**42** Harmonic analysis on Euclidean spaces

**43** Abstract harmonic analysis

**44** Integral transforms, operational calculus

**45** Integral equations

**46** Functional analysis

**47** Operator theory

**49** Calculus of variations and optimal control; optimization

**51** Geometry

**52** Convex and discrete geometry

**53** Differential geometry

**54** General topology

**55** Algebraic topology

**57** Manifolds and cell complexes

**58** Global analysis, analysis on manifolds

**60** Probability theory and stochastic processes

**62** Statistics

**65** Numerical analysis

**68** Computer science

**70** Mechanics of particles and systems

**74** Mechanics of deformable solids

**76** Fluid mechanics

**78** Optics, electromagnetic theory

**80** Classical thermodynamics, heat transfer

**81** Quantum theory

**82** Statistical mechanics, structure of matter

**83** Relativity and gravitational theory

**85** Astronomy and astrophysics

**86** Geophysics

**90** Operations research, mathematical programming

**91** Game theory, economics, social and behavioral sciences

**92** Biology and other natural sciences

**93** Systems theory; control

**94** Information and communication, circuits

**97** Mathematics education

This document is a printed form of MSC2020, an MSC revision produced jointly by the editorial staffs of Mathematical Reviews (MR) and Zentralblatt für Mathematik (zbMATH) in consultation with the mathematical community. The goals of this revision of the Mathematics Subject Classification (MSC) were set out in the announcement of it and call for comments by the Executive Editor of MR and the Chief Editor of zbMATH in July 2016. This document results from the MSC revision process that has been going on since then. MSC2020 will be fully deployed from January 2020.

The editors of MR and zbMATH deploying this revision therefore ask for feedback on remaining errors to help in this work, which should be given through e-mail to feedback@msc2020.org. They are grateful for the many suggestions that were received previously, which have greatly influenced what we have.

# How to use the Mathematics Subject Classification [MSC]

The main purpose of the classification of items in the mathematical literature using the Mathematics Subject Classification scheme is to help users find the items of present or potential interest to them as readily as possible—in products derived from the Mathematical Reviews Database (MRDB) such as MathSciNet, in Zentralblatt MATH (zbMATH), or anywhere else where this classification scheme is used. An item in the mathematical literature should be classified so as to attract the attention of all those possibly interested in it. The item may be something that falls squarely within one clear area of the MSC, or it may involve several areas. Ideally, the MSC codes attached to an item should represent the subjects to which the item contains a contribution. The classification should serve both those closely concerned with specific subject areas, and those familiar enough with subjects to apply their results and methods elsewhere, inside or outside of mathematics. It will be extremely useful for both users and classifiers to familiarize themselves with the entire classification system and thus to become aware of all the classifications of possible interest to them. Every item in the MRDB or zbMATH receives precisely one primary classification, which is simply the MSC code that describes its principal contribution. When an item contains several principal contributions to different areas, the primary classification should cover the most important among them. A paper or book may be assigned one or several secondary classification numbers to cover any remaining principal contributions, ancillary results, motivation or origin of the matters discussed, intended or potential field of application, or other significant aspects worthy of notice. The principal contribution is meant to be the one including the most important part of the work actually done in the item. For example, a paper whose main overall content is the solution of a problem in graph theory, which arose in computer science and whose solution is (perhaps) at present only of interest to computer scientists, would have a primary classification in 05C (Graph Theory) with one or more secondary classifications in 68 (Computer Science); conversely, a paper whose overall content lies mainly in computer science should receive a primary classification in 68, even if it makes heavy use of graph theory and proves several new graph-theoretic results along the way. There are two types of cross-references given at the end of many of the MSC2020 entries in the MSC. The first type is in braces: "{For A, see X}"; if this appears in section Y, it means that contributions described by A should usually be assigned the classification code X, not Y. The other type of cross-reference merely points out related classifications; it is in brackets: "[See also ... ]", "[See mainly ... ]", etc., and the classification codes listed in the brackets may, but need not, be included in the classification codes of a paper, or they may be used in place of the classification where the cross-reference is given. The classifier must judge which classification is the most appropriate for the paper at hand.

# 00-XX General and overarching topics; collections

**00-01** Introductory exposition (textbooks, tutorial papers, etc.) pertaining to mathematics in general

**00-02** Research exposition (monographs, survey articles) pertaining to mathematics in general

## 00Axx General and miscellaneous specific topics

**00A05** Mathematics in general

**00A06** Mathematics for nonmathematicians (engineering, social sciences, etc.)

**00A07** Problem books {For open problems, see 00A27}

**00A08** Recreational mathematics

**00A09** Popularization of mathematics

**00A15** Bibliographies for mathematics in general [See also 01A70 and the classification number –00 in the other sections]

**00A17** External book reviews

**00A20** Dictionaries and other general reference works [See also the classification number –00 in the other sections]

**00A22** Formularies

**00A27** Lists of open problems

**00A30** Philosophy of mathematics [See also 03A05]

**00A35** Methodology of mathematics {For mathematics education, see 97-XX}

**00A64** Mathematics and literature

**00A65** Mathematics and music

**00A66** Mathematics and visual arts

**00A67** Mathematics and architecture

**00A69** General applied mathematics {For physics, see 00A79 and Sections 70 through 86}

**00A71** General theory of mathematical modeling

**00A72** General theory of simulation

**00A79** Physics [Use more specific entries from Sections 70 through 86 when possible]

**00A99** None of the above, but in this section

## 00Bxx Conference proceedings and collections of articles

**00B05** Collections of abstracts of lectures

**00B10** Collections of articles of general interest

**00B15** Collections of articles of miscellaneous specific interest

**00B20** Proceedings of conferences of general interest

**00B25** Proceedings of conferences of miscellaneous specific interest

**00B30** Festschriften

**00B50** Collections of translated articles of general interest

**00B55** Collections of translated articles of miscellaneous specific interest

**00B60** Collections of reprinted articles [See also 01A75]

**00B99** None of the above, but in this section

# 01-XX History and biography [See also the classification number −03 in the other sections]

**01-00** General reference works (handbooks, dictionaries, bibliographies, etc.) pertaining to history and biography

**01-01** Introductory exposition (textbooks, tutorial papers, etc.) pertaining to history and biography

**01-02** Research exposition (monographs, survey articles) pertaining to history and biography

**01-06** Proceedings, conferences, collections, etc. pertaining to history and biography

**01-11** Research data for problems pertaining to history and biography

## 01Axx History of mathematics and mathematicians

**01A05** General histories, source books

**01A07** Ethnomathematics (general)

**01A10** History of mathematics in Paleolithic and Neolithic times

**01A11** History of mathematics of the indigenous cultures of Africa, Asia, and Oceania

**01A12** History of mathematics of the indigenous cultures of the Americas

**01A15** History of mathematics of the indigenous cultures of Europe (pre-Greek, etc.)

**01A16** History of Egyptian mathematics

**01A17** History of Babylonian mathematics

**01A20** History of Greek and Roman mathematics

**01A25** History of Chinese mathematics

**01A27** History of Japanese mathematics

**01A29** History of East and Southeast Asian mathematics (non-Chinese, non-Japanese)

**01A30** History of mathematics in the Golden Age of Islam

**01A32** History of Indian mathematics

**01A35** History of mathematics in Late Antiquity and medieval Europe

**01A40** History of mathematics in the 15th and 16th centuries, Renaissance

**01A45** History of mathematics in the 17th century

**01A50** History of mathematics in the 18th century

**01A55** History of mathematics in the 19th century

**01A60** History of mathematics in the 20th century

**01A61** History of mathematics in the 21st century

**01A65** Development of contemporary mathematics

**01A67** Future perspectives in mathematics

**01A70** Biographies, obituaries, personalia, bibliographies

**01A72** Schools of mathematics

**01A73** History of mathematics at specific universities

**01A74** History of mathematics at institutions and academies (non-university)

**01A75** Collected or selected works; reprintings or translations of classics [See also 00B60]

**01A80** Sociology (and profession) of mathematics

**01A85** Historiography

**01A90** Bibliographic studies

**01A99** None of the above, but in this section

# 03-XX Mathematical logic and foundations

**03-00** General reference works (handbooks, dictionaries, bibliographies, etc.) pertaining to mathematical logic and foundations

**03-01** Introductory exposition (textbooks, tutorial papers, etc.) pertaining to mathematical logic and foundations

**03-02** Research exposition (monographs, survey articles) pertaining to mathematical logic and foundations

**03-03** History of mathematical logic and foundations [Consider also classification numbers from Section 01]

**03-04** Software, source code, etc. for problems pertaining to mathematical logic and foundations

**03-06** Proceedings, conferences, collections, etc. pertaining to mathematical logic and foundations

**03-08** Computational methods for problems pertaining to mathematical logic and foundations

**03-11** Research data for problems pertaining to mathematical logic and foundations

## 03Axx Philosophical aspects of logic and foundations

**03A05** Philosophical and critical aspects of logic and foundations {For philosophy of mathematics, see also 00A30}

**03A10** Logic in the philosophy of science

**03A99** None of the above, but in this section

## 03Bxx General logic

**03B05** Classical propositional logic

**03B10** Classical first-order logic

**03B16** Higher-order logic

**03B20** Subsystems of classical logic (including intuitionistic logic)

**03B22** Abstract deductive systems

**03B25** Decidability of theories and sets of sentences [See also 11U05, 12L05, 20F10]

**03B30** Foundations of classical theories (including reverse mathematics) [See also 03F35]

**03B35** Mechanization of proofs and logical operations [See also 68V15]

**03B38** Type theory

**03B40** Combinatory logic and lambda calculus [See also 68N18]

**03B42** Logics of knowledge and belief (including belief change)

**03B44** Temporal logic

**03B45** Modal logic (including the logic of norms) {For knowledge and belief, see 03B42; for temporal logic, see 03B44; for provability logic, see also 03F45}

**03B47** Substructural logics (including relevance, entailment, linear logic, Lambek calculus, BCK and BCI logics) {For proof-theoretic aspects, see 03F52}

**03B48** Probability and inductive logic [See also 60A05]

**03B50** Many-valued logic

**03B52** Fuzzy logic; logic of vagueness [See also 68T27, 68T37, 94D05]

**03B53** Paraconsistent logics

**03B55** Intermediate logics

**03B60** Other nonclassical logic

**03B62** Combined logics

**03B65** Logic of natural languages [See also 68T50, 91F20]

**03B70** Logic in computer science [See also 68-XX]

**03B80** Other applications of logic

**03B99** None of the above, but in this section

## 03Cxx Model theory

**03C05** Equational classes, universal algebra in model theory [See also 08Axx, 08Bxx, 18C05]

**03C07** Basic properties of first-order languages and structures

**03C10** Quantifier elimination, model completeness, and related topics

**03C13** Model theory of finite structures [See also 68Q15, 68Q19]

**03C15** Model theory of denumerable and separable structures

**03C20** Ultraproducts and related constructions

**03C25** Model-theoretic forcing

**03C30** Other model constructions

**03C35** Categoricity and completeness of theories

**03C40** Interpolation, preservation, definability

**03C45** Classification theory, stability, and related concepts in model theory [See also 03C48]

**03C48** Abstract elementary classes and related topics [See also 03C45]

**03C50** Models with special properties (saturated, rigid, etc.)

**03C52** Properties of classes of models

**03C55** Set-theoretic model theory

**03C57** Computable structure theory, computable model theory [See also 03D45]

**03C60** Model-theoretic algebra [See also 08C10, 12Lxx, 13L05]

**03C62** Models of arithmetic and set theory [See also 03Hxx]

**03C64** Model theory of ordered structures; o-minimality

**03C65** Models of other mathematical theories

**03C66** Continuous model theory, model theory of metric structures

**03C68** Other classical first-order model theory

**03C70** Logic on admissible sets

**03C75** Other infinitary logic

**03C80** Logic with extra quantifiers and operators [See also 03B42, 03B44, 03B45, 03B48]

**03C85** Second- and higher-order model theory

**03C90** Nonclassical models (Boolean-valued, sheaf, etc.)

**03C95** Abstract model theory

**03C98** Applications of model theory [See also 03C60]

**03C99** None of the above, but in this section

## 03Dxx Computability and recursion theory

**03D03** Thue and Post systems, etc.

**03D05** Automata and formal grammars in connection with logical questions [See also 68Q45, 68Q70, 68R15]

**03D10** Turing machines and related notions [See also 68Q04]

**03D15** Complexity of computation (including implicit computational complexity) [See also 68Q15, 68Q17]

**03D20** Recursive functions and relations, subrecursive hierarchies

**03D25** Recursively (computably) enumerable sets and degrees

**03D28** Other Turing degree structures

**03D30** Other degrees and reducibilities in computability and recursion theory

**03D32** Algorithmic randomness and dimension [See also 68Q30]

**03D35** Undecidability and degrees of sets of sentences

**03D40** Word problems, etc. in computability and recursion theory [See also 06B25, 08A50, 20F10, 68R15]

**03D45** Theory of numerations, effectively presented structures [See also 03C57] {For intuitionistic and similar approaches, see 03F55}

**03D50** Recursive equivalence types of sets and structures, isols

**03D55** Hierarchies of computability and definability

**03D60** Computability and recursion theory on ordinals, admissible sets, etc.

**03D65** Higher-type and set recursion theory

**03D70** Inductive definability

**03D75** Abstract and axiomatic computability and recursion theory

**03D78** Computation over the reals, computable analysis {For constructive aspects, see 03F60}

**03D80** Applications of computability and recursion theory

**03D99** None of the above, but in this section

## 03Exx Set theory

**03E02** Partition relations

**03E04** Ordered sets and their cofinalities; pcf theory

**03E05** Other combinatorial set theory

**03E10** Ordinal and cardinal numbers

**03E15** Descriptive set theory [See also 28A05, 54H05]

**03E17** Cardinal characteristics of the continuum

**03E20** Other classical set theory (including functions, relations, and set algebra)

**03E25** Axiom of choice and related propositions

**03E30** Axiomatics of classical set theory and its fragments

**03E35** Consistency and independence results

**03E40** Other aspects of forcing and Boolean-valued models

**03E45** Inner models, including constructibility, ordinal definability, and core models

**03E47** Other notions of set-theoretic definability

**03E50** Continuum hypothesis and Martin's axiom [See also 03E57]

**03E55** Large cardinals

**03E57** Generic absoluteness and forcing axioms [See also 03E50]

**03E60** Determinacy principles

**03E65** Other set-theoretic hypotheses and axioms

**03E70** Nonclassical and second-order set theories

**03E72** Theory of fuzzy sets, etc.

**03E75** Applications of set theory

**03E99** None of the above, but in this section

## 03Fxx Proof theory and constructive mathematics

**03F03** Proof theory in general (including proof-theoretic semantics)

**03F05** Cut-elimination and normal-form theorems

**03F07** Structure of proofs

**03F10** Functionals in proof theory

**03F15** Recursive ordinals and ordinal notations

**03F20** Complexity of proofs

**03F25** Relative consistency and interpretations

**03F30** First-order arithmetic and fragments

**03F35** Second- and higher-order arithmetic and fragments [See also 03B30]

**03F40** Gödel numberings and issues of incompleteness

**03F45** Provability logics and related algebras (e.g., diagonalizable algebras) [See also 03B45, 03G25, 06E25]

**03F50** Metamathematics of constructive systems

**03F52** Proof-theoretic aspects of linear logic and other substructural logics [See also 03B47]

**03F55** Intuitionistic mathematics

**03F60** Constructive and recursive analysis [See also 03B30, 03D45, 03D78, 26E40, 46S30, 47S30]

**03F65** Other constructive mathematics [See also 03D45]

**03F99** None of the above, but in this section

### 03Gxx Algebraic logic

**03G05** Logical aspects of Boolean algebras [See also 06Exx]

**03G10** Logical aspects of lattices and related structures [See also 06Bxx]

**03G12** Quantum logic [See also 06C15, 81P10]

**03G15** Cylindric and polyadic algebras; relation algebras

**03G20** Logical aspects of Łukasiewicz and Post algebras [See also 06D25, 06D30]

**03G25** Other algebras related to logic [See also 03F45, 06D20, 06E25, 06F35]

**03G27** Abstract algebraic logic

**03G30** Categorical logic, topoi [See also 18B25, 18C05, 18C10]

**03G99** None of the above, but in this section

### 03Hxx Nonstandard models [See also 03C62]

**03H05** Nonstandard models in mathematics [See also 26E35, 28E05, 30G06, 46S20, 47S20, 54J05]

**03H10** Other applications of nonstandard models (economics, physics, etc.)

**03H15** Nonstandard models of arithmetic [See also 11U10, 12L15, 13L05]

**03H99** None of the above, but in this section

# 05-XX Combinatorics {For finite fields, see 11Txx}

**05-00** General reference works (handbooks, dictionaries, bibliographies, etc.) pertaining to combinatorics

**05-01** Introductory exposition (textbooks, tutorial papers, etc.) pertaining to combinatorics

**05-02** Research exposition (monographs, survey articles) pertaining to combinatorics

**05-03** History of combinatorics [Consider also classification numbers from Section 01]

**05-04** Software, source code, etc. for problems pertaining to combinatorics

**05-06** Proceedings, conferences, collections, etc. pertaining to combinatorics

**05-08** Computational methods for problems pertaining to combinatorics

**05-11** Research data for problems pertaining to combinatorics

### 05Axx Enumerative combinatorics {For enumeration in graph theory, see 05C30}

**05A05** Permutations, words, matrices

**05A10** Factorials, binomial coefficients, combinatorial functions [See also 11B65, 33Cxx]

**05A15** Exact enumeration problems, generating functions [See also 33Cxx, 33Dxx]

**05A16** Asymptotic enumeration

**05A17** Combinatorial aspects of partitions of integers [See also 11P81, 11P82, 11P83]

**05A18** Partitions of sets

**05A19** Combinatorial identities, bijective combinatorics

**05A20** Combinatorial inequalities

**05A30** *q*-calculus and related topics [See also 33Dxx]

**05A40** Umbral calculus

**05A99** None of the above, but in this section

## 05Bxx Designs and configurations {For applications of design theory, see 94C30}

**05B05** Combinatorial aspects of block designs [See also 51E05, 62K10]

**05B07** Triple systems

**05B10** Combinatorial aspects of difference sets (number-theoretic, group-theoretic, etc.) [See also 11B13]

**05B15** Orthogonal arrays, Latin squares, Room squares

**05B20** Combinatorial aspects of matrices (incidence, Hadamard, etc.)

**05B25** Combinatorial aspects of finite geometries [See also 51D20, 51Exx]

**05B30** Other designs, configurations [See also 51E30]

**05B35** Combinatorial aspects of matroids and geometric lattices [See also 52B40, 90C27]

**05B40** Combinatorial aspects of packing and covering [See also 11H31, 52C15, 52C17]

**05B45** Combinatorial aspects of tessellation and tiling problems [See also 52C20, 52C22]

**05B50** Polyominoes

**05B99** None of the above, but in this section

## 05Cxx Graph theory {For computer science, see 68R10}

**05C05** Trees

**05C07** Vertex degrees [See also 05E30]

**05C09** Graphical indices (Wiener index, Zagreb index, Randić index, etc.)

**05C10** Planar graphs; geometric and topological aspects of graph theory [See also 57K10, 57M15]

**05C12** Distance in graphs

**05C15** Coloring of graphs and hypergraphs

**05C17** Perfect graphs

**05C20** Directed graphs (digraphs), tournaments

**05C21** Flows in graphs

**05C22** Signed and weighted graphs

**05C25** Graphs and abstract algebra (groups, rings, fields, etc.) [See also 20F65]

**05C30** Enumeration in graph theory

**05C31** Graph polynomials

**05C35** Extremal problems in graph theory [See also 90C35]

**05C38** Paths and cycles [See also 90B10]

**05C40** Connectivity

**05C42** Density (toughness, etc.)

**05C45** Eulerian and Hamiltonian graphs

**05C48** Expander graphs

**05C50** Graphs and linear algebra (matrices, eigenvalues, etc.)

**05C51** Graph designs and isomorphic decomposition [See also 05B30]

**05C55** Generalized Ramsey theory [See also 05D10]

**05C57** Games on graphs (graph-theoretic aspects) [See also 91A43, 91A46]

**05C60** Isomorphism problems in graph theory (reconstruction conjecture, etc.) and homomorphisms (subgraph embedding, etc.)

**05C62** Graph representations (geometric and intersection representations, etc.) {For graph drawing, see also 68R10}

**05C63** Infinite graphs

**05C65** Hypergraphs

**05C69** Vertex subsets with special properties (dominating sets, independent sets, cliques, etc.)

**05C70** Edge subsets with special properties (factorization, matching, partitioning, covering and packing, etc.)

**05C72** Fractional graph theory, fuzzy graph theory

**05C75** Structural characterization of families of graphs

**05C76** Graph operations (line graphs, products, etc.)

**05C78** Graph labelling (graceful graphs, bandwidth, etc.)

**05C80** Random graphs (graph-theoretic aspects) [See also 60B20]

**05C81** Random walks on graphs

**05C82** Small world graphs, complex networks (graph-theoretic aspects) [See also 90Bxx, 91D30]

**05C83** Graph minors

**05C85** Graph algorithms (graph-theoretic aspects) [See also 68R10, 68W05]

**05C90** Applications of graph theory [See also 68R10, 81Q30, 82B20, 82C20, 90C35, 92E10, 94C15]

**05C92** Chemical graph theory [See also 92E10]

**05C99** None of the above, but in this section

## 05Dxx Extremal combinatorics

**05D05** Extremal set theory

**05D10** Ramsey theory [See also 05C55]

**05D15** Transversal (matching) theory

**05D40** Probabilistic methods in extremal combinatorics, including polynomial methods (combinatorial Nullstellensatz, etc.)

**05D99** None of the above, but in this section

### 05Exx Algebraic combinatorics

**05E05** Symmetric functions and generalizations

**05E10** Combinatorial aspects of representation theory [See also 20C30]

**05E14** Combinatorial aspects of algebraic geometry [See also 14Nxx]

**05E16** Combinatorial aspects of groups and algebras [See also 22E45, 33C80]

**05E18** Group actions on combinatorial structures

**05E30** Association schemes, strongly regular graphs

**05E40** Combinatorial aspects of commutative algebra

**05E45** Combinatorial aspects of simplicial complexes

**05E99** None of the above, but in this section

# 06-XX Order, lattices, ordered algebraic structures [See also 18B35]

**06-00** General reference works (handbooks, dictionaries, bibliographies, etc.) pertaining to ordered structures

**06-01** Introductory exposition (textbooks, tutorial papers, etc.) pertaining to ordered structures

**06-02** Research exposition (monographs, survey articles) pertaining to ordered structures

**06-03** History of ordered structures [Consider also classification numbers from Section 01]

**06-04** Software, source code, etc. for problems pertaining to ordered structures

**06-06** Proceedings, conferences, collections, etc. pertaining to ordered structures

**06-08** Computational methods for problems pertaining to ordered structures

**06-11** Research data for problems pertaining to ordered structures

### 06Axx Ordered sets

**06A05** Total orders

**06A06** Partial orders, general

**06A07** Combinatorics of partially ordered sets

**06A11** Algebraic aspects of posets

**06A12** Semilattices [See also 20M10] {For topological semilattices, see 22A26}

**06A15** Galois correspondences, closure operators (in relation to ordered sets)

**06A75** Generalizations of ordered sets

**06A99** None of the above, but in this section

# 06Bxx Lattices [See also 03G10]

**06B05** Structure theory of lattices

**06B10** Lattice ideals, congruence relations

**06B15** Representation theory of lattices

**06B20** Varieties of lattices

**06B23** Complete lattices, completions

**06B25** Free lattices, projective lattices, word problems [See also 03D40, 08A50, 20F10]

**06B30** Topological lattices [See also 06F30, 22A26, 54F05, 54H12]

**06B35** Continuous lattices and posets, applications [See also 06B30, 06D10, 06F30, 18B35, 22A26, 68Q55]

**06B75** Generalizations of lattices

**06B99** None of the above, but in this section

# 06Cxx Modular lattices, complemented lattices

**06C05** Modular lattices, Desarguesian lattices

**06C10** Semimodular lattices, geometric lattices

**06C15** Complemented lattices, orthocomplemented lattices and posets [See also 03G12, 81P10]

**06C20** Complemented modular lattices, continuous geometries

**06C99** None of the above, but in this section

# 06Dxx Distributive lattices

**06D05** Structure and representation theory of distributive lattices

**06D10** Complete distributivity

**06D15** Pseudocomplemented lattices

**06D20** Heyting algebras (lattice-theoretic aspects) [See also 03G25]

**06D22** Frames, locales {For topological questions, see 54-XX}

**06D25** Post algebras (lattice-theoretic aspects) [See also 03G20]

**06D30** De Morgan algebras, Łukasiewicz algebras (lattice-theoretic aspects) [See also 03G20]

**06D35** MV-algebras

**06D50** Lattices and duality

**06D72** Fuzzy lattices (soft algebras) and related topics

**06D75** Other generalizations of distributive lattices

**06D99** None of the above, but in this section

## 06Exx Boolean algebras (Boolean rings) [See also 03G05]

**06E05** Structure theory of Boolean algebras

**06E10** Chain conditions, complete algebras

**06E15** Stone spaces (Boolean spaces) and related structures

**06E20** Ring-theoretic properties of Boolean algebras [See also 16E50, 16G30]

**06E25** Boolean algebras with additional operations (diagonalizable algebras, etc.) [See also 03G25, 03F45]

**06E30** Boolean functions [See also 94D10]

**06E75** Generalizations of Boolean algebras

**06E99** None of the above, but in this section

## 06Fxx Ordered structures

**06F05** Ordered semigroups and monoids [See also 20Mxx]

**06F07** Quantales

**06F10** Noether lattices

**06F15** Ordered groups [See also 20F60]

**06F20** Ordered abelian groups, Riesz groups, ordered linear spaces [See also 46A40]

**06F25** Ordered rings, algebras, modules {For ordered fields, see 12J15} [See also 13J25, 16W80]

**06F30** Ordered topological structures [See also 06B30, 22A26, 54F05, 54H12]

**06F35** BCK-algebras, BCI-algebras [See also 03G25]

**06F99** None of the above, but in this section

# 08-XX General algebraic systems

**08-00** General reference works (handbooks, dictionaries, bibliographies, etc.) pertaining to general algebraic systems

**08-01** Introductory exposition (textbooks, tutorial papers, etc.) pertaining to general algebraic systems

**08-02** Research exposition (monographs, survey articles) pertaining to general algebraic systems

**08-03** History of general algebraic systems [Consider also classification numbers from Section 01]

**08-04** Software, source code, etc. for problems pertaining to general algebraic systems

**08-06** Proceedings, conferences, collections, etc. pertaining to general algebraic systems

**08-08** Computational methods for problems pertaining to general algebraic systems

**08-11** Research data for problems pertaining to general algebraic systems

## 08Axx Algebraic structures [See also 03C05]

**08A02** Relational systems, laws of composition

**08A05** Structure theory of algebraic structures

**08A30** Subalgebras, congruence relations

**08A35** Automorphisms and endomorphisms of algebraic structures

**08A40** Operations and polynomials in algebraic structures, primal algebras

**08A45** Equational compactness

**08A50** Word problems (aspects of algebraic structures) [See also 03D40, 06B25, 20F10, 68R15]

**08A55** Partial algebras

**08A60** Unary algebras

**08A62** Finitary algebras

**08A65** Infinitary algebras

**08A68** Heterogeneous algebras

**08A70** Applications of universal algebra in computer science

**08A72** Fuzzy algebraic structures

**08A99** None of the above, but in this section

## 08Bxx Varieties [See also 03C05]

**08B05** Equational logic, Mal'tsev conditions

**08B10** Congruence modularity, congruence distributivity

**08B15** Lattices of varieties

**08B20** Free algebras

**08B25** Products, amalgamated products, and other kinds of limits and colimits [See also 18A30]

**08B26** Subdirect products and subdirect irreducibility

**08B30** Injectives, projectives

**08B99** None of the above, but in this section

## 08Cxx Other classes of algebras

**08C05** Categories of algebras [See also 18C05]

**08C10** Axiomatic model classes [See also 03Cxx, in particular 03C60]

**08C15** Quasivarieties

**08C20** Natural dualities for classes of algebras [See also 06E15, 18A40, 22A30]

**08C99** None of the above, but in this section

# 11-XX Number theory

**11-00** General reference works (handbooks, dictionaries, bibliographies, etc.) pertaining to number theory

**11-01** Introductory exposition (textbooks, tutorial papers, etc.) pertaining to number theory

**11-02** Research exposition (monographs, survey articles) pertaining to number theory

**11-03** History of number theory [Consider also classification numbers from Section 01]

**11-04** Software, source code, etc. for problems pertaining to number theory

**11-06** Proceedings, conferences, collections, etc. pertaining to number theory

**11-11** Research data for problems pertaining to number theory

## 11Axx Elementary number theory {For analogues in number fields, see 11R04}

**11A05** Multiplicative structure; Euclidean algorithm; greatest common divisors

**11A07** Congruences; primitive roots; residue systems

**11A15** Power residues, reciprocity

**11A25** Arithmetic functions; related numbers; inversion formulas

**11A41** Primes

**11A51** Factorization; primality

**11A55** Continued fractions {For approximation results, see 11J70} [See also 11K50, 30B70, 40A15]

**11A63** Radix representation; digital problems {For metric results, see 11K16}

**11A67** Other number representations

**11A99** None of the above, but in this section

## 11Bxx Sequences and sets

**11B05** Density, gaps, topology

**11B13** Additive bases, including sumsets [See also 05B10]

**11B25** Arithmetic progressions [See also 11N13]

**11B30** Arithmetic combinatorics; higher degree uniformity

**11B34** Representation functions

**11B37** Recurrences {For applications to special functions, see 33-XX}

**11B39** Fibonacci and Lucas numbers and polynomials and generalizations

**11B50** Sequences (mod $m$)

**11B57** Farey sequences; the sequences $1^k, 2^k, \ldots$

**11B65** Binomial coefficients; factorials; $q$-identities [See also 05A10, 05A30]

**11B68** Bernoulli and Euler numbers and polynomials

**11B73** Bell and Stirling numbers

**11B75** Other combinatorial number theory

**11B83** Special sequences and polynomials

**11B85** Automata sequences

**11B99** None of the above, but in this section

## 11Cxx Polynomials and matrices

**11C08** Polynomials in number theory [See also 13F20]

**11C20** Matrices, determinants in number theory [See also 15B36]

**11C99** None of the above, but in this section

## 11Dxx Diophantine equations [See also 11Gxx, 14Gxx]

**11D04** Linear Diophantine equations

**11D07** The Frobenius problem

**11D09** Quadratic and bilinear Diophantine equations

**11D25** Cubic and quartic Diophantine equations

**11D41** Higher degree equations; Fermat's equation

**11D45** Counting solutions of Diophantine equations

**11D57** Multiplicative and norm form equations

**11D59** Thue-Mahler equations

**11D61** Exponential Diophantine equations

**11D68** Rational numbers as sums of fractions

**11D72** Diophantine equations in many variables [See also 11P55]

**11D75** Diophantine inequalities [See also 11J25]

**11D79** Congruences in many variables

**11D85** Representation problems [See also 11P55]

**11D88** $p$-adic and power series fields

**11D99** None of the above, but in this section

## 11Exx Forms and linear algebraic groups [See also 19Gxx] {For quadratic forms in linear algebra, see 15A63}

**11E04** Quadratic forms over general fields

**11E08** Quadratic forms over local rings and fields

**11E10** Forms over real fields

**11E12** Quadratic forms over global rings and fields

**11E16** General binary quadratic forms

**11E20** General ternary and quaternary quadratic forms; forms of more than two variables

**11E25** Sums of squares and representations by other particular quadratic forms

**11E39** Bilinear and Hermitian forms

**11E41** Class numbers of quadratic and Hermitian forms

**11E45** Analytic theory (Epstein zeta functions; relations with automorphic forms and functions)

**11E57** Classical groups [See also 14Lxx, 20Gxx]

**11E70** $K$-theory of quadratic and Hermitian forms

**11E72** Galois cohomology of linear algebraic groups [See also 20G10]

**11E76** Forms of degree higher than two

**11E81** Algebraic theory of quadratic forms; Witt groups and rings [See also 19G12, 19G24]

**11E88** Quadratic spaces; Clifford algebras [See also 15A63, 15A66]

**11E95** $p$-adic theory

**11E99** None of the above, but in this section

## 11Fxx Discontinuous groups and automorphic forms [See also 11R39, 11S37, 14Gxx, 14Kxx, 22E50, 22E55, 30F35, 32Nxx] {For relations with quadratic forms, see 11E45}

**11F03** Modular and automorphic functions

**11F06** Structure of modular groups and generalizations; arithmetic groups [See also 20H05, 20H10, 22E40]

**11F11** Holomorphic modular forms of integral weight

**11F12** Automorphic forms, one variable

**11F20** Dedekind eta function, Dedekind sums

**11F22** Relationship to Lie algebras and finite simple groups

**11F23** Relations with algebraic geometry and topology

**11F25** Hecke-Petersson operators, differential operators (one variable)

**11F27** Theta series; Weil representation; theta correspondences

**11F30** Fourier coefficients of automorphic forms

**11F32** Modular correspondences, etc.

**11F33** Congruences for modular and $p$-adic modular forms

**11F37** Forms of half-integer weight; nonholomorphic modular forms

**11F41** Automorphic forms on GL(2); Hilbert and Hilbert-Siegel modular groups and their modular and automorphic forms; Hilbert modular surfaces [See also 14G35]

**11F46** Siegel modular groups; Siegel and Hilbert-Siegel modular and automorphic forms

**11F50** Jacobi forms

**11F52** Modular forms associated to Drinfel'd modules

**11F55** Other groups and their modular and automorphic forms (several variables)

**11F60** Hecke-Petersson operators, differential operators (several variables)

**11F66** Langlands $L$-functions; one variable Dirichlet series and functional equations

**11F67** Special values of automorphic $L$-series, periods of automorphic forms, cohomology, modular symbols

**11F68** Dirichlet series in several complex variables associated to automorphic forms; Weyl group multiple Dirichlet series

**11F70** Representation-theoretic methods; automorphic representations over local and global fields

**11F72** Spectral theory; trace formulas (e.g., that of Selberg)

**11F75** Cohomology of arithmetic groups

**11F77** Automorphic forms and their relations with perfectoid spaces [See also 14G45]

**11F80** Galois representations

**11F85** $p$-adic theory, local fields [See also 14G20, 22E50]

**11F99** None of the above, but in this section

## 11Gxx Arithmetic algebraic geometry (Diophantine geometry) [See also 11Dxx, 14Gxx, 14Kxx]

**11G05** Elliptic curves over global fields [See also 14H52]

**11G07** Elliptic curves over local fields [See also 14G20, 14H52]

**11G09** Drinfel'd modules; higher-dimensional motives, etc. [See also 14L05]

**11G10** Abelian varieties of dimension $> 1$ [See also 14Kxx]

**11G15** Complex multiplication and moduli of abelian varieties [See also 14K22]

**11G16** Elliptic and modular units [See also 11R27]

**11G18** Arithmetic aspects of modular and Shimura varieties [See also 14G35]

**11G20** Curves over finite and local fields [See also 14H25]

**11G25** Varieties over finite and local fields [See also 14G15, 14G20]

**11G30** Curves of arbitrary genus or genus $\neq 1$ over global fields [See also 14H25]

**11G32** Arithmetic aspects of dessins d'enfants, Belyĭ theory

**11G35** Varieties over global fields [See also 14G25]

**11G40** $L$-functions of varieties over global fields; Birch-Swinnerton-Dyer conjecture [See also 14G10]

**11G42** Arithmetic mirror symmetry [See also 14J33]

**11G45** Geometric class field theory [See also 11R37, 14C35, 19F05]

**11G50** Heights [See also 14G40, 37P30]

**11G55** Polylogarithms and relations with $K$-theory

**11G99** None of the above, but in this section

## 11Hxx Geometry of numbers {For applications in coding theory, see 94B75}

**11H06** Lattices and convex bodies (number-theoretic aspects) [See also 11P21, 52C05, 52C07]

**11H16** Nonconvex bodies

**11H31** Lattice packing and covering (number-theoretic aspects) [See also 05B40, 52C15, 52C17]

**11H46** Products of linear forms

**11H50** Minima of forms

**11H55** Quadratic forms (reduction theory, extreme forms, etc.)

**11H56** Automorphism groups of lattices

**11H60** Mean value and transfer theorems

**11H71** Relations with coding theory

**11H99** None of the above, but in this section

## 11Jxx Diophantine approximation, transcendental number theory [See also 11K60]

**11J04** Homogeneous approximation to one number

**11J06** Markov and Lagrange spectra and generalizations

**11J13** Simultaneous homogeneous approximation, linear forms

**11J17** Approximation by numbers from a fixed field

**11J20** Inhomogeneous linear forms

**11J25** Diophantine inequalities [See also 11D75]

**11J54** Small fractional parts of polynomials and generalizations

**11J61** Approximation in non-Archimedean valuations

**11J68** Approximation to algebraic numbers

**11J70** Continued fractions and generalizations [See also 11A55, 11K50]

**11J71** Distribution modulo one [See also 11K06]

**11J72** Irrationality; linear independence over a field

**11J81** Transcendence (general theory)

**11J82** Measures of irrationality and of transcendence

**11J83** Metric theory

**11J85** Algebraic independence; Gel'fond's method

**11J86** Linear forms in logarithms; Baker's method

**11J87** Schmidt Subspace Theorem and applications

**11J89** Transcendence theory of elliptic and abelian functions

**11J91** Transcendence theory of other special functions

**11J93** Transcendence theory of Drinfel'd and $t$-modules

**11J95** Results involving abelian varieties

**11J97** Number-theoretic analogues of methods in Nevanlinna theory (work of Vojta et al.)

**11J99** None of the above, but in this section

## 11Kxx Probabilistic theory: distribution modulo 1; metric theory of algorithms

**11K06** General theory of distribution modulo 1 [See also 11J71]

**11K16** Normal numbers, radix expansions, Pisot numbers, Salem numbers, good lattice points, etc. [See also 11A63]

**11K31** Special sequences

**11K36** Well-distributed sequences and other variations

**11K38** Irregularities of distribution, discrepancy [See also 11Nxx]

**11K41** Continuous, $p$-adic and abstract analogues

**11K45** Pseudo-random numbers; Monte Carlo methods [See also 65C05, 65C10]

**11K50** Metric theory of continued fractions [See also 11A55, 11J70]

**11K55** Metric theory of other algorithms and expansions; measure and Hausdorff dimension [See also 11N99, 28Dxx]

**11K60** Diophantine approximation in probabilistic number theory [See also 11Jxx]

**11K65** Arithmetic functions in probabilistic number theory [See also 11Nxx]

**11K70** Harmonic analysis and almost periodicity in probabilistic number theory

**11K99** None of the above, but in this section

## 11Lxx Exponential sums and character sums {For finite fields, see 11Txx}

**11L03** Trigonometric and exponential sums (general theory)

**11L05** Gauss and Kloosterman sums; generalizations

**11L07** Estimates on exponential sums

**11L10** Jacobsthal and Brewer sums; other complete character sums

**11L15** Weyl sums

**11L20** Sums over primes

**11L26** Sums over arbitrary intervals

**11L40** Estimates on character sums

**11L99** None of the above, but in this section

## 11Mxx Zeta and $L$-functions: analytic theory

**11M06** $\zeta(s)$ and $L(s, \chi)$

**11M20** Real zeros of $L(s, \chi)$; results on $L(1, \chi)$

**11M26** Nonreal zeros of $\zeta(s)$ and $L(s, \chi)$; Riemann and other hypotheses

**11M32** Multiple Dirichlet series and zeta functions and multizeta values

**11M35** Hurwitz and Lerch zeta functions

**11M36** Selberg zeta functions and regularized determinants; applications to spectral theory, Dirichlet series, Eisenstein series, etc. (explicit formulas)

**11M38** Zeta and $L$-functions in characteristic $p$

**11M41** Other Dirichlet series and zeta functions {For local and global ground fields, see 11R42, 11R52, 11S40, 11S45; for algebro-geometric methods, see 14G10} [See also 11E45, 11F66, 11F70, 11F72]

**11M45** Tauberian theorems [See also 40E05]

**11M50** Relations with random matrices

**11M55** Relations with noncommutative geometry

**11M99** None of the above, but in this section

## 11Nxx Multiplicative number theory

**11N05** Distribution of primes

**11N13** Primes in congruence classes

**11N25** Distribution of integers with specified multiplicative constraints

**11N30** Turán theory [See also 30Bxx]

**11N32** Primes represented by polynomials; other multiplicative structures of polynomial values

**11N35** Sieves

**11N36** Applications of sieve methods

**11N37** Asymptotic results on arithmetic functions

**11N45** Asymptotic results on counting functions for algebraic and topological structures

**11N56** Rate of growth of arithmetic functions

**11N60** Distribution functions associated with additive and positive multiplicative functions

**11N64** Other results on the distribution of values or the characterization of arithmetic functions

**11N69** Distribution of integers in special residue classes

**11N75** Applications of automorphic functions and forms to multiplicative problems [See also 11Fxx]

**11N80** Generalized primes and integers

**11N99** None of the above, but in this section

## 11Pxx Additive number theory; partitions

**11P05** Waring's problem and variants

**11P21** Lattice points in specified regions

**11P32** Goldbach-type theorems; other additive questions involving primes

**11P55** Applications of the Hardy-Littlewood method [See also 11D85]

**11P70** Inverse problems of additive number theory, including sumsets

**11P81** Elementary theory of partitions [See also 05A17]

**11P82** Analytic theory of partitions

**11P83** Partitions; congruences and congruential restrictions

**11P84** Partition identities; identities of Rogers-Ramanujan type

**11P99** None of the above, but in this section

## 11Rxx Algebraic number theory: global fields {For complex multiplication, see 11G15}

**11R04** Algebraic numbers; rings of algebraic integers

**11R06** PV-numbers and generalizations; other special algebraic numbers; Mahler measure

**11R09** Polynomials (irreducibility, etc.)

**11R11** Quadratic extensions

**11R16** Cubic and quartic extensions

**11R18** Cyclotomic extensions

**11R20** Other abelian and metabelian extensions

**11R21** Other number fields

**11R23** Iwasawa theory

**11R27** Units and factorization

**11R29** Class numbers, class groups, discriminants

**11R32** Galois theory

**11R33** Integral representations related to algebraic numbers; Galois module structure of rings of integers [See also 20C10]

**11R34** Galois cohomology [See also 12Gxx, 19F05]

**11R37** Class field theory

**11R39** Langlands-Weil conjectures, nonabelian class field theory [See also 11Fxx, 22E55]

**11R42** Zeta functions and $L$-functions of number fields [See also 11M41, 19F27]

**11R44** Distribution of prime ideals [See also 11N05]

**11R45** Density theorems

**11R47** Other analytic theory [See also 11Nxx]

**11R52** Quaternion and other division algebras: arithmetic, zeta functions

**11R54** Other algebras and orders, and their zeta and $L$-functions [See also 11S45, 16Hxx]

**11R56** Adèle rings and groups

**11R58** Arithmetic theory of algebraic function fields [See also 14Gxx, 14H05]

**11R59** Zeta functions and $L$-functions of function fields

**11R60** Cyclotomic function fields (class groups, Bernoulli objects, etc.)

**11R65** Class groups and Picard groups of orders

**11R70** $K$-theory of global fields [See also 19Fxx]

**11R80** Totally real fields [See also 12J15]

**11R99** None of the above, but in this section

## 11Sxx Algebraic number theory: local fields

**11S05** Polynomials

**11S15** Ramification and extension theory

**11S20** Galois theory

**11S23** Integral representations

**11S25** Galois cohomology [See also 12Gxx, 16H05]

**11S31** Class field theory; $p$-adic formal groups [See also 14L05]

**11S37** Langlands-Weil conjectures, nonabelian class field theory [See also 11Fxx, 22E50]

**11S40** Zeta functions and $L$-functions [See also 11M41, 19F27]

**11S45** Algebras and orders, and their zeta functions [See also 11R52, 11R54, 16Hxx, 16Kxx]

**11S70** $K$-theory of local fields [See also 19Fxx]

**11S80** Other analytic theory (analogues of beta and gamma functions, $p$-adic integration, etc.)

**11S82** Non-Archimedean dynamical systems [See mainly 37Pxx]

**11S85** Other nonanalytic theory

**11S90** Prehomogeneous vector spaces

**11S99** None of the above, but in this section

## 11Txx Finite fields and commutative rings (number-theoretic aspects)

**11T06** Polynomials over finite fields

**11T22** Cyclotomy

**11T23** Exponential sums

**11T24** Other character sums and Gauss sums

**11T30** Structure theory for finite fields and commutative rings (number-theoretic aspects)

**11T55** Arithmetic theory of polynomial rings over finite fields

**11T60** Finite upper half-planes

**11T71** Algebraic coding theory; cryptography (number-theoretic aspects)

**11T99** None of the above, but in this section

## 11Uxx Connections of number theory and logic

**11U05** Decidability (number-theoretic aspects) [See also 03B25]

**11U07** Ultraproducts (number-theoretic aspects) [See also 03C20]

**11U09** Model theory (number-theoretic aspects) [See also 03Cxx]

**11U10** Nonstandard arithmetic (number-theoretic aspects) [See also 03H15]

**11U99** None of the above, but in this section

**11Yxx Computational number theory {For software etc., see 11-04} [See also 68W30]**

**11Y05** Factorization

**11Y11** Primality

**11Y16** Number-theoretic algorithms; complexity [See also 68Q25]

**11Y35** Analytic computations

**11Y40** Algebraic number theory computations

**11Y50** Computer solution of Diophantine equations

**11Y55** Calculation of integer sequences

**11Y60** Evaluation of number-theoretic constants

**11Y65** Continued fraction calculations (number-theoretic aspects)

**11Y70** Values of arithmetic functions; tables

**11Y99** None of the above, but in this section

**11Zxx Miscellaneous applications of number theory**

**11Z05** Miscellaneous applications of number theory

**11Z99** None of the above, but in this section

# 12-XX Field theory and polynomials

**12-00** General reference works (handbooks, dictionaries, bibliographies, etc.) pertaining to field theory

**12-01** Introductory exposition (textbooks, tutorial papers, etc.) pertaining to field theory

**12-02** Research exposition (monographs, survey articles) pertaining to field theory

**12-03** History of field theory [Consider also classification numbers from Section 01]

**12-04** Software, source code, etc. for problems pertaining to field theory

**12-06** Proceedings, conferences, collections, etc. pertaining to field theory

**12-08** Computational methods for problems pertaining to field theory [See also 68W30]

**12-11** Research data for problems pertaining to field theory

**12Dxx Real and complex fields**

**12D05** Polynomials in real and complex fields: factorization

**12D10** Polynomials in real and complex fields: location of zeros (algebraic theorems) {For the analytic theory, see 26C10, 30C15}

**12D15** Fields related with sums of squares (formally real fields, Pythagorean fields, etc.) [See also 11Exx]

**12D99** None of the above, but in this section

## 12Exx General field theory

**12E05** Polynomials in general fields (irreducibility, etc.)

**12E10** Special polynomials in general fields

**12E12** Equations in general fields

**12E15** Skew fields, division rings [See also 11R52, 16Kxx]

**12E20** Finite fields (field-theoretic aspects)

**12E25** Hilbertian fields; Hilbert's irreducibility theorem

**12E30** Field arithmetic

**12E99** None of the above, but in this section

## 12Fxx Field extensions

**12F05** Algebraic field extensions

**12F10** Separable extensions, Galois theory

**12F12** Inverse Galois theory

**12F15** Inseparable field extensions

**12F20** Transcendental field extensions

**12F99** None of the above, but in this section

## 12Gxx Homological methods (field theory)

**12G05** Galois cohomology [See also 14F22, 16H05, 16K50]

**12G10** Cohomological dimension of fields

**12G99** None of the above, but in this section

## 12Hxx Differential and difference algebra

**12H05** Differential algebra [See also 13Nxx]

**12H10** Difference algebra [See also 39Axx]

**12H20** Abstract differential equations [See also 34Mxx]

**12H25** $p$-adic differential equations [See also 11S80, 14G20]

**12H99** None of the above, but in this section

## 12Jxx Topological fields

**12J05** Normed fields

**12J10** Valued fields

**12J12** Formally *p*-adic fields

**12J15** Ordered fields

**12J17** Topological semifields

**12J20** General valuation theory for fields [See also 13A18]

**12J25** Non-Archimedean valued fields [See also 30G06, 46S10]

**12J27** Krasner-Tate algebras [See mainly 32P05; see also 46S10, 47S10]

**12J99** None of the above, but in this section

## 12Kxx Generalizations of fields

**12K05** Near-fields [See also 16Y30]

**12K10** Semifields [See also 16Y60]

**12K99** None of the above, but in this section

## 12Lxx Connections between field theory and logic

**12L05** Decidability and field theory [See also 03B25]

**12L10** Ultraproducts and field theory [See also 03C20]

**12L12** Model theory of fields [See also 03C60]

**12L15** Nonstandard arithmetic and field theory [See also 03H15]

**12L99** None of the above, but in this section

# 13-XX Commutative algebra

**13-00** General reference works (handbooks, dictionaries, bibliographies, etc.) pertaining to commutative algebra

**13-01** Introductory exposition (textbooks, tutorial papers, etc.) pertaining to commutative algebra

**13-02** Research exposition (monographs, survey articles) pertaining to commutative algebra

**13-03** History of commutative algebra [Consider also classification numbers from Section 01]

**13-04** Software, source code, etc. for problems pertaining to commutative algebra

**13-06** Proceedings, conferences, collections, etc. pertaining to commutative algebra

**13-11** Research data for problems pertaining to commutative algebra

## 13Axx General commutative ring theory

**13A02** Graded rings [See also 16W50]

**13A05** Divisibility and factorizations in commutative rings [See also 13F15]

**13A15** Ideals and multiplicative ideal theory in commutative rings

**13A18** Valuations and their generalizations for commutative rings [See also 12J20]

**13A30** Associated graded rings of ideals (Rees ring, form ring), analytic spread and related topics

**13A35** Characteristic $p$ methods (Frobenius endomorphism) and reduction to characteristic $p$; tight closure [See also 13B22]

**13A50** Actions of groups on commutative rings; invariant theory [See also 14L24]

**13A70** General commutative ring theory and combinatorics (zero-divisor graphs, annihilating-ideal graphs, etc.) [See also 05C25, 05E40]

**13A99** None of the above, but in this section

## 13Bxx Commutative ring extensions and related topics

**13B02** Extension theory of commutative rings

**13B05** Galois theory and commutative ring extensions

**13B10** Morphisms of commutative rings

**13B21** Integral dependence in commutative rings; going up, going down

**13B22** Integral closure of commutative rings and ideals [See also 13A35]; integrally closed rings, related rings (Japanese, etc.)

**13B25** Polynomials over commutative rings [See also 11C08, 11T06, 13F20, 13M10]

**13B30** Rings of fractions and localization for commutative rings [See also 16S85]

**13B35** Completion of commutative rings [See also 13J10]

**13B40** Étale and flat extensions; Henselization; Artin approximation [See also 13J15, 14B12, 14B25]

**13B99** None of the above, but in this section

## 13Cxx Theory of modules and ideals in commutative rings

**13C05** Structure, classification theorems for modules and ideals in commutative rings

**13C10** Projective and free modules and ideals in commutative rings [See also 19A13]

**13C11** Injective and flat modules and ideals in commutative rings

**13C12** Torsion modules and ideals in commutative rings

**13C13** Other special types of modules and ideals in commutative rings

**13C14** Cohen-Macaulay modules [See also 13H10]

**13C15** Dimension theory, depth, related commutative rings (catenary, etc.)

**13C20** Class groups [See also 11R29]

**13C40** Linkage, complete intersections and determinantal ideals [See also 14M06, 14M10, 14M12]

**13C60** Module categories and commutative rings

**13C70** Theory of modules and ideals in commutative rings described by combinatorial properties [See also 05C25, 05E40]

**13C99** None of the above, but in this section

## 13Dxx Homological methods in commutative ring theory {For noncommutative rings, see 16Exx; for general categories, see 18Gxx}

**13D02** Syzygies, resolutions, complexes and commutative rings

**13D03** (Co)homology of commutative rings and algebras (e.g., Hochschild, André-Quillen, cyclic, dihedral, etc.)

**13D05** Homological dimension and commutative rings

**13D07** Homological functors on modules of commutative rings (Tor, Ext, etc.)

**13D09** Derived categories and commutative rings

**13D10** Deformations and infinitesimal methods in commutative ring theory [See also 14B10, 14B12, 14D15, 32Gxx]

**13D15** Grothendieck groups, $K$-theory and commutative rings [See also 14C35, 18F30, 19Axx, 19D50]

**13D22** Homological conjectures (intersection theorems) in commutative ring theory

**13D30** Torsion theory for commutative rings [See also 13C12, 18E40]

**13D40** Hilbert-Samuel and Hilbert-Kunz functions; Poincaré series

**13D45** Local cohomology and commutative rings [See also 14B15]

**13D99** None of the above, but in this section

## 13Exx Chain conditions, finiteness conditions in commutative ring theory

**13E05** Commutative Noetherian rings and modules

**13E10** Commutative Artinian rings and modules, finite-dimensional algebras

**13E15** Commutative rings and modules of finite generation or presentation; number of generators

**13E99** None of the above, but in this section

## 13Fxx Arithmetic rings and other special commutative rings

**13F05** Dedekind, Prüfer, Krull and Mori rings and their generalizations

**13F07** Euclidean rings and generalizations

**13F10** Principal ideal rings

**13F15** Commutative rings defined by factorization properties (e.g., atomic, factorial, half-factorial) [See also 13A05, 14M05]

**13F20** Polynomial rings and ideals; rings of integer-valued polynomials [See also 11C08, 13B25]

**13F25** Formal power series rings [See also 13J05]

**13F30** Valuation rings [See also 13A18]

**13F35** Witt vectors and related rings

**13F40** Excellent rings

**13F45** Seminormal rings

**13F50** Rings with straightening laws, Hodge algebras

**13F55** Commutative rings defined by monomial ideals; Stanley-Reisner face rings; simplicial complexes [See also 55U10]

**13F60** Cluster algebras

**13F65** Commutative rings defined by binomial ideals, toric rings, etc. [See also 14M25]

**13F70** Other commutative rings defined by combinatorial properties

**13F99** None of the above, but in this section

## 13Gxx Integral domains

**13G05** Integral domains

**13G99** None of the above, but in this section

## 13Hxx Local rings and semilocal rings

**13H05** Regular local rings

**13H10** Special types (Cohen-Macaulay, Gorenstein, Buchsbaum, etc.) [See also 14M05]

**13H15** Multiplicity theory and related topics [See also 14C17]

**13H99** None of the above, but in this section

## 13Jxx Topological rings and modules [See also 16W60, 16W80]

**13J05** Power series rings [See also 13F25]

**13J07** Analytical algebras and rings [See also 32B05]

**13J10** Complete rings, completion [See also 13B35]

**13J15** Henselian rings [See also 13B40]

**13J20** Global topological rings

**13J25** Ordered rings [See also 06F25]

**13J30** Real algebra [See also 12D15, 14Pxx]

**13J99** None of the above, but in this section

## 13Lxx Applications of logic to commutative algebra [See also 03Cxx, 03Hxx]

**13L05** Applications of logic to commutative algebra [See also 03Cxx, 03Hxx]

**13L99** None of the above, but in this section

## 13Mxx Finite commutative rings {For number-theoretic aspects, see 11Txx}

**13M05** Structure of finite commutative rings

**13M10** Polynomials and finite commutative rings

**13M99** None of the above, but in this section

## 13Nxx Differential algebra [See also 12H05, 14F10]

**13N05** Modules of differentials

**13N10** Commutative rings of differential operators and their modules [See also 16S32, 32C38]

**13N15** Derivations and commutative rings

**13N99** None of the above, but in this section

## 13Pxx Computational aspects and applications of commutative rings [See also 14Qxx, 68W30] {For software etc., see 13-04}

**13P05** Polynomials, factorization in commutative rings [See also 12-08]

**13P10** Gröbner bases; other bases for ideals and modules (e.g., Janet and border bases)

**13P15** Solving polynomial systems; resultants

**13P20** Computational homological algebra [See also 13Dxx]

**13P25** Applications of commutative algebra (e.g., to statistics, control theory, optimization, etc.)

**13P99** None of the above, but in this section

# 14-XX Algebraic geometry

**14-00** General reference works (handbooks, dictionaries, bibliographies, etc.) pertaining to algebraic geometry

**14-01** Introductory exposition (textbooks, tutorial papers, etc.) pertaining to algebraic geometry

**14-02** Research exposition (monographs, survey articles) pertaining to algebraic geometry

**14-03** History of algebraic geometry [Consider also classification numbers from Section 01]

**14-04** Software, source code, etc. for problems pertaining to algebraic geometry

**14-06** Proceedings, conferences, collections, etc. pertaining to algebraic geometry

**14-11** Research data for problems pertaining to algebraic geometry

## 14Axx Foundations of algebraic geometry

**14A05** Relevant commutative algebra [See also 13-XX]

**14A10** Varieties and morphisms

**14A15** Schemes and morphisms

**14A20** Generalizations (algebraic spaces, stacks)

**14A21** Logarithmic algebraic geometry, log schemes

**14A22** Noncommutative algebraic geometry [See also 16S38]

**14A23** Geometry over the field with one element

**14A25** Elementary questions in algebraic geometry

**14A30** Fundamental constructions in algebraic geometry involving higher and derived categories (homotopical algebraic geometry, derived algebraic geometry, etc.) {For categorical aspects, see 18Fxx, 18Gxx}

**14A99** None of the above, but in this section

## 14Bxx Local theory in algebraic geometry

**14B05** Singularities in algebraic geometry [See also 14E15, 14H20, 14J17, 32Sxx, 58Kxx]

**14B07** Deformations of singularities [See also 14D15, 32S30]

**14B10** Infinitesimal methods in algebraic geometry [See also 13D10]

**14B12** Local deformation theory, Artin approximation, etc. [See also 13B40, 13D10]

**14B15** Local cohomology and algebraic geometry [See also 13D45, 32C36]

**14B20** Formal neighborhoods in algebraic geometry

**14B25** Local structure of morphisms in algebraic geometry: étale, flat, etc. [See also 13B40]

**14B99** None of the above, but in this section

## 14Cxx Cycles and subschemes

**14C05** Parametrization (Chow and Hilbert schemes)

**14C15** (Equivariant) Chow groups and rings; motives

**14C17** Intersection theory, characteristic classes, intersection multiplicities in algebraic geometry [See also 13H15]

**14C20** Divisors, linear systems, invertible sheaves

**14C21** Pencils, nets, webs in algebraic geometry [See also 53A60]

**14C22** Picard groups

**14C25** Algebraic cycles

**14C30** Transcendental methods, Hodge theory (algebro-geometric aspects) [See also 14D07, 32G20, 32J25, 32S35, 58A14], Hodge conjecture

**14C34** Torelli problem [See also 32G20]

**14C35** Applications of methods of algebraic $K$-theory in algebraic geometry [See also 19Exx]

**14C40** Riemann-Roch theorems [See also 19E20, 19L10]

**14C99** None of the above, but in this section

## 14Dxx Families, fibrations in algebraic geometry

**14D05** Structure of families (Picard-Lefschetz, monodromy, etc.)

**14D06** Fibrations, degenerations in algebraic geometry

**14D07** Variation of Hodge structures (algebro-geometric aspects) [See also 32G20]

**14D10** Arithmetic ground fields (finite, local, global) and families or fibrations

**14D15** Formal methods and deformations in algebraic geometry [See also 13D10, 14B07, 32Gxx]

**14D20** Algebraic moduli problems, moduli of vector bundles {For analytic moduli problems, see 32G13}

**14D21** Applications of vector bundles and moduli spaces in mathematical physics (twistor theory, instantons, quantum field theory) [See also 32L25, 81Txx]

**14D22** Fine and coarse moduli spaces

**14D23** Stacks and moduli problems

**14D24** Geometric Langlands program (algebro-geometric aspects) [See also 22E57]

**14D99** None of the above, but in this section

## 14Exx Birational geometry

**14E05** Rational and birational maps

**14E07** Birational automorphisms, Cremona group and generalizations

**14E08** Rationality questions in algebraic geometry [See also 14M20]

**14E15** Global theory and resolution of singularities (algebro-geometric aspects) [See also 14B05, 32S20, 32S45]

**14E16** McKay correspondence

**14E18** Arcs and motivic integration

**14E20** Coverings in algebraic geometry [See also 14H30]

**14E22** Ramification problems in algebraic geometry [See also 11S15]

**14E25** Embeddings in algebraic geometry

**14E30** Minimal model program (Mori theory, extremal rays)

**14E99** None of the above, but in this section

## 14Fxx (Co)homology theory in algebraic geometry [See also 13Dxx]

**14F06** Sheaves in algebraic geometry [See also 14F08, 14H60, 14J60, 18F20, 32L10, 46M20]

**14F08** Derived categories of sheaves, dg categories, and related constructions in algebraic geometry [See also 14A30, 14F06, 18Gxx]

**14F10** Differentials and other special sheaves; D-modules; Bernstein-Sato ideals and polynomials [See also 13Nxx, 32C38]

**14F17** Vanishing theorems in algebraic geometry [See also 32L20]

**14F18** Multiplier ideals

**14F20** Étale and other Grothendieck topologies and (co)homologies

**14F22** Brauer groups of schemes [See also 12G05, 16K50]

**14F25** Classical real and complex (co)homology in algebraic geometry

**14F30** $p$-adic cohomology, crystalline cohomology

**14F35** Homotopy theory and fundamental groups in algebraic geometry [See also 14H30]

**14F40** de Rham cohomology and algebraic geometry [See also 14C30, 32C35, 32L10]

**14F42** Motivic cohomology; motivic homotopy theory [See also 19E15]

**14F43** Other algebro-geometric (co)homologies (e.g., intersection, equivariant, Lawson, Deligne (co)homologies)

**14F45** Topological properties in algebraic geometry

**14F99** None of the above, but in this section

## 14Gxx Arithmetic problems in algebraic geometry; Diophantine geometry [See also 11Dxx, 11Gxx]

**14G05** Rational points

**14G10** Zeta functions and related questions in algebraic geometry (e.g., Birch-Swinnerton-Dyer conjecture) [See also 11G40]

**14G12** Hasse principle, weak and strong approximation, Brauer-Manin obstruction [See also 14F22]

**14G15** Finite ground fields in algebraic geometry

**14G17** Positive characteristic ground fields in algebraic geometry

**14G20** Local ground fields in algebraic geometry

**14G22** Rigid analytic geometry

**14G25** Global ground fields in algebraic geometry

**14G27** Other nonalgebraically closed ground fields in algebraic geometry

**14G32** Universal profinite groups (relationship to moduli spaces, projective and moduli towers, Galois theory)

**14G35** Modular and Shimura varieties [See also 11F41, 11F46, 11G18]

**14G40** Arithmetic varieties and schemes; Arakelov theory; heights [See also 11G50, 37P30]

**14G45** Perfectoid spaces and mixed characteristic

**14G50** Applications to coding theory and cryptography of arithmetic geometry [See also 94A60, 94B27, 94B40]

**14G99** None of the above, but in this section

## 14Hxx Curves in algebraic geometry

**14H05** Algebraic functions and function fields in algebraic geometry [See also 11R58]

**14H10** Families, moduli of curves (algebraic)

**14H15** Families, moduli of curves (analytic) [See also 30F10, 32G15]

**14H20** Singularities of curves, local rings [See also 13Hxx, 14B05]

**14H25** Arithmetic ground fields for curves [See also 11Dxx, 11G05, 14Gxx]

**14H30** Coverings of curves, fundamental group [See also 14E20, 14F35]

**14H37** Automorphisms of curves

**14H40** Jacobians, Prym varieties [See also 32G20]

**14H42** Theta functions and curves; Schottky problem [See also 14K25, 32G20]

**14H45** Special algebraic curves and curves of low genus

**14H50** Plane and space curves

**14H51** Special divisors on curves (gonality, Brill-Noether theory)

**14H52** Elliptic curves [See also 11G05, 11G07, 14Kxx]

**14H55** Riemann surfaces; Weierstrass points; gap sequences [See also 30Fxx]

**14H57** Dessins d'enfants theory {For arithmetic aspects, see 11G32}

**14H60** Vector bundles on curves and their moduli [See also 14D20, 14F06, 14J60]

**14H70** Relationships between algebraic curves and integrable systems

**14H81** Relationships between algebraic curves and physics

**14H99** None of the above, but in this section

## 14Jxx Surfaces and higher-dimensional varieties {For analytic theory, see 32Jxx}

**14J10** Families, moduli, classification: algebraic theory

**14J15** Moduli, classification: analytic theory; relations with modular forms [See also 32G13]

**14J17** Singularities of surfaces or higher-dimensional varieties [See also 14B05, 14E15, 32S05, 32S25]

**14J20** Arithmetic ground fields for surfaces or higher-dimensional varieties [See also 11Dxx, 11G25, 11G35, 14Gxx]

**14J25** Special surfaces {For Hilbert modular surfaces, see 14G35}

**14J26** Rational and ruled surfaces

**14J27** Elliptic surfaces, elliptic or Calabi-Yau fibrations

**14J28** $K3$ surfaces and Enriques surfaces

**14J29** Surfaces of general type

**14J30** 3-folds

**14J32** Calabi-Yau manifolds (algebro-geometric aspects) [See also 32Q25]

**14J33** Mirror symmetry (algebro-geometric aspects) [See also 11G42, 53D37]

**14J35** 4-folds

**14J40** $n$-folds $(n > 4)$

**14J42** Holomorphic symplectic varieties, hyper-Kähler varieties

**14J45** Fano varieties

**14J50** Automorphisms of surfaces and higher-dimensional varieties

**14J60** Vector bundles on surfaces and higher-dimensional varieties, and their moduli [See also 14D20, 14F06, 14H60, 32Lxx]

**14J70** Hypersurfaces and algebraic geometry

**14J80** Topology of surfaces (Donaldson polynomials, Seiberg-Witten invariants)

**14J81** Relationships between surfaces, higher-dimensional varieties, and physics

**14J99** None of the above, but in this section

## 14Kxx Abelian varieties and schemes

**14K02** Isogeny

**14K05** Algebraic theory of abelian varieties

**14K10** Algebraic moduli of abelian varieties, classification [See also 11G15]

**14K12** Subvarieties of abelian varieties

**14K15** Arithmetic ground fields for abelian varieties [See also 11Dxx, 11Fxx, 11G10, 14Gxx]

**14K20** Analytic theory of abelian varieties; abelian integrals and differentials

**14K22** Complex multiplication and abelian varieties [See also 11G15]

**14K25** Theta functions and abelian varieties [See also 14H42]

**14K30** Picard schemes, higher Jacobians [See also 14H40, 32G20]

**14K99** None of the above, but in this section

## 14Lxx Algebraic groups [See also 11E57] {For Lie algebras, see 17B45; for linear algebraic groups, see 20Gxx}

**14L05** Formal groups, $p$-divisible groups [See also 55N22]

**14L10** Group varieties

**14L15** Group schemes

**14L17** Affine algebraic groups, hyperalgebra constructions [See also 17B45, 18C40]

**14L24** Geometric invariant theory [See also 13A50]

**14L30** Group actions on varieties or schemes (quotients) [See also 13A50, 14L24, 14M17]

**14L35** Classical groups (algebro-geometric aspects) [See also 20Gxx, 51N30]

**14L40** Other algebraic groups (geometric aspects)

**14L99** None of the above, but in this section

## 14Mxx Special varieties

**14M05** Varieties defined by ring conditions (factorial, Cohen-Macaulay, seminormal) [See also 13F15, 13F45, 13H10]

**14M06** Linkage [See also 13C40]

**14M07** Low codimension problems in algebraic geometry

**14M10** Complete intersections [See also 13C40]

**14M12** Determinantal varieties [See also 13C40]

**14M15** Grassmannians, Schubert varieties, flag manifolds [See also 32M10, 51M35]

**14M17** Homogeneous spaces and generalizations [See also 32M10, 53C30, 57T15]

**14M20** Rational and unirational varieties [See also 14E08]

**14M22** Rationally connected varieties

**14M25** Toric varieties, Newton polyhedra, Okounkov bodies [See also 52B20]

**14M27** Compactifications; symmetric and spherical varieties

**14M30** Supervarieties [See also 32C11, 58A50]

**14M35** Character varieties

**14M99** None of the above, but in this section

## 14Nxx Projective and enumerative algebraic geometry [See also 51-XX]

**14N05** Projective techniques in algebraic geometry [See also 51N35]

**14N07** Secant varieties, tensor rank, varieties of sums of powers

**14N10** Enumerative problems (combinatorial problems) in algebraic geometry

**14N15** Classical problems, Schubert calculus

**14N20** Configurations and arrangements of linear subspaces

**14N25** Varieties of low degree

**14N30** Adjunction problems

**14N35** Gromov-Witten invariants, quantum cohomology, Gopakumar-Vafa invariants, Donaldson-Thomas invariants (algebro-geometric aspects) [See also 53D45]

**14N99** None of the above, but in this section

## 14Pxx Real algebraic and real-analytic geometry

**14P05** Real algebraic sets [See also 12D15, 13J30]

**14P10** Semialgebraic sets and related spaces

**14P15** Real-analytic and semi-analytic sets [See also 32B20, 32C05]

**14P20** Nash functions and manifolds [See also 32C07, 58A07]

**14P25** Topology of real algebraic varieties

**14P99** None of the above, but in this section

## 14Qxx Computational aspects in algebraic geometry {For software etc., see 14-04} [See also 12-08, 13Pxx, 68W30]

**14Q05** Computational aspects of algebraic curves [See also 14Hxx]

**14Q10** Computational aspects of algebraic surfaces [See also 14Jxx]

**14Q15** Computational aspects of higher-dimensional varieties [See also 14Jxx, 14Mxx]

**14Q20** Effectivity, complexity and computational aspects of algebraic geometry

**14Q25** Computational algebraic geometry over arithmetic ground fields [See also 14Gxx, 14H25, 14Kxx]

**14Q30** Computational real algebraic geometry [See also 14Pxx]

**14Q65** Geometric aspects of numerical algebraic geometry [See also 65H14]

**14Q99** None of the above, but in this section

### 14Rxx Affine geometry

**14R05** Classification of affine varieties

**14R10** Affine spaces (automorphisms, embeddings, exotic structures, cancellation problem)

**14R15** Jacobian problem [See also 13F20]

**14R20** Group actions on affine varieties [See also 13A50, 14L30]

**14R25** Affine fibrations [See also 14D06]

**14R99** None of the above, but in this section

### 14Txx Tropical geometry [See also 12K10, 14M25, 14N10, 52B20]

**14T10** Foundations of tropical geometry and relations with algebra {For algebraic aspects, see 15A80}

**14T15** Combinatorial aspects of tropical varieties

**14T20** Geometric aspects of tropical varieties

**14T25** Arithmetic aspects of tropical varieties

**14T90** Applications of tropical geometry

**14T99** None of the above, but in this section

# 15-XX Linear and multilinear algebra; matrix theory

**15-00** General reference works (handbooks, dictionaries, bibliographies, etc.) pertaining to linear algebra

**15-01** Introductory exposition (textbooks, tutorial papers, etc.) pertaining to linear algebra

**15-02** Research exposition (monographs, survey articles) pertaining to linear algebra

**15-03** History of linear algebra [Consider also classification numbers from Section 01]

**15-04** Software, source code, etc. for problems pertaining to linear algebra

**15-06** Proceedings, conferences, collections, etc. pertaining to linear algebra

**15-11** Research data for problems pertaining to linear algebra

### 15Axx Basic linear algebra

**15A03** Vector spaces, linear dependence, rank, lineability

**15A04** Linear transformations, semilinear transformations

**15A06** Linear equations (linear algebraic aspects)

**15A09** Theory of matrix inversion and generalized inverses

**15A10** Applications of generalized inverses

**15A12** Conditioning of matrices [See also 65F35]

**15A15** Determinants, permanents, traces, other special matrix functions [See also 19B10, 19B14]

**15A16** Matrix exponential and similar functions of matrices

**15A18** Eigenvalues, singular values, and eigenvectors

**15A20** Diagonalization, Jordan forms

**15A21** Canonical forms, reductions, classification

**15A22** Matrix pencils [See also 47A56]

**15A23** Factorization of matrices

**15A24** Matrix equations and identities

**15A27** Commutativity of matrices

**15A29** Inverse problems in linear algebra

**15A30** Algebraic systems of matrices [See also 16S50, 20Gxx, 20Hxx]

**15A39** Linear inequalities of matrices

**15A42** Inequalities involving eigenvalues and eigenvectors

**15A45** Miscellaneous inequalities involving matrices

**15A54** Matrices over function rings in one or more variables

**15A60** Norms of matrices, numerical range, applications of functional analysis to matrix theory [See also 65F35, 65J05]

**15A63** Quadratic and bilinear forms, inner products [See mainly 11Exx]

**15A66** Clifford algebras, spinors

**15A67** Applications of Clifford algebras to physics, etc.

**15A69** Multilinear algebra, tensor calculus

**15A72** Vector and tensor algebra, theory of invariants [See also 13A50, 14L24]

**15A75** Exterior algebra, Grassmann algebras

**15A78** Other algebras built from modules

**15A80** Max-plus and related algebras

**15A83** Matrix completion problems

**15A86** Linear preserver problems

**15A99** None of the above, but in this section

## 15Bxx Special matrices

**15B05** Toeplitz, Cauchy, and related matrices

**15B10** Orthogonal matrices

**15B15** Fuzzy matrices

**15B30** Matrix Lie algebras

**15B33** Matrices over special rings (quaternions, finite fields, etc.)

**15B34** Boolean and Hadamard matrices

**15B35** Sign pattern matrices

**15B36** Matrices of integers [See also 11C20]

**15B48** Positive matrices and their generalizations; cones of matrices

**15B51** Stochastic matrices

**15B52** Random matrices (algebraic aspects) {For probabilistic aspects, see 60B20}

**15B57** Hermitian, skew-Hermitian, and related matrices

**15B99** None of the above, but in this section

# 16-XX Associative rings and algebras {For the commutative case, see 13-XX}

**16-00** General reference works (handbooks, dictionaries, bibliographies, etc.) pertaining to associative rings and algebras

**16-01** Introductory exposition (textbooks, tutorial papers, etc.) pertaining to associative rings and algebras

**16-02** Research exposition (monographs, survey articles) pertaining to associative rings and algebras

**16-03** History of associative rings and algebras [Consider also classification numbers from Section 01]

**16-04** Software, source code, etc. for problems pertaining to associative rings and algebras

**16-06** Proceedings, conferences, collections, etc. pertaining to associative rings and algebras

**16-11** Research data for problems pertaining to associative rings and algebras

## 16Bxx General and miscellaneous

**16B50** Category-theoretic methods and results in associative algebras (except as in 16D90) [See also 18-XX]

**16B70** Applications of logic in associative algebras [See also 03Cxx]

**16B99** None of the above, but in this section

## 16Dxx Modules, bimodules and ideals in associative algebras

**16D10** General module theory in associative algebras

**16D20** Bimodules in associative algebras

**16D25** Ideals in associative algebras

**16D30** Infinite-dimensional simple rings (except as in 16Kxx)

**16D40** Free, projective, and flat modules and ideals in associative algebras [See also 19A13]

**16D50** Injective modules, self-injective associative rings [See also 16L60]

**16D60** Simple and semisimple modules, primitive rings and ideals in associative algebras

**16D70** Structure and classification for modules, bimodules and ideals (except as in 16Gxx), direct sum decomposition and cancellation in associative algebras)

**16D80** Other classes of modules and ideals in associative algebras [See also 16G50]

**16D90** Module categories in associative algebras [See also 16Gxx, 16S90]; module theory in a category-theoretic context; Morita equivalence and duality

**16D99** None of the above, but in this section

## 16Exx Homological methods in associative algebras {For commutative rings, see 13Dxx; for general categories, see 18Gxx}

**16E05** Syzygies, resolutions, complexes in associative algebras

**16E10** Homological dimension in associative algebras

**16E20** Grothendieck groups, $K$-theory, etc. [See also 18F30, 19Axx, 19D50]

**16E30** Homological functors on modules (Tor, Ext, etc.) in associative algebras

**16E35** Derived categories and associative algebras

**16E40** (Co)homology of rings and associative algebras (e.g., Hochschild, cyclic, dihedral, etc.)

**16E45** Differential graded algebras and applications (associative algebraic aspects)

**16E50** von Neumann regular rings and generalizations (associative algebraic aspects)

**16E60** Semihereditary and hereditary rings, free ideal rings, Sylvester rings, etc.

**16E65** Homological conditions on associative rings (generalizations of regular, Gorenstein, Cohen-Macaulay rings, etc.)

**16E99** None of the above, but in this section

## 16Gxx Representation theory of associative rings and algebras

**16G10** Representations of associative Artinian rings

**16G20** Representations of quivers and partially ordered sets

**16G30** Representations of orders, lattices, algebras over commutative rings [See also 16Hxx]

**16G50** Cohen-Macaulay modules in associative algebras

**16G60** Representation type (finite, tame, wild, etc.) of associative algebras

**16G70** Auslander-Reiten sequences (almost split sequences) and Auslander-Reiten quivers

**16G99** None of the above, but in this section

## 16Hxx Associative algebras and orders {For arithmetic aspects, see 11R52, 11R54, 11S45; for representation theory, see 16G30}

**16H05** Separable algebras (e.g., quaternion algebras, Azumaya algebras, etc.)

**16H10** Orders in separable algebras

**16H15** Commutative orders

**16H20** Lattices over orders

**16H99** None of the above, but in this section

## 16Kxx Division rings and semisimple Artin rings [See also 12E15, 15A30]

**16K20** Finite-dimensional division rings {For crossed products, see 16S35}

**16K40** Infinite-dimensional and general division rings

**16K50** Brauer groups (algebraic aspects) [See also 12G05, 14F22]

**16K99** None of the above, but in this section

## 16Lxx Local rings and generalizations

**16L30** Noncommutative local and semilocal rings, perfect rings

**16L60** Quasi-Frobenius rings [See also 16D50]

**16L99** None of the above, but in this section

## 16Nxx Radicals and radical properties of associative rings

**16N20** Jacobson radical, quasimultiplication

**16N40** Nil and nilpotent radicals, sets, ideals, associative rings

**16N60** Prime and semiprime associative rings [See also 16D60, 16U10]

**16N80** General radicals and associative rings {For radicals in module categories, see 16S90}

**16N99** None of the above, but in this section

## 16Pxx Chain conditions, growth conditions, and other forms of finiteness for associative rings and algebras

**16P10** Finite rings and finite-dimensional associative algebras {For semisimple, see 16K20; for commutative, see 11Txx, 13Mxx}

**16P20** Artinian rings and modules (associative rings and algebras)

**16P40** Noetherian rings and modules (associative rings and algebras)

**16P50** Localization and associative Noetherian rings [See also 16U20]

**16P60** Chain conditions on annihilators and summands: Goldie-type conditions [See also 16U20], Krull dimension (associative rings and algebras)

**16P70** Chain conditions on other classes of submodules, ideals, subrings, etc.; coherence (associative rings and algebras)

**16P90** Growth rate, Gelfand-Kirillov dimension

**16P99** None of the above, but in this section

## 16Rxx Rings with polynomial identity

**16R10** $T$-ideals, identities, varieties of associative rings and algebras

**16R20** Semiprime p.i. rings, rings embeddable in matrices over commutative rings

**16R30** Trace rings and invariant theory (associative rings and algebras)

**16R40** Identities other than those of matrices over commutative rings

**16R50** Other kinds of identities (generalized polynomial, rational, involution)

**16R60** Functional identities (associative rings and algebras)

**16R99** None of the above, but in this section

## 16Sxx Associative rings and algebras arising under various constructions

**16S10** Associative rings determined by universal properties (free algebras, coproducts, adjunction of inverses, etc.)

**16S15** Finite generation, finite presentability, normal forms (diamond lemma, term-rewriting)

**16S20** Centralizing and normalizing extensions

**16S30** Universal enveloping algebras of Lie algebras [See mainly 17B35]

**16S32** Rings of differential operators (associative algebraic aspects) [See also 13N10, 32C38]

**16S34** Group rings [See also 20C05, 20C07], Laurent polynomial rings (associative algebraic aspects)

**16S35** Twisted and skew group rings, crossed products

**16S36** Ordinary and skew polynomial rings and semigroup rings [See also 20M25]

**16S37** Quadratic and Koszul algebras

**16S38** Rings arising from noncommutative algebraic geometry [See also 14A22]

**16S40** Smash products of general Hopf actions [See also 16T05]

**16S50** Endomorphism rings; matrix rings [See also 15-XX]

**16S60** Associative rings of functions, subdirect products, sheaves of rings

**16S70** Extensions of associative rings by ideals

**16S80** Deformations of associative rings [See also 13D10, 14D15]

**16S85** Associative rings of fractions and localizations [See also 13B30]

**16S88** Leavitt path algebras

**16S90** Torsion theories; radicals on module categories (associative algebraic aspects) [See also 13D30, 18E40] {For radicals of rings, see 16Nxx}

**16S99** None of the above, but in this section

## 16Txx Hopf algebras, quantum groups and related topics

**16T05** Hopf algebras and their applications [See also 16S40, 57T05]

**16T10** Bialgebras

**16T15** Coalgebras and comodules; corings

**16T20** Ring-theoretic aspects of quantum groups [See also 17B37, 20G42, 81R50]

**16T25** Yang-Baxter equations

**16T30** Connections of Hopf algebras with combinatorics [See also 05Exx]

**16T99** None of the above, but in this section

## 16Uxx Conditions on elements

**16U10** Integral domains (associative rings and algebras)

**16U20** Ore rings, multiplicative sets, Ore localization

**16U30** Divisibility, noncommutative UFDs

**16U40** Idempotent elements (associative rings and algebras)

**16U60** Units, groups of units (associative rings and algebras)

**16U70** Center, normalizer (invariant elements) (associative rings and algebras)

**16U80** Generalizations of commutativity (associative rings and algebras)

**16U90** Generalized inverses (associative rings and algebras)

**16U99** None of the above, but in this section

## 16Wxx Associative rings and algebras with additional structure

**16W10** Rings with involution; Lie, Jordan and other nonassociative structures [See also 17B60, 17C50, 46Kxx]

**16W20** Automorphisms and endomorphisms

**16W22** Actions of groups and semigroups; invariant theory (associative rings and algebras)

**16W25** Derivations, actions of Lie algebras

**16W50** Graded rings and modules (associative rings and algebras)

**16W55** "Super" (or "skew") structure [See also 17A70, 17Bxx, 17C70] {For exterior algebras, see 15A75; for Clifford algebras, see 11E88, 15A66}

**16W60** Valuations, completions, formal power series and related constructions (associative rings and algebras) [See also 13Jxx]

**16W70** Filtered associative rings; filtrational and graded techniques

**16W80** Topological and ordered rings and modules [See also 06F25, 13Jxx]

**16W99** None of the above, but in this section

## 16Yxx Generalizations {For nonassociative rings, see 17-XX}

**16Y20** Hyperrings

**16Y30** Near-rings [See also 12K05]

**16Y60** Semirings [See also 12K10]

**16Y80** $\Gamma$ and fuzzy structures

**16Y99** None of the above, but in this section

## 16Zxx Computational aspects of associative rings {For software etc., see 16-04}

**16Z05** Computational aspects of associative rings (general theory) [See also 68W30]

**16Z10** Gröbner-Shirshov bases

**16Z99** None of the above, but in this section

# 17-XX Nonassociative rings and algebras

**17-00** General reference works (handbooks, dictionaries, bibliographies, etc.) pertaining to nonassociative rings and algebras

**17-01** Introductory exposition (textbooks, tutorial papers, etc.) pertaining to nonassociative rings and algebras

**17-02** Research exposition (monographs, survey articles) pertaining to nonassociative rings and algebras

**17-03** History of nonassociative rings and algebras [Consider also classification numbers from Section 01]

**17-04** Software, source code, etc. for problems pertaining to nonassociative rings and algebras

**17-06** Proceedings, conferences, collections, etc. pertaining to nonassociative rings and algebras

**17-08** Computational methods for problems pertaining to nonassociative rings and algebras [See also 68W30]

**17-11** Research data for problems pertaining to nonassociative rings and algebras

## 17Axx General nonassociative rings

**17A01** General theory of nonassociative rings and algebras

**17A05** Power-associative rings

**17A15** Noncommutative Jordan algebras

**17A20** Flexible algebras

**17A30** Nonassociative algebras satisfying other identities

**17A32** Leibniz algebras

**17A35** Nonassociative division algebras

**17A36** Automorphisms, derivations, other operators (nonassociative rings and algebras)

**17A40** Ternary compositions

**17A42** Other $n$-ary compositions ($n \geq 3$)

**17A45** Quadratic algebras (but not quadratic Jordan algebras)

**17A50** Free nonassociative algebras

**17A60** Structure theory for nonassociative algebras

**17A61** Gröbner-Shirshov bases in nonassociative algebras

**17A65** Radical theory (nonassociative rings and algebras)

**17A70** Superalgebras

**17A75** Composition algebras

**17A80** Valued algebras

**17A99** None of the above, but in this section

## 17Bxx Lie algebras and Lie superalgebras {For Lie groups, see 22Exx}

**17B01** Identities, free Lie (super)algebras

**17B05** Structure theory for Lie algebras and superalgebras

**17B08** Coadjoint orbits; nilpotent varieties

**17B10** Representations of Lie algebras and Lie superalgebras, algebraic theory (weights)

**17B15** Representations of Lie algebras and Lie superalgebras, analytic theory

**17B20** Simple, semisimple, reductive (super)algebras

**17B22** Root systems

**17B25** Exceptional (super)algebras

**17B30** Solvable, nilpotent (super)algebras

**17B35** Universal enveloping (super)algebras [See also 16S30]

**17B37** Quantum groups (quantized enveloping algebras) and related deformations [See also 16T20, 20G42, 81R50, 82B23]

**17B38** Yang-Baxter equations and Rota-Baxter operators

**17B40** Automorphisms, derivations, other operators for Lie algebras and super algebras

**17B45** Lie algebras of linear algebraic groups [See also 14Lxx and 20Gxx]

**17B50** Modular Lie (super)algebras

**17B55** Homological methods in Lie (super)algebras

**17B56** Cohomology of Lie (super)algebras

**17B60** Lie (super)algebras associated with other structures (associative, Jordan, etc.) [See also 16W10, 17C40, 17C50]

**17B61** Hom-Lie and related algebras

**17B62** Lie bialgebras; Lie coalgebras

**17B63** Poisson algebras

**17B65** Infinite-dimensional Lie (super)algebras [See also 22E65]

**17B66** Lie algebras of vector fields and related (super) algebras

**17B67** Kac-Moody (super)algebras; extended affine Lie algebras; toroidal Lie algebras

**17B68** Virasoro and related algebras

**17B69** Vertex operators; vertex operator algebras and related structures

**17B70** Graded Lie (super)algebras

**17B75** Color Lie (super)algebras

**17B80** Applications of Lie algebras and superalgebras to integrable systems

**17B81** Applications of Lie (super)algebras to physics, etc.

**17B99** None of the above, but in this section

## 17Cxx Jordan algebras (algebras, triples and pairs)

**17C05** Identities and free Jordan structures

**17C10** Structure theory for Jordan algebras

**17C17** Radicals in Jordan algebras

**17C20** Simple, semisimple Jordan algebras

**17C27** Idempotents, Peirce decompositions

**17C30** Associated groups, automorphisms of Jordan algebras

**17C36** Associated manifolds of Jordan algebras

**17C37** Associated geometries of Jordan algebras

**17C40** Exceptional Jordan structures

**17C50** Jordan structures associated with other structures [See also 16W10]

**17C55** Finite-dimensional structures of Jordan algebras

**17C60** Division algebras and Jordan algebras

**17C65** Jordan structures on Banach spaces and algebras [See also 46H70, 46L70]

**17C70** Super structures

**17C90** Applications of Jordan algebras to physics, etc.

**17C99** None of the above, but in this section

## 17Dxx Other nonassociative rings and algebras

**17D05** Alternative rings

**17D10** Mal'tsev rings and algebras

**17D15** Right alternative rings

**17D20** $(\gamma, \delta)$-rings, including $(1, -1)$-rings

**17D25** Lie-admissible algebras

**17D30** (non-Lie) Hom algebras and topics

**17D92** Genetic algebras

**17D99** None of the above, but in this section

# 18-XX Category theory; homological algebra {For commutative rings, see 13Dxx; for associative rings, see 16Exx; for groups, see 20Jxx; for topological groups and related structures, see 57Txx; for algebraic topology, see also 55Nxx, 55Uxx}

**18-00** General reference works (handbooks, dictionaries, bibliographies, etc.) pertaining to category theory

**18-01** Introductory exposition (textbooks, tutorial papers, etc.) pertaining to category theory

**18-02** Research exposition (monographs, survey articles) pertaining to category theory

**18-03** History of category theory [Consider also classification numbers from Section 01]

**18-04** Software, source code, etc. for problems pertaining to category theory

**18-06** Proceedings, conferences, collections, etc. pertaining to category theory

**18-08** Computational methods for problems pertaining to category theory

**18-11** Research data for problems pertaining to category theory

## 18Axx General theory of categories and functors

**18A05** Definitions and generalizations in theory of categories

**18A10** Graphs, diagram schemes, precategories

**18A15** Foundations, relations to logic and deductive systems [See also 03-XX]

**18A20** Epimorphisms, monomorphisms, special classes of morphisms, null morphisms

**18A22** Special properties of functors (faithful, full, etc.)

**18A23** Natural morphisms, dinatural morphisms

**18A25** Functor categories, comma categories

**18A30** Limits and colimits (products, sums, directed limits, pushouts, fiber products, equalizers, kernels, ends and coends, etc.)

**18A32** Factorization systems, substructures, quotient structures, congruences, amalgams

**18A35** Categories admitting limits (complete categories), functors preserving limits, completions

**18A40** Adjoint functors (universal constructions, reflective subcategories, Kan extensions, etc.)

**18A50** Graded categories (general) {For dg categories, see 18G35}

**18A99** None of the above, but in this section

## 18Bxx Special categories

**18B05** Categories of sets, characterizations [See also 03-XX]

**18B10** Categories of spans/cospans, relations, or partial maps

**18B15** Embedding theorems, universal categories [See also 18E20]

**18B20** Categories of machines, automata [See also 03D05, 68Qxx]

**18B25** Topoi [See also 03G30, 18F10]

**18B35** Preorders, orders, domains and lattices (viewed as categories) [See also 06-XX]

**18B40** Groupoids, semigroupoids, semigroups, groups (viewed as categories) [See also 20Axx, 20L05, 20Mxx]

**18B50** Extensive, distributive, and adhesive categories

**18B99** None of the above, but in this section

## 18Cxx Categories and theories

**18C05** Equational categories [See also 03C05, 08C05]

**18C10** Theories (e.g., algebraic theories), structure, and semantics [See also 03G30]

**18C15** Monads (= standard construction, triple or triad), algebras for monads, homology and derived functors for monads [See also 18Gxx] {For functional programming, see also 68N18}

**18C20** Eilenberg-Moore and Kleisli constructions for monads

**18C30** Sketches and generalizations

**18C35** Accessible and locally presentable categories

**18C40** Structured objects in a category (group objects, etc.)

**18C50** Categorical semantics of formal languages [See also 68Q55, 68Q65]

**18C99** None of the above, but in this section

## 18Dxx Categorical structures

**18D15** Closed categories (closed monoidal and Cartesian closed categories, etc.)

**18D20** Enriched categories (over closed or monoidal categories)

**18D25** Actions of a monoidal category, tensorial strength {For functional programming, see also 68N18}

**18D30** Fibered categories

**18D40** Internal categories and groupoids {For double categories, see 18N10; for topological groupoids, see 22A22; for Lie groupoids, see 58H05}

**18D60** Profunctors (= correspondences, distributors, modules)

**18D65** Proarrow equipments, Yoneda structures, KZ doctrines (lax idempotent monads)

**18D70** Formal category theory

**18D99** None of the above, but in this section

## 18Exx Categorical algebra

**18E05** Preadditive, additive categories

**18E08** Regular categories, Barr-exact categories

**18E10** Abelian categories, Grothendieck categories

**18E13** Protomodular categories, semi-abelian categories, Mal'tsev categories [See also 08B05 and 18B10]

**18E20** Categorical embedding theorems [See also 18B15]

**18E35** Localization of categories, calculus of fractions {For homotopical aspects, see also 18N55, 55P60}

**18E40** Torsion theories, radicals [See also 13D30, 16S90]

**18E45** Definable subcategories and connections with model theory [See also 13C60]

**18E50** Categorical Galois theory

**18E99** None of the above, but in this section

## 18Fxx Categories in geometry and topology

**18F05** Local categories and functors

**18F10** Grothendieck topologies and Grothendieck topoi [See also 14F20, 18B25]

**18F15** Abstract manifolds and fiber bundles (category-theoretic aspects) [See also 55Rxx, 57Pxx]

**18F20** Presheaves and sheaves, stacks, descent conditions (category-theoretic aspects) [See also 14F06, 14F08, 32C35, 32L10, 54B40, 55N30]

**18F25** Algebraic $K$-theory and $L$-theory (category-theoretic aspects) [See also 11Exx, 11R70, 11S70, 12-XX, 13D15, 14Cxx, 16E20, 19-XX, 46L80, 57R65, 57R67]

**18F30** Grothendieck groups (category-theoretic aspects) [See also 13D15, 16E20, 19Axx]

**18F40** Synthetic differential geometry, tangent categories, differential categories

**18F50** Goodwillie calculus and functor calculus

**18F60** Categories of topological spaces and continuous mappings [See also 54-XX]

**18F70** Frames and locales, pointfree topology, Stone duality [See also 06D22, 18B35]

**18F75** Quantales [See also 06F07, 18B35]

**18F99** None of the above, but in this section

## 18Gxx Homological algebra in category theory, derived categories and functors [See also 13Dxx, 16Exx, 20Jxx, 55Nxx, 55Uxx, 57Txx]

**18G05** Projectives and injectives (category-theoretic aspects) [See also 13C10, 13C11, 16D40, 16D50]

**18G10** Resolutions; derived functors (category-theoretic aspects) [See also 13D02, 16E05, 18Gxx]

**18G15** Ext and Tor, generalizations, Künneth formula (category-theoretic aspects) [See also 55U25]

**18G20** Homological dimension (category-theoretic aspects) [See also 13D05, 16E10]

**18G25** Relative homological algebra, projective classes (category-theoretic aspects)

**18G31** Simplicial modules and Dold-Kan correspondence

**18G35** Chain complexes (category-theoretic aspects), dg categories [See also 14F08, 18G80, 55U15]

**18G40** Spectral sequences, hypercohomology [See also 55Txx]

**18G45** 2-groups, crossed modules, crossed complexes

**18G50** Nonabelian homological algebra (category-theoretic aspects)

**18G65** Stable module categories [See also 20C20]

**18G70** $A_\infty$-categories, relations with homological mirror symmetry [See also 14F08, 14J33, 53D37]

**18G80** Derived categories, triangulated categories

**18G85** Graph complexes and graph homology {For relations with deformation quantization, see 53D55}

**18G90** Other (co)homology theories (category-theoretic aspects) [See also 19D55, 46L80, 58J20, 58J22]

**18G99** None of the above, but in this section

## 18Mxx Monoidal categories and operads

**18M05** Monoidal categories, symmetric monoidal categories [See also 19D23]

**18M10** Traced monoidal categories, compact closed categories, star-autonomous categories

**18M15** Braided monoidal categories and ribbon categories {For applications to knot theory, see also 57Kxx; for applications to quantum groups, see also 16T20, 17B37, 81R50}

**18M20** Fusion categories, modular tensor categories, modular functors {For applications to topological quantum field theories, see also 57R56; for applications to conformal field theories, see also 81T40}

**18M25** Tannakian categories {For applications to motives, see also 14C15, 19E15}

**18M30** String diagrams and graphical calculi

**18M35** Categories of networks and processes, compositionality

**18M40** Dagger categories, categorical quantum mechanics [See also 81P68]

**18M45** Categorical aspects of linear logic [See also 03B47]

**18M50** Bimonoidal, skew-monoidal, duoidal categories

**18M60** Operads (general)

**18M65** Non-symmetric operads, multicategories, generalized multicategories

**18M70** Algebraic operads, cooperads, and Koszul duality

**18M75** Topological and simplicial operads [See also 18N60]

**18M80** Species, Hopf monoids, operads in combinatorics

**18M85** Polycategories/dioperads, properads, PROPs, cyclic operads, modular operads

**18M90** Globular operads

**18M99** None of the above, but in this section

## 18Nxx Higher categories and homotopical algebra

**18N10** 2-categories, bicategories, double categories

**18N15** 2-dimensional monad theory [See also 18C15]

**18N20** Tricategories, weak $n$-categories, coherence, semi-strictification

**18N25** Categorification

**18N30** Strict omega-categories, computads, polygraphs

**18N40** Homotopical algebra, Quillen model categories, derivators [See also 55U35]

**18N45** Categories of fibrations, relations to $K$-theory, relations to type theory

**18N50** Simplicial sets, simplicial objects [See also 55U10]

**18N55** Localizations (e.g., simplicial localization, Bousfield localization) [See also 18E35, 55P60]

**18N60** $(\infty, 1)$-categories (quasi-categories, Segal spaces, etc.); $\infty$-topoi, stable $\infty$-categories [See also 55U35, 55U40]

**18N65** $(\infty, n)$-categories and $(\infty, \infty)$-categories

**18N70** $\infty$-operads and higher algebra [See also 18M75]

**18N99** None of the above, but in this section

# 19-XX $K$-theory [See also 16E20, 18F25]

**19-00** General reference works (handbooks, dictionaries, bibliographies, etc.) pertaining to $K$-theory

**19-01** Introductory exposition (textbooks, tutorial papers, etc.) pertaining to $K$-theory

**19-02** Research exposition (monographs, survey articles) pertaining to $K$-theory

**19-03** History of $K$-theory [Consider also classification numbers from Section 01]

**19-04** Software, source code, etc. for problems pertaining to $K$-theory

**19-06** Proceedings, conferences, collections, etc. pertaining to $K$-theory

**19-08** Computational methods for problems pertaining to $K$-theory

**19-11** Research data for problems pertaining to $K$-theory

## 19Axx Grothendieck groups and $K_0$ [See also 13D15, 18F30]

**19A13** Stability for projective modules [See also 13C10]

**19A15** Efficient generation of modules

**19A22** Frobenius induction, Burnside and representation rings

**19A31** $K_0$ of group rings and orders

**19A49** $K_0$ of other rings

**19A99** None of the above, but in this section

## 19Bxx Whitehead groups and $K_1$

**19B10** Stable range conditions

**19B14** Stability for linear groups

**19B28** $K_1$ of group rings and orders [See also 57Q10]

**19B37** Congruence subgroup problems [See also 20H05]

**19B99** None of the above, but in this section

## 19Cxx Steinberg groups and $K_2$

**19C09** Central extensions and Schur multipliers

**19C20** Symbols, presentations and stability of $K_2$

**19C30** $K_2$ and the Brauer group

**19C40** Excision for $K_2$

**19C99** None of the above, but in this section

## 19Dxx Higher algebraic $K$-theory

**19D06** $Q$- and plus-constructions

**19D10** Algebraic $K$-theory of spaces

**19D23** Symmetric monoidal categories [See also 18M05]

**19D25** Karoubi-Villamayor-Gersten $K$-theory

**19D35** Negative $K$-theory, NK and Nil

**19D45** Higher symbols, Milnor $K$-theory

**19D50** Computations of higher $K$-theory of rings [See also 13D15, 16E20]

**19D55** $K$-theory and homology; cyclic homology and cohomology [See also 18G90]

**19D99** None of the above, but in this section

## 19Exx $K$-theory in geometry

**19E08** $K$-theory of schemes [See also 14C35]

**19E15** Algebraic cycles and motivic cohomology ($K$-theoretic aspects) [See also 14C25, 14C35, 14F42]

**19E20** Relations of $K$-theory with cohomology theories [See also 14Fxx]

**19E99** None of the above, but in this section

## 19Fxx $K$-theory in number theory [See also 11R70, 11S70]

**19F05** Generalized class field theory ($K$-theoretic aspects) [See also 11G45]

**19F15** Symbols and arithmetic ($K$-theoretic aspects) [See also 11R37]

**19F27** Étale cohomology, higher regulators, zeta and $L$-functions ($K$-theoretic aspects) [See also 11G40, 11R42, 11S40, 14F20, 14G10]

**19F99** None of the above, but in this section

## 19Gxx $K$-theory of forms [See also 11Exx]

**19G05** Stability for quadratic modules

**19G12** Witt groups of rings [See also 11E81]

**19G24** $L$-theory of group rings [See also 11E81]

**19G38** Hermitian $K$-theory, relations with $K$-theory of rings

**19G99** None of the above, but in this section

## 19Jxx Obstructions from topology

**19J05** Finiteness and other obstructions in $K_0$

**19J10** Whitehead (and related) torsion

**19J25** Surgery obstructions ($K$-theoretic aspects) [See also 57R67]

**19J35** Obstructions to group actions ($K$-theoretic aspects)

**19J99** None of the above, but in this section

## 19Kxx $K$-theory and operator algebras [See mainly 46L80, and also 46M20]

**19K14** $K_0$ as an ordered group, traces

**19K33** Ext and $K$-homology [See also 55N22]

**19K35** Kasparov theory ($KK$-theory) [See also 58J22]

**19K56** Index theory [See also 58J20, 58J22]

**19K99** None of the above, but in this section

## 19Lxx Topological $K$-theory [See also 55N15, 55R50, 55S25]

**19L10** Riemann-Roch theorems, Chern characters

**19L20** $J$-homomorphism, Adams operations [See also 55Q50]

**19L41** Connective $K$-theory, cobordism [See also 55N22]

**19L47** Equivariant $K$-theory [See also 55N91, 55P91, 55Q91, 55R91, 55S91]

**19L50** Twisted $K$-theory; differential $K$-theory

**19L64** Geometric applications of topological $K$-theory

**19L99** None of the above, but in this section

## 19Mxx Miscellaneous applications of $K$-theory

**19M05** Miscellaneous applications of $K$-theory

**19M99** None of the above, but in this section

# 20-XX Group theory and generalizations

**20-00** General reference works (handbooks, dictionaries, bibliographies, etc.) pertaining to group theory

**20-01** Introductory exposition (textbooks, tutorial papers, etc.) pertaining to group theory

**20-02** Research exposition (monographs, survey articles) pertaining to group theory

**20-03** History of group theory [Consider also classification numbers from Section 01]

**20-04** Software, source code, etc. for problems pertaining to group theory

**20-06** Proceedings, conferences, collections, etc. pertaining to group theory

**20-08** Computational methods for problems pertaining to group theory

**20-11** Research data for problems pertaining to group theory

## 20Axx Foundations

**20A05** Axiomatics and elementary properties of groups

**20A10** Metamathematical considerations in group theory {For word problems, see 20F10}

**20A15** Applications of logic to group theory

**20A99** None of the above, but in this section

## 20Bxx Permutation groups

**20B05** General theory for finite permutation groups

**20B07** General theory for infinite permutation groups

**20B10** Characterization theorems for permutation groups

**20B15** Primitive groups

**20B20** Multiply transitive finite groups

**20B22** Multiply transitive infinite groups

**20B25** Finite automorphism groups of algebraic, geometric, or combinatorial structures [See also 05Bxx, 12F10, 20G40, 20H30, 51-XX]

**20B27** Infinite automorphism groups [See also 12F10]

**20B30** Symmetric groups

**20B35** Subgroups of symmetric groups

**20B99** None of the above, but in this section

## 20Cxx Representation theory of groups {For representation rings and Burnside rings, see also 19A22}

**20C05** Group rings of finite groups and their modules (group-theoretic aspects) [See also 16S34]

**20C07** Group rings of infinite groups and their modules (group-theoretic aspects) [See also 16S34]

**20C08** Hecke algebras and their representations

**20C10** Integral representations of finite groups

**20C11** $p$-adic representations of finite groups

**20C12** Integral representations of infinite groups

**20C15** Ordinary representations and characters

**20C20** Modular representations and characters

**20C25** Projective representations and multipliers

**20C30** Representations of finite symmetric groups

**20C32** Representations of infinite symmetric groups

**20C33** Representations of finite groups of Lie type

**20C34** Representations of sporadic groups

**20C35** Applications of group representations to physics and other areas of science

**20C99** None of the above, but in this section

## 20Dxx Abstract finite groups

**20D05** Finite simple groups and their classification

**20D06** Simple groups: alternating groups and groups of Lie type [See also 20Gxx]

**20D08** Simple groups: sporadic groups

**20D10** Finite solvable groups, theory of formations, Schunck classes, Fitting classes, $\pi$-length, ranks [See also 20F17]

**20D15** Finite nilpotent groups, $p$-groups

**20D20** Sylow subgroups, Sylow properties, $\pi$-groups, $\pi$-structure

**20D25** Special subgroups (Frattini, Fitting, etc.)

**20D30** Series and lattices of subgroups

**20D35** Subnormal subgroups of abstract finite groups

**20D40** Products of subgroups of abstract finite groups

**20D45** Automorphisms of abstract finite groups

**20D60** Arithmetic and combinatorial problems involving abstract finite groups

**20D99** None of the above, but in this section

## 20Exx Structure and classification of infinite or finite groups

**20E05** Free nonabelian groups

**20E06** Free products of groups, free products with amalgamation, Higman-Neumann-Neumann extensions, and generalizations

**20E07** Subgroup theorems; subgroup growth

**20E08** Groups acting on trees [See also 20F65]

**20E10** Quasivarieties and varieties of groups

**20E15** Chains and lattices of subgroups, subnormal subgroups [See also 20F22]

**20E18** Limits, profinite groups

**20E22** Extensions, wreath products, and other compositions of groups [See also 20J05]

**20E25** Local properties of groups

**20E26** Residual properties and generalizations; residually finite groups

**20E28** Maximal subgroups

**20E32** Simple groups [See also 20D05]

**20E34** General structure theorems for groups

**20E36** Automorphisms of infinite groups {For automorphisms of finite groups, see 20D45}

**20E42** Groups with a $BN$-pair; buildings [See also 51E24]

**20E45** Conjugacy classes for groups

**20E99** None of the above, but in this section

# 20Fxx Special aspects of infinite or finite groups

**20F05** Generators, relations, and presentations of groups

**20F06** Cancellation theory of groups; application of van Kampen diagrams [See also 57M05]

**20F10** Word problems, other decision problems, connections with logic and automata (group-theoretic aspects) [See also 03B25, 03D05, 03D40, 06B25, 08A50, 20M05, 68Q70]

**20F11** Groups of finite Morley rank [See also 03C45, 03C60]

**20F12** Commutator calculus

**20F14** Derived series, central series, and generalizations for groups

**20F16** Solvable groups, supersolvable groups [See also 20D10]

**20F17** Formations of groups, Fitting classes [See also 20D10]

**20F18** Nilpotent groups [See also 20D15]

**20F19** Generalizations of solvable and nilpotent groups

**20F22** Other classes of groups defined by subgroup chains

**20F24** FC-groups and their generalizations

**20F28** Automorphism groups of groups [See also 20E36]

**20F29** Representations of groups as automorphism groups of algebraic systems

**20F34** Fundamental groups and their automorphisms (group-theoretic aspects) [See also 57M05, 57Sxx]

**20F36** Braid groups; Artin groups

**20F38** Other groups related to topology or analysis

**20F40** Associated Lie structures for groups

**20F45** Engel conditions

**20F50** Periodic groups; locally finite groups

**20F55** Reflection and Coxeter groups (group-theoretic aspects) [See also 22E40, 51F15]

**20F60** Ordered groups (group-theoretic aspects) [See mainly 06F15]

**20F65** Geometric group theory [See also 05C25, 20E08, 57Mxx]

**20F67** Hyperbolic groups and nonpositively curved groups

**20F69** Asymptotic properties of groups

**20F70** Algebraic geometry over groups; equations over groups

**20F99** None of the above, but in this section

**20Gxx Linear algebraic groups and related topics {For arithmetic theory, see 11E57, 11H56; for geometric theory, see 14Lxx, 22Exx; for other methods in representation theory, see 15A30, 22E45, 22E46, 22E47, 22E50, 22E55}**

**20G05** Representation theory for linear algebraic groups

**20G07** Structure theory for linear algebraic groups

**20G10** Cohomology theory for linear algebraic groups

**20G15** Linear algebraic groups over arbitrary fields

**20G20** Linear algebraic groups over the reals, the complexes, the quaternions

**20G25** Linear algebraic groups over local fields and their integers

**20G30** Linear algebraic groups over global fields and their integers

**20G35** Linear algebraic groups over adèles and other rings and schemes

**20G40** Linear algebraic groups over finite fields

**20G41** Exceptional groups

**20G42** Quantum groups (quantized function algebras) and their representations [See also 16T20, 17B37, 81R50]

**20G43** Schur and $q$-Schur algebras

**20G44** Kac-Moody groups

**20G45** Applications of linear algebraic groups to the sciences

**20G99** None of the above, but in this section

## 20Hxx Other groups of matrices [See also 15A30]

**20H05** Unimodular groups, congruence subgroups (group-theoretic aspects) [See also 11F06, 19B37, 22E40, 51F20]

**20H10** Fuchsian groups and their generalizations (group-theoretic aspects) [See also 11F06, 22E40, 30F35, 32Nxx]

**20H15** Other geometric groups, including crystallographic groups [See also 51-XX, especially 51F15, and 82D25]

**20H20** Other matrix groups over fields

**20H25** Other matrix groups over rings

**20H30** Other matrix groups over finite fields

**20H99** None of the above, but in this section

## 20Jxx Connections of group theory with homological algebra and category theory

**20J05** Homological methods in group theory

**20J06** Cohomology of groups

**20J15** Category of groups

**20J99** None of the above, but in this section

## 20Kxx Abelian groups

**20K01** Finite abelian groups {For sumsets, see 11B13, 11P70}

**20K10** Torsion groups, primary groups and generalized primary groups

**20K15** Torsion-free groups, finite rank

**20K20** Torsion-free groups, infinite rank

**20K21** Mixed groups

**20K25** Direct sums, direct products, etc. for abelian groups

**20K27** Subgroups of abelian groups

**20K30** Automorphisms, homomorphisms, endomorphisms, etc. for abelian groups

**20K35** Extensions of abelian groups

**20K40** Homological and categorical methods for abelian groups

**20K45** Topological methods for abelian groups [See also 22A05, 22B05]

**20K99** None of the above, but in this section

## 20Lxx Groupoids (i.e. small categories in which all morphisms are isomorphisms) {For sets with a single binary operation, see 20N02; for topological groupoids, see 22A22, 58H05}

**20L05** Groupoids (i.e. small categories in which all morphisms are isomorphisms) {For sets with a single binary operation, see 20N02; for topological groupoids, see 22A22, 58H05}

**20L99** None of the above, but in this section

## 20Mxx Semigroups

**20M05** Free semigroups, generators and relations, word problems [See also 03D40, 08A50, 20F10]

**20M07** Varieties and pseudovarieties of semigroups

**20M10** General structure theory for semigroups

**20M11** Radical theory for semigroups

**20M12** Ideal theory for semigroups

**20M13** Arithmetic theory of semigroups

**20M14** Commutative semigroups

**20M15** Mappings of semigroups

**20M17** Regular semigroups

**20M18** Inverse semigroups

**20M19** Orthodox semigroups

**20M20** Semigroups of transformations, relations, partitions, etc. [See also 47D03, 47H20, 54H15]

**20M25** Semigroup rings, multiplicative semigroups of rings [See also 16S36, 16Y60]

**20M30** Representation of semigroups; actions of semigroups on sets

**20M32** Algebraic monoids

**20M35** Semigroups in automata theory, linguistics, etc. [See also 03D05, 68Q70, 68T50]

**20M50** Connections of semigroups with homological algebra and category theory

**20M75** Generalizations of semigroups

**20M99** None of the above, but in this section

## 20Nxx Other generalizations of groups

**20N02** Sets with a single binary operation (groupoids) {For groupoids in connection with category theory, see 20L05; for topological groupoids, see 22A22, 58H05}

**20N05** Loops, quasigroups [See also 05Bxx]

**20N10** Ternary systems (heaps, semiheaps, heapoids, etc.)

**20N15** $n$-ary systems ($n \geq 3$)

**20N20** Hypergroups

**20N25** Fuzzy groups [See also 03E72]

**20N99** None of the above, but in this section

## 20Pxx Probabilistic methods in group theory [See also 60Bxx]

**20P05** Probabilistic methods in group theory [See also 60Bxx]

**20P99** None of the above, but in this section

# 22-XX Topological groups, Lie groups {For transformation groups, see 54H15, 57Sxx, 58-XX; for abstract harmonic analysis, see 43-XX}

**22-00** General reference works (handbooks, dictionaries, bibliographies, etc.) pertaining to topological groups

**22-01** Introductory exposition (textbooks, tutorial papers, etc.) pertaining to topological groups

**22-02** Research exposition (monographs, survey articles) pertaining to topological groups

**22-03** History of topological groups [Consider also classification numbers from Section 01]

**22-04** Software, source code, etc. for problems pertaining to topological groups

**22-06** Proceedings, conferences, collections, etc. pertaining to topological groups

**22-08** Computational methods for problems pertaining to topological groups

**22-11** Research data for problems pertaining to topological groups

# 22Axx Topological and differentiable algebraic systems {For topological rings and fields, see 12Jxx, 13Jxx, 16W80}

**22A05** Structure of general topological groups

**22A10** Analysis on general topological groups

**22A15** Structure of topological semigroups

**22A20** Analysis on topological semigroups

**22A22** Topological groupoids (including differentiable and Lie groupoids) [See also 58H05]

**22A25** Representations of general topological groups and semigroups

**22A26** Topological semilattices, lattices and applications [See also 06B30, 06B35, 06F30]

**22A30** Other topological algebraic systems and their representations

**22A99** None of the above, but in this section

# 22Bxx Locally compact abelian groups (LCA groups)

**22B05** General properties and structure of LCA groups

**22B10** Structure of group algebras of LCA groups

**22B99** None of the above, but in this section

# 22Cxx Compact groups

**22C05** Compact groups

**22C99** None of the above, but in this section

# 22Dxx Locally compact groups and their algebras

**22D05** General properties and structure of locally compact groups

**22D10** Unitary representations of locally compact groups

**22D12** Other representations of locally compact groups

**22D15** Group algebras of locally compact groups

**22D20** Representations of group algebras

**22D25** $C^*$-algebras and $W^*$-algebras in relation to group representations [See also 46Lxx]

**22D30** Induced representations for locally compact groups

**22D35** Duality theorems for locally compact groups

**22D40** Ergodic theory on groups [See also 28Dxx]

**22D45** Automorphism groups of locally compact groups

**22D50** Rigidity in locally compact groups

**22D55** Kazhdan's property (T), the Haagerup property, and generalizations

**22D99** None of the above, but in this section

## 22Exx Lie groups {For the topology of Lie groups and homogeneous spaces, see 57Sxx, 57Txx; for analysis thereon, see 43A80, 43A85, 43A90}

**22E05** Local Lie groups [See also 34-XX, 35-XX, 58H05]

**22E10** General properties and structure of complex Lie groups [See also 32M05]

**22E15** General properties and structure of real Lie groups

**22E20** General properties and structure of other Lie groups

**22E25** Nilpotent and solvable Lie groups

**22E27** Representations of nilpotent and solvable Lie groups (special orbital integrals, non-type I representations, etc.)

**22E30** Analysis on real and complex Lie groups [See also 33C80, 43-XX]

**22E35** Analysis on $p$-adic Lie groups

**22E40** Discrete subgroups of Lie groups [See also 20Hxx, 32Nxx]

**22E41** Continuous cohomology of Lie groups [See also 57R32, 57Txx, 58H10]

**22E43** Structure and representation of the Lorentz group

**22E45** Representations of Lie and linear algebraic groups over real fields: analytic methods {For the purely algebraic theory, see 20G05}

**22E46** Semisimple Lie groups and their representations

**22E47** Representations of Lie and real algebraic groups: algebraic methods (Verma modules, etc.) [See also 17B10]

**22E50** Representations of Lie and linear algebraic groups over local fields [See also 11F70, 20G05]

**22E55** Representations of Lie and linear algebraic groups over global fields and adèle rings [See also 11F70, 20G05]

**22E57** Geometric Langlands program: representation-theoretic aspects [See also 14D24]

**22E60** Lie algebras of Lie groups {For the algebraic theory of Lie algebras, see 17Bxx}

**22E65** Infinite-dimensional Lie groups and their Lie algebras: general properties [See also 17B65, 58B25, 58D05 58H05]

**22E66** Analysis on and representations of infinite-dimensional Lie groups

**22E67** Loop groups and related constructions, group-theoretic treatment [See also 58D05]

**22E70** Applications of Lie groups to the sciences; explicit representations [See also 81R05, 81R10]

**22E99** None of the above, but in this section

## 22Fxx Noncompact transformation groups

**22F05** General theory of group and pseudogroup actions {For topological properties of spaces with an action, see 57S20}

**22F10** Measurable group actions [See also 22D40, 28Dxx, 37Axx]

**22F30** Homogeneous spaces {For general actions on manifolds or preserving geometrical structures, see 57M60, 57Sxx; for discrete subgroups of Lie groups, see especially 22E40}

**22F50** Groups as automorphisms of other structures

**22F99** None of the above, but in this section

# 26-XX Real functions [See also 54C30]

**26-00** General reference works (handbooks, dictionaries, bibliographies, etc.) pertaining to real functions

**26-01** Introductory exposition (textbooks, tutorial papers, etc.) pertaining to real functions

**26-02** Research exposition (monographs, survey articles) pertaining to real functions

**26-03** History of real functions [Consider also classification numbers from Section 01]

**26-04** Software, source code, etc. for problems pertaining to real functions

**26-06** Proceedings, conferences, collections, etc. pertaining to real functions

**26-08** Computational methods for problems pertaining to real functions

**26-11** Research data for problems pertaining to real functions

## 26Axx Functions of one variable

**26A03** Foundations: limits and generalizations, elementary topology of the line

**26A06** One-variable calculus

**26A09** Elementary functions

**26A12** Rate of growth of functions, orders of infinity, slowly varying functions [See also 26A48]

**26A15** Continuity and related questions (modulus of continuity, semicontinuity, discontinuities, etc.) for real functions in one variable {For properties determined by Fourier coefficients, see 42A16; for those determined by approximation properties, see 41A25, 41A27}

**26A16** Lipschitz (Hölder) classes

**26A18** Iteration of real functions in one variable [See also 37Bxx, 37Cxx, 37Exx, 39B12, 47H10, 54H25]

**26A21** Classification of real functions; Baire classification of sets and functions [See also 03E15, 28A05, 54C50, 54H05]

**26A24** Differentiation (real functions of one variable): general theory, generalized derivatives, mean value theorems [See also 28A15]

**26A27** Nondifferentiability (nondifferentiable functions, points of nondifferentiability), discontinuous derivatives

**26A30** Singular functions, Cantor functions, functions with other special properties

**26A33** Fractional derivatives and integrals

**26A36** Antidifferentiation

**26A39** Denjoy and Perron integrals, other special integrals

**26A42** Integrals of Riemann, Stieltjes and Lebesgue type [See also 28-XX]

**26A45** Functions of bounded variation, generalizations

**26A46** Absolutely continuous real functions in one variable

**26A48** Monotonic functions, generalizations

**26A51** Convexity of real functions in one variable, generalizations

**26A99** None of the above, but in this section

## 26Bxx Functions of several variables

**26B05** Continuity and differentiation questions

**26B10** Implicit function theorems, Jacobians, transformations with several variables

**26B12** Calculus of vector functions

**26B15** Integration of real functions of several variables: length, area, volume [See also 28A75, 51M25]

**26B20** Integral formulas of real functions of several variables (Stokes, Gauss, Green, etc.)

**26B25** Convexity of real functions of several variables, generalizations

**26B30** Absolutely continuous real functions of several variables, functions of bounded variation

**26B35** Special properties of functions of several variables, Hölder conditions, etc.

**26B40** Representation and superposition of functions

**26B99** None of the above, but in this section

## 26Cxx Polynomials, rational functions in real analysis

**26C05** Real polynomials: analytic properties, etc. [See also 12Dxx, 12Exx]

**26C10** Real polynomials: location of zeros {For algebraic theory, see 12D10; for complex methods, see 30C15; for numerical methods, see 65H05}

**26C15** Real rational functions [See also 14Pxx]

**26C99** None of the above, but in this section

## 26Dxx Inequalities in real analysis {For maximal function inequalities, see 42B25; for functional inequalities, see 39B72; for probabilistic inequalities, see 60E15}

**26D05** Inequalities for trigonometric functions and polynomials

**26D07** Inequalities involving other types of functions

**26D10** Inequalities involving derivatives and differential and integral operators

**26D15** Inequalities for sums, series and integrals

**26D20** Other analytical inequalities

**26D99** None of the above, but in this section

## 26Exx Miscellaneous topics in real functions [See also 58Cxx]

**26E05** Real-analytic functions [See also 32B05, 32C05]

**26E10** $C^\infty$-functions, quasi-analytic functions [See also 58C25]

**26E15** Calculus of functions on infinite-dimensional spaces [See also 46G05, 58Cxx]

**26E20** Calculus of functions taking values in infinite-dimensional spaces [See also 46E40, 46G10, 58Cxx]

**26E25** Set-valued functions [See also 28B20, 49J53, 54C60] {For nonsmooth analysis, see 49J52, 58Cxx, 90Cxx}

**26E30** Non-Archimedean analysis [See also 12J25]

**26E35** Nonstandard analysis [See also 03H05, 28E05, 54J05]

**26E40** Constructive real analysis [See also 03F60]

**26E50** Fuzzy real analysis [See also 03E72, 28E10]

**26E60** Means [See also 47A64]

**26E70** Real analysis on time scales or measure chains {For dynamic equations on time scales or measure chains, see 34N05}

**26E99** None of the above, but in this section

# 28-XX Measure and integration {For analysis on manifolds, see 58-XX}

**28-00** General reference works (handbooks, dictionaries, bibliographies, etc.) pertaining to measure and integration

**28-01** Introductory exposition (textbooks, tutorial papers, etc.) pertaining to measure and integration

**28-02** Research exposition (monographs, survey articles) pertaining to measure and integration

**28-03** History of measure and integration [Consider also classification numbers from Section 01]

**28-04** Software, source code, etc. for problems pertaining to measure and integration

**28-06** Proceedings, conferences, collections, etc. pertaining to measure and integration

**28-08** Computational methods for problems pertaining to measure and integration

**28-11** Research data for problems pertaining to measure and integration

## 28Axx Classical measure theory

**28A05** Classes of sets (Borel fields, $\sigma$-rings, etc.), measurable sets, Suslin sets, analytic sets [See also 03E15, 26A21, 54H05]

**28A10** Real- or complex-valued set functions

**28A12** Contents, measures, outer measures, capacities

**28A15** Abstract differentiation theory, differentiation of set functions [See also 26A24]

**28A20** Measurable and nonmeasurable functions, sequences of measurable functions, modes of convergence

**28A25** Integration with respect to measures and other set functions

**28A33** Spaces of measures, convergence of measures [See also 46E27, 60Bxx]

**28A35** Measures and integrals in product spaces

**28A50** Integration and disintegration of measures

**28A51** Lifting theory [See also 46G15]

**28A60** Measures on Boolean rings, measure algebras [See also 54H10]

**28A75** Length, area, volume, other geometric measure theory [See also 26B15, 49Q15]

**28A78** Hausdorff and packing measures

**28A80** Fractals [See also 37Fxx]

**28A99** None of the above, but in this section

## 28Bxx Set functions, measures and integrals with values in abstract spaces

**28B05** Vector-valued set functions, measures and integrals [See also 46G10]

**28B10** Group- or semigroup-valued set functions, measures and integrals

**28B15** Set functions, measures and integrals with values in ordered spaces

**28B20** Set-valued set functions and measures; integration of set-valued functions; measurable selections [See also 26E25, 54C60, 54C65, 91B14]

**28B99** None of the above, but in this section

## 28Cxx Set functions and measures on spaces with additional structure [See also 46G12, 58C35, 58D20]

**28C05** Integration theory via linear functionals (Radon measures, Daniell integrals, etc.), representing set functions and measures

**28C10** Set functions and measures on topological groups or semigroups, Haar measures, invariant measures [See also 22Axx, 43A05]

**28C15** Set functions and measures on topological spaces (regularity of measures, etc.)

**28C20** Set functions and measures and integrals in infinite-dimensional spaces (Wiener measure, Gaussian measure, etc.) [See also 46G12, 58C35, 58D20, 60B11]

**28C99** None of the above, but in this section

## 28Dxx Measure-theoretic ergodic theory [See also 11K50, 11K55, 22D40, 37Axx, 47A35, 60Fxx, 60G10]

**28D05** Measure-preserving transformations {For measure-preserving transformations and dynamical systems, see 37A05}

**28D10** One-parameter continuous families of measure-preserving transformations {For dynamical systems aspect, see 37A10}

**28D15** General groups of measure-preserving transformations {For dynamical systems aspects, see 37A15}

**28D20** Entropy and other invariants

**28D99** None of the above, but in this section

## 28Exx Miscellaneous topics in measure theory

**28E05** Nonstandard measure theory [See also 03H05, 26E35]

**28E10** Fuzzy measure theory [See also 03E72, 26E50, 94D05]

**28E15** Other connections with logic and set theory

**28E99** None of the above, but in this section

# 30-XX Functions of a complex variable

**30-00** General reference works (handbooks, dictionaries, bibliographies, etc.) pertaining to functions of a complex variable

**30-01** Introductory exposition (textbooks, tutorial papers, etc.) pertaining to functions of a complex variable

**30-02** Research exposition (monographs, survey articles) pertaining to functions of a complex variable

**30-03** History of functions of a complex variable [Consider also classification numbers from Section 01]

**30-04** Software, source code, etc. for problems pertaining to functions of a complex variable

**30-06** Proceedings, conferences, collections, etc. pertaining to functions of a complex variable

**30-08** Computational methods for problems pertaining to functions of a complex variable [See also 65Exx]

**30-11** Research data for problems pertaining to functions of a complex variable

## 30Axx General properties of functions of one complex variable

**30A05** Monogenic and polygenic functions of one complex variable

**30A10** Inequalities in the complex plane

**30A99** None of the above, but in this section

## 30Bxx Series expansions of functions of one complex variable

**30B10** Power series (including lacunary series) in one complex variable

**30B20** Random power series in one complex variable

**30B30** Boundary behavior of power series in one complex variable; over-convergence

**30B40** Analytic continuation of functions of one complex variable

**30B50** Dirichlet series, exponential series and other series in one complex variable [See also 11M41, 42-XX]

**30B60** Completeness problems, closure of a system of functions of one complex variable

**30B70** Continued fractions; complex-analytic aspects [See also 11A55, 40A15]

**30B99** None of the above, but in this section

## 30Cxx Geometric function theory

**30C10** Polynomials and rational functions of one complex variable

**30C15** Zeros of polynomials, rational functions, and other analytic functions of one complex variable (e.g., zeros of functions with bounded Dirichlet integral) {For algebraic theory, see 12D10; for real methods, see 26C10}

**30C20** Conformal mappings of special domains

**30C25** Covering theorems in conformal mapping theory

**30C30** Schwarz-Christoffel-type mappings [See also 65E10]

**30C35** General theory of conformal mappings

**30C40** Kernel functions in one complex variable and applications

**30C45** Special classes of univalent and multivalent functions of one complex variable (starlike, convex, bounded rotation, etc.)

**30C50** Coefficient problems for univalent and multivalent functions of one complex variable

**30C55** General theory of univalent and multivalent functions of one complex variable

**30C62** Quasiconformal mappings in the complex plane

**30C65** Quasiconformal mappings in $\mathbb{R}^n$, other generalizations

**30C70** Extremal problems for conformal and quasiconformal mappings, variational methods

**30C75** Extremal problems for conformal and quasiconformal mappings, other methods

**30C80** Maximum principle, Schwarz's lemma, Lindelöf principle, analogues and generalizations; subordination

**30C85** Capacity and harmonic measure in the complex plane [See also 31A15]

**30C99** None of the above, but in this section

## 30Dxx Entire and meromorphic functions of one complex variable, and related topics

**30D05** Functional equations in the complex plane, iteration and composition of analytic functions of one complex variable [See also 34Mxx, 37Fxx, 39-XX]

**30D10** Representations of entire functions of one complex variable by series and integrals

**30D15** Special classes of entire functions of one complex variable and growth estimates

**30D20** Entire functions of one complex variable (general theory)

**30D30** Meromorphic functions of one complex variable (general theory)

**30D35** Value distribution of meromorphic functions of one complex variable, Nevanlinna theory

**30D40** Cluster sets, prime ends, boundary behavior

**30D45** Normal functions of one complex variable, normal families

**30D60** Quasi-analytic and other classes of functions of one complex variable

**30D99** None of the above, but in this section

## 30Exx Miscellaneous topics of analysis in the complex plane

**30E05** Moment problems and interpolation problems in the complex plane

**30E10** Approximation in the complex plane

**30E15** Asymptotic representations in the complex plane

**30E20** Integration, integrals of Cauchy type, integral representations of analytic functions in the complex plane [See also 45Exx]

**30E25** Boundary value problems in the complex plane [See also 45Exx]

**30E99** None of the above, but in this section

## 30Fxx Riemann surfaces

**30F10** Compact Riemann surfaces and uniformization [See also 14H15, 32G15]

**30F15** Harmonic functions on Riemann surfaces

**30F20** Classification theory of Riemann surfaces

**30F25** Ideal boundary theory for Riemann surfaces

**30F30** Differentials on Riemann surfaces

**30F35** Fuchsian groups and automorphic functions (aspects of compact Riemann surfaces and uniformization) [See also 11Fxx, 20H10, 22E40, 32Gxx, 32Nxx]

**30F40** Kleinian groups (aspects of compact Riemann surfaces and uniformization) [See also 20H10]

**30F45** Conformal metrics (hyperbolic, Poincaré, distance functions)

**30F50** Klein surfaces

**30F60** Teichmüller theory for Riemann surfaces [See also 32G15]

**30F99** None of the above, but in this section

## 30Gxx Generalized function theory

**30G06** Non-Archimedean function theory [See also 12J25]; nonstandard function theory [See also 03H05]

**30G12** Finely holomorphic functions and topological function theory

**30G20** Generalizations of Bers and Vekua type (pseudoanalytic, $p$-analytic, etc.)

**30G25** Discrete analytic functions

**30G30** Other generalizations of analytic functions (including abstract-valued functions)

**30G35** Functions of hypercomplex variables and generalized variables

**30G99** None of the above, but in this section

## 30Hxx Spaces and algebras of analytic functions of one complex variable

**30H05** Spaces of bounded analytic functions of one complex variable

**30H10** Hardy spaces [See also 42B30, 46E30]

**30H15** Nevanlinna spaces and Smirnov spaces

**30H20** Bergman spaces and Fock spaces [See also 46E30, 46E35]

**30H25** Besov spaces and $Q_p$-spaces

**30H30** Bloch spaces

**30H35** BMO-spaces

**30H40** Zygmund spaces

**30H45** de Branges-Rovnyak spaces

**30H50** Algebras of analytic functions of one complex variable

**30H80** Corona theorems

**30H99** None of the above, but in this section

### 30Jxx Function theory on the disc

**30J05** Inner functions of one complex variable

**30J10** Blaschke products

**30J15** Singular inner functions of one complex variable

**30J99** None of the above, but in this section

### 30Kxx Universal holomorphic functions of one complex variable

**30K05** Universal Taylor series in one complex variable

**30K10** Universal Dirichlet series in one complex variable

**30K15** Universal functions of one complex variable

**30K20** Compositional universality

**30K99** None of the above, but in this section

### 30Lxx Analysis on metric spaces

**30L05** Geometric embeddings of metric spaces

**30L10** Quasiconformal mappings in metric spaces

**30L15** Inequalities in metric spaces

**30L99** None of the above, but in this section

# 31-XX Potential theory {For probabilistic potential theory, see 60J45}

**31-00** General reference works (handbooks, dictionaries, bibliographies, etc.) pertaining to potential theory

**31-01** Introductory exposition (textbooks, tutorial papers, etc.) pertaining to potential theory

**31-02** Research exposition (monographs, survey articles) pertaining to potential theory

**31-03** History of potential theory [Consider also classification numbers from Section 01]

**31-04** Software, source code, etc. for problems pertaining to potential theory

**31-06** Proceedings, conferences, collections, etc. pertaining to potential theory

**31-08** Computational methods for problems pertaining to potential theory [See also 65Exx]

**31-11** Research data for problems pertaining to potential theory

### 31Axx Two-dimensional potential theory

**31A05** Harmonic, subharmonic, superharmonic functions in two dimensions

**31A10** Integral representations, integral operators, integral equations methods in two dimensions

**31A15** Potentials and capacity, harmonic measure, extremal length and related notions in two dimensions [See also 30C85]

**31A20** Boundary behavior (theorems of Fatou type, etc.) of harmonic functions in two dimensions

**31A25** Boundary value and inverse problems for harmonic functions in two dimensions

**31A30** Biharmonic, polyharmonic functions and equations, Poisson's equation in two dimensions

**31A35** Connections of harmonic functions with differential equations in two dimensions

**31A99** None of the above, but in this section

## 31Bxx Higher-dimensional potential theory

**31B05** Harmonic, subharmonic, superharmonic functions in higher dimensions

**31B10** Integral representations, integral operators, integral equations methods in higher dimensions

**31B15** Potentials and capacities, extremal length and related notions in higher dimensions

**31B20** Boundary value and inverse problems for harmonic functions in higher dimensions

**31B25** Boundary behavior of harmonic functions in higher dimensions

**31B30** Biharmonic and polyharmonic equations and functions in higher dimensions

**31B35** Connections of harmonic functions with differential equations in higher dimensions

**31B99** None of the above, but in this section

## 31Cxx Generalizations of potential theory

**31C05** Harmonic, subharmonic, superharmonic functions on other spaces

**31C10** Pluriharmonic and plurisubharmonic functions [See also 32U05]

**31C12** Potential theory on Riemannian manifolds and other spaces [See also 53C20] {For Hodge theory, see 58A14}

**31C15** Potentials and capacities on other spaces

**31C20** Discrete potential theory

**31C25** Dirichlet forms

**31C35** Martin boundary theory [See also 60J50]

**31C40** Fine potential theory; fine properties of sets and functions

**31C45** Other generalizations (nonlinear potential theory, etc.)

**31C99** None of the above, but in this section

## 31Dxx Axiomatic potential theory

**31D05** Axiomatic potential theory

**31D99** None of the above, but in this section

## 31Exx Potential theory on fractals and metric spaces

**31E05** Potential theory on fractals and metric spaces

**31E99** None of the above, but in this section

# 32-XX Several complex variables and analytic spaces {For infinite-dimensional holomorphy, see also 46G20, 58B12}

**32-00** General reference works (handbooks, dictionaries, bibliographies, etc.) pertaining to several complex variables and analytic spaces

**32-01** Introductory exposition (textbooks, tutorial papers, etc.) pertaining to several complex variables and analytic spaces

**32-02** Research exposition (monographs, survey articles) pertaining to several complex variables and analytic spaces

**32-03** History of several complex variables and analytic spaces [Consider also classification numbers from Section 01]

**32-04** Software, source code, etc. for problems pertaining to several complex variables and analytic spaces

**32-06** Proceedings, conferences, collections, etc. pertaining to several complex variables and analytic spaces

**32-08** Computational methods for problems pertaining to several complex variables and analytic spaces [See also 65Exx]

**32-11** Research data for problems pertaining to several complex variables and analytic spaces

## 32Axx Holomorphic functions of several complex variables

**32A05** Power series, series of functions of several complex variables

**32A08** Polynomials and rational functions of several complex variables

**32A10** Holomorphic functions of several complex variables

**32A12** Multifunctions of several complex variables

**32A15** Entire functions of several complex variables

**32A17** Special families of functions of several complex variables

**32A18** Bloch functions, normal functions of several complex variables

**32A19** Normal families of holomorphic functions, mappings of several complex variables, and related topics (taut manifolds etc.)

**32A20** Meromorphic functions of several complex variables

**32A22** Nevanlinna theory; growth estimates; other inequalities of several complex variables {For geometric theory, see 32H25, 32H30}

**32A25** Integral representations; canonical kernels (Szegő, Bergman, etc.)

**32A26** Integral representations, constructed kernels (e.g., Cauchy, Fantappiè-type kernels)

**32A27** Residues for several complex variables [See also 32C30]

**32A30** Other generalizations of function theory of one complex variable [Should also be assigned at least one classification number from Section 30] {For functions of several hypercomplex variables, see 30G35}

**32A35** $H^p$-spaces, Nevanlinna spaces of functions in several complex variables [See also 32M15, 42B30, 43A85, 46J15]

**32A36** Bergman spaces of functions in several complex variables

**32A37** Other spaces of holomorphic functions of several complex variables (e.g., bounded mean oscillation (BMOA), vanishing mean oscillation (VMOA)) [See also 46Exx]

**32A38** Algebras of holomorphic functions of several complex variables [See also 46J10, 46J15]

**32A40** Boundary behavior of holomorphic functions of several complex variables

**32A45** Hyperfunctions [See also 46F15]

**32A50** Harmonic analysis of several complex variables [See mainly 43-XX]

**32A55** Singular integrals of functions in several complex variables

**32A60** Zero sets of holomorphic functions of several complex variables

**32A65** Banach algebra techniques applied to functions of several complex variables [See also 46Jxx]

**32A70** Functional analysis techniques applied to functions of several complex variables [See also 46Exx]

**32A99** None of the above, but in this section

## 32Bxx Local analytic geometry [See also 13-XX, 14-XX]

**32B05** Analytic algebras and generalizations, preparation theorems

**32B10** Germs of analytic sets, local parametrization

**32B15** Analytic subsets of affine space

**32B20** Semi-analytic sets, subanalytic sets, and generalizations [See also 14P15]

**32B25** Triangulation and topological properties of semi-analytic andsubanalytic sets, and related questions

**32B99** None of the above, but in this section

## 32Cxx Analytic spaces

**32C05** Real-analytic manifolds, real-analytic spaces [See also 14Pxx, 58A07]

**32C07** Real-analytic sets, complex Nash functions [See also 14P15, 14P20]

**32C09** Embedding of real-analytic manifolds

**32C11** Complex supergeometry [See also 14A22, 14M30, 58A50]

**32C15** Complex spaces

**32C18** Topology of analytic spaces

**32C20** Normal analytic spaces

**32C22** Embedding of analytic spaces

**32C25** Analytic subsets and submanifolds

**32C30** Integration on analytic sets and spaces, currents [See also 32A25, 32A27]

**32C35** Analytic sheaves and cohomology groups [See also 14Fxx, 18F20, 55N30]

**32C36** Local cohomology of analytic spaces

**32C37** Duality theorems for analytic spaces

**32C38** Sheaves of differential operators and their modules, *D*-modules [See also 13N10, 14F10, 16S32, 35A27, 35S35, 58J15]

**32C55** The Levi problem in complex spaces; generalizations

**32C81** Applications of analytic spaces to physics and other areas of science

**32C99** None of the above, but in this section

## 32Dxx Analytic continuation

**32D05** Domains of holomorphy

**32D10** Envelopes of holomorphy

**32D15** Continuation of analytic objects in several complex variables

**32D20** Removable singularities in several complex variables

**32D26** Riemann domains

**32D99** None of the above, but in this section

## 32Exx Holomorphic convexity

**32E05** Holomorphically convex complex spaces, reduction theory

**32E10** Stein spaces

**32E20** Polynomial convexity, rational convexity, meromorphic convexity in several complex variables

**32E30** Holomorphic, polynomial and rational approximation, and interpolation in several complex variables; Runge pairs

**32E35** Global boundary behavior of holomorphic functions of several complex variables

**32E40** The Levi problem

**32E99** None of the above, but in this section

## 32Fxx Geometric convexity in several complex variables

**32F10** $q$-convexity, $q$-concavity

**32F17** Other notions of convexity in relation to several complex variables

**32F18** Finite-type conditions for the boundary of a domain

**32F27** Topological consequences of geometric convexity

**32F32** Analytical consequences of geometric convexity (vanishing theorems, etc.)

**32F45** Invariant metrics and pseudodistances in several complex variables

**32F99** None of the above, but in this section

## 32Gxx Deformations of analytic structures

**32G05** Deformations of complex structures [See also 13D10, 16S80, 58H10, 58H15]

**32G07** Deformations of special (e.g., CR) structures

**32G08** Deformations of fiber bundles

**32G10** Deformations of submanifolds and subspaces

**32G13** Complex-analytic moduli problems {For algebraic moduli problems, see 14D20, 14D22, 14H10, 14J10} [See also 14H15, 14J15]

**32G15** Moduli of Riemann surfaces, Teichmüller theory (complex-analytic aspects in several variables) [See also 14H15, 30Fxx]

**32G20** Period matrices, variation of Hodge structure; degenerations [See also 14D05, 14D07, 14K30]

**32G34** Moduli and deformations for ordinary differential equations (e.g., Knizhnik-Zamolodchikov equation) [See also 34Mxx]

**32G81** Applications of deformations of analytic structures to the sciences

**32G99** None of the above, but in this section

## 32Hxx Holomorphic mappings and correspondences

**32H02** Holomorphic mappings, (holomorphic) embeddings and related questions in several complex variables

**32H04** Meromorphic mappings in several complex variables

**32H12** Boundary uniqueness of mappings in several complex variables

**32H25** Picard-type theorems and generalizations for several complex variables {For function-theoretic properties, see 32A22}

**32H30** Value distribution theory in higher dimensions {For function-theoretic properties, see 32A22}

**32H35** Proper holomorphic mappings, finiteness theorems

**32H40** Boundary regularity of mappings in several complex variables

**32H50** Iteration of holomorphic maps, fixed points of holomorphic maps and related problems for several complex variables

**32H99** None of the above, but in this section

## 32Jxx Compact analytic spaces {For Riemann surfaces, see 14Hxx, 30Fxx; for algebraic theory, see 14Jxx}

**32J05** Compactification of analytic spaces

**32J10** Algebraic dependence theorems

**32J15** Compact complex surfaces

**32J17** Compact complex 3-folds

**32J18** Compact complex $n$-folds

**32J25** Transcendental methods of algebraic geometry (complex-analytic aspects) [See also 14C30]

**32J27** Compact Kähler manifolds: generalizations, classification

**32J81** Applications of compact analytic spaces to the sciences

**32J99** None of the above, but in this section

## 32Kxx Generalizations of analytic spaces

**32K05** Banach analytic manifolds and spaces [See also 46G20, 58Bxx]

**32K07** Formal and graded complex spaces [See also 58C50]

**32K12** Holomorphic maps with infinite-dimensional arguments or values [See also 46G20]

**32K15** Differentiable functions on analytic spaces, differentiable spaces [See also 58C25]

**32K99** None of the above, but in this section

## 32Lxx Holomorphic fiber spaces [See also 55Rxx]

**32L05** Holomorphic bundles and generalizations

**32L10** Sheaves and cohomology of sections of holomorphic vector bundles, general results [See also 14F06, 14H60, 14J60, 18F20, 55N30]

**32L15** Bundle convexity [See also 32F10]

**32L20** Vanishing theorems

**32L25** Twistor theory, double fibrations (complex-analytic aspects) [See also 53C28]

**32L81** Applications of holomorphic fiber spaces to the sciences

**32L99** None of the above, but in this section

## 32Mxx Complex spaces with a group of automorphisms

**32M05** Complex Lie groups, group actions on complex spaces [See also 22E10]

**32M10** Homogeneous complex manifolds [See also 14M17, 57T15]

**32M12** Almost homogeneous manifolds and spaces [See also 14M17]

**32M15** Hermitian symmetric spaces, bounded symmetric domains, Jordan algebras (complex-analytic aspects) [See also 22E10, 22E40, 53C35, 57T15]

**32M17** Automorphism groups of $\mathbb{C}^n$ and affine manifolds

**32M18** Automorphism groups of other complex spaces

**32M25** Complex vector fields, holomorphic foliations, $\mathbb{C}$-actions

**32M99** None of the above, but in this section

## 32Nxx Automorphic functions [See also 11Fxx, 20H10, 22E40, 30F35]

**32N05** General theory of automorphic functions of several complex variables

**32N10** Automorphic forms in several complex variables

**32N15** Automorphic functions in symmetric domains

**32N99** None of the above, but in this section

## 32Pxx Non-Archimedean analysis [Should also be assigned at least one other classification number from Section 32 describing the type of problem]

**32P05** Non-Archimedean analysis [Should also be assigned at least one other classification number from Section 32 describing the type of problem]

**32P99** None of the above, but in this section

## 32Qxx Complex manifolds

**32Q02** Special domains (Reinhardt, Hartogs, circular, tube, etc.) in $\mathbb{C}^n$ and complex manifolds

**32Q05** Negative curvature complex manifolds

**32Q10** Positive curvature complex manifolds

**32Q15** Kähler manifolds

**32Q20** Kähler-Einstein manifolds [See also 53Cxx]

**32Q25** Calabi-Yau theory (complex-analytic aspects) [See also 14J32]

**32Q26** Notions of stability for complex manifolds

**32Q28** Stein manifolds

**32Q30** Uniformization of complex manifolds

**32Q35** Complex manifolds as subdomains of Euclidean space

**32Q40** Embedding theorems for complex manifolds

**32Q45** Hyperbolic and Kobayashi hyperbolic manifolds

**32Q55** Topological aspects of complex manifolds

**32Q56** Oka principle and Oka manifolds

**32Q57** Classification theorems for complex manifolds

**32Q60** Almost complex manifolds

**32Q65** Pseudoholomorphic curves

**32Q99** None of the above, but in this section

## 32Sxx Complex singularities [See also 58Kxx]

**32S05** Local complex singularities [See also 14J17]

**32S10** Invariants of analytic local rings

**32S15** Equisingularity (topological and analytic) [See also 14E15]

**32S20** Global theory of complex singularities; cohomological properties [See also 14E15]

**32S22** Relations with arrangements of hyperplanes [See also 52C35]

**32S25** Complex surface and hypersurface singularities [See also 14J17]

**32S30** Deformations of complex singularities; vanishing cycles [See also 14B07]

**32S35** Mixed Hodge theory of singular varieties (complex-analytic aspects) [See also 14C30, 14D07]

**32S40** Monodromy; relations with differential equations and $D$-modules (complex-analytic aspects)

**32S45** Modifications; resolution of singularities (complex-analytic aspects) [See also 14E15]

**32S50** Topological aspects of complex singularities: Lefschetz theorems, topological classification, invariants

**32S55** Milnor fibration; relations with knot theory [See also 57K10, 57K45]

**32S60** Stratifications; constructible sheaves; intersection cohomology (complex-analytic aspects) [See also 58Kxx]

**32S65** Singularities of holomorphic vector fields and foliations

**32S70** Other operations on complex singularities

**32S99** None of the above, but in this section

## 32Txx Pseudoconvex domains

**32T05** Domains of holomorphy

**32T15** Strongly pseudoconvex domains

**32T20** Worm domains

**32T25** Finite-type domains

**32T27** Geometric and analytic invariants on weakly pseudoconvex boundaries

**32T35** Exhaustion functions

**32T40** Peak functions

**32T99** None of the above, but in this section

## 32Uxx Pluripotential theory

**32U05** Plurisubharmonic functions and generalizations [See also 31C10]

**32U10** Plurisubharmonic exhaustion functions

**32U15** General pluripotential theory

**32U20** Capacity theory and generalizations

**32U25** Lelong numbers

**32U30** Removable sets in pluripotential theory

**32U35** Plurisubharmonic extremal functions, pluricomplex Green functions

**32U40** Currents

**32U99** None of the above, but in this section

### 32Vxx CR manifolds

**32V05** CR structures, CR operators, and generalizations

**32V10** CR functions

**32V15** CR manifolds as boundaries of domains

**32V20** Analysis on CR manifolds

**32V25** Extension of functions and other analytic objects from CR manifolds

**32V30** Embeddings of CR manifolds

**32V35** Finite-type conditions on CR manifolds

**32V40** Real submanifolds in complex manifolds

**32V99** None of the above, but in this section

### 32Wxx Differential operators in several variables

**32W05** $\overline{\partial}$ and $\overline{\partial}$-Neumann operators

**32W10** $\overline{\partial}_b$ and $\overline{\partial}_b$-Neumann operators

**32W20** Complex Monge-Ampère operators

**32W25** Pseudodifferential operators in several complex variables

**32W30** Heat kernels in several complex variables

**32W50** Other partial differential equations of complex analysis in several variables

**32W99** None of the above, but in this section

# 33-XX Special functions (33-XX deals with the properties of functions as functions) {For orthogonal functions, see 42Cxx; for aspects of combinatorics, see 05Axx; for number-theoretic aspects, see 11-XX; for representation theory, see 22Exx}

**33-00** General reference works (handbooks, dictionaries, bibliographies, etc.) pertaining to special functions

**33-01** Introductory exposition (textbooks, tutorial papers, etc.) pertaining to special functions

**33-02** Research exposition (monographs, survey articles) pertaining to special functions

**33-03** History of special functions [Consider also classification numbers from Section 01]

**33-04** Software, source code, etc. for problems pertaining to special functions

**33-06** Proceedings, conferences, collections, etc. pertaining to special functions

**33-11** Research data for problems pertaining to special functions

## 33Bxx Elementary classical functions

**33B10** Exponential and trigonometric functions

**33B15** Gamma, beta and polygamma functions

**33B20** Incomplete beta and gamma functions (error functions, probability integral, Fresnel integrals)

**33B30** Higher logarithm functions

**33B99** None of the above, but in this section

## 33Cxx Hypergeometric functions

**33C05** Classical hypergeometric functions, $_2F_1$

**33C10** Bessel and Airy functions, cylinder functions, $_0F_1$

**33C15** Confluent hypergeometric functions, Whittaker functions, $_1F_1$

**33C20** Generalized hypergeometric series, $_pF_q$

**33C45** Orthogonal polynomials and functions of hypergeometric type (Jacobi, Laguerre, Hermite, Askey scheme, etc.) {For general orthogonal polynomials and functions, see also 42C05}

**33C47** Other special orthogonal polynomials and functions

**33C50** Orthogonal polynomials and functions in several variables expressible in terms of special functions in one variable

**33C52** Orthogonal polynomials and functions associated with root systems

**33C55** Spherical harmonics

**33C60** Hypergeometric integrals and functions defined by them ($E$, $G$, $H$ and $I$ functions)

**33C65** Appell, Horn and Lauricella functions

**33C67** Hypergeometric functions associated with root systems

**33C70** Other hypergeometric functions and integrals in several variables

**33C75** Elliptic integrals as hypergeometric functions

**33C80** Connections of hypergeometric functions with groups and algebras, and related topics

**33C90** Applications of hypergeometric functions

**33C99** None of the above, but in this section

## 33Dxx Basic hypergeometric functions

**33D05** $q$-gamma functions, $q$-beta functions and integrals

**33D15** Basic hypergeometric functions in one variable, $_r\phi_s$

**33D45** Basic orthogonal polynomials and functions (Askey-Wilson polynomials, etc.)

**33D50** Orthogonal polynomials and functions in several variables expressible in terms of basic hypergeometric functions in one variable

**33D52** Basic orthogonal polynomials and functions associated with root systems (Macdonald polynomials, etc.)

**33D60** Basic hypergeometric integrals and functions defined by them

**33D65** Bibasic functions and multiple bases

**33D67** Basic hypergeometric functions associated with root systems

**33D70** Other basic hypergeometric functions and integrals in several variables

**33D80** Connections of basic hypergeometric functions with quantum groups, Chevalley groups, $p$-adic groups, Hecke algebras, and related topics

**33D90** Applications of basic hypergeometric functions

**33D99** None of the above, but in this section

## 33Exx Other special functions

**33E05** Elliptic functions and integrals

**33E10** Lamé, Mathieu, and spheroidal wave functions

**33E12** Mittag-Leffler functions and generalizations

**33E15** Other wave functions

**33E17** Painlevé-type functions

**33E20** Other functions defined by series and integrals

**33E30** Other functions coming from differential, difference and integral equations

**33E50** Special functions in characteristic $p$ (gamma functions, etc.)

**33E99** None of the above, but in this section

## 33Fxx Computational aspects of special functions {For software etc., see 33-04}

**33F05** Numerical approximation and evaluation of special functions [See also 65D20]

**33F10** Symbolic computation of special functions (Gosper and Zeilberger algorithms, etc.) [See also 68W30]

**33F99** None of the above, but in this section

# 34-XX Ordinary differential equations

**34-00** General reference works (handbooks, dictionaries, bibliographies, etc.) pertaining to ordinary differential equations

**34-01** Introductory exposition (textbooks, tutorial papers, etc.) pertaining to ordinary differential equations

**34-02** Research exposition (monographs, survey articles) pertaining to ordinary differential equations

**34-03** History of ordinary differential equations [Consider also classification numbers from Section 01]

**34-04** Software, source code, etc. for problems pertaining to ordinary differential equations

**34-06** Proceedings, conferences, collections, etc. pertaining to ordinary differential equations

**34-11** Research data for problems pertaining to ordinary differential equations

## 34Axx General theory for ordinary differential equations

**34A05** Explicit solutions, first integrals of ordinary differential equations

**34A06** Generalized ordinary differential equations (measure-differential equations, set-valued differential equations, etc.)

**34A07** Fuzzy ordinary differential equations

**34A08** Fractional ordinary differential equations

**34A09** Implicit ordinary differential equations, differential-algebraic equations

**34A12** Initial value problems, existence, uniqueness, continuous dependence and continuation of solutions to ordinary differential equations

**34A25** Analytical theory of ordinary differential equations: series, transformations, transforms, operational calculus, etc. [See also 44-XX]

**34A26** Geometric methods in ordinary differential equations

**34A30** Linear ordinary differential equations and systems

**34A33** Ordinary lattice differential equations

**34A34** Nonlinear ordinary differential equations and systems

**34A35** Ordinary differential equations of infinite order

**34A36** Discontinuous ordinary differential equations

**34A37** Ordinary differential equations with impulses

**34A38** Hybrid systems of ordinary differential equations

**34A40** Differential inequalities involving functions of a single real variable [See also 26D20]

**34A45** Theoretical approximation of solutions to ordinary differential equations {For numerical analysis, see 65Lxx}

**34A55** Inverse problems involving ordinary differential equations

**34A60** Ordinary differential inclusions [See also 49J21, 49K21]

**34A99** None of the above, but in this section

## 34Bxx Boundary value problems for ordinary differential equations {For ordinary differential operators, see 34Lxx}

**34B05** Linear boundary value problems for ordinary differential equations

**34B07** Linear boundary value problems for ordinary differential equations with nonlinear dependence on the spectral parameter

**34B08** Parameter dependent boundary value problems for ordinary differential equations

**34B09** Boundary eigenvalue problems for ordinary differential equations

**34B10** Nonlocal and multipoint boundary value problems for ordinary differential equations

**34B15** Nonlinear boundary value problems for ordinary differential equations

**34B16** Singular nonlinear boundary value problems for ordinary differential equations

**34B18** Positive solutions to nonlinear boundary value problems for ordinary differential equations

**34B20** Weyl theory and its generalizations for ordinary differential equations

**34B24** Sturm-Liouville theory [See also 34Lxx]

**34B27** Green's functions for ordinary differential equations

**34B30** Special ordinary differential equations (Mathieu, Hill, Bessel, etc.)

**34B37** Boundary value problems with impulses for ordinary differential equations

**34B40** Boundary value problems on infinite intervals for ordinary differential equations

**34B45** Boundary value problems on graphs and networks for ordinary differential equations

**34B60** Applications of boundary value problems involving ordinary differential equations

**34B99** None of the above, but in this section

## 34Cxx Qualitative theory for ordinary differential equations [See also 37-XX]

**34C05** Topological structure of integral curves, singular points, limit cycles of ordinary differential equations

**34C07** Theory of limit cycles of polynomial and analytic vector fields (existence, uniqueness, bounds, Hilbert's 16th problem and ramifications) for ordinary differential equations

**34C08** Ordinary differential equations and connections with real algebraic geometry (fewnomials, desingularization, zeros of abelian integrals, etc.)

**34C10** Oscillation theory, zeros, disconjugacy and comparison theory for ordinary differential equations

**34C11** Growth and boundedness of solutions to ordinary differential equations

**34C12** Monotone systems involving ordinary differential equations

**34C14** Symmetries, invariants of ordinary differential equations [See also 37C79]

**34C15** Nonlinear oscillations and coupled oscillators for ordinary differential equations

**34C20** Transformation and reduction of ordinary differential equations and systems, normal forms

**34C23** Bifurcation theory for ordinary differential equations [See also 37Gxx]

**34C25** Periodic solutions to ordinary differential equations

**34C26** Relaxation oscillations for ordinary differential equations

**34C27** Almost and pseudo-almost periodic solutions to ordinary differential equations

**34C28** Complex behavior and chaotic systems of ordinary differential equations [See also 37Dxx]

**34C29** Averaging method for ordinary differential equations

**34C37** Homoclinic and heteroclinic solutions to ordinary differential equations

**34C40** Ordinary differential equations and systems on manifolds

**34C41** Equivalence and asymptotic equivalence of ordinary differential equations

**34C45** Invariant manifolds for ordinary differential equations

**34C46** Multifrequency systems of ordinary differential equations

**34C55** Hysteresis for ordinary differential equations

**34C60** Qualitative investigation and simulation of ordinary differential equation models

**34C99** None of the above, but in this section

## 34Dxx Stability theory for ordinary differential equations [See also 37C75, 93Dxx]

**34D05** Asymptotic properties of solutions to ordinary differential equations

**34D06** Synchronization of solutions to ordinary differential equations

**34D08** Characteristic and Lyapunov exponents of ordinary differential equations

**34D09** Dichotomy, trichotomy of solutions to ordinary differential equations

**34D10** Perturbations of ordinary differential equations

**34D15** Singular perturbations of ordinary differential equations

**34D20** Stability of solutions to ordinary differential equations

**34D23** Global stability of solutions to ordinary differential equations

**34D30** Structural stability and analogous concepts of solutions to ordinary differential equations [See also 37C20]

**34D35** Stability of manifolds of solutions to ordinary differential equations

**34D45** Attractors of solutions to ordinary differential equations [See also 37C70, 37D45]

**34D99** None of the above, but in this section

## 34Exx Asymptotic theory for ordinary differential equations

**34E05** Asymptotic expansions of solutions to ordinary differential equations

**34E10** Perturbations, asymptotics of solutions to ordinary differential equations

**34E13** Multiple scale methods for ordinary differential equations

**34E15** Singular perturbations for ordinary differential equations

**34E17** Canard solutions to ordinary differential equations

**34E18** Methods of nonstandard analysis for ordinary differential equations

**34E20** Singular perturbations, turning point theory, WKB methods for ordinary differential equations

**34E99** None of the above, but in this section

## 34Fxx Ordinary differential equations and systems with randomness [See also 34K50, 60H10, 93E03]

**34F05** Ordinary differential equations and systems with randomness

**34F10** Bifurcation of solutions to ordinary differential equations involving randomness

**34F15** Resonance phenomena for ordinary differential equations involving randomness

**34F99** None of the above, but in this section

## 34Gxx Differential equations in abstract spaces [See also 34K30, 47Jxx, 58D25]

**34G10** Linear differential equations in abstract spaces [See also 47D06, 47D09]

**34G20** Nonlinear differential equations in abstract spaces [See also 34K30, 47Jxx]

**34G25** Evolution inclusions

**34G99** None of the above, but in this section

# 34Hxx Control problems involving ordinary differential equations [See also 49J15, 49K15, 93C15]

**34H05** Control problems involving ordinary differential equations

**34H10** Chaos control for problems involving ordinary differential equations

**34H15** Stabilization of solutions to ordinary differential equations

**34H20** Bifurcation control of ordinary differential equations

**34H99** None of the above, but in this section

# 34Kxx Functional-differential equations (including equations with delayed, advanced or state-dependent argument)

**34K04** Symmetries, invariants of functional-differential equations [See also 37C79]

**34K05** General theory of functional-differential equations

**34K06** Linear functional-differential equations

**34K07** Theoretical approximation of solutions to functional-differential equations

**34K08** Spectral theory of functional-differential operators

**34K09** Functional-differential inclusions

**34K10** Boundary value problems for functional-differential equations

**34K11** Oscillation theory of functional-differential equations

**34K12** Growth, boundedness, comparison of solutions to functional-differential equations [See also 37C35]

**34K13** Periodic solutions to functional-differential equations [See also 37C27]

**34K14** Almost and pseudo-almost periodic solutions to functional-differential equations

**34K16** Heteroclinic and homoclinic orbits of functional-differential equations [See also 37C29]

**34K17** Transformation and reduction of functional-differential equations and systems, normal forms

**34K18** Bifurcation theory of functional-differential equations [See also 37Gxx]

**34K19** Invariant manifolds of functional-differential equations

**34K20** Stability theory of functional-differential equations [See also 37C75]

**34K21** Stationary solutions of functional-differential equations

**34K23** Complex (chaotic) behavior of solutions to functional-differential equations [See also 37D45]

**34K24** Synchronization of functional-differential equations

**34K25** Asymptotic theory of functional-differential equations

**34K26** Singular perturbations of functional-differential equations

**34K27** Perturbations of functional-differential equations

**34K29** Inverse problems for functional-differential equations

**34K30** Functional-differential equations in abstract spaces [See also 34Gxx, 35R09, 35R10, 47Jxx]

**34K31** Lattice functional-differential equations

**34K32** Implicit functional-differential equations

**34K33** Averaging for functional-differential equations

**34K34** Hybrid systems of functional-differential equations

**34K35** Control problems for functional-differential equations [See also 49J21, 49K21, 93C23]

**34K36** Fuzzy functional-differential equations

**34K37** Functional-differential equations with fractional derivatives

**34K38** Functional-differential inequalities

**34K39** Discontinuous functional-differential equations

**34K40** Neutral functional-differential equations

**34K41** Functional-differential equations in the complex domain

**34K42** Functional-differential equations on time scales or measure chains

**34K43** Functional-differential equations with state-dependent arguments

**34K45** Functional-differential equations with impulses

**34K50** Stochastic functional-differential equations [See also 34Fxx, 60Hxx]

**34K60** Qualitative investigation and simulation of models involving functional-differential equations

**34K99** None of the above, but in this section

## 34Lxx Ordinary differential operators [See also 47E05]

**34L05** General spectral theory of ordinary differential operators

**34L10** Eigenfunctions, eigenfunction expansions, completeness of eigenfunctions of ordinary differential operators

**34L15** Eigenvalues, estimation of eigenvalues, upper and lower bounds of ordinary differential operators

**34L16** Numerical approximation of eigenvalues and of other parts of the spectrum of ordinary differential operators

**34L20** Asymptotic distribution of eigenvalues, asymptotic theory of eigenfunctions for ordinary differential operators

**34L25** Scattering theory, inverse scattering involving ordinary differential operators

**34L30** Nonlinear ordinary differential operators

**34L40** Particular ordinary differential operators (Dirac, one-dimensional Schrödinger, etc.)

**34L99** None of the above, but in this section

### 34Mxx Ordinary differential equations in the complex domain [See also 30Dxx, 32G34]

**34M03** Linear ordinary differential equations and systems in the complex domain

**34M04** Nonlinear ordinary differential equations and systems in the complex domain

**34M05** Entire and meromorphic solutions to ordinary differential equations in the complex domain

**34M10** Oscillation, growth of solutions to ordinary differential equations in the complex domain

**34M15** Algebraic aspects (differential-algebraic, hypertranscendence, group-theoretical) of ordinary differential equations in the complex domain

**34M25** Formal solutions and transform techniques for ordinary differential equations in the complex domain

**34M30** Asymptotics and summation methods for ordinary differential equations in the complex domain

**34M35** Singularities, monodromy and local behavior of solutions to ordinary differential equations in the complex domain, normal forms

**34M40** Stokes phenomena and connection problems (linear and nonlinear) for ordinary differential equations in the complex domain

**34M45** Ordinary differential equations on complex manifolds

**34M46** Spectral theory for ordinary differential operators in the complex domain

**34M50** Inverse problems (Riemann-Hilbert, inverse differential Galois, etc.) for ordinary differential equations in the complex domain

**34M55** Painlevé and other special ordinary differential equations in the complex domain; classification, hierarchies

**34M56** Isomonodromic deformations for ordinary differential equations in the complex domain

**34M60** Singular perturbation problems for ordinary differential equations in the complex domain (complex WKB, turning points, steepest descent) [See also 34E20]

**34M65** Topological structure of trajectories of ordinary differential equations in the complex domain

**34M99** None of the above, but in this section

### 34Nxx Dynamic equations on time scales or measure chains {For real analysis on time scales, see 26E70}

**34N05** Dynamic equations on time scales or measure chains {For real analysis on time scales or measure chains, see 26E70}

**34N99** None of the above, but in this section

# 35-XX Partial differential equations

**35-00** General reference works (handbooks, dictionaries, bibliographies, etc.) pertaining to partial differential equations

**35-01** Introductory exposition (textbooks, tutorial papers, etc.) pertaining to partial differential equations

**35-02** Research exposition (monographs, survey articles) pertaining to partial differential equations

**35-03** History of partial differential equations [Consider also classification numbers from Section 01]

**35-04** Software, source code, etc. for problems pertaining to partial differential equations

**35-06** Proceedings, conferences, collections, etc. pertaining to partial differential equations

**35-11** Research data for problems pertaining to partial differential equations

## 35Axx General topics in partial differential equations

**35A01** Existence problems for PDEs: global existence, local existence, non-existence

**35A02** Uniqueness problems for PDEs: global uniqueness, local uniqueness, non-uniqueness

**35A08** Fundamental solutions to PDEs

**35A09** Classical solutions to PDEs

**35A10** Cauchy-Kovalevskaya theorems

**35A15** Variational methods applied to PDEs

**35A16** Topological and monotonicity methods applied to PDEs

**35A17** Parametrices in context of PDEs

**35A18** Wave front sets in context of PDEs

**35A20** Analyticity in context of PDEs

**35A21** Singularity in context of PDEs

**35A22** Transform methods (e.g., integral transforms) applied to PDEs

**35A23** Inequalities applied to PDEs involving derivatives, differential and integral operators, or integrals

**35A24** Methods of ordinary differential equations applied to PDEs

**35A25** Other special methods applied to PDEs

**35A27** Microlocal methods and methods of sheaf theory and homological algebra applied to PDEs [See also 32C38, 58J15]

**35A30** Geometric theory, characteristics, transformations in context of PDEs [See also 58J70, 58J72]

**35A35** Theoretical approximation in context of PDEs {For numerical analysis, see 65Mxx, 65Nxx}

**35A99** None of the above, but in this section

## 35Bxx Qualitative properties of solutions to partial differential equations

**35B05** Oscillation, zeros of solutions, mean value theorems, etc. in context of PDEs

**35B06** Symmetries, invariants, etc. in context of PDEs

**35B07** Axially symmetric solutions to PDEs

**35B08** Entire solutions to PDEs

**35B09** Positive solutions to PDEs

**35B10** Periodic solutions to PDEs

**35B15** Almost and pseudo-almost periodic solutions to PDEs

**35B20** Perturbations in context of PDEs

**35B25** Singular perturbations in context of PDEs

**35B27** Homogenization in context of PDEs; PDEs in media with periodic structure [See also 74Q05, 74Q10, 76M50, 78M40, 80M40]

**35B30** Dependence of solutions to PDEs on initial and/or boundary data and/or on parameters of PDEs [See also 37Cxx]

**35B32** Bifurcations in context of PDEs [See also 34C23, 34F10, 34H20, 37F46, 37Gxx, 37H20, 37J20, 37K50, 37L10, 37M20, 47J15, 58E05, 58E07, 58J55]

**35B33** Critical exponents in context of PDEs

**35B34** Resonance in context of PDEs [See also 34F15, 70J40, 70K28, 70K30, 81U24]

**35B35** Stability in context of PDEs [See also 34Dxx, 37B25, 37C20, 37C75, 37F15, 37J25, 37K45, 37L15, 49K40, 58K25, 93Dxx]

**35B36** Pattern formations in context of PDEs [See also 92C15]

**35B38** Critical points of functionals in context of PDEs (e.g., energy functionals) [See also 57R70, 58K05, 58E05]

**35B40** Asymptotic behavior of solutions to PDEs

**35B41** Attractors [See also 34D45, 37B35, 37C70, 37D45, 37G35, 37L30, 37M22]

**35B42** Inertial manifolds [See also 37L25]

**35B44** Blow-up in context of PDEs

**35B45** A priori estimates in context of PDEs

**35B50** Maximum principles in context of PDEs

**35B51** Comparison principles in context of PDEs

**35B53** Liouville theorems and Phragmén-Lindelöf theorems in context of PDEs

**35B60** Continuation and prolongation of solutions to PDEs [See also 58A15, 58A17, 58Hxx]

**35B65** Smoothness and regularity of solutions to PDEs

**35B99** None of the above, but in this section

## 35Cxx Representations of solutions to partial differential equations

**35C05** Solutions to PDEs in closed form

**35C06** Self-similar solutions to PDEs

**35C07** Traveling wave solutions

**35C08** Soliton solutions [See also 37K40]

**35C09** Trigonometric solutions to PDEs

**35C10** Series solutions to PDEs

**35C11** Polynomial solutions to PDEs

**35C15** Integral representations of solutions to PDEs

**35C20** Asymptotic expansions of solutions to PDEs

**35C99** None of the above, but in this section

## 35Dxx Generalized solutions to partial differential equations

**35D30** Weak solutions to PDEs

**35D35** Strong solutions to PDEs

**35D40** Viscosity solutions to PDEs

**35D99** None of the above, but in this section

## 35Exx Partial differential equations and systems of partial differential equations with constant coefficients [See also 35N05]

**35E05** Fundamental solutions to PDEs and systems of PDEs with constant coefficients

**35E10** Convexity properties of solutions to PDEs with constant coefficients

**35E15** Initial value problems for PDEs and systems of PDEs with constant coefficients

**35E20** General theory of PDEs and systems of PDEs with constant coefficients

**35E99** None of the above, but in this section

## 35Fxx General first-order partial differential equations and systems of first-order partial differential equations

**35F05** Linear first-order PDEs

**35F10** Initial value problems for linear first-order PDEs

**35F15** Boundary value problems for linear first-order PDEs

**35F16** Initial-boundary value problems for linear first-order PDEs

**35F20** Nonlinear first-order PDEs

**35F21** Hamilton-Jacobi equations {For calculus of variations and optimal control, see 49Lxx; for mechanics of particles and systems, see 70H20}

**35F25** Initial value problems for nonlinear first-order PDEs

**35F30** Boundary value problems for nonlinear first-order PDEs

**35F31** Initial-boundary value problems for nonlinear first-order PDEs

**35F35** Systems of linear first-order PDEs

**35F40** Initial value problems for systems of linear first-order PDEs

**35F45** Boundary value problems for systems of linear first-order PDEs

**35F46** Initial-boundary value problems for systems of linear first-order PDEs

**35F50** Systems of nonlinear first-order PDEs

**35F55** Initial value problems for systems of nonlinear first-order PDEs

**35F60** Boundary value problems for systems of nonlinear first-order PDEs

**35F61** Initial-boundary value problems for systems of nonlinear first-order PDEs

**35F99** None of the above, but in this section

## 35Gxx General higher-order partial differential equations and systems of higher-order partial differential equations

**35G05** Linear higher-order PDEs

**35G10** Initial value problems for linear higher-order PDEs

**35G15** Boundary value problems for linear higher-order PDEs

**35G16** Initial-boundary value problems for linear higher-order PDEs

**35G20** Nonlinear higher-order PDEs

**35G25** Initial value problems for nonlinear higher-order PDEs

**35G30** Boundary value problems for nonlinear higher-order PDEs

**35G31** Initial-boundary value problems for nonlinear higher-order PDEs

**35G35** Systems of linear higher-order PDEs

**35G40** Initial value problems for systems of linear higher-order PDEs

**35G45** Boundary value problems for systems of linear higher-order PDEs

**35G46** Initial-boundary value problems for systems of linear higher-order PDEs

**35G50** Systems of nonlinear higher-order PDEs

**35G55** Initial value problems for systems of nonlinear higher-order PDEs

**35G60** Boundary value problems for systems of nonlinear higher-order PDEs

**35G61** Initial-boundary value problems for systems of nonlinear higher-order PDEs

**35G99** None of the above, but in this section

## 35Hxx Close-to-elliptic equations

**35H10** Hypoelliptic equations

**35H20** Subelliptic equations

**35H30** Quasielliptic equations

**35H99** None of the above, but in this section

## 35Jxx Elliptic equations and elliptic systems {For global analysis, analysis on manifolds, see 58J10, 58J20}

**35J05** Laplace operator, Helmholtz equation (reduced wave equation), Poisson equation [See also 31Axx, 31Bxx]

**35J08** Green's functions for elliptic equations

**35J10** Schrödinger operator, Schrödinger equation {For ordinary differential equations, see 34L40; for operator theory, see 47D08; for quantum theory, see 81Q05; for statistical mechanics, see 82B44}

**35J15** Second-order elliptic equations

**35J20** Variational methods for second-order elliptic equations

**35J25** Boundary value problems for second-order elliptic equations

**35J30** Higher-order elliptic equations [See also 31A30, 31B30]

**35J35** Variational methods for higher-order elliptic equations

**35J40** Boundary value problems for higher-order elliptic equations

**35J46** First-order elliptic systems

**35J47** Second-order elliptic systems

**35J48** Higher-order elliptic systems

**35J50** Variational methods for elliptic systems

**35J56** Boundary value problems for first-order elliptic systems

**35J57** Boundary value problems for second-order elliptic systems

**35J58** Boundary value problems for higher-order elliptic systems

**35J60** Nonlinear elliptic equations

**35J61** Semilinear elliptic equations

**35J62** Quasilinear elliptic equations

**35J65** Nonlinear boundary value problems for linear elliptic equations

**35J66** Nonlinear boundary value problems for nonlinear elliptic equations

**35J67** Boundary values of solutions to elliptic equations and elliptic systems

**35J70** Degenerate elliptic equations

**35J75** Singular elliptic equations

**35J86** Unilateral problems for linear elliptic equations and variational inequalities with linear elliptic operators [See also 35R35, 49J40]

**35J87** Unilateral problems for nonlinear elliptic equations and variational inequalities with nonlinear elliptic operators [See also 35R35, 49J40]

**35J88** Unilateral problems for elliptic systems and systems of variational inequalities with elliptic operators [See also 35R35, 49J40]

**35J91** Semilinear elliptic equations with Laplacian, bi-Laplacian or poly-Laplacian

**35J92** Quasilinear elliptic equations with $p$-Laplacian

**35J93** Quasilinear elliptic equations with mean curvature operator

**35J94** Elliptic equations with infinity-Laplacian

**35J96** Monge-Ampère equations {For complex Monge-Ampère operators, see 32W20; for parabolic Monge-Ampère equations, see 35K96}

**35J99** None of the above, but in this section

## 35Kxx Parabolic equations and parabolic systems {For global analysis, analysis on manifolds, see 58J35}

**35K05** Heat equation

**35K08** Heat kernel

**35K10** Second-order parabolic equations

**35K15** Initial value problems for second-order parabolic equations

**35K20** Initial-boundary value problems for second-order parabolic equations

**35K25** Higher-order parabolic equations

**35K30** Initial value problems for higher-order parabolic equations

**35K35** Initial-boundary value problems for higher-order parabolic equations

**35K40** Second-order parabolic systems

**35K41** Higher-order parabolic systems

**35K45** Initial value problems for second-order parabolic systems

**35K46** Initial value problems for higher-order parabolic systems

**35K51** Initial-boundary value problems for second-order parabolic systems

**35K52** Initial-boundary value problems for higher-order parabolic systems

**35K55** Nonlinear parabolic equations

**35K57** Reaction-diffusion equations {For diffusion processes and reaction effects, see 47D07, 58J65, 60J60, 60J70, 74N25, 76R50, 76V05, 80A23, 82B24, 82C24, 92E20}

**35K58** Semilinear parabolic equations

**35K59** Quasilinear parabolic equations

**35K60** Nonlinear initial, boundary and initial-boundary value problems for linear parabolic equations

**35K61** Nonlinear initial, boundary and initial-boundary value problems for nonlinear parabolic equations

**35K65** Degenerate parabolic equations

**35K67** Singular parabolic equations

**35K70** Ultraparabolic equations, pseudoparabolic equations, etc.

**35K85** Unilateral problems for linear parabolic equations and variational inequalities with linear parabolic operators [See also 35R35, 49J40]

**35K86** Unilateral problems for nonlinear parabolic equations and variational inequalities with nonlinear parabolic operators [See also 35R35, 49J40]

**35K87** Unilateral problems for parabolic systems and systems of variational inequalities with parabolic operators [See also 35R35, 49J40]

**35K90** Abstract parabolic equations

**35K91** Semilinear parabolic equations with Laplacian, bi-Laplacian or poly-Laplacian

**35K92** Quasilinear parabolic equations with $p$-Laplacian

**35K93** Quasilinear parabolic equations with mean curvature operator

**35K96** Parabolic Monge-Ampère equations

**35K99** None of the above, but in this section

## 35Lxx Hyperbolic equations and hyperbolic systems {For global analysis, see 58J45}

**35L02** First-order hyperbolic equations

**35L03** Initial value problems for first-order hyperbolic equations

**35L04** Initial-boundary value problems for first-order hyperbolic equations

**35L05** Wave equation

**35L10** Second-order hyperbolic equations

**35L15** Initial value problems for second-order hyperbolic equations

**35L20** Initial-boundary value problems for second-order hyperbolic equations

**35L25** Higher-order hyperbolic equations

**35L30** Initial value problems for higher-order hyperbolic equations

**35L35** Initial-boundary value problems for higher-order hyperbolic equations

**35L40** First-order hyperbolic systems

**35L45** Initial value problems for first-order hyperbolic systems

**35L50** Initial-boundary value problems for first-order hyperbolic systems

**35L51** Second-order hyperbolic systems

**35L52** Initial value problems for second-order hyperbolic systems

**35L53** Initial-boundary value problems for second-order hyperbolic systems

**35L55** Higher-order hyperbolic systems

**35L56** Initial value problems for higher-order hyperbolic systems

**35L57** Initial-boundary value problems for higher-order hyperbolic systems

**35L60** First-order nonlinear hyperbolic equations

**35L65** Hyperbolic conservation laws

**35L67** Shocks and singularities for hyperbolic equations [See also 58Kxx, 74J40, 76L05]

**35L70** Second-order nonlinear hyperbolic equations

**35L71** Second-order semilinear hyperbolic equations

**35L72** Second-order quasilinear hyperbolic equations

**35L75** Higher-order nonlinear hyperbolic equations

**35L76** Higher-order semilinear hyperbolic equations

**35L77** Higher-order quasilinear hyperbolic equations

**35L80** Degenerate hyperbolic equations

**35L81** Singular hyperbolic equations

**35L82** Pseudohyperbolic equations

**35L85** Unilateral problems for linear hyperbolic equations and variational inequalities with linear hyperbolic operators [See also 35R35, 49J40]

**35L86** Unilateral problems for nonlinear hyperbolic equations and variational inequalities with nonlinear hyperbolic operators [See also 35R35, 49J40]

**35L87** Unilateral problems for hyperbolic systems and systems of variational inequalities with hyperbolic operators [See also 35R35, 49J40]

**35L90** Abstract hyperbolic equations

**35L99** None of the above, but in this section

## 35Mxx Partial differential equations of mixed type and mixed-type systems of partial differential equations

**35M10** PDEs of mixed type

**35M11** Initial value problems for PDEs of mixed type

**35M12** Boundary value problems for PDEs of mixed type

**35M13** Initial-boundary value problems for PDEs of mixed type

**35M30** Mixed-type systems of PDEs

**35M31** Initial value problems for mixed-type systems of PDEs

**35M32** Boundary value problems for mixed-type systems of PDEs

**35M33** Initial-boundary value problems for mixed-type systems of PDEs

**35M85** Unilateral problems for linear PDEs of mixed type and variational inequalities with partial differential operators of mixed type [See also 35R35, 49J40]

**35M86** Unilateral problems for nonlinear PDEs of mixed type and variational inequalities with nonlinear partial differential operators of mixed type [See also 35R35, 49J40]

**35M87** Unilateral problems for mixed-type systems of PDEs and systems of variational inequalities with partial differential operators of mixed type [See also 35R35, 49J40]

**35M99** None of the above, but in this section

## 35Nxx Overdetermined problems for partial differential equations and systems of partial differential equations {For global analysis, see 58Hxx, 58J10, 58J15}

**35N05** Overdetermined systems of PDEs with constant coefficients

**35N10** Overdetermined systems of PDEs with variable coefficients

**35N15** $\overline{\partial}$-Neumann problems and formal complexes in context of PDEs [See also 32W05, 32W10, 58J10]

**35N20** Overdetermined initial value problems for PDEs and systems of PDEs

**35N25** Overdetermined boundary value problems for PDEs and systems of PDEs

**35N30** Overdetermined initial-boundary value problems for PDEs and systems of PDEs

**35N99** None of the above, but in this section

## 35Pxx Spectral theory and eigenvalue problems for partial differential equations {For operator theory, see 47Axx, 47Bxx, 47F05}

**35P05** General topics in linear spectral theory for PDEs

**35P10** Completeness of eigenfunctions and eigenfunction expansions in context of PDEs

**35P15** Estimates of eigenvalues in context of PDEs

**35P20** Asymptotic distributions of eigenvalues in context of PDEs

**35P25** Scattering theory for PDEs [See also 47A40]

**35P30** Nonlinear eigenvalue problems and nonlinear spectral theory for PDEs

**35P99** None of the above, but in this section

## 35Qxx Partial differential equations of mathematical physics and other areas of application [See also 35J05, 35J10, 35K05, 35L05]

**35Q05** Euler-Poisson-Darboux equations

**35Q07** Fuchsian PDEs

**35Q15** Riemann-Hilbert problems in context of PDEs [See also 30E25, 31A25, 31B20]

**35Q20** Boltzmann equations {For fluid mechanics, see 76P05; for statistical mechanics, see 82B40, 82C40, 82D05}

**35Q30** Navier-Stokes equations {For fluid mechanics, see 76D05, 76D07, 76N10}

**35Q31** Euler equations {For fluid mechanics, see 76D05, 76D07, 76N10}

**35Q35** PDEs in connection with fluid mechanics

**35Q40** PDEs in connection with quantum mechanics

**35Q41** Time-dependent Schrödinger equations and Dirac equations {For quantum theory, see 81Q05; for relativity and gravitational theory, see 83A05, 83C10}

**35Q49** Transport equations {For calculus of variations and optimal control, see 49Q22; for fluid mechanics, see 76F25; for statistical mechanics, see 82C70, 82D75; for operations research, see 90B06; for mathematical programming, see 90C08}

**35Q51** Soliton equations {For dynamical systems and ergodic theory, see 37K40}

**35Q53** KdV equations (Korteweg-de Vries equations) {For dynamical systems and ergodic theory, see 37K10}

**35Q55** NLS equations (nonlinear Schrödinger equations) {For dynamical systems and ergodic theory, see 37K10}

**35Q56** Ginzburg-Landau equations {For optics and electromagnetic theory, see 78A25}

**35Q60** PDEs in connection with optics and electromagnetic theory

**35Q61** Maxwell equations {For optics and electromagnetic theory, see 78A25; for relativity and gravitational theory, see 83C22}

**35Q62** PDEs in connection with statistics

**35Q68** PDEs in connection with computer science

**35Q70** PDEs in connection with mechanics of particles and systems of particles

**35Q74** PDEs in connection with mechanics of deformable solids

**35Q75** PDEs in connection with relativity and gravitational theory

**35Q76** Einstein equations {For several complex variables and analytic spaces, see 32Q40; for differential geometry, see 53C07; for relativity and gravitational theory, see 83C05, 83C25, 83D05}

**35Q79** PDEs in connection with classical thermodynamics and heat transfer

**35Q81** PDEs in connection with semiconductor devices {For statistical mechanics, see 82D37}

**35Q82** PDEs in connection with statistical mechanics

**35Q83** Vlasov equations {For statistical mechanics, see 82C70, 82D75}

**35Q84** Fokker-Planck equations {For fluid mechanics, see 76X05, 76W05; for statistical mechanics, see 82C31}

**35Q85** PDEs in connection with astronomy and astrophysics

**35Q86** PDEs in connection with geophysics

**35Q89** PDEs in connection with mean field game theory {For calculus of variations and optimal control, see 49N80; for game theory, see 91A16}

**35Q90** PDEs in connection with mathematical programming

**35Q91** PDEs in connection with game theory, economics, social and behavioral sciences

**35Q92** PDEs in connection with biology, chemistry and other natural sciences

**35Q93** PDEs in connection with control and optimization

**35Q94** PDEs in connection with information and communication

**35Q99** None of the above, but in this section

## 35Rxx Miscellaneous topics in partial differential equations {For equations on manifolds, see 32Wxx, 58Jxx; for manifolds of solutions, see 58Bxx; for stochastic PDEs, see 60H15}

**35R01** PDEs on manifolds [See also 32Wxx, 53Cxx, 58Jxx]

**35R02** PDEs on graphs and networks (ramified or polygonal spaces)

**35R03** PDEs on Heisenberg groups, Lie groups, Carnot groups, etc.

**35R05** PDEs with low regular coefficients and/or low regular data

**35R06** PDEs with measure

**35R07** PDEs on time scales

**35R09** Integro-partial differential equations [See also 34K30, 45K05]

**35R10** Partial functional-differential equations

**35R11** Fractional partial differential equations

**35R12** Impulsive partial differential equations

**35R13** Fuzzy partial differential equations

**35R15** PDEs on infinite-dimensional (e.g., function) spaces (= PDEs in infinitely many variables) [See also 46Gxx, 58D25]

**35R20** Operator partial differential equations (= PDEs on finite-dimensional spaces for abstract space valued functions) [See also 34Gxx, 47A50, 47D03, 47D06, 47D09, 47H20, 47Jxx]

**35R25** Ill-posed problems for PDEs

**35R30** Inverse problems for PDEs

**35R35** Free boundary problems for PDEs

**35R37** Moving boundary problems for PDEs

**35R45** Partial differential inequalities and systems of partial differential inequalities

**35R50** PDEs of infinite order

**35R60** PDEs with randomness, stochastic partial differential equations [See also 60H15]

**35R70** PDEs with multivalued right-hand sides

**35R99** None of the above, but in this section

## 35Sxx Pseudodifferential operators and other generalizations of partial differential operators {For operator theory, see 47G30, 58J40}

**35S05** Pseudodifferential operators as generalizations of partial differential operators [See also 32W25, 47G30, 47L80, 58J40]

**35S10** Initial value problems for PDEs with pseudodifferential operators

**35S15** Boundary value problems for PDEs with pseudodifferential operators

**35S16** Initial-boundary value problems for PDEs with pseudodifferential operators

**35S30** Fourier integral operators applied to PDEs [See also 43A32, 58J40]

**35S35** Topological aspects for pseudodifferential operators in context of PDEs: intersection cohomology, stratified sets, etc. [See also 32C38, 32S40, 32S60, 58J15]

**35S50** Paradifferential operators as generalizations of partial differential operators in context of PDEs

**35S99** None of the above, but in this section

# 37-XX Dynamical systems and ergodic theory [See also 26A18, 28Dxx, 34Cxx, 34Dxx, 35Bxx, 46Lxx, 58Jxx, 70-XX]

**37-00** General reference works (handbooks, dictionaries, bibliographies, etc.) pertaining to dynamical systems and ergodic theory

**37-01** Introductory exposition (textbooks, tutorial papers, etc.) pertaining to dynamical systems and ergodic theory

**37-02** Research exposition (monographs, survey articles) pertaining to dynamical systems and ergodic theory

**37-03** History of dynamical systems and ergodic theory [Consider also classification numbers from Section 01]

**37-04** Software, source code, etc. for problems pertaining to dynamical systems and ergodic theory

**37-06** Proceedings, conferences, collections, etc. pertaining to dynamical systems and ergodic theory

**37-11** Research data for problems pertaining to dynamical systems and ergodic theory

## 37Axx Ergodic theory [See also 28Dxx]

**37A05** Dynamical aspects of measure-preserving transformations

**37A10** Dynamical systems involving one-parameter continuous families of measure-preserving transformations

**37A15** General groups of measure-preserving transformations and dynamical systems [See mainly 22Fxx]

**37A17** Homogeneous flows [See also 22Fxx]

**37A20** Algebraic ergodic theory, cocycles, orbit equivalence, ergodic equivalence relations

**37A25** Ergodicity, mixing, rates of mixing

**37A30** Ergodic theorems, spectral theory, Markov operators {For operator ergodic theory, see mainly 47A35}

**37A35** Entropy and other invariants, isomorphism, classification in ergodic theory

**37A40** Nonsingular (and infinite-measure preserving) transformations

**37A44** Relations between ergodic theory and number theory [See also 11Kxx]

**37A46** Relations between ergodic theory and harmonic analysis

**37A50** Dynamical systems and their relations with probability theory and stochastic processes [See also 60Fxx, 60G10]

**37A55** Dynamical systems and the theory of $C^*$-algebras [See mainly 46L55]

**37A60** Dynamical aspects of statistical mechanics [See also 82Cxx]

**37A99** None of the above, but in this section

## 37Bxx Topological dynamics

**37B02** Dynamics in general topological spaces

**37B05** Dynamical systems involving transformations and group actions with special properties (minimality, distality, proximality, expansivity, etc.)

**37B10** Symbolic dynamics

**37B15** Dynamical aspects of cellular automata {For computational aspects, see 68Q80}

**37B20** Notions of recurrence and recurrent behavior in topological dynamical systems

**37B25** Stability of topological dynamical systems

**37B30** Index theory for dynamical systems, Morse-Conley indices

**37B35** Gradient-like behavior; isolated (locally maximal) invariant sets; attractors, repellers for topological dynamical systems

**37B40** Topological entropy

**37B45** Continua theory in dynamics

**37B51** Multidimensional shifts of finite type

**37B52** Tiling dynamics

**37B55** Topological dynamics of nonautonomous systems

**37B65** Approximate trajectories, pseudotrajectories, shadowing and related notions for topological dynamical systems

**37B99** None of the above, but in this section

## 37Cxx Smooth dynamical systems: general theory [See also 34Cxx, 34Dxx]

**37C05** Dynamical systems involving smooth mappings and diffeomorphisms

**37C10** Dynamics induced by flows and semiflows

**37C15** Topological and differentiable equivalence, conjugacy, moduli, classification of dynamical systems

**37C20** Generic properties, structural stability of dynamical systems

**37C25** Fixed points and periodic points of dynamical systems; fixed-point index theory; local dynamics

**37C27** Periodic orbits of vector fields and flows

**37C29** Homoclinic and heteroclinic orbits for dynamical systems

**37C30** Functional analytic techniques in dynamical systems; zeta functions, (Ruelle-Frobenius) transfer operators, etc.

**37C35** Orbit growth in dynamical systems

**37C40** Smooth ergodic theory, invariant measures for smooth dynamical systems [See also 37Dxx]

**37C45** Dimension theory of smooth dynamical systems

**37C50** Approximate trajectories (pseudotrajectories, shadowing, etc.) in smooth dynamics

**37C55** Periodic and quasi-periodic flows and diffeomorphisms

**37C60** Nonautonomous smooth dynamical systems

**37C65** Monotone flows as dynamical systems

**37C70** Attractors and repellers of smooth dynamical systems and their topological structure

**37C75** Stability theory for smooth dynamical systems

**37C79** Symmetries and invariants of dynamical systems [See also 34C14, 34K04]

**37C81** Equivariant dynamical systems

**37C83** Dynamical systems with singularities (billiards, etc.)

**37C85** Dynamics induced by group actions other than $\mathbb{Z}$ and $\mathbb{R}$, and $\mathbb{C}$ [See mainly 22Fxx, and also 32M25, 57R30, 57Sxx]

**37C86** Foliations generated by dynamical systems

**37C99** None of the above, but in this section

## 37Dxx Dynamical systems with hyperbolic behavior

**37D05** Dynamical systems with hyperbolic orbits and sets

**37D10** Invariant manifold theory for dynamical systems

**37D15** Morse-Smale systems

**37D20** Uniformly hyperbolic systems (expanding, Anosov, Axiom A, etc.)

**37D25** Nonuniformly hyperbolic systems (Lyapunov exponents, Pesin theory, etc.)

**37D30** Partially hyperbolic systems and dominated splittings

**37D35** Thermodynamic formalism, variational principles, equilibrium states for dynamical systems

**37D40** Dynamical systems of geometric origin and hyperbolicity (geodesic and horocycle flows, etc.)

**37D45** Strange attractors, chaotic dynamics of systems with hyperbolic behavior

**37D99** None of the above, but in this section

## 37Exx Low-dimensional dynamical systems

**37E05** Dynamical systems involving maps of the interval

**37E10** Dynamical systems involving maps of the circle

**37E15** Combinatorial dynamics (types of periodic orbits)

**37E20** Universality and renormalization of dynamical systems [See also 37F25]

**37E25** Dynamical systems involving maps of trees and graphs

**37E30** Dynamical systems involving homeomorphisms and diffeomorphisms of planes and surfaces

**37E35** Flows on surfaces

**37E40** Dynamical aspects of twist maps

**37E45** Rotation numbers and vectors

**37E99** None of the above, but in this section

## 37Fxx Dynamical systems over complex numbers [See also 30D05, 32H50]

**37F05** Dynamical systems involving relations and correspondences in one complex variable

**37F10** Dynamics of complex polynomials, rational maps, entire and meromorphic functions; Fatou and Julia sets [See also 32A10, 32A20, 32H02, 32H04]

**37F12** Critical orbits for holomorphic dynamical systems

**37F15** Expanding holomorphic maps; hyperbolicity; structural stability of holomorphic dynamical systems

**37F20** Combinatorics and topology in relation with holomorphic dynamical systems

**37F25** Renormalization of holomorphic dynamical systems

**37F31** Quasiconformal methods in holomorphic dynamics; quasiconformal dynamics

**37F32** Fuchsian and Kleinian groups as dynamical systems

**37F34** Teichmüller theory; moduli spaces of holomorphic dynamical systems

**37F35** Conformal densities and Hausdorff dimension for holomorphic dynamical systems

**37F40** Geometric limits in holomorphic dynamics

**37F44** Holomorphic families of dynamical systems; holomorphic motions; semigroups of holomorphic maps

**37F46** Bifurcations; parameter spaces in holomorphic dynamics; the Mandelbrot and Multibrot sets

**37F50** Small divisors, rotation domains and linearization in holomorphic dynamics

**37F75** Dynamical aspects of holomorphic foliations and vector fields [See also 32M25, 32S65, 34Mxx]

**37F80** Higher-dimensional holomorphic and meromorphic dynamics

**37F99** None of the above, but in this section

## 37Gxx Local and nonlocal bifurcation theory for dynamical systems [See also 34C23, 34K18]

**37G05** Normal forms for dynamical systems

**37G10** Bifurcations of singular points in dynamical systems

**37G15** Bifurcations of limit cycles and periodic orbits in dynamical systems

**37G20** Hyperbolic singular points with homoclinic trajectories in dynamical systems

**37G25** Bifurcations connected with nontransversal intersection in dynamical systems

**37G30** Infinite nonwandering sets arising in bifurcations of dynamical systems

**37G35** Dynamical aspects of attractors and their bifurcations

**37G40** Dynamical aspects of symmetries, equivariant bifurcation theory

**37G99** None of the above, but in this section

## 37Hxx Random dynamical systems [See also 15B52, 34Fxx, 47B80, 70L05, 82C05, 93Exx]

**37H05** General theory of random and stochastic dynamical systems

**37H10** Generation, random and stochastic difference and differential equations [See also 34F05, 34K50, 60H10, 60H15]

**37H12** Random iteration

**37H15** Random dynamical systems aspects of multiplicative ergodic theory, Lyapunov exponents [See also 34Fxx, 37Axx, 37Cxx, 37Dxx]

**37H20** Bifurcation theory for random and stochastic dynamical systems [See also 37Gxx]

**37H30** Stability theory for random and stochastic dynamical systems

**37H99** None of the above, but in this section

## 37Jxx Dynamical aspects of finite-dimensional Hamiltonian and Lagrangian systems [See also 53Dxx, 70Fxx, 70Hxx]

**37J06** General theory of finite-dimensional Hamiltonian and Lagrangian systems, Hamiltonian and Lagrangian structures, symmetries, invariants

**37J11** Symplectic and canonical mappings

**37J12** Fixed points and periodic points of finite-dimensional Hamiltonian and Lagrangian systems

**37J20** Bifurcation problems for finite-dimensional Hamiltonian and Lagrangian systems

**37J25** Stability problems for finite-dimensional Hamiltonian and Lagrangian systems

**37J30** Obstructions to integrability for finite-dimensional Hamiltonian and Lagrangian systems (nonintegrability criteria)

**37J35** Completely integrable finite-dimensional Hamiltonian systems, integration methods, integrability tests

**37J37** Relations of finite-dimensional Hamiltonian and Lagrangian systems with Lie algebras and other algebraic structures

**37J38** Relations of finite-dimensional Hamiltonian and Lagrangian systems with algebraic geometry, complex analysis, special functions

**37J39** Relations of finite-dimensional Hamiltonian and Lagrangian systems with topology, geometry and differential geometry (symplectic geometry, Poisson geometry, etc.) [See also 53D20]

**37J40** Perturbations of finite-dimensional Hamiltonian systems, normal forms, small divisors, KAM theory, Arnol'd diffusion

**37J46** Periodic, homoclinic and heteroclinic orbits of finite-dimensional Hamiltonian systems

**37J51** Action-minimizing orbits and measures for finite-dimensional Hamiltonian and Lagrangian systems; variational principles; degree-theoretic methods

**37J55** Contact systems [See also 53D10]

**37J60** Nonholonomic dynamical systems [See also 70F25]

**37J65** Nonautonomous Hamiltonian dynamical systems (Painlevé equations, etc.) [See also 34M55]

**37J70** Completely integrable discrete dynamical systems

**37J99** None of the above, but in this section

## 37Kxx Dynamical system aspects of infinite-dimensional Hamiltonian and Lagrangian systems [See also 35Axx, 35Qxx]

**37K06** General theory of infinite-dimensional Hamiltonian and Lagrangian systems, Hamiltonian and Lagrangian structures, symmetries, conservation laws

**37K10** Completely integrable infinite-dimensional Hamiltonian and Lagrangian systems, integration methods, integrability tests, integrable hierarchies (KdV, KP, Toda, etc.)

**37K15** Inverse spectral and scattering methods for infinite-dimensional Hamiltonian and Lagrangian systems

**37K20** Relations of infinite-dimensional Hamiltonian and Lagrangian dynamical systems with algebraic geometry, complex analysis, and special functions [See also 14H70]

**37K25** Relations of infinite-dimensional Hamiltonian and Lagrangian dynamical systems with topology, geometry and differential geometry

**37K30** Relations of infinite-dimensional Hamiltonian and Lagrangian dynamical systems with infinite-dimensional Lie algebras and other algebraic structures

**37K35** Lie-Bäcklund and other transformations for infinite-dimensional Hamiltonian and Lagrangian systems

**37K40** Soliton theory, asymptotic behavior of solutions of infinite-dimensional Hamiltonian systems

**37K45** Stability problems for infinite-dimensional Hamiltonian and Lagrangian systems

**37K50** Bifurcation problems for infinite-dimensional Hamiltonian and Lagrangian systems

**37K55** Perturbations, KAM theory for infinite-dimensional Hamiltonian and Lagrangian systems

**37K58** Variational principles and methods for infinite-dimensional Hamiltonian and Lagrangian systems

**37K60** Lattice dynamics; integrable lattice equations [See also 37L60]

**37K65** Hamiltonian systems on groups of diffeomorphisms and on manifolds of mappings and metrics

**37K99** None of the above, but in this section

## 37Lxx Infinite-dimensional dissipative dynamical systems [See also 35Bxx, 35Qxx]

**37L05** General theory of infinite-dimensional dissipative dynamical systems, nonlinear semigroups, evolution equations

**37L10** Normal forms, center manifold theory, bifurcation theory for infinite-dimensional dissipative dynamical systems

**37L15** Stability problems for infinite-dimensional dissipative dynamical systems

**37L20** Symmetries of infinite-dimensional dissipative dynamical systems

**37L25** Inertial manifolds and other invariant attracting sets of infinite-dimensional dissipative dynamical systems

**37L30** Attractors and their dimensions, Lyapunov exponents for infinite-dimensional dissipative dynamical systems

**37L40** Invariant measures for infinite-dimensional dissipative dynamical systems

**37L45** Hyperbolicity, Lyapunov functions for infinite-dimensional dissipative dynamical systems

**37L50** Noncompact semigroups, dispersive equations, perturbations of infinite-dimensional dissipative dynamical systems

**37L55** Infinite-dimensional random dynamical systems; stochastic equations [See also 35R60, 60H10, 60H15]

**37L60** Lattice dynamics and infinite-dimensional dissipative dynamical systems [See also 37K60]

**37L65** Special approximation methods (nonlinear Galerkin, etc.) for infinite-dimensional dissipative dynamical systems

**37L99** None of the above, but in this section

## 37Mxx Approximation methods and numerical treatment of dynamical systems {For numerical analysis, see also 65Pxx; for software etc., see 37-04}

**37M05** Simulation of dynamical systems

**37M10** Time series analysis of dynamical systems

**37M15** Discretization methods and integrators (symplectic, variational, geometric, etc.) for dynamical systems

**37M20** Computational methods for bifurcation problems in dynamical systems

**37M21** Computational methods for invariant manifolds of dynamical systems

**37M22** Computational methods for attractors of dynamical systems

**37M25** Computational methods for ergodic theory (approximation of invariant measures, computation of Lyapunov exponents, entropy, etc.)

**37M99** None of the above, but in this section

## 37Nxx Applications of dynamical systems

**37N05** Dynamical systems in classical and celestial mechanics [See mainly 70Fxx, 70Hxx, 70Kxx]

**37N10** Dynamical systems in fluid mechanics, oceanography and meteorology [See mainly 76-XX, especially 76D05, 76F20, 86A05, 86A10]

**37N15** Dynamical systems in solid mechanics [See mainly 74Hxx]

**37N20** Dynamical systems in other branches of physics (quantum mechanics, general relativity, laser physics)

**37N25** Dynamical systems in biology [See also 92-XX]

**37N30** Dynamical systems in numerical analysis [See also 65-XX]

**37N35** Dynamical systems in control [See also 93-XX]

**37N40** Dynamical systems in optimization and economics [See also 90-XX, 91-XX]

**37N99** None of the above, but in this section

## 37Pxx Arithmetic and non-Archimedean dynamical systems [See also 11S82, 37A44]

**37P05** Arithmetic and non-Archimedean dynamical systems involving polynomial and rational maps

**37P10** Arithmetic and non-Archimedean dynamical systems involving analytic and meromorphic maps

**37P15** Dynamical systems over global ground fields

**37P20** Dynamical systems over non-Archimedean local ground fields

**37P25** Dynamical systems over finite ground fields

**37P30** Height functions; Green functions; invariant measures in arithmetic and non-Archimedean dynamical systems [See also 11G50, 14G40]

**37P35** Arithmetic properties of periodic points

**37P40** Non-Archimedean Fatou and Julia sets

**37P45** Families and moduli spaces in arithmetic and non-Archimedean dynamical systems

**37P50** Dynamical systems on Berkovich spaces

**37P55** Arithmetic dynamics on general algebraic varieties

**37P99** None of the above, but in this section

# 39-XX Difference and functional equations

**39-00** General reference works (handbooks, dictionaries, bibliographies, etc.) pertaining to difference and functional equations

**39-01** Introductory exposition (textbooks, tutorial papers, etc.) pertaining to difference and functional equations

**39-02** Research exposition (monographs, survey articles) pertaining to difference and functional equations

**39-03** History of difference and functional equations [Consider also classification numbers from Section 01]

**39-04** Software, source code, etc. for problems pertaining to difference and functional equations

**39-06** Proceedings, conferences, collections, etc. pertaining to difference and functional equations

**39-08** Computational methods for problems pertaining to difference and functional equations

**39-11** Research data for problems pertaining to difference and functional equations

# 39Axx Difference equations {For dynamic equations on time scales, see 34N05; for dynamical systems, see 37-XX}

**39A05** General theory of difference equations

**39A06** Linear difference equations

**39A10** Additive difference equations

**39A12** Discrete version of topics in analysis

**39A13** Difference equations, scaling ($q$-differences) [See also 33Dxx]

**39A14** Partial difference equations

**39A20** Multiplicative and other generalized difference equations

**39A21** Oscillation theory for difference equations

**39A22** Growth, boundedness, comparison of solutions to difference equations

**39A23** Periodic solutions of difference equations

**39A24** Almost periodic solutions of difference equations

**39A26** Fuzzy difference equations

**39A27** Boundary value problems for difference equations

**39A28** Bifurcation theory for difference equations

**39A30** Stability theory for difference equations

**39A33** Chaotic behavior of solutions of difference equations

**39A36** Integrable difference and lattice equations; integrability tests

**39A45** Difference equations in the complex domain

**39A50** Stochastic difference equations

**39A60** Applications of difference equations

**39A70** Difference operators [See also 47B39]

**39A99** None of the above, but in this section

## 39Bxx Functional equations and inequalities [See also 30D05]

**39B05** General theory of functional equations and inequalities

**39B12** Iteration theory, iterative and composite equations [See also 26A18, 30D05, 37-XX]

**39B22** Functional equations for real functions [See also 26A51, 26B25]

**39B32** Functional equations for complex functions [See also 30D05]

**39B42** Matrix and operator functional equations [See also 47Jxx]

**39B52** Functional equations for functions with more general domains and/or ranges

**39B55** Orthogonal additivity and other conditional functional equations

**39B62** Functional inequalities, including subadditivity, convexity, etc. [See also 26A51, 26B25, 26Dxx]

**39B72** Systems of functional equations and inequalities

**39B82** Stability, separation, extension, and related topics for functional equations [See also 46A22]

**39B99** None of the above, but in this section

# 40-XX Sequences, series, summability

**40-00** General reference works (handbooks, dictionaries, bibliographies, etc.) pertaining to sequences, series, summability

**40-01** Introductory exposition (textbooks, tutorial papers, etc.) pertaining to sequences, series, summability

**40-02** Research exposition (monographs, survey articles) pertaining to sequences, series, summability

**40-03** History of sequences, series, summability [Consider also classification numbers from Section 01]

**40-04** Software, source code, etc. for problems pertaining to sequences, series, summability

**40-06** Proceedings, conferences, collections, etc. pertaining to sequences, series, summability

**40-08** Computational methods for problems pertaining to sequences, series, summability

**40-11** Research data for problems pertaining to sequences, series, summability

## 40Axx Convergence and divergence of infinite limiting processes

**40A05** Convergence and divergence of series and sequences

**40A10** Convergence and divergence of integrals

**40A15** Convergence and divergence of continued fractions [See also 30B70]

**40A20** Convergence and divergence of infinite products

**40A25** Approximation to limiting values (summation of series, etc.) {For the Euler-Maclaurin summation formula, see 65B15}

**40A30** Convergence and divergence of series and sequences of functions

**40A35** Ideal and statistical convergence [See also 40G15]

**40A99** None of the above, but in this section

## 40Bxx Multiple sequences and series

**40B05** Multiple sequences and series [Should also be assigned at least one other classification number in this section]

**40B99** None of the above, but in this section

## 40Cxx General summability methods

**40C05** Matrix methods for summability

**40C10** Integral methods for summability

**40C15** Function-theoretic methods (including power series methods and semicontinuous methods) for summability

**40C99** None of the above, but in this section

## 40Dxx Direct theorems on summability

**40D05** General theorems on summability

**40D09** Structure of summability fields

**40D10** Tauberian constants and oscillation limits in summability theory

**40D15** Convergence factors and summability factors

**40D20** Summability and bounded fields of methods

**40D25** Inclusion and equivalence theorems in summability theory

**40D99** None of the above, but in this section

## 40Exx Inversion theorems

**40E05** Tauberian theorems

**40E10** Growth estimates

**40E15** Lacunary inversion theorems

**40E20** Tauberian constants

**40E99** None of the above, but in this section

## 40Fxx Absolute and strong summability [Should also be assigned at least one other classification number in Section 40]

**40F05** Absolute and strong summability [Should also be assigned at least one other classification number in Section 40]

**40F99** None of the above, but in this section

## 40Gxx Special methods of summability

**40G05** Cesàro, Euler, Nörlund and Hausdorff methods

**40G10** Abel, Borel and power series methods

**40G15** Summability methods using statistical convergence [See also 40A35]

**40G99** None of the above, but in this section

## 40Hxx Functional analytic methods in summability

**40H05** Functional analytic methods in summability

**40H99** None of the above, but in this section

## 40Jxx Summability in abstract structures [Should also be assigned at least one other classification number from Section 40] [See also 43A55, 46A35, 46B15]

**40J05** Summability in abstract structures [Should also be assigned at least one other classification number from Section 40] [See also 43A55, 46A35, 46B15]

**40J99** None of the above, but in this section

# 41-XX Approximations and expansions {For approximation theory in the complex domain, see 30E05, 30E10; for trigonometric approximation and interpolation, see 42A10, 42A15; for numerical approximation, see 65Dxx}

**41-00** General reference works (handbooks, dictionaries, bibliographies, etc.) pertaining to approximations and expansions

**41-01** Introductory exposition (textbooks, tutorial papers, etc.) pertaining to approximations and expansions

**41-02** Research exposition (monographs, survey articles) pertaining to approximations and expansions

**41-03** History of approximations and expansions [Consider also classification numbers from Section 01]

**41-04** Software, source code, etc. for problems pertaining to approximations and expansions

**41-06** Proceedings, conferences, collections, etc. pertaining to approximations and expansions

**41-11** Research data for problems pertaining to approximations and expansions

## 41Axx Approximations and expansions {For approximation theory in the complex domain, see 30E05, 30E10; for trigonometric approximation and interpolation, see 42A10, 42A15; for numerical approximation, see 65Dxx}

**41A05** Interpolation in approximation theory [See also 42A15, 65D05]

**41A10** Approximation by polynomials {For approximation by trigonometric polynomials, see 42A10}

**41A15** Spline approximation

**41A17** Inequalities in approximation (Bernstein, Jackson, Nikol'skiĭ-type inequalities)

**41A20** Approximation by rational functions

**41A21** Padé approximation

**41A25** Rate of convergence, degree of approximation

**41A27** Inverse theorems in approximation theory

**41A28** Simultaneous approximation

**41A29** Approximation with constraints

**41A30** Approximation by other special function classes

**41A35** Approximation by operators (in particular, by integral operators)

**41A36** Approximation by positive operators

**41A40** Saturation in approximation theory

**41A44** Best constants in approximation theory

**41A45** Approximation by arbitrary linear expressions

**41A46** Approximation by arbitrary nonlinear expressions; widths and entropy

**41A50** Best approximation, Chebyshev systems

**41A52** Uniqueness of best approximation

**41A55** Approximate quadratures

**41A58** Series expansions (e.g., Taylor, Lidstone series, but not Fourier series)

**41A60** Asymptotic approximations, asymptotic expansions (steepest descent, etc.) [See also 30E15]

**41A63** Multidimensional problems [Should also be assigned at least one other classification number from Section 41]

**41A65** Abstract approximation theory (approximation in normed linear spaces and other abstract spaces)

**41A80** Remainders in approximation formulas

**41A81** Weighted approximation

**41A99** None of the above, but in this section

# 42-XX Harmonic analysis on Euclidean spaces

**42-00** General reference works (handbooks, dictionaries, bibliographies, etc.) pertaining to harmonic analysis on Euclidean spaces

**42-01** Introductory exposition (textbooks, tutorial papers, etc.) pertaining to harmonic analysis on Euclidean spaces

**42-02** Research exposition (monographs, survey articles) pertaining to harmonic analysis on Euclidean spaces

**42-03** History of harmonic analysis on Euclidean spaces [Consider also classification numbers from Section 01]

**42-04** Software, source code, etc. for problems pertaining to harmonic analysis on Euclidean spaces

**42-06** Proceedings, conferences, collections, etc. pertaining to harmonic analysis on Euclidean spaces

**42-08** Computational methods for problems pertaining to harmonic analysis on Euclidean spaces

**42-11** Research data for problems pertaining to harmonic analysis on Euclidean spaces

## 42Axx Harmonic analysis in one variable

**42A05** Trigonometric polynomials, inequalities, extremal problems

**42A10** Trigonometric approximation

**42A15** Trigonometric interpolation

**42A16** Fourier coefficients, Fourier series of functions with special properties, special Fourier series {For automorphic theory, see mainly 11F30}

**42A20** Convergence and absolute convergence of Fourier and trigonometric series

**42A24** Summability and absolute summability of Fourier and trigonometric series

**42A32** Trigonometric series of special types (positive coefficients, monotonic coefficients, etc.)

**42A38** Fourier and Fourier-Stieltjes transforms and other transforms of Fourier type

**42A45** Multipliers in one variable harmonic analysis

**42A50** Conjugate functions, conjugate series, singular integrals

**42A55** Lacunary series of trigonometric and other functions; Riesz products

**42A61** Probabilistic methods for one variable harmonic analysis

**42A63** Uniqueness of trigonometric expansions, uniqueness of Fourier expansions, Riemann theory, localization

**42A65** Completeness of sets of functions in one variable harmonic analysis

**42A70** Trigonometric moment problems in one variable harmonic analysis

**42A75** Classical almost periodic functions, mean periodic functions [See also 43A60]

**42A82** Positive definite functions in one variable harmonic analysis

**42A85** Convolution, factorization for one variable harmonic analysis

**42A99** None of the above, but in this section

## 42Bxx Harmonic analysis in several variables {For automorphic theory, see mainly 11F30}

**42B05** Fourier series and coefficients in several variables

**42B08** Summability in several variables

**42B10** Fourier and Fourier-Stieltjes transforms and other transforms of Fourier type

**42B15** Multipliers for harmonic analysis in several variables

**42B20** Singular and oscillatory integrals (Calderón-Zygmund, etc.)

**42B25** Maximal functions, Littlewood-Paley theory

**42B30** $H^p$-spaces

**42B35** Function spaces arising in harmonic analysis

**42B37** Harmonic analysis and PDEs [See also 35-XX]

**42B99** None of the above, but in this section

## 42Cxx Nontrigonometric harmonic analysis

**42C05** Orthogonal functions and polynomials, general theory of nontrigonometric harmonic analysis [See also 33C45, 33C50, 33D45]

**42C10** Fourier series in special orthogonal functions (Legendre polynomials, Walsh functions, etc.)

**42C15** General harmonic expansions, frames

**42C20** Other transformations of harmonic type

**42C25** Uniqueness and localization for orthogonal series

**42C30** Completeness of sets of functions in nontrigonometric harmonic analysis

**42C40** Nontrigonometric harmonic analysis involving wavelets and other special systems

**42C99** None of the above, but in this section

# 43-XX Abstract harmonic analysis {For other analysis on topological and Lie groups, see 22Exx}

**43-00** General reference works (handbooks, dictionaries, bibliographies, etc.) pertaining to abstract harmonic analysis

**43-01** Introductory exposition (textbooks, tutorial papers, etc.) pertaining to abstract harmonic analysis

**43-02** Research exposition (monographs, survey articles) pertaining to abstract harmonic analysis

**43-03** History of abstract harmonic analysis [Consider also classification numbers from Section 01]

**43-04** Software, source code, etc. for problems pertaining to abstract harmonic analysis

**43-06** Proceedings, conferences, collections, etc. pertaining to abstract harmonic analysis

**43-08** Computational methods for problems pertaining to abstract harmonic analysis

**43-11** Research data for problems pertaining to abstract harmonic analysis

## 43Axx Abstract harmonic analysis {For other analysis on topological and Lie groups, see 22Exx}

**43A05** Measures on groups and semigroups, etc.

**43A07** Means on groups, semigroups, etc.; amenable groups

**43A10** Measure algebras on groups, semigroups, etc.

**43A15** $L^p$-spaces and other function spaces on groups, semigroups, etc.

**43A17** Analysis on ordered groups, $H^p$-theory

**43A20** $L^1$-algebras on groups, semigroups, etc.

**43A22** Homomorphisms and multipliers of function spaces on groups, semigroups, etc.

**43A25** Fourier and Fourier-Stieltjes transforms on locally compact and other abelian groups

**43A30** Fourier and Fourier-Stieltjes transforms on nonabelian groups and on semigroups, etc.

**43A32** Other transforms and operators of Fourier type

**43A35** Positive definite functions on groups, semigroups, etc.

**43A40** Character groups and dual objects

**43A45** Spectral synthesis on groups, semigroups, etc.

**43A46** Special sets (thin sets, Kronecker sets, Helson sets, Ditkin sets, Sidon sets, etc.)

**43A50** Convergence of Fourier series and of inverse transforms

**43A55** Summability methods on groups, semigroups, etc. [See also 40J05]

**43A60** Almost periodic functions on groups and semigroups and their generalizations (recurrent functions, distal functions, etc.); almost automorphic functions

**43A62** Harmonic analysis on hypergroups

**43A65** Representations of groups, semigroups, etc. (aspects of abstract harmonic analysis) [See also 22A10, 22A20, 22Dxx, 22E45]

**43A70** Analysis on specific locally compact and other abelian groups [See also 11R56, 22B05]

**43A75** Harmonic analysis on specific compact groups

**43A77** Harmonic analysis on general compact groups

**43A80** Analysis on other specific Lie groups [See also 22Exx]

**43A85** Harmonic analysis on homogeneous spaces

**43A90** Harmonic analysis and spherical functions [See also 22E45, 22E46, 33C55]

**43A95** Categorical methods for abstract harmonic analysis [See also 46Mxx]

**43A99** None of the above, but in this section

# 44-XX Integral transforms, operational calculus {For fractional derivatives and integrals, see 26A33; for Fourier transforms, see 42A38, 42B10; for integral transforms in distribution spaces, see 46F12; for numerical methods, see 65R10}

**44-00** General reference works (handbooks, dictionaries, bibliographies, etc.) pertaining to integral transforms

**44-01** Introductory exposition (textbooks, tutorial papers, etc.) pertaining to integral transforms

**44-02** Research exposition (monographs, survey articles) pertaining to integral transforms

**44-03** History of integral transforms [Consider also classification numbers from Section 01]

**44-04** Software, source code, etc. for problems pertaining to integral transforms

**44-06** Proceedings, conferences, collections, etc. pertaining to integral transforms

**44-11** Research data for problems pertaining to integral transforms

## 44Axx Integral transforms, operational calculus {For fractional derivatives and integrals, see 26A33; for Fourier transforms, see 42A38, 42B10; for integral transforms in distribution spaces, see 46F12; for numerical methods, see 65R10}

**44A05** General integral transforms [See also 42A38]

**44A10** Laplace transform

**44A12** Radon transform [See also 92C55]

**44A15** Special integral transforms (Legendre, Hilbert, etc.)

**44A20** Integral transforms of special functions

**44A30** Multiple integral transforms

**44A35** Convolution as an integral transform

**44A40** Calculus of Mikusiński and other operational calculi

**44A45** Classical operational calculus

**44A55** Discrete operational calculus

**44A60** Moment problems {For trigonometric moment problems, see 42A70}

**44A99** None of the above, but in this section

# 45-XX Integral equations

**45-00** General reference works (handbooks, dictionaries, bibliographies, etc.) pertaining to integral equations

**45-01** Introductory exposition (textbooks, tutorial papers, etc.) pertaining to integral equations

**45-02** Research exposition (monographs, survey articles) pertaining to integral equations

**45-03** History of integral equations [Consider also classification numbers from Section 01]

**45-04** Software, source code, etc. for problems pertaining to integral equations

**45-06** Proceedings, conferences, collections, etc. pertaining to integral equations

**45-11** Research data for problems pertaining to integral equations

## 45Axx Linear integral equations

**45A05** Linear integral equations

**45A99** None of the above, but in this section

## 45Bxx Fredholm integral equations

**45B05** Fredholm integral equations

**45B99** None of the above, but in this section

## 45Cxx Eigenvalue problems for integral equations [See also 34Lxx, 35Pxx, 45P05, 47A75]

**45C05** Eigenvalue problems for integral equations [See also 34Lxx, 35Pxx, 45P05, 47A75]

**45C99** None of the above, but in this section

## 45Dxx Volterra integral equations [See also 34A12]

**45D05** Volterra integral equations [See also 34A12]

**45D99** None of the above, but in this section

## 45Exx Singular integral equations [See also 30E20, 30E25, 44A15, 44A35]

**45E05** Integral equations with kernels of Cauchy type [See also 35J15]

**45E10** Integral equations of the convolution type (Abel, Picard, Toeplitz and Wiener-Hopf type) [See also 47B35]

**45E99** None of the above, but in this section

## 45Fxx Systems of linear integral equations

**45F05** Systems of nonsingular linear integral equations

**45F10** Dual, triple, etc., integral and series equations

**45F15** Systems of singular linear integral equations

**45F99** None of the above, but in this section

## 45Gxx Nonlinear integral equations [See also 47H30, 47Jxx]

**45G05** Singular nonlinear integral equations

**45G10** Other nonlinear integral equations

**45G15** Systems of nonlinear integral equations

**45G99** None of the above, but in this section

## 45Hxx Integral equations with miscellaneous special kernels [See also 44A15]

**45H05** Integral equations with miscellaneous special kernels [See also 44A15]

**45H99** None of the above, but in this section

## 45Jxx Integro-ordinary differential equations [See also 34K05, 34K30, 47G20]

**45J05** Integro-ordinary differential equations [See also 34K05, 34K30, 47G20]

**45J99** None of the above, but in this section

## 45Kxx Integro-partial differential equations [See also 34K30, 35R09, 35R10, 47G20]

**45K05** Integro-partial differential equations [See also 34K30, 35R09, 35R10, 47G20]

**45K99** None of the above, but in this section

## 45Lxx Theoretical approximation of solutions to integral equations {For numerical analysis, see 65Rxx}

**45L05** Theoretical approximation of solutions to integral equations {For numerical analysis, see 65Rxx}

**45L99** None of the above, but in this section

## 45Mxx Qualitative behavior of solutions to integral equations

**45M05** Asymptotics of solutions to integral equations

**45M10** Stability theory for integral equations

**45M15** Periodic solutions of integral equations

**45M20** Positive solutions of integral equations

**45M99** None of the above, but in this section

## 45Nxx Abstract integral equations, integral equations in abstract spaces

**45N05** Abstract integral equations, integral equations in abstract spaces

**45N99** None of the above, but in this section

## 45Pxx Integral operators [See also 47B38, 47G10]

**45P05** Integral operators [See also 47B38, 47G10]

**45P99** None of the above, but in this section

## 45Qxx Inverse problems for integral equations

**45Q05** Inverse problems for integral equations

**45Q99** None of the above, but in this section

## 45Rxx Random integral equations [See also 60H20]

**45R05** Random integral equations [See also 60H20]

**45R99** None of the above, but in this section

# 46-XX Functional analysis {For manifolds modeled on topological linear spaces, see 57Nxx, 58Bxx}

**46-00** General reference works (handbooks, dictionaries, bibliographies, etc.) pertaining to functional analysis

**46-01** Introductory exposition (textbooks, tutorial papers, etc.) pertaining to functional analysis

**46-02** Research exposition (monographs, survey articles) pertaining to functional analysis

**46-03** History of functional analysis [Consider also classification numbers from Section 01]

**46-04** Software, source code, etc. for problems pertaining to functional analysis

**46-06** Proceedings, conferences, collections, etc. pertaining to functional analysis

**46-08** Computational methods for problems pertaining to functional analysis

**46-11** Research data for problems pertaining to functional analysis

## 46Axx Topological linear spaces and related structures {For function spaces, see 46Exx}

**46A03** General theory of locally convex spaces

**46A04** Locally convex Fréchet spaces and (DF)-spaces

**46A08** Barrelled spaces, bornological spaces

**46A11** Spaces determined by compactness or summability properties (nuclear spaces, Schwartz spaces, Montel spaces, etc.)

**46A13** Spaces defined by inductive or projective limits (LB, LF, etc.) [See also 46M40]

**46A16** Not locally convex spaces (metrizable topological linear spaces, locally bounded spaces, quasi-Banach spaces, etc.)

**46A17** Bornologies and related structures; Mackey convergence, etc.

**46A19** Other "topological" linear spaces (convergence spaces, ranked spaces, spaces with a metric taking values in an ordered structure more general than $\mathbb{R}$, etc.)

**46A20** Duality theory for topological vector spaces

**46A22** Theorems of Hahn-Banach type; extension and lifting of functionals and operators [See also 46M10]

**46A25** Reflexivity and semi-reflexivity [See also 46B10]

**46A30** Open mapping and closed graph theorems; completeness (including $B$-, $B_r$-completeness)

**46A32** Spaces of linear operators; topological tensor products; approximation properties [See also 46B28, 46M05, 47L05, 47L20]

**46A35** Summability and bases in topological vector spaces [See also 46B15]

**46A40** Ordered topological linear spaces, vector lattices [See also 06F20, 46B40, 46B42]

**46A45** Sequence spaces (including Köthe sequence spaces) [See also 46B45]

**46A50** Compactness in topological linear spaces; angelic spaces, etc.

**46A55** Convex sets in topological linear spaces; Choquet theory [See also 52A07]

**46A61** Graded Fréchet spaces and tame operators

**46A63** Topological invariants ((DN), ($\Omega$), etc.) for locally convex spaces

**46A70** Saks spaces and their duals (strict topologies, mixed topologies, two-norm spaces, co-Saks spaces, etc.)

**46A80** Modular spaces

**46A99** None of the above, but in this section

## 46Bxx Normed linear spaces and Banach spaces; Banach lattices {For function spaces, see 46Exx}

**46B03** Isomorphic theory (including renorming) of Banach spaces

**46B04** Isometric theory of Banach spaces

**46B06** Asymptotic theory of Banach spaces [See also 52A23]

**46B07** Local theory of Banach spaces

**46B08** Ultraproduct techniques in Banach space theory [See also 46M07]

**46B09** Probabilistic methods in Banach space theory [See also 60Bxx]

**46B10** Duality and reflexivity in normed linear and Banach spaces [See also 46A25]

**46B15** Summability and bases; functional analytic aspects of frames in Banach and Hilbert spaces [See also 46A35, 42C15]

**46B20** Geometry and structure of normed linear spaces

**46B22** Radon-Nikodým, Kreĭn-Milman and related properties [See also 46G10]

**46B25** Classical Banach spaces in the general theory

**46B26** Nonseparable Banach spaces

**46B28** Spaces of operators; tensor products; approximation properties [See also 46A32, 46M05, 47L05, 47L20]

**46B40** Ordered normed spaces [See also 46A40, 46B42]

**46B42** Banach lattices [See also 46A40, 46B40]

**46B45** Banach sequence spaces [See also 46A45]

**46B50** Compactness in Banach (or normed) spaces

**46B70** Interpolation between normed linear spaces [See also 46M35]

**46B80** Nonlinear classification of Banach spaces; nonlinear quotients

**46B85** Embeddings of discrete metric spaces into Banach spaces; applications in topology and computer science [See also 05C12, 68Rxx]

**46B87** Lineability in functional analysis [See also 15A03]

**46B99** None of the above, but in this section

## 46Cxx Inner product spaces and their generalizations, Hilbert spaces {For function spaces, see 46Exx}

**46C05** Hilbert and pre-Hilbert spaces: geometry and topology (including spaces with semidefinite inner product)

**46C07** Hilbert subspaces (= operator ranges); complementation (Aronszajn, de Branges, etc.) [See also 46B70, 46M35]

**46C15** Characterizations of Hilbert spaces

**46C20** Spaces with indefinite inner product (Kreĭn spaces, Pontryagin spaces, etc.) [See also 47B50]

**46C50** Generalizations of inner products (semi-inner products, partial inner products, etc.)

**46C99** None of the above, but in this section

## 46Exx Linear function spaces and their duals [See also 30H05, 32A38, 46F05] {For function algebras, see 46J10}

**46E05** Lattices of continuous, differentiable or analytic functions

**46E10** Topological linear spaces of continuous, differentiable or analytic functions

**46E15** Banach spaces of continuous, differentiable or analytic functions

**46E20** Hilbert spaces of continuous, differentiable or analytic functions

**46E22** Hilbert spaces with reproducing kernels (= (proper) functional Hilbert spaces, including de Branges-Rovnyak and other structured spaces) [See also 47B32]

**46E25** Rings and algebras of continuous, differentiable or analytic functions {For Banach function algebras, see 46J10, 46J15}

**46E27** Spaces of measures [See also 28A33, 46Gxx]

**46E30** Spaces of measurable functions ($L^p$-spaces, Orlicz spaces, Köthe function spaces, Lorentz spaces, rearrangement invariant spaces, ideal spaces, etc.)

**46E35** Sobolev spaces and other spaces of "smooth" functions, embedding theorems, trace theorems

**46E36** Sobolev (and similar kinds of) spaces of functions on metric spaces; analysis on metric spaces

**46E39** Sobolev (and similar kinds of) spaces of functions of discrete variables

**46E40** Spaces of vector- and operator-valued functions

**46E50** Spaces of differentiable or holomorphic functions on infinite-dimensional spaces [See also 46G20, 46G25, 47H60]

**46E99** None of the above, but in this section

## 46Fxx Distributions, generalized functions, distribution spaces [See also 46T30]

**46F05** Topological linear spaces of test functions, distributions and ultradistributions [See also 46E10, 46E35]

**46F10** Operations with distributions and generalized functions

**46F12** Integral transforms in distribution spaces [See also 42-XX, 44-XX]

**46F15** Hyperfunctions, analytic functionals [See also 32A25, 32A45, 32C35, 58J15]

**46F20** Distributions and ultradistributions as boundary values of analytic functions [See also 30D40, 30E25, 32A40]

**46F25** Distributions on infinite-dimensional spaces [See also 58C35]

**46F30** Generalized functions for nonlinear analysis (Rosinger, Colombeau, nonstandard, etc.)

**46F99** None of the above, but in this section

## 46Gxx Measures, integration, derivative, holomorphy (all involving infinite-dimensional spaces) [See also 28-XX, 46Txx]

**46G05** Derivatives of functions in infinite-dimensional spaces [See also 46T20, 58C20, 58C25]

**46G10** Vector-valued measures and integration [See also 26E20, 28B05, 46B22]

**46G12** Measures and integration on abstract linear spaces [See also 28C20, 46T12]

**46G15** Functional analytic lifting theory [See also 28A51]

**46G20** Infinite-dimensional holomorphy [See also 32-XX, 46E50, 46T25, 58B12, 58C10]

**46G25** (Spaces of) multilinear mappings, polynomials [See also 46E50, 46G20, 47H60]

**46G99** None of the above, but in this section

## 46Hxx Topological algebras, normed rings and algebras, Banach algebras {For group algebras, convolution algebras and measure algebras, see 43A10, 43A20}

**46H05** General theory of topological algebras

**46H10** Ideals and subalgebras

**46H15** Representations of topological algebras

**46H20** Structure, classification of topological algebras

**46H25** Normed modules and Banach modules, topological modules (if not placed in 13-XX or 16-XX)

**46H30** Functional calculus in topological algebras [See also 47A60]

**46H35** Topological algebras of operators [See mainly 47Lxx]

**46H40** Automatic continuity

**46H70** Nonassociative topological algebras [See also 46K70, 46L70]

**46H99** None of the above, but in this section

## 46Jxx Commutative Banach algebras and commutative topological algebras [See also 46E25]

**46J05** General theory of commutative topological algebras

**46J10** Banach algebras of continuous functions, function algebras [See also 46E25]

**46J15** Banach algebras of differentiable or analytic functions, $H^p$-spaces [See also 30H10, 32A35, 32A37, 32A38, 42B30]

**46J20** Ideals, maximal ideals, boundaries

**46J25** Representations of commutative topological algebras

**46J30** Subalgebras of commutative topological algebras

**46J40** Structure and classification of commutative topological algebras

**46J45** Radical Banach algebras

**46J99** None of the above, but in this section

## 46Kxx Topological (rings and) algebras with an involution [See also 16W10]

**46K05** General theory of topological algebras with involution

**46K10** Representations of topological algebras with involution

**46K15** Hilbert algebras

**46K50** Nonselfadjoint (sub)algebras in algebras with involution

**46K70** Nonassociative topological algebras with an involution [See also 46H70, 46L70]

**46K99** None of the above, but in this section

## 46Lxx Selfadjoint operator algebras ($C^*$-algebras, von Neumann ($W^*$-) algebras, etc.) [See also 22D25, 47Lxx]

**46L05** General theory of $C^*$-algebras

**46L06** Tensor products of $C^*$-algebras

**46L07** Operator spaces and completely bounded maps [See also 47L25]

**46L08** $C^*$-modules

**46L09** Free products of $C^*$-algebras

**46L10** General theory of von Neumann algebras

**46L30** States of selfadjoint operator algebras

**46L35** Classifications of $C^*$-algebras

**46L36** Classification of factors

**46L37** Subfactors and their classification

**46L40** Automorphisms of selfadjoint operator algebras

**46L45** Decomposition theory for $C^*$-algebras

**46L51** Noncommutative measure and integration

**46L52** Noncommutative function spaces

**46L53** Noncommutative probability and statistics

**46L54** Free probability and free operator algebras

**46L55** Noncommutative dynamical systems [See also 28Dxx, 37Kxx, 37Lxx, 37A55]

**46L57** Derivations, dissipations and positive semigroups in $C^*$-algebras

**46L60** Applications of selfadjoint operator algebras to physics [See also 46N50, 46N55, 47L90, 81T05, 82B10, 82C10]

**46L65** Quantizations, deformations for selfadjoint operator algebras

**46L67** Quantum groups (operator algebraic aspects)

**46L70** Nonassociative selfadjoint operator algebras [See also 46H70, 46K70]

**46L80** $K$-theory and operator algebras (including cyclic theory) [See also 18F25, 19Kxx, 46M20, 55Rxx, 58J22]

**46L85** Noncommutative topology [See also 58B32, 58B34, 58J22]

**46L87** Noncommutative differential geometry [See also 58B32, 58B34, 58J22]

**46L89** Other "noncommutative" mathematics based on $C^*$-algebra theory [See also 58B32, 58B34, 58J22]

**46L99** None of the above, but in this section

## 46Mxx Methods of category theory in functional analysis [See also 18-XX]

**46M05** Tensor products in functional analysis [See also 46A32, 46B28, 47A80]

**46M07** Ultraproducts in functional analysis [See also 46B08, 46S20]

**46M10** Projective and injective objects in functional analysis [See also 46A22]

**46M15** Categories, functors in functional analysis {For $K$-theory, Ext, etc., see 19K33, 46L80, 46M18, 46M20}

**46M18** Homological methods in functional analysis (exact sequences, right inverses, lifting, etc.)

**46M20** Methods of algebraic topology in functional analysis (cohomology, sheaf and bundle theory, etc.) [See also 14F06, 18Fxx, 19Kxx, 32Cxx, 32Lxx, 46L80, 46M15, 46M18, 55Rxx]

**46M35** Abstract interpolation of topological vector spaces [See also 46B70]

**46M40** Inductive and projective limits in functional analysis [See also 46A13]

**46M99** None of the above, but in this section

## 46Nxx Miscellaneous applications of functional analysis [See also 47Nxx]

**46N10** Applications of functional analysis in optimization, convex analysis, mathematical programming, economics

**46N20** Applications of functional analysis to differential and integral equations

**46N30** Applications of functional analysis in probability theory and statistics

**46N40** Applications of functional analysis in numerical analysis [See also 65Jxx]

**46N50** Applications of functional analysis in quantum physics

**46N55** Applications of functional analysis in statistical physics

**46N60** Applications of functional analysis in biology and other sciences

**46N99** None of the above, but in this section

## 46Sxx Other (nonclassical) types of functional analysis [See also 47Sxx]

**46S05** Quaternionic functional analysis

**46S10** Functional analysis over fields other than $\mathbb{R}$ or $\mathbb{C}$ or the quaternions; non-Archimedean functional analysis [See also 12J25, 32P05]

**46S20** Nonstandard functional analysis [See also 03H05]

**46S30** Constructive functional analysis [See also 03F60]

**46S40** Fuzzy functional analysis [See also 03E72]

**46S50** Functional analysis in probabilistic metric linear spaces

**46S60** Functional analysis on superspaces (supermanifolds) or graded spaces [See also 58A50, 58C50]

**46S99** None of the above, but in this section

## 46Txx Nonlinear functional analysis [See also 47Hxx, 47Jxx, 58Cxx, 58Dxx]

**46T05** Infinite-dimensional manifolds [See also 53Axx, 57N20, 58Bxx, 58Dxx]

**46T10** Manifolds of mappings

**46T12** Measure (Gaussian, cylindrical, etc.) and integrals (Feynman, path, Fresnel, etc.) on manifolds [See also 28Cxx, 46G12, 60-XX]

**46T20** Continuous and differentiable maps in nonlinear functional analysis [See also 46G05]

**46T25** Holomorphic maps in nonlinear functional analysis [See also 46G20]

**46T30** Distributions and generalized functions on nonlinear spaces [See also 46Fxx]

**46T99** None of the above, but in this section

# 47-XX Operator theory

**47-00** General reference works (handbooks, dictionaries, bibliographies, etc.) pertaining to operator theory

**47-01** Introductory exposition (textbooks, tutorial papers, etc.) pertaining to operator theory

**47-02** Research exposition (monographs, survey articles) pertaining to operator theory

**47-03** History of operator theory [Consider also classification numbers from Section 01]

**47-04** Software, source code, etc. for problems pertaining to operator theory

**47-06** Proceedings, conferences, collections, etc. pertaining to operator theory

**47-08** Computational methods for problems pertaining to operator theory

**47-11** Research data for problems pertaining to operator theory

## 47Axx General theory of linear operators

**47A05** General (adjoints, conjugates, products, inverses, domains, ranges, etc.)

**47A06** Linear relations (multivalued linear operators)

**47A07** Forms (bilinear, sesquilinear, multilinear)

**47A08** Operator matrices [See also 47A13]

**47A10** Spectrum, resolvent

**47A11** Local spectral properties of linear operators

**47A12** Numerical range, numerical radius

**47A13** Several-variable operator theory (spectral, Fredholm, etc.)

**47A15** Invariant subspaces of linear operators [See also 47A46]

**47A16** Cyclic vectors, hypercyclic and chaotic operators

**47A20** Dilations, extensions, compressions of linear operators

**47A25** Spectral sets of linear operators

**47A30** Norms (inequalities, more than one norm, etc.) of linear operators

**47A35** Ergodic theory of linear operators [See also 28Dxx, 37Axx]

**47A40** Scattering theory of linear operators [See also 34L25, 35P25, 37K15, 58J50, 81Uxx]

**47A45** Canonical models for contractions and nonselfadjoint linear operators

**47A46** Chains (nests) of projections or of invariant subspaces, integrals along chains, etc.

**47A48** Operator colligations (= nodes), vessels, linear systems, characteristic functions, realizations, etc.

**47A50** Equations and inequalities involving linear operators, with vector unknowns

**47A52** Linear operators and ill-posed problems, regularization [See also 35R25, 47J06, 65F22, 65J20, 65L08, 65M30, 65R30]

**47A53** (Semi-) Fredholm operators; index theories [See also 58B15, 58J20]

**47A55** Perturbation theory of linear operators [See also 47H14, 58J37, 70H09, 81Q15]

**47A56** Functions whose values are linear operators (operator- and matrix-valued functions, etc., including analytic and meromorphic ones)

**47A57** Linear operator methods in interpolation, moment and extension problems [See also 30E05, 42A70, 42A82, 44A60]

**47A58** Linear operator approximation theory

**47A60** Functional calculus for linear operators

**47A62** Equations involving linear operators, with operator unknowns

**47A63** Linear operator inequalities

**47A64** Operator means involving linear operators, shorted linear operators, etc.

**47A65** Structure theory of linear operators

**47A66** Quasitriangular and nonquasitriangular, quasidiagonal and nonquasidiagonal linear operators

**47A67** Representation theory of linear operators

**47A68** Factorization theory (including Wiener-Hopf and spectral factorizations) of linear operators

**47A70** (Generalized) eigenfunction expansions of linear operators; rigged Hilbert spaces

**47A75** Eigenvalue problems for linear operators [See also 47J10, 49R05]

**47A80** Tensor products of linear operators [See also 46M05]

**47A99** None of the above, but in this section

## 47Bxx Special classes of linear operators

**47B01** Operators on Banach spaces

**47B02** Operators on Hilbert spaces (general)

**47B06** Riesz operators; eigenvalue distributions; approximation numbers, $s$-numbers, Kolmogorov numbers, entropy numbers, etc. of operators

**47B07** Linear operators defined by compactness properties

**47B10** Linear operators belonging to operator ideals (nuclear, $p$-summing, in the Schatten-von Neumann classes, etc.) [See also 47L20]

**47B12** Sectorial operators

**47B13** Cowen-Douglas operators

**47B15** Hermitian and normal operators (spectral measures, functional calculus, etc.)

**47B20** Subnormal operators, hyponormal operators, etc.

**47B25** Linear symmetric and selfadjoint operators (unbounded)

**47B28** Nonselfadjoint operators [See also 47A45, 81Q12]

**47B32** Linear operators in reproducing-kernel Hilbert spaces (including de Branges, de Branges-Rovnyak, and other structured spaces) [See also 46E22]

**47B33** Linear composition operators

**47B34** Kernel operators

**47B35** Toeplitz operators, Hankel operators, Wiener-Hopf operators {For other integral operators, see also 45P05, 47G10} [See also 32A25, 32M15]

**47B36** Jacobi (tridiagonal) operators (matrices) and generalizations

**47B37** Linear operators on special spaces (weighted shifts, operators on sequence spaces, etc.)

**47B38** Linear operators on function spaces (general)

**47B39** Linear difference operators [See also 39A70]

**47B40** Spectral operators, decomposable operators, well-bounded operators, etc.

**47B44** Linear accretive operators, dissipative operators, etc.

**47B47** Commutators, derivations, elementary operators, etc.

**47B48** Linear operators on Banach algebras

**47B49** Transformers, preservers (linear operators on spaces of linear operators)

**47B50** Linear operators on spaces with an indefinite metric [See also 46C20]

**47B60** Linear operators on ordered spaces

**47B65** Positive linear operators and order-bounded operators

**47B80** Random linear operators [See also 47H40, 60H25]

**47B90** Operator theory and harmonic analysis [See also 42-XX, 43-XX, 44-XX]

**47B91** Operators on complex function spaces

**47B92** Operators on real function spaces

**47B93** Operators arising in mathematical physics

**47B99** None of the above, but in this section

## 47Cxx Individual linear operators as elements of algebraic systems

**47C05** Linear operators in algebras

**47C10** Linear operators in $*$-algebras

**47C15** Linear operators in $C^*$- or von Neumann algebras

**47C99** None of the above, but in this section

## 47Dxx Groups and semigroups of linear operators, their generalizations and applications

**47D03** Groups and semigroups of linear operators [See also 20M20] {For nonlinear operators, see 47H20}

**47D06** One-parameter semigroups and linear evolution equations [See also 34G10, 34K30]

**47D07** Markov semigroups and applications to diffusion processes {For Markov processes, see 60Jxx}

**47D08** Schrödinger and Feynman-Kac semigroups

**47D09** Operator sine and cosine functions and higher-order Cauchy problems [See also 34G10]

**47D60** $C$-semigroups, regularized semigroups

**47D62** Integrated semigroups

**47D99** None of the above, but in this section

## 47Exx Ordinary differential operators [See also 34Bxx, 34Lxx]

**47E05** General theory of ordinary differential operators [Should also be assigned at least one other classification number in Section 47] [See also 34Bxx, 34Lxx]

**47E07** Functional-differential and differential-difference operators [See also 34K08]

**47E99** None of the above, but in this section

## 47Fxx Partial differential operators [See also 35Pxx, 58Jxx]

**47F05** General theory of partial differential operators [Should also be assigned at least one other classification number in Section 47] [See also 35Pxx, 58Jxx]

**47F10** Elliptic operators and their generalizations {For elliptic complexes, see 58J10}

**47F99** None of the above, but in this section

## 47Gxx Integral, integro-differential, and pseudodifferential operators [See also 58Jxx]

**47G10** Integral operators [See also 45P05]

**47G20** Integro-differential operators [See also 34K30, 35R09, 35R10, 45J05, 45K05]

**47G30** Pseudodifferential operators [See also 35Sxx, 58Jxx]

**47G40** Potential operators [See also 31-XX]

**47G99** None of the above, but in this section

## 47Hxx Nonlinear operators and their properties {For global and geometric aspects, see 49J53, 58-XX, especially 58Cxx}

**47H04** Set-valued operators [See also 28B20, 54C60, 58C06]

**47H05** Monotone operators and generalizations

**47H06** Nonlinear accretive operators, dissipative operators, etc.

**47H07** Monotone and positive operators on ordered Banach spaces or other ordered topological vector spaces

**47H08** Measures of noncompactness and condensing mappings, $K$-set contractions, etc.

**47H09** Contraction-type mappings, nonexpansive mappings, $A$-proper mappings, etc.

**47H10** Fixed-point theorems [See also 37C25, 54H25, 55M20, 58C30]

**47H11** Degree theory for nonlinear operators [See also 55M25, 58C30]

**47H14** Perturbations of nonlinear operators [See also 47A55, 58J37, 70H09, 70K60, 81Q15]

**47H20** Semigroups of nonlinear operators [See also 37L05, 47J35, 54H15, 58D07]

**47H25** Nonlinear ergodic theorems [See also 28Dxx, 37Axx, 47A35]

**47H30** Particular nonlinear operators (superposition, Hammerstein, Nemytskiĭ, Uryson, etc.) [See also 45Gxx, 45P05]

**47H40** Random nonlinear operators [See also 47B80, 60H25]

**47H60** Multilinear and polynomial operators [See also 46G25]

**47H99** None of the above, but in this section

## 47Jxx Equations and inequalities involving nonlinear operators [See also 46Txx] {For global and geometric aspects, see 58-XX}

**47J05** Equations involving nonlinear operators (general) [See also 47H10, 47J25]

**47J06** Nonlinear ill-posed problems [See also 35R25, 47A52, 65F22, 65J20, 65L08, 65M30, 65R30]

**47J07** Abstract inverse mapping and implicit function theorems involving nonlinear operators [See also 46T20, 58C15]

**47J10** Nonlinear spectral theory, nonlinear eigenvalue problems [See also 49R05]

**47J15** Abstract bifurcation theory involving nonlinear operators [See also 34C23, 37Gxx, 58E07, 58E09]

**47J20** Variational and other types of inequalities involving nonlinear operators (general) [See also 49J40]

**47J22** Variational and other types of inclusions [See also 34A60, 49J21, 49K21]

**47J25** Iterative procedures involving nonlinear operators [See also 47J26, 65J15]

**47J26** Fixed-point iterations [See also 47J25]

**47J30** Variational methods involving nonlinear operators [See also 58Exx]

**47J35** Nonlinear evolution equations [See also 34G20, 35K90, 35L90, 35Qxx, 35R20, 37Kxx, 37Lxx, 47H20, 58D25]

**47J40** Equations with nonlinear hysteresis operators [See also 34C55, 74N30]

**47J99** None of the above, but in this section

## 47Lxx Linear spaces and algebras of operators [See also 46Lxx]

**47L05** Linear spaces of operators [See also 46A32, 46B28]

**47L07** Convex sets and cones of operators [See also 46A55]

**47L10** Algebras of operators on Banach spaces and other topological linear spaces

**47L15** Operator algebras with symbol structure

**47L20** Operator ideals [See also 47B10]

**47L22** Ideals of polynomials and of multilinear mappings in operator theory

**47L25** Operator spaces (= matricially normed spaces) [See also 46L07]

**47L30** Abstract operator algebras on Hilbert spaces

**47L35** Nest algebras, CSL algebras

**47L40** Limit algebras, subalgebras of $C^*$-algebras

**47L45** Dual algebras; weakly closed singly generated operator algebras

**47L50** Dual spaces of operator algebras

**47L55** Representations of (nonselfadjoint) operator algebras

**47L60** Algebras of unbounded operators; partial algebras of operators

**47L65** Crossed product algebras (analytic crossed products)

**47L70** Nonassociative nonselfadjoint operator algebras

**47L75** Other nonselfadjoint operator algebras

**47L80** Algebras of specific types of operators (Toeplitz, integral, pseudodifferential, etc.)

**47L90** Applications of operator algebras to the sciences

**47L99** None of the above, but in this section

## 47Nxx Miscellaneous applications of operator theory [See also 46Nxx]

**47N10** Applications of operator theory in optimization, convex analysis, mathematical programming, economics

**47N20** Applications of operator theory to differential and integral equations

**47N30** Applications of operator theory in probability theory and statistics

**47N40** Applications of operator theory in numerical analysis [See also 65Jxx]

**47N50** Applications of operator theory in the physical sciences

**47N60** Applications of operator theory in chemistry and life sciences

**47N70** Applications of operator theory in systems, signals, circuits, and control theory

**47N99** None of the above, but in this section

## 47Sxx Other (nonclassical) types of operator theory [See also 46Sxx]

**47S05** Quaternionic operator theory

**47S10** Operator theory over fields other than $\mathbb{R}$, $\mathbb{C}$ or the quaternions; non-Archimedean operator theory

**47S20** Nonstandard operator theory [See also 03H05]

**47S30** Constructive operator theory [See also 03F60]

**47S40** Fuzzy operator theory [See also 03E72]

**47S50** Operator theory in probabilistic metric linear spaces [See also 54E70]

**47S99** None of the above, but in this section

# 49-XX Calculus of variations and optimal control; optimization [See also 34H05, 34K35, 65Kxx, 90Cxx, 93-XX]

**49-00** General reference works (handbooks, dictionaries, bibliographies, etc.) pertaining to calculus of variations and optimal control

**49-01** Introductory exposition (textbooks, tutorial papers, etc.) pertaining to calculus of variations and optimal control

**49-02** Research exposition (monographs, survey articles) pertaining to calculus of variations and optimal control

**49-03** History of calculus of variations and optimal control [Consider also classification numbers from Section 01]

**49-04** Software, source code, etc. for problems pertaining to calculus of variations and optimal control

**49-06** Proceedings, conferences, collections, etc. pertaining to calculus of variations and optimal control

**49-11** Research data for problems pertaining to calculus of variations and optimal control

## 49Jxx Existence theories in calculus of variations and optimal control

**49J05** Existence theories for free problems in one independent variable

**49J10** Existence theories for free problems in two or more independent variables

**49J15** Existence theories for optimal control problems involving ordinary differential equations

**49J20** Existence theories for optimal control problems involving partial differential equations

**49J21** Existence theories for optimal control problems involving relations other than differential equations

**49J27** Existence theories for problems in abstract spaces [See also 90C48, 93C25]

**49J30** Existence of optimal solutions belonging to restricted classes (Lipschitz controls, bang-bang controls, etc.)

**49J35** Existence of solutions for minimax problems

**49J40** Variational inequalities [See also 47J20]

**49J45** Methods involving semicontinuity and convergence; relaxation

**49J50** Fréchet and Gateaux differentiability in optimization [See also 46G05, 58C20]

**49J52** Nonsmooth analysis [See also 46G05, 58C50, 90C56]

**49J53** Set-valued and variational analysis [See also 28B20, 47H04, 54C60, 58C06]

**49J55** Existence of optimal solutions to problems involving randomness [See also 93E20]

**49J99** None of the above, but in this section

## 49Kxx Optimality conditions

**49K05** Optimality conditions for free problems in one independent variable

**49K10** Optimality conditions for free problems in two or more independent variables

**49K15** Optimality conditions for problems involving ordinary differential equations

**49K20** Optimality conditions for problems involving partial differential equations

**49K21** Optimality conditions for problems involving relations other than differential equations

**49K27** Optimality conditions for problems in abstract spaces [See also 90C48, 93C25]

**49K30** Optimality conditions for solutions belonging to restricted classes (Lipschitz controls, bang-bang controls, etc.)

**49K35** Optimality conditions for minimax problems

**49K40** Sensitivity, stability, well-posedness [See also 90C31]

**49K45** Optimality conditions for problems involving randomness [See also 93E20]

**49K99** None of the above, but in this section

## 49Lxx Hamilton-Jacobi theories [See also 70H20, 35F21]

**49L12** Hamilton-Jacobi equations in optimal control and differential games

**49L20** Dynamic programming in optimal control and differential games

**49L25** Viscosity solutions to Hamilton-Jacobi equations in optimal control and differential games

**49L99** None of the above, but in this section

## 49Mxx Numerical methods in optimal control [See also 65Kxx, 90-08, 90Cxx]

**49M05** Numerical methods based on necessary conditions

**49M15** Newton-type methods [See also 90C53]

**49M20** Numerical methods of relaxation type

**49M25** Discrete approximations in optimal control

**49M27** Decomposition methods

**49M29** Numerical methods involving duality

**49M37** Numerical methods based on nonlinear programming [See also 65Kxx, 90C30]

**49M41** PDE constrained optimization (numerical aspects)

**49M99** None of the above, but in this section

## 49Nxx Miscellaneous topics in calculus of variations and optimal control

**49N05** Linear optimal control problems [See also 93C05]

**49N10** Linear-quadratic optimal control problems

**49N15** Duality theory (optimization) [See also 90C46]

**49N20** Periodic optimal control problems

**49N25** Impulsive optimal control problems

**49N30** Problems with incomplete information (optimization) [See also 93C41]

**49N35** Optimal feedback synthesis [See also 93B52]

**49N45** Inverse problems in optimal control

**49N60** Regularity of solutions in optimal control

**49N70** Differential games and control [See also 91A23]

**49N75** Pursuit and evasion games [See also 91A24]

**49N80** Mean field games and control {For partial differential equations, see 35Q89; for game theory, see 91A16}

**49N90** Applications of optimal control and differential games [See also 90C90, 91A80, 93C95]

**49N99** None of the above, but in this section

## 49Qxx Manifolds and measure-geometric topics [See also 58Exx]

**49Q05** Minimal surfaces and optimization [See also 53A10, 58E12]

**49Q10** Optimization of shapes other than minimal surfaces [See also 90C90]

**49Q12** Sensitivity analysis for optimization problems on manifolds

**49Q15** Geometric measure and integration theory, integral and normal currents in optimization [See also 28A75, 32C30, 58A25, 58C35]

**49Q20** Variational problems in a geometric measure-theoretic setting

**49Q22** Optimal transportation [See also 90B06]

**49Q99** None of the above, but in this section

**49Rxx Variational methods for eigenvalues of operators [Should also be assigned at least one other classification number in Section 49] [See also 47A75]**

**49R05** Variational methods for eigenvalues of operators [Should also be assigned at least one other classification number in Section 49] [See also 47A75]

**49R99** None of the above, but in this section

**49Sxx Variational principles of physics [Should also be assigned at least one other classification number in Section 49]**

**49S05** Variational principles of physics [Should also be assigned at least one other classification number in Section 49]

**49S99** None of the above, but in this section

# 51-XX Geometry {For algebraic geometry, see 14-XX; for differential geometry, see 53-XX}

**51-00** General reference works (handbooks, dictionaries, bibliographies, etc.) pertaining to geometry

**51-01** Introductory exposition (textbooks, tutorial papers, etc.) pertaining to geometry

**51-02** Research exposition (monographs, survey articles) pertaining to geometry

**51-03** History of geometry [Consider also classification numbers from Section 01]

**51-04** Software, source code, etc. for problems pertaining to geometry

**51-06** Proceedings, conferences, collections, etc. pertaining to geometry

**51-08** Computational methods for problems pertaining to geometry

**51-11** Research data for problems pertaining to geometry

## 51Axx Linear incidence geometry

**51A05** General theory of linear incidence geometry and projective geometries

**51A10** Homomorphism, automorphism and dualities in linear incidence geometry

**51A15** Linear incidence geometric structures with parallelism

**51A20** Configuration theorems in linear incidence geometry

**51A25** Algebraization in linear incidence geometry [See also 12Kxx, 20N05]

**51A30** Desarguesian and Pappian geometries

**51A35** Non-Desarguesian affine and projective planes

**51A40** Translation planes and spreads in linear incidence geometry

**51A45** Incidence structures embeddable into projective geometries

**51A50** Polar geometry, symplectic spaces, orthogonal spaces

**51A99** None of the above, but in this section

## 51Bxx Nonlinear incidence geometry

**51B05** General theory of nonlinear incidence geometry

**51B10** Möbius geometries

**51B15** Laguerre geometries

**51B20** Minkowski geometries in nonlinear incidence geometry

**51B25** Lie geometries in nonlinear incidence geometry

**51B99** None of the above, but in this section

## 51Cxx Ring geometry (Hjelmslev, Barbilian, etc.)

**51C05** Ring geometry (Hjelmslev, Barbilian, etc.)

**51C99** None of the above, but in this section

## 51Dxx Geometric closure systems

**51D05** Abstract (Maeda) geometries

**51D10** Abstract geometries with exchange axiom

**51D15** Abstract geometries with parallelism

**51D20** Combinatorial geometries and geometric closure systems [See also 05B25, 05B35]

**51D25** Lattices of subspaces and geometric closure systems [See also 05B35]

**51D30** Continuous geometries, geometric closure systems and related topics [See also 06Cxx]

**51D99** None of the above, but in this section

## 51Exx Finite geometry and special incidence structures

**51E05** General block designs in finite geometry [See also 05B05]

**51E10** Steiner systems in finite geometry [See also 05B05]

**51E12** Generalized quadrangles and generalized polygons in finite geometry

**51E14** Finite partial geometries (general), nets, partial spreads

**51E15** Finite affine and projective planes (geometric aspects)

**51E20** Combinatorial structures in finite projective spaces [See also 05Bxx]

**51E21** Blocking sets, ovals, $k$-arcs

**51E22** Linear codes and caps in Galois spaces [See also 94B05]

**51E23** Spreads and packing problems in finite geometry

**51E24** Buildings and the geometry of diagrams

**51E25** Other finite nonlinear geometries

**51E26** Other finite linear geometries

**51E30** Other finite incidence structures (geometric aspects) [See also 05B30]

**51E99** None of the above, but in this section

## 51Fxx Metric geometry

**51F05** Absolute planes in metric geometry

**51F10** Absolute spaces in metric geometry

**51F15** Reflection groups, reflection geometries [See also 20H10, 20H15] {For Coxeter groups, see 20F55}

**51F20** Congruence and orthogonality in metric geometry [See also 20H05]

**51F25** Orthogonal and unitary groups in metric geometry [See also 20H05]

**51F30** Lipschitz and coarse geometry of metric spaces [See also 53C23]

**51F99** None of the above, but in this section

## 51Gxx Ordered geometries (ordered incidence structures, etc.)

**51G05** Ordered geometries (ordered incidence structures, etc.)

**51G99** None of the above, but in this section

## 51Hxx Topological geometry

**51H05** General theory of topological geometry

**51H10** Topological linear incidence structures

**51H15** Topological nonlinear incidence structures

**51H20** Topological geometries on manifolds [See also 57-XX]

**51H25** Geometries with differentiable structure [See also 53Cxx, especially 53C70]

**51H30** Geometries with algebraic manifold structure [See also 14-XX]

**51H99** None of the above, but in this section

## 51Jxx Incidence groups

**51J05** General theory of incidence groups

**51J10** Projective incidence groups

**51J15** Kinematic spaces

**51J20** Representation by near-fields and near-algebras [See also 12K05, 16Y30]

**51J99** None of the above, but in this section

## 51Kxx Distance geometry

**51K05** General theory of distance geometry

**51K10** Synthetic differential geometry

**51K99** None of the above, but in this section

## 51Lxx Geometric order structures [See also 53C75]

**51L05** Geometry of orders of nondifferentiable curves

**51L10** Directly differentiable curves in geometric order structures

**51L15** $n$-vertex theorems via direct methods

**51L20** Geometry of orders of surfaces

**51L99** None of the above, but in this section

## 51Mxx Real and complex geometry

**51M04** Elementary problems in Euclidean geometries

**51M05** Euclidean geometries (general) and generalizations

**51M09** Elementary problems in hyperbolic and elliptic geometries

**51M10** Hyperbolic and elliptic geometries (general) and generalizations

**51M15** Geometric constructions in real or complex geometry

**51M16** Inequalities and extremum problems in real or complex geometry {For convex problems, see 52A40}

**51M20** Polyhedra and polytopes; regular figures, division of spaces [See also 51F15]

**51M25** Length, area and volume in real or complex geometry [See also 26B15]

**51M30** Line geometries and their generalizations [See also 53A25]

**51M35** Synthetic treatment of fundamental manifolds in projective geometries (Grassmannians, Veronesians and their generalizations) [See also 14M15]

**51M99** None of the above, but in this section

## 51Nxx Analytic and descriptive geometry

**51N05** Descriptive geometry [See also 65D17, 68U07]

**51N10** Affine analytic geometry

**51N15** Projective analytic geometry

**51N20** Euclidean analytic geometry

**51N25** Analytic geometry with other transformation groups

**51N30** Geometry of classical groups [See also 14L35, 20Gxx]

**51N35** Questions of classical algebraic geometry [See also 14Nxx]

**51N99** None of the above, but in this section

## 51Pxx Classical or axiomatic geometry and physics [Should also be assigned at least one other classification number from Sections 70 through 86]

**51P05** Classical or axiomatic geometry and physics [Should also be assigned at least one other classification number from Sections 70 through 86]

**51P99** None of the above, but in this section

# 52-XX Convex and discrete geometry

**52-00** General reference works (handbooks, dictionaries, bibliographies, etc.) pertaining to convex and discrete geometry

**52-01** Introductory exposition (textbooks, tutorial papers, etc.) pertaining to convex and discrete geometry

**52-02** Research exposition (monographs, survey articles) pertaining to convex and discrete geometry

**52-03** History of convex and discrete geometry [Consider also classification numbers from Section 01]

**52-04** Software, source code, etc. for problems pertaining to convex and discrete geometry

**52-06** Proceedings, conferences, collections, etc. pertaining to convex and discrete geometry

**52-08** Computational methods for problems pertaining to convex and discrete geometry

**52-11** Research data for problems pertaining to convex and discrete geometry

## 52Axx General convexity

**52A01** Axiomatic and generalized convexity

**52A05** Convex sets without dimension restrictions (aspects of convex geometry)

**52A07** Convex sets in topological vector spaces (aspects of convex geometry) [See also 46A55]

**52A10** Convex sets in 2 dimensions (including convex curves) [See also 53A04]

**52A15** Convex sets in 3 dimensions (including convex surfaces) [See also 53A05, 53C45]

**52A20** Convex sets in $n$ dimensions (including convex hypersurfaces) [See also 53A07, 53C45]

**52A21** Convexity and finite-dimensional Banach spaces (including special norms, zonoids, etc.) (aspects of convex geometry) [See also 46Bxx]

**52A22** Random convex sets and integral geometry (aspects of convex geometry) [See also 53C65, 60D05]

**52A23** Asymptotic theory of convex bodies [See also 46B06]

**52A27** Approximation by convex sets

**52A30** Variants of convex sets (star-shaped, $(m, n)$-convex, etc.)

**52A35** Helly-type theorems and geometric transversal theory

**52A37** Other problems of combinatorial convexity

**52A38** Length, area, volume and convex sets (aspects of convex geometry) [See also 26B15, 28A75, 49Q20]

**52A39** Mixed volumes and related topics in convex geometry

**52A40** Inequalities and extremum problems involving convexity in convex geometry

**52A41** Convex functions and convex programs in convex geometry [See also 26B25, 90C25]

**52A55** Spherical and hyperbolic convexity

**52A99** None of the above, but in this section

## 52Bxx Polytopes and polyhedra

**52B05** Combinatorial properties of polytopes and polyhedra (number of faces, shortest paths, etc.) [See also 05Cxx]

**52B10** Three-dimensional polytopes

**52B11** $n$-dimensional polytopes

**52B12** Special polytopes (linear programming, centrally symmetric, etc.)

**52B15** Symmetry properties of polytopes

**52B20** Lattice polytopes in convex geometry (including relations with commutative algebra and algebraic geometry) [See also 06A11, 13F20, 13F55, 13Hxx, 52C05, 52C07]

**52B22** Shellability for polytopes and polyhedra

**52B35** Gale and other diagrams

**52B40** Matroids in convex geometry (realizations in the context of convex polytopes, convexity in combinatorial structures, etc.) [See also 05B35, 52Cxx]

**52B45** Dissections and valuations (Hilbert's third problem, etc.)

**52B55** Computational aspects related to convexity {For computational methods, see 52-08; for computational geometry and algorithms, see 68Q25, 68U05; for numerical algorithms, see 65Yxx} [See also 68Uxx]

**52B60** Isoperimetric problems for polytopes

**52B70** Polyhedral manifolds

**52B99** None of the above, but in this section

## 52Cxx Discrete geometry

**52C05** Lattices and convex bodies in 2 dimensions (aspects of discrete geometry) [See also 11H06, 11H31, 11P21]

**52C07** Lattices and convex bodies in $n$ dimensions (aspects of discrete geometry) [See also 11H06, 11H31, 11P21]

**52C10** Erdős problems and related topics of discrete geometry [See also 11Hxx]

**52C15** Packing and covering in 2 dimensions (aspects of discrete geometry) [See also 05B40, 11H31]

**52C17** Packing and covering in $n$ dimensions (aspects of discrete geometry) [See also 05B40, 11H31]

**52C20** Tilings in 2 dimensions (aspects of discrete geometry) [See also 05B45, 51M20]

**52C22** Tilings in $n$ dimensions (aspects of discrete geometry) [See also 05B45, 51M20]

**52C23** Quasicrystals and aperiodic tilings in discrete geometry

**52C25** Rigidity and flexibility of structures (aspects of discrete geometry) [See also 70B15]

**52C26** Circle packings and discrete conformal geometry

**52C30** Planar arrangements of lines and pseudolines (aspects of discrete geometry)

**52C35** Arrangements of points, flats, hyperplanes (aspects of discrete geometry) [See also 14N20, 32S22]

**52C40** Oriented matroids in discrete geometry

**52C45** Combinatorial complexity of geometric structures [See also 68U05]

**52C99** None of the above, but in this section

# 53-XX Differential geometry {For differential topology, see 57Rxx; for foundational questions of differentiable manifolds, see 58Axx}

**53-00** General reference works (handbooks, dictionaries, bibliographies, etc.) pertaining to differential geometry

**53-01** Introductory exposition (textbooks, tutorial papers, etc.) pertaining to differential geometry

**53-02** Research exposition (monographs, survey articles) pertaining to differential geometry

**53-03** History of differential geometry [Consider also classification numbers from Section 01]

**53-04** Software, source code, etc. for problems pertaining to differential geometry

**53-06** Proceedings, conferences, collections, etc. pertaining to differential geometry

**53-08** Computational methods for problems pertaining to differential geometry

**53-11** Research data for problems pertaining to differential geometry

## 53Axx Classical differential geometry

**53A04** Curves in Euclidean and related spaces

**53A05** Surfaces in Euclidean and related spaces

**53A07** Higher-dimensional and -codimensional surfaces in Euclidean and related $n$-spaces

**53A10** Minimal surfaces in differential geometry, surfaces with prescribed mean curvature [See also 49Q05, 49Q10, 53C42]

**53A15** Affine differential geometry

**53A17** Differential geometric aspects in kinematics

**53A20** Projective differential geometry

**53A25** Differential line geometry

**53A31** Differential geometry of submanifolds of Möbius space

**53A35** Non-Euclidean differential geometry

**53A40** Other special differential geometries

**53A45** Differential geometric aspects in vector and tensor analysis

**53A55** Differential invariants (local theory), geometric objects

**53A60** Differential geometry of webs [See also 14C21, 20N05]

**53A70** Discrete differential geometry

**53A99** None of the above, but in this section

## 53Bxx Local differential geometry

**53B05** Linear and affine connections

**53B10** Projective connections

**53B12** Differential geometric aspects of statistical manifolds and information geometry

**53B15** Other connections

**53B20** Local Riemannian geometry

**53B21** Methods of local Riemannian geometry

**53B25** Local submanifolds [See also 53C40]

**53B30** Local differential geometry of Lorentz metrics, indefinite metrics

**53B35** Local differential geometry of Hermitian and Kählerian structures [See also 32Qxx]

**53B40** Local differential geometry of Finsler spaces and generalizations (areal metrics)

**53B50** Applications of local differential geometry to the sciences

**53B99** None of the above, but in this section

## 53Cxx Global differential geometry [See also 51H25, 58-XX] {For related bundle theory, see 55Rxx, 57Rxx}

**53C05** Connections (general theory)

**53C07** Special connections and metrics on vector bundles (Hermite-Einstein, Yang-Mills) [See also 32Q20]

**53C08** Differential geometric aspects of gerbes and differential characters

**53C10** $G$-structures

**53C12** Foliations (differential geometric aspects) [See also 57R30, 57R32]

**53C15** General geometric structures on manifolds (almost complex, almost product structures, etc.)

**53C17** Sub-Riemannian geometry

**53C18** Conformal structures on manifolds

**53C20** Global Riemannian geometry, including pinching [See also 31C12, 58B20]

**53C21** Methods of global Riemannian geometry, including PDE methods; curvature restrictions [See also 58J60]

**53C22** Geodesics in global differential geometry [See also 58E10]

**53C23** Global geometric and topological methods (à la Gromov); differential geometric analysis on metric spaces

**53C24** Rigidity results

**53C25** Special Riemannian manifolds (Einstein, Sasakian, etc.)

**53C26** Hyper-Kähler and quaternionic Kähler geometry, "special" geometry

**53C27** Spin and Spin$^c$ geometry

**53C28** Twistor methods in differential geometry [See also 32L25]

**53C29** Issues of holonomy in differential geometry

**53C30** Differential geometry of homogeneous manifolds [See also 14M15, 14M17, 32M10, 57T15]

**53C35** Differential geometry of symmetric spaces [See also 32M15, 57T15]

**53C38** Calibrations and calibrated geometries

**53C40** Global submanifolds [See also 53B25]

**53C42** Differential geometry of immersions (minimal, prescribed curvature, tight, etc.) [See also 49Q05, 49Q10, 53A10, 57R40, 57R42]

**53C43** Differential geometric aspects of harmonic maps [See also 58E20]

**53C45** Global surface theory (convex surfaces à la A. D. Aleksandrov)

**53C50** Global differential geometry of Lorentz manifolds, manifolds with indefinite metrics

**53C55** Global differential geometry of Hermitian and Kählerian manifolds [See also 32Qxx]

**53C56** Other complex differential geometry [See also 32Qxx]

**53C60** Global differential geometry of Finsler spaces and generalizations (areal metrics) [See also 58B20]

**53C65** Integral geometry [See also 52A22, 60D05]; differential forms, currents, etc. [See mainly 58Axx]

**53C70** Direct methods (*G*-spaces of Busemann, etc.)

**53C75** Geometric orders, order geometry [See also 51Lxx]

**53C80** Applications of global differential geometry to the sciences

**53C99** None of the above, but in this section

## 53Dxx Symplectic geometry, contact geometry [See also 37Jxx, 70Gxx, 70Hxx]

**53D05** Symplectic manifolds (general theory)

**53D10** Contact manifolds (general theory)

**53D12** Lagrangian submanifolds; Maslov index

**53D15** Almost contact and almost symplectic manifolds

**53D17** Poisson manifolds; Poisson groupoids and algebroids

**53D18** Generalized geometries (à la Hitchin)

**53D20** Momentum maps; symplectic reduction

**53D22** Canonical transformations in symplectic and contact geometry

**53D25** Geodesic flows in symplectic geometry and contact geometry

**53D30** Symplectic structures of moduli spaces

**53D35** Global theory of symplectic and contact manifolds [See also 57Rxx]

**53D37** Symplectic aspects of mirror symmetry, homological mirror symmetry, and Fukaya category [See also 14J33]

**53D40** Symplectic aspects of Floer homology and cohomology

**53D42** Symplectic field theory; contact homology

**53D45** Gromov-Witten invariants, quantum cohomology, Frobenius manifolds [See also 14N35]

**53D50** Geometric quantization

**53D55** Deformation quantization, star products

**53D99** None of the above, but in this section

## 53Exx Geometric evolution equations

**53E10** Flows related to mean curvature

**53E20** Ricci flows

**53E30** Flows related to complex manifolds (e.g., Kähler-Ricci flows, Chern-Ricci flows)

**53E40** Higher-order geometric flows

**53E50** Flows related to symplectic and contact structures

**53E99** None of the above, but in this section

## 53Zxx Applications of differential geometry to sciences and engineering

**53Z05** Applications of differential geometry to physics

**53Z10** Applications of differential geometry to biology

**53Z15** Applications of differential geometry to chemistry

**53Z30** Applications of differential geometry to engineering

**53Z50** Applications of differential geometry to data and computer science

**53Z99** None of the above, but in this section

# 54-XX General topology {For the topology of manifolds of all dimensions, see 57Nxx}

**54-00** General reference works (handbooks, dictionaries, bibliographies, etc.) pertaining to general topology

**54-01** Introductory exposition (textbooks, tutorial papers, etc.) pertaining to general topology

**54-02** Research exposition (monographs, survey articles) pertaining to general topology

**54-03** History of general topology [Consider also classification numbers from Section 01]

**54-04** Software, source code, etc. for problems pertaining to general topology

**54-06** Proceedings, conferences, collections, etc. pertaining to general topology

**54-08** Computational methods for problems pertaining to general topology

**54-11** Research data for problems pertaining to general topology

## 54Axx Generalities in topology

**54A05** Topological spaces and generalizations (closure spaces, etc.)

**54A10** Several topologies on one set (change of topology, comparison of topologies, lattices of topologies)

**54A15** Syntopogeneous structures

**54A20** Convergence in general topology (sequences, filters, limits, convergence spaces, nets, etc.)

**54A25** Cardinality properties (cardinal functions and inequalities, discrete subsets) [See also 03Exx] {For ultrafilters, see 54D80}

**54A35** Consistency and independence results in general topology [See also 03E35]

**54A40** Fuzzy topology [See also 03E72]

**54A99** None of the above, but in this section

## 54Bxx Basic constructions in general topology

**54B05** Subspaces in general topology

**54B10** Product spaces in general topology

**54B15** Quotient spaces, decompositions in general topology

**54B17** Adjunction spaces and similar constructions in general topology

**54B20** Hyperspaces in general topology

**54B30** Categorical methods in general topology [See also 18F60]

**54B35** Spectra in general topology

**54B40** Presheaves and sheaves in general topology [See also 18F20]

**54B99** None of the above, but in this section

## 54Cxx Maps and general types of topological spaces defined by maps

**54C05** Continuous maps

**54C08** Weak and generalized continuity

**54C10** Special maps on topological spaces (open, closed, perfect, etc.)

**54C15** Retraction

**54C20** Extension of maps

**54C25** Embedding

**54C30** Real-valued functions in general topology [See also 26-XX]

**54C35** Function spaces in general topology [See also 46Exx, 58D15]

**54C40** Algebraic properties of function spaces in general topology [See also 46Exx]

**54C45** $C$- and $C^*$-embedding

**54C50** Topology of special sets defined by functions [See also 26A21]

**54C55** Absolute neighborhood extensor, absolute extensor, absolute neighborhood retract (ANR), absolute retract spaces (general properties) [See also 55M15]

**54C56** Shape theory in general topology [See also 55P55, 57N25]

**54C60** Set-valued maps in general topology [See also 26E25, 28B20, 47H04, 58C06]

**54C65** Selections in general topology [See also 28B20]

**54C70** Entropy in general topology

**54C99** None of the above, but in this section

## 54Dxx Fairly general properties of topological spaces

**54D05** Connected and locally connected spaces (general aspects)

**54D10** Lower separation axioms ($T_0$–$T_3$, etc.)

**54D15** Higher separation axioms (completely regular, normal, perfectly or collectionwise normal, etc.)

**54D20** Noncompact covering properties (paracompact, Lindelöf, etc.)

**54D25** "$P$-minimal" and "$P$-closed" spaces

**54D30** Compactness

**54D35** Extensions of spaces (compactifications, supercompactifications, completions, etc.)

**54D40** Remainders in general topology

**54D45** Local compactness, $\sigma$-compactness

**54D50** $k$-spaces

**54D55** Sequential spaces

**54D60** Realcompactness and realcompactification

**54D65** Separability of topological spaces

**54D70** Base properties of topological spaces

**54D80** Special constructions of topological spaces (spaces of ultrafilters, etc.)

**54D99** None of the above, but in this section

## 54Exx Topological spaces with richer structures

**54E05** Proximity structures and generalizations

**54E15** Uniform structures and generalizations

**54E17** Nearness spaces

**54E18** $p$-spaces, $M$-spaces, $\sigma$-spaces, etc.

**54E20** Stratifiable spaces, cosmic spaces, etc.

**54E25** Semimetric spaces

**54E30** Moore spaces

**54E35** Metric spaces, metrizability

**54E40** Special maps on metric spaces

**54E45** Compact (locally compact) metric spaces

**54E50** Complete metric spaces

**54E52** Baire category, Baire spaces

**54E55** Bitopologies

**54E70** Probabilistic metric spaces

**54E99** None of the above, but in this section

## 54Fxx Special properties of topological spaces

**54F05** Linearly ordered topological spaces, generalized ordered spaces, and partially ordered spaces [See also 06B30, 06F30]

**54F15** Continua and generalizations

**54F16** Hyperspaces of continua

**54F17** Inverse limits of set-valued functions

**54F35** Higher-dimensional local connectedness [See also 55Mxx, 55Nxx]

**54F45** Dimension theory in general topology [See also 55M10]

**54F50** Topological spaces of dimension $\leq 1$; curves, dendrites [See also 26A03]

**54F55** Unicoherence, multicoherence

**54F65** Topological characterizations of particular spaces

**54F99** None of the above, but in this section

## 54Gxx Peculiar topological spaces

**54G05** Extremally disconnected spaces, $F$-spaces, etc.

**54G10** $P$-spaces

**54G12** Scattered spaces

**54G15** Pathological topological spaces

**54G20** Counterexamples in general topology

**54G99** None of the above, but in this section

## 54Hxx Connections of general topology with other structures, applications

**54H05** Descriptive set theory (topological aspects of Borel, analytic, projective, etc. sets) [See also 03E15, 26A21, 28A05]

**54H10** Topological representations of algebraic systems [See also 22-XX]

**54H11** Topological groups (topological aspects) [See also 22A05]

**54H12** Topological lattices, etc. (topological aspects) [See also 06B30, 06F30]

**54H13** Topological fields, rings, etc. (topological aspects) [See also 12Jxx] {For algebraic aspects, see 13Jxx, 16W80}

**54H15** Transformation groups and semigroups (topological aspects) [See also 20M20, 22-XX, 57Sxx]

**54H25** Fixed-point and coincidence theorems (topological aspects) [See also 47H10, 55M20]

**54H30** Applications of general topology to computer science (e.g., digital topology, image processing) [See also 68U03]

**54H99** None of the above, but in this section

## 54Jxx Nonstandard topology [See also 03H05]

**54J05** Nonstandard topology [See also 03H05]

**54J99** None of the above, but in this section

# 55-XX Algebraic topology

**55-00** General reference works (handbooks, dictionaries, bibliographies, etc.) pertaining to algebraic topology

**55-01** Introductory exposition (textbooks, tutorial papers, etc.) pertaining to algebraic topology

**55-02** Research exposition (monographs, survey articles) pertaining to algebraic topology

**55-03** History of algebraic topology [Consider also classification numbers from Section 01]

**55-04** Software, source code, etc. for problems pertaining to algebraic topology

**55-06** Proceedings, conferences, collections, etc. pertaining to algebraic topology

**55-08** Computational methods for problems pertaining to algebraic topology

**55-11** Research data for problems pertaining to algebraic topology

## 55Mxx Classical topics in algebraic topology {For the topology of Euclidean spaces and manifolds, see 57Nxx}

**55M05** Duality in algebraic topology

**55M10** Dimension theory in algebraic topology [See also 54F45]

**55M15** Absolute neighborhood retracts [See also 54C55]

**55M20** Fixed points and coincidences in algebraic topology [See also 54H25]

**55M25** Degree, winding number

**55M30** Lyusternik-Shnirel'man category of a space, topological complexity à la Farber, topological robotics (topological aspects)

**55M35** Finite groups of transformations in algebraic topology (including Smith theory) [See also 57S17]

**55M99** None of the above, but in this section

## 55Nxx Homology and cohomology theories in algebraic topology {For homology and cohomology of topological groups and related structures, see 57Txx}

**55N05** Čech types

**55N07** Steenrod-Sitnikov homologies

**55N10** Singular homology and cohomology theory

**55N15** Topological $K$-theory [See also 19Lxx] {For algebraic $K$-theory, see 18F25, 19-XX}

**55N20** Generalized (extraordinary) homology and cohomology theories in algebraic topology

**55N22** Bordism and cobordism theories and formal group laws in algebraic topology [See also 14L05, 19L41, 57R75, 57R77, 57R85, 57R90]

**55N25** Homology with local coefficients, equivariant cohomology

**55N30** Sheaf cohomology in algebraic topology [See also 18F20, 32C35, 32L10]

**55N31** Persistent homology and applications, topological data analysis [See also 62R40, 68T09]

**55N32** Orbifold cohomology

**55N33** Intersection homology and cohomology in algebraic topology

**55N34** Elliptic cohomology

**55N35** Other homology theories in algebraic topology

**55N40** Axioms for homology theory and uniqueness theorems in algebraic topology

**55N45** Products and intersections in homology and cohomology

**55N91** Equivariant homology and cohomology in algebraic topology [See also 19L47]

**55N99** None of the above, but in this section

## 55Pxx Homotopy theory {For simple homotopy type, see 57Q10}

**55P05** Homotopy extension properties, cofibrations in algebraic topology

**55P10** Homotopy equivalences in algebraic topology

**55P15** Classification of homotopy type

**55P20** Eilenberg-Mac Lane spaces

**55P25** Spanier-Whitehead duality

**55P30** Eckmann-Hilton duality

**55P35** Loop spaces

**55P40** Suspensions

**55P42** Stable homotopy theory, spectra

**55P43** Spectra with additional structure ($E_\infty$, $A_\infty$, ring spectra, etc.)

**55P45** $H$-spaces and duals

**55P47** Infinite loop spaces

**55P48** Loop space machines and operads in algebraic topology [See also 18Mxx]

**55P50** String topology

**55P55** Shape theory [See also 54C56, 55Q07]

**55P57** Proper homotopy theory

**55P60** Localization and completion in homotopy theory

**55P62** Rational homotopy theory

**55P65** Homotopy functors in algebraic topology

**55P91** Equivariant homotopy theory in algebraic topology [See also 19L47]

**55P92** Relations between equivariant and nonequivariant homotopy theory in algebraic topology

**55P99** None of the above, but in this section

## 55Qxx Homotopy groups

**55Q05** Homotopy groups, general; sets of homotopy classes

**55Q07** Shape groups

**55Q10** Stable homotopy groups

**55Q15** Whitehead products and generalizations

**55Q20** Homotopy groups of wedges, joins, and simple spaces

**55Q25** Hopf invariants

**55Q35** Operations in homotopy groups

**55Q40** Homotopy groups of spheres

**55Q45** Stable homotopy of spheres

**55Q50** $J$-morphism [See also 19L20]

**55Q51** $v_n$-periodicity

**55Q52** Homotopy groups of special spaces

**55Q55** Cohomotopy groups

**55Q70** Homotopy groups of special types [See also 55N05, 55N07]

**55Q91** Equivariant homotopy groups [See also 19L47]

**55Q99** None of the above, but in this section

## 55Rxx Fiber spaces and bundles in algebraic topology [See also 18F15, 32Lxx, 46M20, 57R20, 57R22, 57R25]

**55R05** Fiber spaces in algebraic topology

**55R10** Fiber bundles in algebraic topology

**55R12** Transfer for fiber spaces and bundles in algebraic topology

**55R15** Classification of fiber spaces or bundles in algebraic topology

**55R20** Spectral sequences and homology of fiber spaces in algebraic topology [See also 55Txx]

**55R25** Sphere bundles and vector bundles in algebraic topology

**55R35** Classifying spaces of groups and $H$-spaces in algebraic topology

**55R37** Maps between classifying spaces in algebraic topology

**55R40** Homology of classifying spaces and characteristic classes in algebraic topology [See also 57Txx, 57R20]

**55R45** Homology and homotopy of $B$O and $B$U; Bott periodicity

**55R50** Stable classes of vector space bundles in algebraic topology and relations to $K$-theory [See also 19Lxx] {For algebraic $K$-theory, see 18F25, 19-XX}

**55R55** Fiberings with singularities in algebraic topology

**55R60** Microbundles and block bundles in algebraic topology [See also 57N55, 57Q50]

**55R65** Generalizations of fiber spaces and bundles in algebraic topology

**55R70** Fibrewise topology

**55R80** Discriminantal varieties and configuration spaces in algebraic topology

**55R91** Equivariant fiber spaces and bundles in algebraic topology [See also 19L47]

**55R99** None of the above, but in this section

## 55Sxx Operations and obstructions in algebraic topology

**55S05** Primary cohomology operations in algebraic topology

**55S10** Steenrod algebra

**55S12** Dyer-Lashof operations

**55S15** Symmetric products and cyclic products in algebraic topology

**55S20** Secondary and higher cohomology operations in algebraic topology

**55S25** $K$-theory operations and generalized cohomology operations in algebraic topology [See also 19D55, 19Lxx]

**55S30** Massey products

**55S35** Obstruction theory in algebraic topology

**55S36** Extension and compression of mappings in algebraic topology

**55S37** Classification of mappings in algebraic topology

**55S40** Sectioning fiber spaces and bundles in algebraic topology

**55S45** Postnikov systems, $k$-invariants

**55S91** Equivariant operations and obstructions in algebraic topology [See also 19L47]

**55S99** None of the above, but in this section

## 55Txx Spectral sequences in algebraic topology [See also 18G40, 55R20]

**55T05** General theory of spectral sequences in algebraic topology

**55T10** Serre spectral sequences

**55T15** Adams spectral sequences

**55T20** Eilenberg-Moore spectral sequences [See also 57T35]

**55T25** Generalized cohomology and spectral sequences in algebraic topology

**55T99** None of the above, but in this section

## 55Uxx Applied homological algebra and category theory in algebraic topology [See also 18Gxx]

**55U05** Abstract complexes in algebraic topology

**55U10** Simplicial sets and complexes in algebraic topology

**55U15** Chain complexes in algebraic topology

**55U20** Universal coefficient theorems, Bockstein operator

**55U25** Homology of a product, Künneth formula

**55U30** Duality in applied homological algebra and category theory (aspects of algebraic topology)

**55U35** Abstract and axiomatic homotopy theory in algebraic topology

**55U40** Topological categories, foundations of homotopy theory

**55U99** None of the above, but in this section

# 57-XX Manifolds and cell complexes {For complex manifolds, see 32Qxx}

**57-00** General reference works (handbooks, dictionaries, bibliographies, etc.) pertaining to manifolds and cell complexes

**57-01** Introductory exposition (textbooks, tutorial papers, etc.) pertaining to manifolds and cell complexes

**57-02** Research exposition (monographs, survey articles) pertaining to manifolds and cell complexes

**57-03** History of manifolds and cell complexes [Consider also classification numbers from Section 01]

**57-04** Software, source code, etc. for problems pertaining to manifolds and cell complexes

**57-06** Proceedings, conferences, collections, etc. pertaining to manifolds and cell complexes

**57-08** Computational methods for problems pertaining to manifolds and cell complexes

**57-11** Research data for problems pertaining to manifolds and cell complexes

## 57Kxx Low-dimensional topology in specific dimensions

**57K10** Knot theory

**57K12** Generalized knots (virtual knots, welded knots, quandles, etc.)

**57K14** Knot polynomials

**57K16** Finite-type and quantum invariants, topological quantum field theories (TQFT)

**57K18** Homology theories in knot theory (Khovanov, Heegaard-Floer, etc.)

**57K20** 2-dimensional topology (including mapping class groups of surfaces, Teichmüller theory, curve complexes, etc.)

**57K30** General topology of 3-manifolds

**57K31** Invariants of 3-manifolds (including skein modules, character varieties)

**57K32** Hyperbolic 3-manifolds

**57K33** Contact structures in 3 dimensions [See also 57R17]

**57K35** Other geometric structures on 3-manifolds

**57K40** General topology of 4-manifolds

**57K41** Invariants of 4-manifolds (including Donaldson and Seiberg-Witten invariants)

**57K43** Symplectic structures in 4 dimensions [See also 57R17]

**57K45** Higher-dimensional knots and links

**57K50** Low-dimensional manifolds of specific dimension 5 or higher

**57K99** None of the above, but in this section

## 57Mxx General low-dimensional topology

**57M05** Fundamental group, presentations, free differential calculus

**57M07** Topological methods in group theory

**57M10** Covering spaces and low-dimensional topology

**57M12** Low-dimensional topology of special (e.g., branched) coverings

**57M15** Relations of low-dimensional topology with graph theory [See also 05C10]

**57M30** Wild embeddings

**57M50** General geometric structures on low-dimensional manifolds

**57M60** Group actions on manifolds and cell complexes in low dimensions

**57M99** None of the above, but in this section

## 57Nxx Topological manifolds

**57N16** Geometric structures on manifolds of high or arbitrary dimension [See also 57M50]

**57N17** Topology of topological vector spaces

**57N20** Topology of infinite-dimensional manifolds [See also 58Bxx]

**57N25** Shapes (aspects of topological manifolds) [See also 54C56, 55P55, 55Q07]

**57N30** Engulfing in topological manifolds

**57N35** Embeddings and immersions in topological manifolds

**57N37** Isotopy and pseudo-isotopy

**57N40** Neighborhoods of submanifolds

**57N45** Flatness and tameness of topological manifolds

**57N50** $S^{n-1} \subset E^n$, Schoenflies problem

**57N55** Microbundles and block bundles [See also 55R60, 57Q50]

**57N60** Cellularity in topological manifolds

**57N65** Algebraic topology of manifolds

**57N70** Cobordism and concordance in topological manifolds

**57N75** General position and transversality

**57N80** Stratifications in topological manifolds

**57N99** None of the above, but in this section

## 57Pxx Generalized manifolds [See also 18F15]

**57P05** Local properties of generalized manifolds

**57P10** Poincaré duality spaces

**57P99** None of the above, but in this section

## 57Qxx PL-topology

**57Q05** General topology of complexes

**57Q10** Simple homotopy type, Whitehead torsion, Reidemeister-Franz torsion, etc. [See also 19B28]

**57Q12** Wall finiteness obstruction for CW-complexes

**57Q15** Triangulating manifolds

**57Q20** Cobordism in PL-topology

**57Q25** Comparison of PL-structures: classification, Hauptvermutung

**57Q30** Engulfing

**57Q35** Embeddings and immersions in PL-topology

**57Q37** Isotopy in PL-topology

**57Q40** Regular neighborhoods in PL-topology

**57Q50** Microbundles and block bundles [See also 55R60, 57N55]

**57Q55** Approximations in PL-topology

**57Q60** Cobordism and concordance in PL-topology

**57Q65** General position and transversality

**57Q70** Discrete Morse theory and related ideas in manifold topology

**57Q91** Equivariant PL-topology

**57Q99** None of the above, but in this section

# 57Rxx Differential topology {For foundational questions of differentiable manifolds, see 58Axx; for infinite-dimensional manifolds, see 58Bxx}

**57R05** Triangulating

**57R10** Smoothing in differential topology

**57R12** Smooth approximations in differential topology

**57R15** Specialized structures on manifolds (spin manifolds, framed manifolds, etc.)

**57R17** Symplectic and contact topology in high or arbitrary dimension {For dimensions 3 and 4, see 57K33, 57K43}

**57R18** Topology and geometry of orbifolds

**57R19** Algebraic topology on manifolds and differential topology

**57R20** Characteristic classes and numbers in differential topology

**57R22** Topology of vector bundles and fiber bundles [See also 55Rxx]

**57R25** Vector fields, frame fields in differential topology

**57R27** Controllability of vector fields on $C^\infty$ and real-analytic manifolds [See also 49Qxx, 37C10, 93B05]

**57R30** Foliations in differential topology; geometric theory [See also 53C12]

**57R32** Classifying spaces for foliations; Gelfand-Fuks cohomology [See also 58H10]

**57R35** Differentiable mappings in differential topology

**57R40** Embeddings in differential topology

**57R42** Immersions in differential topology

**57R45** Singularities of differentiable mappings in differential topology

**57R50** Differential topological aspects of diffeomorphisms

**57R52** Isotopy in differential topology

**57R55** Differentiable structures in differential topology

**57R56** Topological quantum field theories (aspects of differential topology)

**57R57** Applications of global analysis to structures on manifolds [See also 57K41, 58-XX]

**57R58** Floer homology

**57R60** Homotopy spheres, Poincaré conjecture

**57R65** Surgery and handlebodies

**57R67** Surgery obstructions, Wall groups [See also 19J25]

**57R70** Critical points and critical submanifolds in differential topology

**57R75** O- and SO-cobordism

**57R77** Complex cobordism (U- and SU-cobordism) [See also 55N22]

**57R80** $h$- and $s$-cobordism

**57R85** Equivariant cobordism

**57R90** Other types of cobordism [See also 55N22]

**57R91** Equivariant algebraic topology of manifolds

**57R95** Realizing cycles by submanifolds

**57R99** None of the above, but in this section

## 57Sxx Topological transformation groups [See also 20F34, 22-XX, 37-XX, 54H15, 58D05]

**57S05** Topological properties of groups of homeomorphisms or diffeomorphisms

**57S10** Compact groups of homeomorphisms

**57S12** Toric topology

**57S15** Compact Lie groups of differentiable transformations

**57S17** Finite transformation groups

**57S20** Noncompact Lie groups of transformations

**57S25** Groups acting on specific manifolds

**57S30** Discontinuous groups of transformations

**57S99** None of the above, but in this section

## 57Txx Homology and homotopy of topological groups and related structures

**57T05** Hopf algebras (aspects of homology and homotopy of topological groups) [See also 16T05]

**57T10** Homology and cohomology of Lie groups

**57T15** Homology and cohomology of homogeneous spaces of Lie groups

**57T20** Homotopy groups of topological groups and homogeneous spaces

**57T25** Homology and cohomology of $H$-spaces

**57T30** Bar and cobar constructions [See also 18N40, 55Uxx]

**57T35** Applications of Eilenberg-Moore spectral sequences [See also 55R20, 55T20]

**57T99** None of the above, but in this section

## 57Zxx Relations of manifolds and cell complexes with science and engineering

**57Z05** Relations of manifolds and cell complexes with physics

**57Z10** Relations of manifolds and cell complexes with biology

**57Z15** Relations of manifolds and cell complexes with chemistry

**57Z20** Relations of manifolds and cell complexes with engineering

**57Z25** Relations of manifolds and cell complexes with computer and data science

**57Z99** None of the above, but in this section

# 58-XX Global analysis, analysis on manifolds [See also 32Cxx, 32Fxx, 32Wxx, 46-XX, 53Cxx] {For nonlinear operators, see 47Hxx; for geometric integration theory, see 49Q15}

**58-00** General reference works (handbooks, dictionaries, bibliographies, etc.) pertaining to global analysis

**58-01** Introductory exposition (textbooks, tutorial papers, etc.) pertaining to global analysis

**58-02** Research exposition (monographs, survey articles) pertaining to global analysis

**58-03** History of global analysis [Consider also classification numbers from Section 01]

**58-04** Software, source code, etc. for problems pertaining to global analysis

**58-06** Proceedings, conferences, collections, etc. pertaining to global analysis

**58-08** Computational methods for problems pertaining to global analysis

**58-11** Research data for problems pertaining to global analysis

## 58Axx General theory of differentiable manifolds [See also 32Cxx]

**58A03** Topos-theoretic approach to differentiable manifolds

**58A05** Differentiable manifolds, foundations

**58A07** Real-analytic and Nash manifolds [See also 14P20, 32C07]

**58A10** Differential forms in global analysis

**58A12** de Rham theory in global analysis [See also 14Fxx]

**58A14** Hodge theory in global analysis [See also 14C30, 14Fxx, 32J25, 32S35]

**58A15** Exterior differential systems (Cartan theory)

**58A17** Pfaffian systems

**58A20** Jets in global analysis

**58A25** Currents in global analysis [See also 32C30, 53C65]

**58A30** Vector distributions (subbundles of the tangent bundles)

**58A32** Natural bundles

**58A35** Stratified sets [See also 32S60]

**58A40** Differential spaces

**58A50** Supermanifolds and graded manifolds [See also 14A22, 32C11]

**58A99** None of the above, but in this section

## 58Bxx Infinite-dimensional manifolds

**58B05** Homotopy and topological questions for infinite-dimensional manifolds

**58B10** Differentiability questions for infinite-dimensional manifolds

**58B12** Questions of holomorphy and infinite-dimensional manifolds [See also 32-XX, 46G20]

**58B15** Fredholm structures on infinite-dimensional manifolds [See also 47A53]

**58B20** Riemannian, Finsler and other geometric structures on infinite-dimensional manifolds [See also 53C20, 53C60]

**58B25** Group structures and generalizations on infinite-dimensional manifolds [See also 22E65, 58D05]

**58B32** Geometry of quantum groups

**58B34** Noncommutative geometry (à la Connes)

**58B99** None of the above, but in this section

## 58Cxx Calculus on manifolds; nonlinear operators [See also 46Txx, 47Hxx, 47Jxx]

**58C05** Real-valued functions on manifolds

**58C06** Set-valued and function-space-valued mappings on manifolds [See also 47H04, 54C60]

**58C07** Continuity properties of mappings on manifolds

**58C10** Holomorphic maps on manifolds [See also 32-XX]

**58C15** Implicit function theorems; global Newton methods on manifolds

**58C20** Differentiation theory (Gateaux, Fréchet, etc.) on manifolds [See also 26Exx, 46G05]

**58C25** Differentiable maps on manifolds

**58C30** Fixed-point theorems on manifolds [See also 47H10]

**58C35** Integration on manifolds; measures on manifolds [See also 28Cxx]

**58C40** Spectral theory; eigenvalue problems on manifolds [See also 47J10, 58E07]

**58C50** Analysis on supermanifolds or graded manifolds

**58C99** None of the above, but in this section

## 58Dxx Spaces and manifolds of mappings (including nonlinear versions of 46Exx) [See also 46Txx, 53Cxx]

**58D05** Groups of diffeomorphisms and homeomorphisms as manifolds [See also 22E65, 57S05]

**58D07** Groups and semigroups of nonlinear operators [See also 17B65, 47H20]

**58D10** Spaces of embeddings and immersions

**58D15** Manifolds of mappings [See also 46T10, 54C35]

**58D17** Manifolds of metrics (especially Riemannian)

**58D19** Group actions and symmetry properties

**58D20** Measures (Gaussian, cylindrical, etc.) on manifolds of maps [See also 28Cxx, 46T12]

**58D25** Equations in function spaces; evolution equations [See also 34Gxx, 35K90, 35L90, 35R15, 37Lxx, 47Jxx]

**58D27** Moduli problems for differential geometric structures

**58D29** Moduli problems for topological structures

**58D30** Applications of manifolds of mappings to the sciences

**58D99** None of the above, but in this section

## 58Exx Variational problems in infinite-dimensional spaces

**58E05** Abstract critical point theory (Morse theory, Lyusternik-Shnirel'man theory, etc.) in infinite-dimensional spaces

**58E07** Variational problems in abstract bifurcation theory in infinite-dimensional spaces

**58E09** Group-invariant bifurcation theory in infinite-dimensional spaces

**58E10** Variational problems in applications to the theory of geodesics (problems in one independent variable)

**58E11** Critical metrics

**58E12** Variational problems concerning minimal surfaces (problems in two independent variables) [See also 49Q05]

**58E15** Variational problems concerning extremal problems in several variables; Yang-Mills functionals [See also 81T13], etc.

**58E17** Multiobjective variational problems, Pareto optimality, applications to economics, etc. [See also 90C29, 91Bxx]

**58E20** Harmonic maps, etc. [See also 53C43]

**58E25** Applications of variational problems to control theory [See also 49-XX, 93-XX]

**58E30** Variational principles in infinite-dimensional spaces

**58E35** Variational inequalities (global problems) in infinite-dimensional spaces

**58E40** Variational aspects of group actions in infinite-dimensional spaces

**58E50** Applications of variational problems in infinite-dimensional spaces to the sciences

**58E99** None of the above, but in this section

## 58Hxx Pseudogroups, differentiable groupoids and general structures on manifolds

**58H05** Pseudogroups and differentiable groupoids [See also 22A22, 22E65]

**58H10** Cohomology of classifying spaces for pseudogroup structures (Spencer, Gelfand-Fuks, etc.) [See also 57R32]

**58H15** Deformations of general structures on manifolds [See also 32Gxx, 58J10]

**58H99** None of the above, but in this section

## 58Jxx Partial differential equations on manifolds; differential operators [See also 32Wxx, 35-XX, 53Cxx]

**58J05** Elliptic equations on manifolds, general theory [See also 35Jxx]

**58J10** Differential complexes [See also 35Nxx]; elliptic complexes

**58J15** Relations of PDEs on manifolds with hyperfunctions

**58J20** Index theory and related fixed-point theorems on manifolds [See also 19K56, 46L80]

**58J22** Exotic index theories on manifolds [See also 19K56, 46L05, 46L10, 46L80, 46M20]

**58J26** Elliptic genera

**58J28** Eta-invariants, Chern-Simons invariants

**58J30** Spectral flows

**58J32** Boundary value problems on manifolds

**58J35** Heat and other parabolic equation methods for PDEs on manifolds

**58J37** Perturbations of PDEs on manifolds; asymptotics

**58J40** Pseudodifferential and Fourier integral operators on manifolds [See also 35Sxx]

**58J42** Noncommutative global analysis, noncommutative residues

**58J45** Hyperbolic equations on manifolds [See also 35Lxx]

**58J47** Propagation of singularities; initial value problems on manifolds

**58J50** Spectral problems; spectral geometry; scattering theory on manifolds [See also 35Pxx]

**58J51** Relations between spectral theory and ergodic theory, e.g., quantum unique ergodicity

**58J52** Determinants and determinant bundles, analytic torsion

**58J53** Isospectrality

**58J55** Bifurcation theory for PDEs on manifolds [See also 35B32]

**58J60** Relations of PDEs with special manifold structures (Riemannian, Finsler, etc.)

**58J65** Diffusion processes and stochastic analysis on manifolds [See also 35R60, 60H10, 60J60]

**58J70** Invariance and symmetry properties for PDEs on manifolds [See also 35A30]

**58J72** Correspondences and other transformation methods (e.g., Lie-Bäcklund) for PDEs on manifolds [See also 35A22]

**58J90** Applications of PDEs on manifolds

**58J99** None of the above, but in this section

**58Kxx Theory of singularities and catastrophe theory** [See also 32Sxx, 37-XX]

**58K05** Critical points of functions and mappings on manifolds

**58K10** Monodromy on manifolds

**58K15** Topological properties of mappings on manifolds

**58K20** Algebraic and analytic properties of mappings on manifolds

**58K25** Stability theory for manifolds

**58K30** Global theory of singularities

**58K35** Catastrophe theory

**58K40** Classification; finite determinacy of map germs

**58K45** Singularities of vector fields, topological aspects

**58K50** Normal forms on manifolds

**58K55** Asymptotic behavior of solutions to equations on manifolds

**58K60** Deformation of singularities

**58K65** Topological invariants on manifolds

**58K70** Symmetries, equivariance on manifolds

**58K99** None of the above, but in this section

## 58Zxx Applications of global analysis to the sciences

**58Z05** Applications of global analysis to the sciences

**58Z99** None of the above, but in this section

# 60-XX Probability theory and stochastic processes {For additional applications, see 05Cxx, 11Kxx, 34-XX, 35-XX, 62-XX, 76-XX, 81-XX, 82-XX, 90-XX, 91-XX, 92-XX, 93-XX, 94-XX}

**60-00** General reference works (handbooks, dictionaries, bibliographies, etc.) pertaining to probability theory

**60-01** Introductory exposition (textbooks, tutorial papers, etc.) pertaining to probability theory

**60-02** Research exposition (monographs, survey articles) pertaining to probability theory

**60-03** History of probability theory [Consider also classification numbers from Section 01]

**60-04** Software, source code, etc. for problems pertaining to probability theory

**60-06** Proceedings, conferences, collections, etc. pertaining to probability theory

**60-08** Computational methods for problems pertaining to probability theory

**60-11** Research data for problems pertaining to probability theory

## 60Axx Foundations of probability theory

**60A05** Axioms; other general questions in probability

**60A10** Probabilistic measure theory {For ergodic theory, see 28Dxx, 60Fxx}

**60A86** Fuzzy probability

**60A99** None of the above, but in this section

## 60Bxx Probability theory on algebraic and topological structures

**60B05** Probability measures on topological spaces

**60B10** Convergence of probability measures

**60B11** Probability theory on linear topological spaces [See also 28C20]

**60B12** Limit theorems for vector-valued random variables (infinite-dimensional case)

**60B15** Probability measures on groups or semigroups, Fourier transforms, factorization

**60B20** Random matrices (probabilistic aspects) {For algebraic aspects, see 15B52}

**60B99** None of the above, but in this section

## 60Cxx Combinatorial probability

**60C05** Combinatorial probability

**60C99** None of the above, but in this section

## 60Dxx Geometric probability and stochastic geometry [See also 52A22, 53C65]

**60D05** Geometric probability and stochastic geometry [See also 52A22, 53C65]

**60D99** None of the above, but in this section

## 60Exx Distribution theory [See also 62Exx, 62Hxx]

**60E05** Probability distributions: general theory

**60E07** Infinitely divisible distributions; stable distributions

**60E10** Characteristic functions; other transforms

**60E15** Inequalities; stochastic orderings

**60E99** None of the above, but in this section

## 60Fxx Limit theorems in probability theory [See also 28Dxx, 60B12]

**60F05** Central limit and other weak theorems

**60F10** Large deviations

**60F15** Strong limit theorems

**60F17** Functional limit theorems; invariance principles

**60F20** Zero-one laws

**60F25** $L^p$-limit theorems

**60F99** None of the above, but in this section

## 60Gxx Stochastic processes

**60G05** Foundations of stochastic processes

**60G07** General theory of stochastic processes

**60G09** Exchangeability for stochastic processes

**60G10** Stationary stochastic processes

**60G12** General second-order stochastic processes

**60G15** Gaussian processes

**60G17** Sample path properties

**60G18** Self-similar stochastic processes

**60G20** Generalized stochastic processes

**60G22** Fractional processes, including fractional Brownian motion

**60G25** Prediction theory (aspects of stochastic processes) [See also 62M20]

**60G30** Continuity and singularity of induced measures

**60G35** Signal detection and filtering (aspects of stochastic processes) [See also 62M20, 93E10, 93E11, 94Axx]

**60G40** Stopping times; optimal stopping problems; gambling theory [See also 62L15, 91A60]

**60G42** Martingales with discrete parameter

**60G44** Martingales with continuous parameter

**60G46** Martingales and classical analysis

**60G48** Generalizations of martingales

**60G50** Sums of independent random variables; random walks

**60G51** Processes with independent increments; Lévy processes

**60G52** Stable stochastic processes

**60G53** Feller processes

**60G55** Point processes (e.g., Poisson, Cox, Hawkes processes)

**60G57** Random measures

**60G60** Random fields

**60G65** Nonlinear processes (e.g., $G$-Brownian motion, $G$-Lévy processes)

**60G70** Extreme value theory; extremal stochastic processes

**60G99** None of the above, but in this section

## 60Hxx Stochastic analysis [See also 58J65]

**60H05** Stochastic integrals

**60H07** Stochastic calculus of variations and the Malliavin calculus

**60H10** Stochastic ordinary differential equations (aspects of stochastic analysis) [See also 34F05]

**60H15** Stochastic partial differential equations (aspects of stochastic analysis) [See also 35R60]

**60H17** Singular stochastic partial differential equations

**60H20** Stochastic integral equations

**60H25** Random operators and equations (aspects of stochastic analysis) [See also 47B80]

**60H30** Applications of stochastic analysis (to PDEs, etc.)

**60H35** Computational methods for stochastic equations (aspects of stochastic analysis) [See also 65C30]

**60H40** White noise theory

**60H50** Regularization by noise

**60H99** None of the above, but in this section

## 60Jxx Markov processes

**60J05** Discrete-time Markov processes on general state spaces

**60J10** Markov chains (discrete-time Markov processes on discrete state spaces)

**60J20** Applications of Markov chains and discrete-time Markov processes on general state spaces (social mobility, learning theory, industrial processes, etc.) [See also 90B30, 91D10, 91E40]

**60J22** Computational methods in Markov chains [See also 65C40]

**60J25** Continuous-time Markov processes on general state spaces

**60J27** Continuous-time Markov processes on discrete state spaces

**60J28** Applications of continuous-time Markov processes on discrete state spaces

**60J35** Transition functions, generators and resolvents [See also 47D03, 47D07]

**60J40** Right processes

**60J45** Probabilistic potential theory [See also 31Cxx, 31D05]

**60J46** Dirichlet form methods in Markov processes

**60J50** Boundary theory for Markov processes

**60J55** Local time and additive functionals

**60J57** Multiplicative functionals and Markov processes

**60J60** Diffusion processes [See also 58J65]

**60J65** Brownian motion [See also 58J65]

**60J67** Stochastic (Schramm-)Loewner evolution (SLE)

**60J68** Superprocesses

**60J70** Applications of Brownian motions and diffusion theory (population genetics, absorption problems, etc.) [See also 92Dxx]

**60J74** Jump processes on discrete state spaces

**60J76** Jump processes on general state spaces

**60J80** Branching processes (Galton-Watson, birth-and-death, etc.)

**60J85** Applications of branching processes [See also 92Dxx]

**60J90** Coalescent processes

**60J95** Applications of coalescent processes [See also 92Dxx]

**60J99** None of the above, but in this section

## 60Kxx Special processes

**60K05** Renewal theory

**60K10** Applications of renewal theory (reliability, demand theory, etc.)

**60K15** Markov renewal processes, semi-Markov processes

**60K20** Applications of Markov renewal processes (reliability, queueing networks, etc.) [See also 90Bxx]

**60K25** Queueing theory (aspects of probability theory) [See also 68M20, 90B22]

**60K30** Applications of queueing theory (congestion, allocation, storage, traffic, etc.) [See also 90Bxx]

**60K35** Interacting random processes; statistical mechanics type models; percolation theory [See also 82B43, 82C43]

**60K37** Processes in random environments

**60K40** Other physical applications of random processes

**60K50** Anomalous diffusion models (subdiffusion, superdiffusion, continuous-time random walks, etc.) [See also 60G22, 60G55, 60J74, 60J76] {For applications to physics and the sciences, see 76-XX, 82Cxx, 92-XX}

**60K99** None of the above, but in this section

## 60Lxx Rough analysis

**60L10** Signatures and data streams

**60L20** Rough paths

**60L30** Regularity structures

**60L40** Paracontrolled distributions and alternative approaches

**60L50** Rough partial differential equations

**60L70** Algebraic structures and computation

**60L90** Applications of rough analysis

**60L99** None of the above, but in this section

# 62-XX Statistics

**62-00** General reference works (handbooks, dictionaries, bibliographies, etc.) pertaining to statistics

**62-01** Introductory exposition (textbooks, tutorial papers, etc.) pertaining to statistics

**62-02** Research exposition (monographs, survey articles) pertaining to statistics

**62-03** History of statistics [Consider also classification numbers from Section 01]

**62-04** Software, source code, etc. for problems pertaining to statistics

**62-06** Proceedings, conferences, collections, etc. pertaining to statistics

**62-08** Computational methods for problems pertaining to statistics

**62-11** Research data for problems pertaining to statistics

## 62Axx Foundational topics in statistics

**62A01** Foundations and philosophical topics in statistics

**62A09** Graphical methods in statistics

**62A86** Fuzzy analysis in statistics

**62A99** None of the above, but in this section

## 62Bxx Sufficiency and information

**62B05** Sufficient statistics and fields

**62B10** Statistical aspects of information-theoretic topics [See also 94A17]

**62B11** Information geometry (statistical aspects) {For differential geometric aspects, see 53B12}

**62B15** Theory of statistical experiments

**62B86** Statistical aspects of fuzziness, sufficiency, and information

**62B99** None of the above, but in this section

## 62Cxx Statistical decision theory [See also 90B50, 91B06] {For game theory, see 91A35}

**62C05** General considerations in statistical decision theory

**62C07** Complete class results in statistical decision theory

**62C10** Bayesian problems; characterization of Bayes procedures

**62C12** Empirical decision procedures; empirical Bayes procedures

**62C15** Admissibility in statistical decision theory

**62C20** Minimax procedures in statistical decision theory

**62C25** Compound decision problems in statistical decision theory

**62C86** Statistical decision theory and fuzziness

**62C99** None of the above, but in this section

## 62Dxx Statistical sampling theory and related topics

**62D05** Sampling theory, sample surveys

**62D10** Missing data

**62D20** Causal inference from observational studies

**62D99** None of the above, but in this section

## 62Exx Statistical distribution theory [See also 60Exx]

**62E10** Characterization and structure theory of statistical distributions

**62E15** Exact distribution theory in statistics

**62E17** Approximations to statistical distributions (nonasymptotic)

**62E20** Asymptotic distribution theory in statistics

**62E86** Fuzziness in connection with statistical distributions

**62E99** None of the above, but in this section

## 62Fxx Parametric inference

**62F03** Parametric hypothesis testing

**62F05** Asymptotic properties of parametric tests

**62F07** Statistical ranking and selection procedures

**62F10** Point estimation

**62F12** Asymptotic properties of parametric estimators

**62F15** Bayesian inference

**62F25** Parametric tolerance and confidence regions

**62F30** Parametric inference under constraints

**62F35** Robustness and adaptive procedures (parametric inference)

**62F40** Bootstrap, jackknife and other resampling methods

**62F86** Parametric inference and fuzziness

**62F99** None of the above, but in this section

## 62Gxx Nonparametric inference

**62G05** Nonparametric estimation

**62G07** Density estimation

**62G08** Nonparametric regression and quantile regression

**62G09** Nonparametric statistical resampling methods

**62G10** Nonparametric hypothesis testing

**62G15** Nonparametric tolerance and confidence regions

**62G20** Asymptotic properties of nonparametric inference

**62G30** Order statistics; empirical distribution functions

**62G32** Statistics of extreme values; tail inference

**62G35** Nonparametric robustness

**62G86** Nonparametric inference and fuzziness

**62G99** None of the above, but in this section

## 62Hxx Multivariate analysis [See also 60Exx]

**62H05** Characterization and structure theory for multivariate probability distributions; copulas

**62H10** Multivariate distribution of statistics

**62H11** Directional data; spatial statistics

**62H12** Estimation in multivariate analysis

**62H15** Hypothesis testing in multivariate analysis

**62H17** Contingency tables

**62H20** Measures of association (correlation, canonical correlation, etc.)

**62H22** Probabilistic graphical models

**62H25** Factor analysis and principal components; correspondence analysis

**62H30** Classification and discrimination; cluster analysis (statistical aspects) [See also 68T10, 91C20]; mixture models

**62H35** Image analysis in multivariate analysis

**62H86** Multivariate analysis and fuzziness

**62H99** None of the above, but in this section

## 62Jxx Linear inference, regression

**62J02** General nonlinear regression

**62J05** Linear regression; mixed models

**62J07** Ridge regression; shrinkage estimators (Lasso)

**62J10** Analysis of variance and covariance (ANOVA)

**62J12** Generalized linear models (logistic models)

**62J15** Paired and multiple comparisons; multiple testing

**62J20** Diagnostics, and linear inference and regression

**62J86** Fuzziness, and linear inference and regression

**62J99** None of the above, but in this section

## 62Kxx Design of statistical experiments [See also 05Bxx]

**62K05** Optimal statistical designs

**62K10** Statistical block designs

**62K15** Factorial statistical designs

**62K20** Response surface designs

**62K25** Robust parameter designs

**62K86** Fuzziness and design of statistical experiments

**62K99** None of the above, but in this section

## 62Lxx Sequential statistical methods

**62L05** Sequential statistical design

**62L10** Sequential statistical analysis

**62L12** Sequential estimation

**62L15** Optimal stopping in statistics [See also 60G40, 91A60]

**62L20** Stochastic approximation

**62L86** Fuzziness and sequential statistical methods

**62L99** None of the above, but in this section

## 62Mxx Inference from stochastic processes

**62M02** Markov processes: hypothesis testing

**62M05** Markov processes: estimation; hidden Markov models

**62M07** Non-Markovian processes: hypothesis testing

**62M09** Non-Markovian processes: estimation

**62M10** Time series, auto-correlation, regression, etc. in statistics (GARCH) [See also 91B84]

**62M15** Inference from stochastic processes and spectral analysis

**62M20** Inference from stochastic processes and prediction [See also 60G25]; filtering [See also 60G35, 93E10, 93E11]

**62M30** Inference from spatial processes

**62M40** Random fields; image analysis

**62M45** Neural nets and related approaches to inference from stochastic processes

**62M86** Inference from stochastic processes and fuzziness

**62M99** None of the above, but in this section

## 62Nxx Survival analysis and censored data

**62N01** Censored data models

**62N02** Estimation in survival analysis and censored data

**62N03** Testing in survival analysis and censored data

**62N05** Reliability and life testing [See also 90B25]

**62N86** Fuzziness, and survival analysis and censored data

**62N99** None of the above, but in this section

## 62Pxx Applications of statistics [See also 90-XX, 91-XX, 92-XX]

**62P05** Applications of statistics to actuarial sciences and financial mathematics

**62P10** Applications of statistics to biology and medical sciences; meta analysis

**62P12** Applications of statistics to environmental and related topics

**62P15** Applications of statistics to psychology

**62P20** Applications of statistics to economics [See also 91Bxx]

**62P25** Applications of statistics to social sciences

**62P30** Applications of statistics in engineering and industry; control charts

**62P35** Applications of statistics to physics

**62P99** None of the above, but in this section

## 62Qxx Statistical tables

**62Q05** Statistical tables

**62Q99** None of the above, but in this section

## 62Rxx Statistics on algebraic and topological structures

**62R01** Algebraic statistics

**62R07** Statistical aspects of big data and data science {For computer science aspects, see 68T09; for information-theoretic aspects, see 94A16}

**62R10** Functional data analysis

**62R20** Statistics on metric spaces

**62R30** Statistics on manifolds

**62R40** Topological data analysis [See also 55N31]

**62R99** None of the above, but in this section

# 65-XX Numerical analysis

**65-00** General reference works (handbooks, dictionaries, bibliographies, etc.) pertaining to numerical analysis

**65-01** Introductory exposition (textbooks, tutorial papers, etc.) pertaining to numerical analysis

**65-02** Research exposition (monographs, survey articles) pertaining to numerical analysis

**65-03** History of numerical analysis [Consider also classification numbers from Section 01]

**65-04** Software, source code, etc. for problems pertaining to numerical analysis

**65-06** Proceedings, conferences, collections, etc. pertaining to numerical analysis

**65-11** Research data for problems pertaining to numerical analysis

## 65Axx Tables in numerical analysis

**65A05** Tables in numerical analysis

**65A99** None of the above, but in this section

## 65Bxx Acceleration of convergence in numerical analysis

**65B05** Extrapolation to the limit, deferred corrections

**65B10** Numerical summation of series

**65B15** Euler-Maclaurin formula in numerical analysis

**65B99** None of the above, but in this section

## 65Cxx Probabilistic methods, stochastic differential equations

**65C05** Monte Carlo methods [See also 82M31]

**65C10** Random number generation in numerical analysis [See also 11K45]

**65C20** Probabilistic models, generic numerical methods in probability and statistics [See also 60-08, 62-08]

**65C30** Numerical solutions to stochastic differential and integral equations {For theoretical aspects, see 60H35} [See also 65M75, 65N75]

**65C35** Stochastic particle methods [See also 82M60]

**65C40** Numerical analysis or methods applied to Markov chains [See also 60J22]

**65C99** None of the above, but in this section

## 65Dxx Numerical approximation and computational geometry (primarily algorithms) {For theoretical aspects, see 41-XX, 68Uxx}

**65D05** Numerical interpolation

**65D07** Numerical computation using splines

**65D10** Numerical smoothing, curve fitting

**65D12** Numerical radial basis function approximation

**65D15** Algorithms for approximation of functions

**65D17** Computer-aided design (modeling of curves and surfaces) [See also 68U07]

**65D18** Numerical aspects of computer graphics, image analysis, and computational geometry [See also 51N05, 68U05]

**65D19** Computational issues in computer and robotic vision

**65D20** Computation of special functions and constants, construction of tables [See also 33F05]

**65D25** Numerical differentiation

**65D30** Numerical integration

**65D32** Numerical quadrature and cubature formulas

**65D40** Numerical approximation of high-dimensional functions; sparse grids

**65D99** None of the above, but in this section

## 65Exx Numerical methods in complex analysis (potential theory, etc.)

**65E05** General theory of numerical methods in complex analysis (potential theory, etc.) [See also 30-08, 31-08, 32-08]

**65E10** Numerical methods in conformal mappings [See also 30C30]

**65E99** None of the above, but in this section

## 65Fxx Numerical linear algebra

**65F05** Direct numerical methods for linear systems and matrix inversion

**65F08** Preconditioners for iterative methods

**65F10** Iterative numerical methods for linear systems [See also 65N22]

**65F15** Numerical computation of eigenvalues and eigenvectors of matrices

**65F18** Numerical solutions to inverse eigenvalue problems

**65F20** Numerical solutions to overdetermined systems, pseudoinverses

**65F22** Ill-posedness and regularization problems in numerical linear algebra

**65F25** Orthogonalization in numerical linear algebra

**65F35** Numerical computation of matrix norms, conditioning, scaling [See also 15A12, 15A60]

**65F40** Numerical computation of determinants

**65F45** Numerical methods for matrix equations

**65F50** Computational methods for sparse matrices

**65F55** Numerical methods for low-rank matrix approximation; matrix compression

**65F60** Numerical computation of matrix exponential and similar matrix functions

**65F99** None of the above, but in this section

## 65Gxx Error analysis and interval analysis

**65G20** Algorithms with automatic result verification

**65G30** Interval and finite arithmetic

**65G40** General methods in interval analysis

**65G50** Roundoff error

**65G99** None of the above, but in this section

## 65Hxx Nonlinear algebraic or transcendental equations

**65H04** Numerical computation of roots of polynomial equations

**65H05** Numerical computation of solutions to single equations

**65H10** Numerical computation of solutions to systems of equations

**65H14** Numerical algebraic geometry

**65H17** Numerical solution of nonlinear eigenvalue and eigenvector problems [See also 47Hxx, 47Jxx, 58C40, 58E07, 90C30]

**65H20** Global methods, including homotopy approaches to the numerical solution of nonlinear equations [See also 58C30, 90C30]

**65H99** None of the above, but in this section

## 65Jxx Numerical analysis in abstract spaces

**65J05** General theory of numerical analysis in abstract spaces

**65J08** Numerical solutions to abstract evolution equations

**65J10** Numerical solutions to equations with linear operators [do not use 65Fxx]

**65J15** Numerical solutions to equations with nonlinear operators [do not use 65Hxx]

**65J20** Numerical solutions of ill-posed problems in abstract spaces; regularization

**65J22** Numerical solution to inverse problems in abstract spaces

**65J99** None of the above, but in this section

## 65Kxx Numerical methods for mathematical programming, optimization and variational techniques

**65K05** Numerical mathematical programming methods [See also 90Cxx]

**65K10** Numerical optimization and variational techniques [See also 49Mxx, 93-08]

**65K15** Numerical methods for variational inequalities and related problems

**65K99** None of the above, but in this section

## 65Lxx Numerical methods for ordinary differential equations

**65L03** Numerical methods for functional-differential equations

**65L04** Numerical methods for stiff equations

**65L05** Numerical methods for initial value problems involving ordinary differential equations

**65L06** Multistep, Runge-Kutta and extrapolation methods for ordinary differential equations

**65L07** Numerical investigation of stability of solutions to ordinary differential equations

**65L08** Numerical solution of ill-posed problems involving ordinary differential equations

**65L09** Numerical solution of inverse problems involving ordinary differential equations

**65L10** Numerical solution of boundary value problems involving ordinary differential equations

**65L11** Numerical solution of singularly perturbed problems involving ordinary differential equations

**65L12** Finite difference and finite volume methods for ordinary differential equations

**65L15** Numerical solution of eigenvalue problems involving ordinary differential equations

**65L20** Stability and convergence of numerical methods for ordinary differential equations

**65L50** Mesh generation, refinement, and adaptive methods for ordinary differential equations

**65L60** Finite element, Rayleigh-Ritz, Galerkin and collocation methods for ordinary differential equations

**65L70** Error bounds for numerical methods for ordinary differential equations

**65L80** Numerical methods for differential-algebraic equations

**65L99** None of the above, but in this section

## 65Mxx Numerical methods for partial differential equations, initial value and time-dependent initial-boundary value problems

**65M06** Finite difference methods for initial value and initial-boundary value problems involving PDEs

**65M08** Finite volume methods for initial value and initial-boundary value problems involving PDEs

**65M12** Stability and convergence of numerical methods for initial value and initial-boundary value problems involving PDEs

**65M15** Error bounds for initial value and initial-boundary value problems involving PDEs

**65M20** Method of lines for initial value and initial-boundary value problems involving PDEs

**65M22** Numerical solution of discretized equations for initial value and initial-boundary value problems involving PDEs [See also 65Fxx, 65Hxx]

**65M25** Numerical aspects of the method of characteristics for initial value and initial-boundary value problems involving PDEs

**65M30** Numerical methods for ill-posed problems for initial value and initial-boundary value problems involving PDEs

**65M32** Numerical methods for inverse problems for initial value and initial-boundary value problems involving PDEs

**65M38** Boundary element methods for initial value and initial-boundary value problems involving PDEs

**65M50** Mesh generation, refinement, and adaptive methods for the numerical solution of initial value and initial-boundary value problems involving PDEs

**65M55** Multigrid methods; domain decomposition for initial value and initial-boundary value problems involving PDEs

**65M60** Finite element, Rayleigh-Ritz and Galerkin methods for initial value and initial-boundary value problems involving PDEs

**65M70** Spectral, collocation and related methods for initial value and initial-boundary value problems involving PDEs

**65M75** Probabilistic methods, particle methods, etc. for initial value and initial-boundary value problems involving PDEs

**65M80** Fundamental solutions, Green's function methods, etc. for initial value and initial-boundary value problems involving PDEs

**65M85** Fictitious domain methods for initial value and initial-boundary value problems involving PDEs

**65M99** None of the above, but in this section

## 65Nxx Numerical methods for partial differential equations, boundary value problems

**65N06** Finite difference methods for boundary value problems involving PDEs

**65N08** Finite volume methods for boundary value problems involving PDEs

**65N12** Stability and convergence of numerical methods for boundary value problems involving PDEs

**65N15** Error bounds for boundary value problems involving PDEs

**65N20** Numerical methods for ill-posed problems for boundary value problems involving PDEs

**65N21** Numerical methods for inverse problems for boundary value problems involving PDEs

**65N22** Numerical solution of discretized equations for boundary value problems involving PDEs [See also 65Fxx, 65Hxx]

**65N25** Numerical methods for eigenvalue problems for boundary value problems involving PDEs

**65N30** Finite element, Rayleigh-Ritz and Galerkin methods for boundary value problems involving PDEs

**65N35** Spectral, collocation and related methods for boundary value problems involving PDEs

**65N38** Boundary element methods for boundary value problems involving PDEs

**65N40** Method of lines for boundary value problems involving PDEs

**65N45** Method of contraction of the boundary for boundary value problems involving PDEs

**65N50** Mesh generation, refinement, and adaptive methods for boundary value problems involving PDEs

**65N55** Multigrid methods; domain decomposition for boundary value problems involving PDEs

**65N75** Probabilistic methods, particle methods, etc. for boundary value problems involving PDEs

**65N80** Fundamental solutions, Green's function methods, etc. for boundary value problems involving PDEs

**65N85** Fictitious domain methods for boundary value problems involving PDEs

**65N99** None of the above, but in this section

## 65Pxx Numerical problems in dynamical systems [See also 37Mxx]

**65P10** Numerical methods for Hamiltonian systems including symplectic integrators

**65P20** Numerical chaos

**65P30** Numerical bifurcation problems

**65P40** Numerical nonlinear stabilities in dynamical systems

**65P99** None of the above, but in this section

## 65Qxx Numerical methods for difference and functional equations, recurrence relations

**65Q10** Numerical methods for difference equations

**65Q20** Numerical methods for functional equations

**65Q30** Numerical aspects of recurrence relations

**65Q99** None of the above, but in this section

## 65Rxx Numerical methods for integral equations, integral transforms

**65R10** Numerical methods for integral transforms

**65R15** Numerical methods for eigenvalue problems in integral equations

**65R20** Numerical methods for integral equations

**65R30** Numerical methods for ill-posed problems for integral equations

**65R32** Numerical methods for inverse problems for integral equations

**65R99** None of the above, but in this section

## 65Sxx Graphical methods in numerical analysis

**65S05** Graphical methods in numerical analysis

**65S99** None of the above, but in this section

## 65Txx Numerical methods in Fourier analysis

**65T40** Numerical methods for trigonometric approximation and interpolation

**65T50** Numerical methods for discrete and fast Fourier transforms

**65T60** Numerical methods for wavelets

**65T99** None of the above, but in this section

## 65Yxx Computer aspects of numerical algorithms

**65Y04** Numerical algorithms for computer arithmetic, etc. [See also 68M07]

**65Y05** Parallel numerical computation

**65Y10** Numerical algorithms for specific classes of architectures

**65Y15** Packaged methods for numerical algorithms

**65Y20** Complexity and performance of numerical algorithms [See also 68Q25]

**65Y99** None of the above, but in this section

## 65Zxx Applications to the sciences

**65Z05** Applications to the sciences

**65Z99** None of the above, but in this section

# 68-XX Computer science {For papers containing software, source code, etc. in a specific mathematical area, see the classification number −04 in that area}

**68-00** General reference works (handbooks, dictionaries, bibliographies, etc.) pertaining to computer science

**68-01** Introductory exposition (textbooks, tutorial papers, etc.) pertaining to computer science

**68-02** Research exposition (monographs, survey articles) pertaining to computer science

**68-03** History of computer science [Consider also classification numbers from Section 01]

**68-04** Software, source code, etc. for problems pertaining to computer science

**68-06** Proceedings, conferences, collections, etc. pertaining to computer science

**68-11** Research data for problems pertaining to computer science

## 68Mxx Computer system organization

**68M01** General theory of computer systems

**68M07** Mathematical problems of computer architecture [See also 68W35]

**68M10** Network design and communication in computer systems [See also 68R10, 90B18]

**68M11** Internet topics [See also 68U35]

**68M12** Network protocols

**68M14** Distributed systems

**68M15** Reliability, testing and fault tolerance of networks and computer systems

**68M18** Wireless sensor networks as related to computer science [See also 90B18, 90B80]

**68M20** Performance evaluation, queueing, and scheduling in the context of computer systems [See also 60K20, 60K25, 90B22, 90B35, 90B36]

**68M25** Computer security

**68M99** None of the above, but in this section

## 68Nxx Theory of software

**68N01** General topics in the theory of software

**68N15** Theory of programming languages

**68N17** Logic programming

**68N18** Functional programming and lambda calculus [See also 03B40]

**68N19** Other programming paradigms (object-oriented, sequential, concurrent, automatic, etc.)

**68N20** Theory of compilers and interpreters

**68N25** Theory of operating systems

**68N30** Mathematical aspects of software engineering (specification, verification, metrics, requirements, etc.)

**68N99** None of the above, but in this section

## 68Pxx Theory of data

**68P01** General topics in the theory of data

**68P05** Data structures

**68P10** Searching and sorting

**68P15** Database theory

**68P20** Information storage and retrieval of data

**68P25** Data encryption (aspects in computer science) [See also 81P94, 94A60]

**68P27** Privacy of data

**68P30** Coding and information theory (compaction, compression, models of communication, encoding schemes, etc.) (aspects in computer science) [See also 94Axx, 94Bxx]

**68P99** None of the above, but in this section

## 68Qxx Theory of computing

**68Q01** General topics in the theory of computing

**68Q04** Classical models of computation (Turing machines, etc.) [See also 03D10]

**68Q06** Networks and circuits as models of computation; circuit complexity [See also 94C11]

**68Q07** Biologically inspired models of computation (DNA computing, membrane computing, etc.)

**68Q09** Other nonclassical models of computation {For quantum computing, see mainly 68Q12, 81P68}

**68Q10** Modes of computation (nondeterministic, parallel, interactive, probabilistic, etc.) [See also 68Q85]

**68Q11** Communication complexity, information complexity

**68Q12** Quantum algorithms and complexity in the theory of computing [See also 68Q09, 81P68]

**68Q15** Complexity classes (hierarchies, relations among complexity classes, etc.) [See also 03D15, 68Q17, 68Q19]

**68Q17** Computational difficulty of problems (lower bounds, completeness, difficulty of approximation, etc.) [See also 68Q15]

**68Q19** Descriptive complexity and finite models [See also 03C13]

**68Q25** Analysis of algorithms and problem complexity [See also 68W40]

**68Q27** Parameterized complexity, tractability and kernelization

**68Q30** Algorithmic information theory (Kolmogorov complexity, etc.) [See also 03D32]

**68Q32** Computational learning theory [See also 68T05]

**68Q42** Grammars and rewriting systems

**68Q45** Formal languages and automata [See also 03D05, 68Q70, 94A45]

**68Q55** Semantics in the theory of computing [See also 03B70, 06B35, 18C50]

**68Q60** Specification and verification (program logics, model checking, etc.) [See also 03B70]

**68Q65** Abstract data types; algebraic specification [See also 18C50]

**68Q70** Algebraic theory of languages and automata [See also 18B20, 20M35]

**68Q80** Cellular automata (computational aspects) {For cellular automata as dynamical systems, see 37B15}

**68Q85** Models and methods for concurrent and distributed computing (process algebras, bisimulation, transition nets, etc.)

**68Q87** Probability in computer science (algorithm analysis, random structures, phase transitions, etc.) [See also 68W20, 68W40]

**68Q99** None of the above, but in this section

## 68Rxx Discrete mathematics in relation to computer science

**68R01** General topics of discrete mathematics in relation to computer science

**68R05** Combinatorics in computer science

**68R07** Computational aspects of satisfiability [See also 68T20]

**68R10** Graph theory (including graph drawing) in computer science [See also 05Cxx, 90B10, 90C35]

**68R12** Metric embeddings as related to computational problems and algorithms

**68R15** Combinatorics on words

**68R99** None of the above, but in this section

## 68Txx Artificial intelligence

**68T01** General topics in artificial intelligence

**68T05** Learning and adaptive systems in artificial intelligence [See also 68Q32]

**68T07** Artificial neural networks and deep learning

**68T09** Computational aspects of data analysis and big data [See also 62R07] {For homological aspects, see 55N31}

**68T10** Pattern recognition, speech recognition {For cluster analysis, see 62H30}

**68T20** Problem solving in the context of artificial intelligence (heuristics, search strategies, etc.)

**68T27** Logic in artificial intelligence

**68T30** Knowledge representation

**68T35** Theory of languages and software systems (knowledge-based systems, expert systems, etc.) for artificial intelligence

**68T37** Reasoning under uncertainty in the context of artificial intelligence

**68T40** Artificial intelligence for robotics [See also 93C85]

**68T42** Agent technology and artificial intelligence

**68T45** Machine vision and scene understanding

**68T50** Natural language processing [See also 03B65, 91F20]

**68T99** None of the above, but in this section

## 68Uxx Computing methodologies and applications

**68U01** General topics in computing methodologies

**68U03** Computational aspects of digital topology {For topological aspects, see 54H30; for homological aspects, see 55-XX}

**68U05** Computer graphics; computational geometry (digital and algorithmic aspects) {For methods of numerical mathematics, see 65D18}

**68U07** Computer science aspects of computer-aided design {For methods of numerical mathematics, see 65D17}

**68U10** Computing methodologies for image processing

**68U15** Computing methodologies for text processing; mathematical typography

**68U35** Computing methodologies for information systems (hypertext navigation, interfaces, decision support, etc.) [See also 68M11]

**68U99** None of the above, but in this section

## 68Vxx Computer science support for mathematical research and practice

**68V05** Computer assisted proofs of proofs-by-exhaustion type {For rigorous numerics, see 65Gxx; for proofs employing automated or interactive theorem provers, see 68V15}

**68V15** Theorem proving (automated and interactive theorem provers, deduction, resolution, etc.) [See also 03B35]

**68V20** Formalization of mathematics in connection with theorem provers [See also 03B35, 68V15]

**68V25** Presentation and content markup for mathematics

**68V30** Mathematical knowledge management

**68V35** Digital mathematics libraries and repositories

**68V99** None of the above, but in this section

## 68Wxx Algorithms in computer science {For numerical algorithms, see 65-XX; for combinatorics and graph theory, see 05C85, 68Rxx}

**68W01** General topics in the theory of algorithms

**68W05** Nonnumerical algorithms

**68W10** Parallel algorithms in computer science

**68W15** Distributed algorithms

**68W20** Randomized algorithms

**68W25** Approximation algorithms

**68W27** Online algorithms; streaming algorithms

**68W30** Symbolic computation and algebraic computation [See also 11Yxx, 12-08, 13Pxx, 14Qxx, 16Z05, 17-08, 33F10]

**68W32** Algorithms on strings

**68W35** Hardware implementations of nonnumerical algorithms (VLSI algorithms, etc.) [See also 68M07]

**68W40** Analysis of algorithms [See also 68Q25]

**68W50** Evolutionary algorithms, genetic algorithms (computational aspects) [See also 68T05, 68T20, 90C59]

**68W99** None of the above, but in this section

# 70-XX Mechanics of particles and systems {For relativistic mechanics, see 83A05, 83C10; for statistical mechanics, see 82-XX}

**70-00** General reference works (handbooks, dictionaries, bibliographies, etc.) pertaining to mechanics of particles and systems

**70-01** Introductory exposition (textbooks, tutorial papers, etc.) pertaining to mechanics of particles and systems

**70-02** Research exposition (monographs, survey articles) pertaining to mechanics of particles and systems

**70-03** History of mechanics of particles and systems [Consider also classification numbers from Section 01]

**70-04** Software, source code, etc. for problems pertaining to mechanics of particles and systems

**70-05** Experimental work for problems pertaining to mechanics of particles and systems

**70-06** Proceedings, conferences, collections, etc. pertaining to mechanics of particles and systems

**70-08** Computational methods for problems pertaining to mechanics of particles and systems

**70-10** Mathematical modeling or simulation for problems pertaining to mechanics of particles and systems

**70-11** Research data for problems pertaining to mechanics of particles and systems

## 70Axx Axiomatics, foundations

**70A05** Axiomatics, foundations

**70A99** None of the above, but in this section

## 70Bxx Kinematics [See also 53A17]

**70B05** Kinematics of a particle

**70B10** Kinematics of a rigid body

**70B15** Kinematics of mechanisms and robots [See also 68T40, 70Q05, 93C85]

**70B99** None of the above, but in this section

## 70Cxx Statics

**70C20** Statics

**70C99** None of the above, but in this section

## 70Exx Dynamics of a rigid body and of multibody systems

**70E05** Motion of the gyroscope

**70E15** Free motion of a rigid body [See also 70M20]

**70E17** Motion of a rigid body with a fixed point

**70E18** Motion of a rigid body in contact with a solid surface [See also 70F25]

**70E20** Perturbation methods for rigid body dynamics

**70E40** Integrable cases of motion in rigid body dynamics

**70E45** Higher-dimensional generalizations in rigid body dynamics

**70E50** Stability problems in rigid body dynamics

**70E55** Dynamics of multibody systems

**70E60** Robot dynamics and control of rigid bodies [See also 68T40, 70Q05, 93C85]

**70E99** None of the above, but in this section

## 70Fxx Dynamics of a system of particles, including celestial mechanics

**70F05** Two-body problems

**70F07** Three-body problems

**70F10** $n$-body problems

**70F15** Celestial mechanics

**70F16** Collisions in celestial mechanics, regularization

**70F17** Inverse problems for systems of particles

**70F20** Holonomic systems related to the dynamics of a system of particles

**70F25** Nonholonomic systems related to the dynamics of a system of particles

**70F35** Collision of rigid or pseudo-rigid bodies

**70F40** Problems involving a system of particles with friction

**70F45** The dynamics of infinite particle systems

**70F99** None of the above, but in this section

## 70Gxx General models, approaches, and methods in mechanics of particles and systems [See also 37-XX]

**70G10** Generalized coordinates; event, impulse-energy, configuration, state, or phase space for problems in mechanics

**70G40** Topological and differential topological methods for problems in mechanics

**70G45** Differential geometric methods (tensors, connections, symplectic, Poisson, contact, Riemannian, nonholonomic, etc.) for problems in mechanics [See also 53Cxx, 53Dxx, 58Axx]

**70G55** Algebraic geometry methods for problems in mechanics

**70G60** Dynamical systems methods for problems in mechanics

**70G65** Symmetries, Lie group and Lie algebra methods for problems in mechanics

**70G70** Functional analytic methods for problems in mechanics

**70G75** Variational methods for problems in mechanics

**70G99** None of the above, but in this section

## 70Hxx Hamiltonian and Lagrangian mechanics [See also 37Jxx]

**70H03** Lagrange's equations

**70H05** Hamilton's equations

**70H06** Completely integrable systems and methods of integration for problems in Hamiltonian and Lagrangian mechanics

**70H07** Nonintegrable systems for problems in Hamiltonian and Lagrangian mechanics

**70H08** Nearly integrable Hamiltonian systems, KAM theory

**70H09** Perturbation theories for problems in Hamiltonian and Lagrangian mechanics

**70H11** Adiabatic invariants for problems in Hamiltonian and Lagrangian mechanics

**70H12** Periodic and almost periodic solutions for problems in Hamiltonian and Lagrangian mechanics

**70H14** Stability problems for problems in Hamiltonian and Lagrangian mechanics

**70H15** Canonical and symplectic transformations for problems in Hamiltonian and Lagrangian mechanics

**70H20** Hamilton-Jacobi equations in mechanics

**70H25** Hamilton's principle

**70H30** Other variational principles in mechanics

**70H33** Symmetries and conservation laws, reverse symmetries, invariant manifolds and their bifurcations, reduction for problems in Hamiltonian and Lagrangian mechanics

**70H40** Relativistic dynamics for problems in Hamiltonian and Lagrangian mechanics

**70H45** Constrained dynamics, Dirac's theory of constraints [See also 70F20, 70F25, 70Gxx]

**70H50** Higher-order theories for problems in Hamiltonian and Lagrangian mechanics

**70H99** None of the above, but in this section

## 70Jxx Linear vibration theory

**70J10** Modal analysis in linear vibration theory

**70J25** Stability for problems in linear vibration theory

**70J30** Free motions in linear vibration theory

**70J35** Forced motions in linear vibration theory

**70J40** Parametric resonances in linear vibration theory

**70J50** Systems arising from the discretization of structural vibration problems

**70J99** None of the above, but in this section

## 70Kxx Nonlinear dynamics in mechanics [See also 34Cxx, 37-XX]

**70K05** Phase plane analysis, limit cycles for nonlinear problems in mechanics

**70K20** Stability for nonlinear problems in mechanics

**70K25** Free motions for nonlinear problems in mechanics

**70K28** Parametric resonances for nonlinear problems in mechanics

**70K30** Nonlinear resonances for nonlinear problems in mechanics

**70K40** Forced motions for nonlinear problems in mechanics

**70K42** Equilibria and periodic trajectories for nonlinear problems in mechanics

**70K43** Quasi-periodic motions and invariant tori for nonlinear problems in mechanics

**70K44** Homoclinic and heteroclinic trajectories for nonlinear problems in mechanics

**70K45** Normal forms for nonlinear problems in mechanics

**70K50** Bifurcations and instability for nonlinear problems in mechanics

**70K55** Transition to stochasticity (chaotic behavior) for nonlinear problems in mechanics [See also 37D45]

**70K60** General perturbation schemes for nonlinear problems in mechanics

**70K65** Averaging of perturbations for nonlinear problems in mechanics

**70K70** Systems with slow and fast motions for nonlinear problems in mechanics

**70K75** Nonlinear modes

**70K99** None of the above, but in this section

## 70Lxx Random and stochastic aspects of the mechanics of particles and systems

**70L05** Random vibrations in mechanics of particles and systems [See also 74H50]

**70L10** Stochastic geometric mechanics

**70L99** None of the above, but in this section

## 70Mxx Orbital mechanics

**70M20** Orbital mechanics

**70M99** None of the above, but in this section

## 70Pxx Variable mass, rockets

**70P05** Variable mass, rockets

**70P99** None of the above, but in this section

## 70Qxx Control of mechanical systems [See also 60Gxx, 60Jxx]

**70Q05** Control of mechanical systems

**70Q99** None of the above, but in this section

**70Sxx Classical field theories** [See also 37Kxx, 37Lxx, 78-XX, 81Txx, 83-XX]

**70S05** Lagrangian formalism and Hamiltonian formalism in mechanics of particles and systems

**70S10** Symmetries and conservation laws in mechanics of particles and systems

**70S15** Yang-Mills and other gauge theories in mechanics of particles and systems

**70S20** More general nonquantum field theories in mechanics of particles and systems

**70S99** None of the above, but in this section

# 74-XX Mechanics of deformable solids

**74-00** General reference works (handbooks, dictionaries, bibliographies, etc.) pertaining to mechanics of deformable solids

**74-01** Introductory exposition (textbooks, tutorial papers, etc.) pertaining to mechanics of deformable solids

**74-02** Research exposition (monographs, survey articles) pertaining to mechanics of deformable solids

**74-03** History of mechanics of deformable solids [Consider also classification numbers from Section 01]

**74-04** Software, source code, etc. for problems pertaining to mechanics of deformable solids

**74-05** Experimental work for problems pertaining to mechanics of deformable solids

**74-06** Proceedings, conferences, collections, etc. pertaining to mechanics of deformable solids

**74-10** Mathematical modeling or simulation for problems pertaining to mechanics of deformable solids

**74-11** Research data for problems pertaining to mechanics of deformable solids

## 74Axx Generalities, axiomatics, foundations of continuum mechanics of solids

**74A05** Kinematics of deformation

**74A10** Stress

**74A15** Thermodynamics in solid mechanics

**74A20** Theory of constitutive functions in solid mechanics

**74A25** Molecular, statistical, and kinetic theories in solid mechanics

**74A30** Nonsimple materials

**74A35** Polar materials

**74A40** Random materials and composite materials

**74A45** Theories of fracture and damage

**74A50** Structured surfaces and interfaces, coexistent phases

**74A55** Theories of friction (tribology)

**74A60** Micromechanical theories

**74A65** Reactive materials

**74A70** Peridynamics

**74A99** None of the above, but in this section

## 74Bxx Elastic materials

**74B05** Classical linear elasticity

**74B10** Linear elasticity with initial stresses

**74B15** Equations linearized about a deformed state (small deformations superposed on large)

**74B20** Nonlinear elasticity

**74B99** None of the above, but in this section

## 74Cxx Plastic materials, materials of stress-rate and internal-variable type

**74C05** Small-strain, rate-independent theories of plasticity (including rigid-plastic and elasto-plastic materials)

**74C10** Small-strain, rate-dependent theories of plasticity (including theories of viscoplasticity)

**74C15** Large-strain, rate-independent theories of plasticity (including nonlinear plasticity)

**74C20** Large-strain, rate-dependent theories of plasticity

**74C99** None of the above, but in this section

## 74Dxx Materials of strain-rate type and history type, other materials with memory (including elastic materials with viscous damping, various viscoelastic materials)

**74D05** Linear constitutive equations for materials with memory

**74D10** Nonlinear constitutive equations for materials with memory

**74D99** None of the above, but in this section

## 74Exx Material properties given special treatment

**74E05** Inhomogeneity in solid mechanics

**74E10** Anisotropy in solid mechanics

**74E15** Crystalline structure

**74E20** Granularity

**74E25** Texture in solid mechanics

**74E30** Composite and mixture properties

**74E35** Random structure in solid mechanics

**74E40** Chemical structure in solid mechanics

**74E99** None of the above, but in this section

## 74Fxx Coupling of solid mechanics with other effects

**74F05** Thermal effects in solid mechanics

**74F10** Fluid-solid interactions (including aero- and hydro-elasticity, porosity, etc.)

**74F15** Electromagnetic effects in solid mechanics

**74F20** Mixture effects in solid mechanics

**74F25** Chemical and reactive effects in solid mechanics

**74F99** None of the above, but in this section

## 74Gxx Equilibrium (steady-state) problems in solid mechanics

**74G05** Explicit solutions of equilibrium problems in solid mechanics

**74G10** Analytic approximation of solutions (perturbation methods, asymptotic methods, series, etc.) of equilibrium problems in solid mechanics

**74G15** Numerical approximation of solutions of equilibrium problems in solid mechanics

**74G22** Existence of solutions of equilibrium problems in solid mechanics

**74G30** Uniqueness of solutions of equilibrium problems in solid mechanics

**74G35** Multiplicity of solutions of equilibrium problems in solid mechanics

**74G40** Regularity of solutions of equilibrium problems in solid mechanics

**74G45** Bounds for solutions of equilibrium problems in solid mechanics

**74G50** Saint-Venant's principle

**74G55** Qualitative behavior of solutions of equilibrium problems in solid mechanics

**74G60** Bifurcation and buckling

**74G65** Energy minimization in equilibrium problems in solid mechanics

**74G70** Stress concentrations, singularities in solid mechanics

**74G75** Inverse problems in equilibrium solid mechanics

**74G99** None of the above, but in this section

## 74Hxx Dynamical problems in solid mechanics

**74H05** Explicit solutions of dynamical problems in solid mechanics

**74H10** Analytic approximation of solutions (perturbation methods, asymptotic methods, series, etc.) of dynamical problems in solid mechanics

**74H15** Numerical approximation of solutions of dynamical problems in solid mechanics

**74H20** Existence of solutions of dynamical problems in solid mechanics

**74H25** Uniqueness of solutions of dynamical problems in solid mechanics

**74H30** Regularity of solutions of dynamical problems in solid mechanics

**74H35** Singularities, blow-up, stress concentrations for dynamical problems in solid mechanics

**74H40** Long-time behavior of solutions for dynamical problems in solid mechanics

**74H45** Vibrations in dynamical problems in solid mechanics

**74H50** Random vibrations in dynamical problems in solid mechanics

**74H55** Stability of dynamical problems in solid mechanics

**74H60** Dynamical bifurcation of solutions to dynamical problems in solid mechanics

**74H65** Chaotic behavior of solutions to dynamical problems in solid mechanics

**74H75** Inverse problems in dynamical solid mechanics

**74H80** Energy minimization in dynamical problems in solid mechanics

**74H99** None of the above, but in this section

## 74Jxx Waves in solid mechanics

**74J05** Linear waves in solid mechanics

**74J10** Bulk waves in solid mechanics

**74J15** Surface waves in solid mechanics

**74J20** Wave scattering in solid mechanics

**74J25** Inverse problems for waves in solid mechanics

**74J30** Nonlinear waves in solid mechanics

**74J35** Solitary waves in solid mechanics

**74J40** Shocks and related discontinuities in solid mechanics

**74J99** None of the above, but in this section

## 74Kxx Thin bodies, structures

**74K05** Strings

**74K10** Rods (beams, columns, shafts, arches, rings, etc.)

**74K15** Membranes

**74K20** Plates

**74K25** Shells

**74K30** Junctions

**74K35** Thin films

**74K99** None of the above, but in this section

## 74Lxx Special subfields of solid mechanics

**74L05** Geophysical solid mechanics [See also 86-XX]

**74L10** Soil and rock mechanics

**74L15** Biomechanical solid mechanics [See also 92C10]

**74L99** None of the above, but in this section

## 74Mxx Special kinds of problems in solid mechanics

**74M05** Control, switches and devices ("smart materials") in solid mechanics [See also 93Cxx]

**74M10** Friction in solid mechanics

**74M15** Contact in solid mechanics

**74M20** Impact in solid mechanics

**74M25** Micromechanics of solids

**74M99** None of the above, but in this section

## 74Nxx Phase transformations in solids [See also 74A50, 80A22, 82B26, 82C26]

**74N05** Crystals in solids

**74N10** Displacive transformations in solids

**74N15** Analysis of microstructure in solids

**74N20** Dynamics of phase boundaries in solids

**74N25** Transformations involving diffusion in solids

**74N30** Problems involving hysteresis in solids

**74N99** None of the above, but in this section

## 74Pxx Optimization problems in solid mechanics [See also 49Qxx]

**74P05** Compliance or weight optimization in solid mechanics

**74P10** Optimization of other properties in solid mechanics

**74P15** Topological methods for optimization problems in solid mechanics

**74P20** Geometrical methods for optimization problems in solid mechanics

**74P99** None of the above, but in this section

## 74Qxx Homogenization, determination of effective properties in solid mechanics

**74Q05** Homogenization in equilibrium problems of solid mechanics

**74Q10** Homogenization and oscillations in dynamical problems of solid mechanics

**74Q15** Effective constitutive equations in solid mechanics

**74Q20** Bounds on effective properties in solid mechanics

**74Q99** None of the above, but in this section

## 74Rxx Fracture and damage

**74R05** Brittle damage

**74R10** Brittle fracture

**74R15** High-velocity fracture

**74R20** Anelastic fracture and damage

**74R99** None of the above, but in this section

**74Sxx Numerical and other methods in solid mechanics** [See also 65-XX, 74G15, 74H15]

**74S05** Finite element methods applied to problems in solid mechanics

**74S10** Finite volume methods applied to problems in solid mechanics

**74S15** Boundary element methods applied to problems in solid mechanics

**74S20** Finite difference methods applied to problems in solid mechanics

**74S22** Isogeometric methods applied to problems in solid mechanics

**74S25** Spectral and related methods applied to problems in solid mechanics

**74S40** Applications of fractional calculus in solid mechanics

**74S50** Applications of graph theory in solid mechanics

**74S60** Stochastic and other probabilistic methods applied to problems in solid mechanics

**74S70** Complex-variable methods applied to problems in solid mechanics

**74S99** None of the above, but in this section

# 76-XX Fluid mechanics {For general continuum mechanics, see 74Axx, or other parts of 74-XX}

**76-00** General reference works (handbooks, dictionaries, bibliographies, etc.) pertaining to fluid mechanics

**76-01** Introductory exposition (textbooks, tutorial papers, etc.) pertaining to fluid mechanics

**76-02** Research exposition (monographs, survey articles) pertaining to fluid mechanics

**76-03** History of fluid mechanics [Consider also classification numbers from Section 01]

**76-04** Software, source code, etc. for problems pertaining to fluid mechanics

**76-05** Experimental work for problems pertaining to fluid mechanics

**76-06** Proceedings, conferences, collections, etc. pertaining to fluid mechanics

**76-10** Mathematical modeling or simulation for problems pertaining to fluid mechanics

**76-11** Research data for problems pertaining to fluid mechanics

## 76Axx Foundations, constitutive equations, rheology, hydrodynamical models of non-fluid phenomena

**76A02** Foundations of fluid mechanics

**76A05** Non-Newtonian fluids

**76A10** Viscoelastic fluids

**76A15** Liquid crystals [See also 82D30]

**76A20** Thin fluid films

**76A25** Superfluids (classical aspects)

**76A30** Traffic and pedestrian flow models

**76A99** None of the above, but in this section

## 76Bxx Incompressible inviscid fluids

**76B03** Existence, uniqueness, and regularity theory for incompressible inviscid fluids [See also 35Q35]

**76B07** Free-surface potential flows for incompressible inviscid fluids

**76B10** Jets and cavities, cavitation, free-streamline theory, water-entry problems, airfoil and hydrofoil theory, sloshing

**76B15** Water waves, gravity waves; dispersion and scattering, nonlinear interaction [See also 35Q30]

**76B20** Ship waves

**76B25** Solitary waves for incompressible inviscid fluids [See also 35C11]

**76B45** Capillarity (surface tension) for incompressible inviscid fluids [See also 76D45]

**76B47** Vortex flows for incompressible inviscid fluids

**76B55** Internal waves for incompressible inviscid fluids

**76B70** Stratification effects in inviscid fluids

**76B75** Flow control and optimization for incompressible inviscid fluids [See also 49Q10, 93C20, 93C95]

**76B99** None of the above, but in this section

## 76Dxx Incompressible viscous fluids

**76D03** Existence, uniqueness, and regularity theory for incompressible viscous fluids [See also 35Q30]

**76D05** Navier-Stokes equations for incompressible viscous fluids [See also 35Q30]

**76D06** Statistical solutions of Navier-Stokes and related equations [See also 60H30, 76M35]

**76D07** Stokes and related (Oseen, etc.) flows

**76D08** Lubrication theory

**76D09** Viscous-inviscid interaction

**76D10** Boundary-layer theory, separation and reattachment, higher-order effects

**76D17** Viscous vortex flows

**76D25** Wakes and jets

**76D27** Other free boundary flows; Hele-Shaw flows

**76D33** Waves for incompressible viscous fluids

**76D45** Capillarity (surface tension) for incompressible viscous fluids [See also 76B45]

**76D50** Stratification effects in viscous fluids

**76D55** Flow control and optimization for incompressible viscous fluids [See also 49Q10, 93C20, 93C95]

**76D99** None of the above, but in this section

## 76Exx Hydrodynamic stability

**76E05** Parallel shear flows in hydrodynamic stability

**76E06** Convection in hydrodynamic stability

**76E07** Rotation in hydrodynamic stability

**76E09** Stability and instability of nonparallel flows in hydrodynamic stability

**76E15** Absolute and convective instability and stability in hydrodynamic stability

**76E17** Interfacial stability and instability in hydrodynamic stability

**76E19** Compressibility effects in hydrodynamic stability

**76E20** Stability and instability of geophysical and astrophysical flows

**76E25** Stability and instability of magnetohydrodynamic and electrohydrodynamic flows

**76E30** Nonlinear effects in hydrodynamic stability

**76E99** None of the above, but in this section

## 76Fxx Turbulence [See also 37-XX, 60Gxx, 60Jxx]

**76F02** Fundamentals of turbulence

**76F05** Isotropic turbulence; homogeneous turbulence

**76F06** Transition to turbulence

**76F10** Shear flows and turbulence

**76F20** Dynamical systems approach to turbulence [See also 37-XX]

**76F25** Turbulent transport, mixing

**76F30** Renormalization and other field-theoretical methods for turbulence [See also 81T99]

**76F35** Convective turbulence [See also 76E15, 76Rxx]

**76F40** Turbulent boundary layers

**76F45** Stratification effects in turbulence

**76F50** Compressibility effects in turbulence

**76F55** Statistical turbulence modeling [See also 76M35]

**76F60** $k$-$\varepsilon$ modeling in turbulence

**76F65** Direct numerical and large eddy simulation of turbulence

**76F70** Control of turbulent flows

**76F80** Turbulent combustion; reactive turbulence

**76F99** None of the above, but in this section

## 76Gxx General aerodynamics and subsonic flows

**76G25** General aerodynamics and subsonic flows

**76G99** None of the above, but in this section

## 76Hxx Transonic flows

**76H05** Transonic flows

**76H99** None of the above, but in this section

## 76Jxx Supersonic flows

**76J20** Supersonic flows

**76J99** None of the above, but in this section

## 76Kxx Hypersonic flows

**76K05** Hypersonic flows

**76K99** None of the above, but in this section

## 76Lxx Shock waves and blast waves in fluid mechanics [See also 35L67]

**76L05** Shock waves and blast waves in fluid mechanics [See also 35L67]

**76L99** None of the above, but in this section

## 76Mxx Basic methods in fluid mechanics [See also 65-XX]

**76M10** Finite element methods applied to problems in fluid mechanics

**76M12** Finite volume methods applied to problems in fluid mechanics

**76M15** Boundary element methods applied to problems in fluid mechanics

**76M20** Finite difference methods applied to problems in fluid mechanics

**76M21** Inverse problems in fluid mechanics

**76M22** Spectral methods applied to problems in fluid mechanics

**76M23** Vortex methods applied to problems in fluid mechanics

**76M27** Visualization algorithms applied to problems in fluid mechanics

**76M28** Particle methods and lattice-gas methods

**76M30** Variational methods applied to problems in fluid mechanics

**76M35** Stochastic analysis applied to problems in fluid mechanics

**76M40** Complex variables methods applied to problems in fluid mechanics

**76M45** Asymptotic methods, singular perturbations applied to problems in fluid mechanics

**76M50** Homogenization applied to problems in fluid mechanics

**76M55** Dimensional analysis and similarity applied to problems in fluid mechanics

**76M60** Symmetry analysis, Lie group and Lie algebra methods applied to problems in fluid mechanics

**76M99** None of the above, but in this section

## 76Nxx Compressible fluids and gas dynamics

**76N06** Compressible Navier-Stokes equations

**76N10** Existence, uniqueness, and regularity theory for compressible fluids and gas dynamics [See also 35L60, 35L65, 35Q30]

**76N15** Gas dynamics (general theory)

**76N17** Viscous-inviscid interaction for compressible fluids and gas dynamics

**76N20** Boundary-layer theory for compressible fluids and gas dynamics

**76N25** Flow control and optimization for compressible fluids and gas dynamics

**76N30** Waves in compressible fluids

**76N99** None of the above, but in this section

## 76Pxx Rarefied gas flows, Boltzmann equation in fluid mechanics [See also 82B40, 82C40, 82D05]

**76P05** Rarefied gas flows, Boltzmann equation in fluid mechanics [See also 82B40, 82C40, 82D05]

**76P99** None of the above, but in this section

## 76Qxx Hydro- and aero-acoustics

**76Q05** Hydro- and aero-acoustics

**76Q99** None of the above, but in this section

## 76Rxx Diffusion and convection

**76R05** Forced convection

**76R10** Free convection

**76R50** Diffusion [See also 60J60]

**76R99** None of the above, but in this section

## 76Sxx Flows in porous media; filtration; seepage

**76S05** Flows in porous media; filtration; seepage

**76S99** None of the above, but in this section

## 76Txx Multiphase and multicomponent flows

**76T06** Liquid-liquid two component flows

**76T10** Liquid-gas two-phase flows, bubbly flows

**76T15** Dusty-gas two-phase flows

**76T17** Two gas multicomponent flows

**76T20** Suspensions

**76T25** Granular flows [See also 74C99, 74E20]

**76T30** Three or more component flows

**76T99** None of the above, but in this section

## 76Uxx Rotating fluids

**76U05** General theory of rotating fluids

**76U60** Geophysical flows [See also 86A05, 86A10]

**76U65** Rossby waves [See also 86A05, 86A10]

**76U99** None of the above, but in this section

## 76Vxx Reaction effects in flows [See also 80A32]

**76V05** Reaction effects in flows [See also 80A32]

**76V99** None of the above, but in this section

## 76Wxx Magnetohydrodynamics and electrohydrodynamics

**76W05** Magnetohydrodynamics and electrohydrodynamics

**76W99** None of the above, but in this section

## 76Xxx Ionized gas flow in electromagnetic fields; plasmic flow [See also 82D10]

**76X05** Ionized gas flow in electromagnetic fields; plasmic flow [See also 82D10]

**76X99** None of the above, but in this section

## 76Yxx Quantum hydrodynamics and relativistic hydrodynamics [See also 82D50, 83C55, 85A30]

**76Y05** Quantum hydrodynamics and relativistic hydrodynamics [See also 82D50, 83C55, 85A30]

**76Y99** None of the above, but in this section

## 76Zxx Biological fluid mechanics [See also 74F10, 74L15, 92Cxx]

**76Z05** Physiological flows [See also 92C35]

**76Z10** Biopropulsion in water and in air

**76Z99** None of the above, but in this section

# 78-XX Optics, electromagnetic theory {For quantum optics, see 81V80}

**78-00** General reference works (handbooks, dictionaries, bibliographies, etc.) pertaining to optics and electromagnetic theory

**78-01** Introductory exposition (textbooks, tutorial papers, etc.) pertaining to optics and electromagnetic theory

**78-02** Research exposition (monographs, survey articles) pertaining to optics and electromagnetic theory

**78-03** History of optics and electromagnetic theory [Consider also classification numbers from Section 01]

**78-04** Software, source code, etc. for problems pertaining to optics and electromagnetic theory

**78-05** Experimental work for problems pertaining to optics and electromagnetic theory

**78-06** Proceedings, conferences, collections, etc. pertaining to optics and electromagnetic theory

**78-10** Mathematical modeling or simulation for problems pertaining to optics and electromagnetic theory

**78-11** Research data for problems pertaining to optics and electromagnetic theory

## 78Axx General topics in optics and electromagnetic theory

**78A02** Foundations in optics and electromagnetic theory

**78A05** Geometric optics

**78A10** Physical optics

**78A15** Electron optics

**78A20** Space charge waves

**78A25** Electromagnetic theory (general)

**78A30** Electro- and magnetostatics

**78A35** Motion of charged particles

**78A37** Ion traps

**78A40** Waves and radiation in optics and electromagnetic theory

**78A45** Diffraction, scattering {For WKB methods, see also 34E20}

**78A46** Inverse problems (including inverse scattering) in optics and electromagnetic theory

**78A48** Composite media; random media in optics and electromagnetic theory

**78A50** Antennas, waveguides in optics and electromagnetic theory

**78A55** Technical applications of optics and electromagnetic theory

**78A57** Electrochemistry

**78A60** Lasers, masers, optical bistability, nonlinear optics [See also 81V80]

**78A70** Biological applications of optics and electromagnetic theory [See also 92-XX]

**78A97** Mathematically heuristic optics and electromagnetic theory (must also be assigned at least one other classification number in Section 78)

**78A99** None of the above, but in this section

## 78Mxx Basic methods for problems in optics and electromagnetic theory [See also 65-XX]

**78M05** Method of moments applied to problems in optics and electromagnetic theory

**78M10** Finite element, Galerkin and related methods applied to problems in optics and electromagnetic theory

**78M12** Finite volume methods, finite integration techniques applied to problems in optics and electromagnetic theory

**78M15** Boundary element methods applied to problems in optics and electromagnetic theory

**78M16** Multipole methods applied to problems in optics and electromagnetic theory

**78M20** Finite difference methods applied to problems in optics and electromagnetic theory

**78M22** Spectral, collocation and related methods applied to problems in optics and electromagnetic theory

**78M30** Variational methods applied to problems in optics and electromagnetic theory

**78M31** Monte Carlo methods applied to problems in optics and electromagnetic theory

**78M32** Neural and heuristic methods applied to problems in optics and electromagnetic theory

**78M34** Model reduction in optics and electromagnetic theory

**78M35** Asymptotic analysis in optics and electromagnetic theory

**78M40** Homogenization in optics and electromagnetic theory

**78M50** Optimization problems in optics and electromagnetic theory

**78M99** None of the above, but in this section

# 80-XX Classical thermodynamics, heat transfer {For thermodynamics of solids, see 74A15}

**80-00** General reference works (handbooks, dictionaries, bibliographies, etc.) pertaining to classical thermodynamics

**80-01** Introductory exposition (textbooks, tutorial papers, etc.) pertaining to classical thermodynamics

**80-02** Research exposition (monographs, survey articles) pertaining to classical thermodynamics

**80-03** History of classical thermodynamics [Consider also classification numbers from Section 01]

**80-04** Software, source code, etc. for problems pertaining to classical thermodynamics

**80-05** Experimental work for problems pertaining to classical thermodynamics

**80-06** Proceedings, conferences, collections, etc. pertaining to classical thermodynamics

**80-10** Mathematical modeling or simulation for problems pertaining to classical thermodynamics

**80-11** Research data for problems pertaining to classical thermodynamics

## 80Axx Thermodynamics and heat transfer

**80A05** Foundations of thermodynamics and heat transfer

**80A10** Classical and relativistic thermodynamics

**80A17** Thermodynamics of continua [See also 74A15]

**80A19** Diffusive and convective heat and mass transfer, heat flow

**80A21** Radiative heat transfer

**80A22** Stefan problems, phase changes, etc. [See also 74Nxx]

**80A23** Inverse problems in thermodynamics and heat transfer

**80A25** Combustion

**80A30** Chemical kinetics in thermodynamics and heat transfer [See also 76V05, 92C45, 92E20]

**80A32** Chemically reacting flows [See also 92C45, 92E20]

**80A50** Chemistry (general) in thermodynamics and heat transfer [See mainly 92Exx]

**80A99** None of the above, but in this section

**80Mxx Basic methods in thermodynamics and heat transfer [See also 65-XX]**

**80M10** Finite element, Galerkin and related methods applied to problems in thermodynamics and heat transfer

**80M12** Finite volume methods applied to problems in thermodynamics and heat transfer

**80M15** Boundary element methods applied to problems in thermodynamics and heat transfer

**80M20** Finite difference methods applied to problems in thermodynamics and heat transfer

**80M22** Spectral, collocation and related (meshless) methods applied to problems in thermodynamics and heat transfer

**80M30** Variational methods applied to problems in thermodynamics and heat transfer

**80M31** Monte Carlo methods applied to problems in thermodynamics and heat transfer

**80M35** Asymptotic analysis for problems in thermodynamics and heat transfer

**80M40** Homogenization for problems in thermodynamics and heat transfer

**80M50** Optimization problems in thermodynamics and heat transfer

**80M60** Stochastic analysis in thermodynamics and heat transfer

**80M99** None of the above, but in this section

# 81-XX Quantum theory

**81-00** General reference works (handbooks, dictionaries, bibliographies, etc.) pertaining to quantum theory

**81-01** Introductory exposition (textbooks, tutorial papers, etc.) pertaining to quantum theory

**81-02** Research exposition (monographs, survey articles) pertaining to quantum theory

**81-03** History of quantum theory [Consider also classification numbers from Section 01]

**81-04** Software, source code, etc. for problems pertaining to quantum theory

**81-05** Experimental work for problems pertaining to quantum theory

**81-06** Proceedings, conferences, collections, etc. pertaining to quantum theory

**81-08** Computational methods for problems pertaining to quantum theory

**81-10** Mathematical modeling or simulation for problems pertaining to quantum theory

**81-11** Research data for problems pertaining to quantum theory

## 81Pxx Foundations, quantum information and its processing, quantum axioms, and philosophy

**81P05** General and philosophical questions in quantum theory

**81P10** Logical foundations of quantum mechanics; quantum logic (quantum-theoretic aspects) [See also 03G12, 06C15]

**81P13** Contextuality in quantum theory

**81P15** Quantum measurement theory, state operations, state preparations

**81P16** Quantum state spaces, operational and probabilistic concepts

**81P17** Quantum entropies

**81P18** Quantum state tomography, quantum state discrimination

**81P20** Stochastic mechanics (including stochastic electrodynamics)

**81P40** Quantum coherence, entanglement, quantum correlations

**81P42** Entanglement measures, concurrencies, separability criteria

**81P43** Quantum discord

**81P45** Quantum information, communication, networks (quantum-theoretic aspects) [See also 94A15, 94A17]

**81P47** Quantum channels, fidelity [See also 94A40]

**81P48** LOCC, teleportation, dense coding, remote state operations, distillation

**81P50** Quantum state estimation, approximate cloning

**81P55** Special bases (entangled, mutual unbiased, etc.)

**81P65** Quantum gates

**81P68** Quantum computation [See also 68Q09] {For algorithmic aspects, see 68Q12}

**81P70** Quantum coding (general)

**81P73** Computational stability and error-correcting codes for quantum computation and communication processing

**81P94** Quantum cryptography (quantum-theoretic aspects) [See also 94A60]

**81P99** None of the above, but in this section

## 81Qxx General mathematical topics and methods in quantum theory

**81Q05** Closed and approximate solutions to the Schrödinger, Dirac, Klein-Gordon and other equations of quantum mechanics

**81Q10** Selfadjoint operator theory in quantum theory, including spectral analysis

**81Q12** Nonselfadjoint operator theory in quantum theory including creation and destruction operators

**81Q15** Perturbation theories for operators and differential equations in quantum theory

**81Q20** Semiclassical techniques, including WKB and Maslov methods applied to problems in quantum theory

**81Q30** Feynman integrals and graphs; applications of algebraic topology and algebraic geometry [See also 05Cxx, 14D05, 32S40]

**81Q35** Quantum mechanics on special spaces: manifolds, fractals, graphs, lattices [See also 81R20]

**81Q37** Quantum dots, waveguides, ratchets, etc. [See also 82D20, 82D77]

**81Q40** Bethe-Salpeter and other integral equations arising in quantum theory

**81Q50** Quantum chaos [See also 37D45]

**81Q60** Supersymmetry and quantum mechanics

**81Q65** Alternative quantum mechanics (including hidden variables, etc.)

**81Q70** Differential geometric methods, including holonomy, Berry and Hannay phases, Aharonov-Bohm effect, etc. in quantum theory

**81Q80** Special quantum systems, such as solvable systems

**81Q93** Quantum control

**81Q99** None of the above, but in this section

## 81Rxx Groups and algebras in quantum theory

**81R05** Finite-dimensional groups and algebras motivated by physics and their representations [See also 20C35, 22E70]

**81R10** Infinite-dimensional groups and algebras motivated by physics, including Virasoro, Kac-Moody, $W$-algebras and other current algebras and their representations [See also 17B65, 17B67, 22E65, 22E67, 22E70]

**81R12** Groups and algebras in quantum theory and relations with integrable systems [See also 17Bxx, 37J35]

**81R15** Operator algebra methods applied to problems in quantum theory [See also 46Lxx, 81T05]

**81R20** Covariant wave equations in quantum theory, relativistic quantum mechanics [See also 81Q35]

**81R25** Spinor and twistor methods applied to problems in quantum theory [See also 32L25]

**81R30** Coherent states [See also 22E45]; squeezed states in quantum theory [See also 81V80]

**81R40** Symmetry breaking in quantum theory

**81R50** Quantum groups and related algebraic methods applied to problems in quantum theory [See also 16T20, 17B37]

**81R60** Noncommutative geometry in quantum theory

**81R99** None of the above, but in this section

## 81Sxx General quantum mechanics and problems of quantization

**81S05** Commutation relations and statistics as related to quantum mechanics (general)

**81S07** Uncertainty relations, also entropic

**81S08** Canonical quantization

**81S10** Geometry and quantization, symplectic methods [See also 53D50]

**81S20** Stochastic quantization

**81S22** Open systems, reduced dynamics, master equations, decoherence [See also 82C31]

**81S25** Quantum stochastic calculus

**81S30** Phase-space methods including Wigner distributions, etc. applied to problems in quantum mechanics

**81S40** Path integrals in quantum mechanics [See also 58D30, 81Q30, 81T18]

**81S99** None of the above, but in this section

## 81Txx Quantum field theory; related classical field theories [See also 70Sxx]

**81T05** Axiomatic quantum field theory; operator algebras

**81T08** Constructive quantum field theory

**81T10** Model quantum field theories

**81T11** Higher spin theories

**81T12** Effective quantum field theories

**81T13** Yang-Mills and other gauge theories in quantum field theory [See also 53C07, 58E15]

**81T15** Perturbative methods of renormalization applied to problems in quantum field theory

**81T16** Nonperturbative methods of renormalization applied to problems in quantum field theory

**81T17** Renormalization group methods applied to problems in quantum field theory

**81T18** Feynman diagrams

**81T20** Quantum field theory on curved space or space-time backgrounds

**81T25** Quantum field theory on lattices

**81T27** Continuum limits in quantum field theory

**81T28** Thermal quantum field theory [See also 82B30]

**81T30** String and superstring theories; other extended objects (e.g., branes) in quantum field theory [See also 83E30]

**81T32** Matrix models and tensor models for quantum field theory

**81T33** Dimensional compactification in quantum field theory

**81T35** Correspondence, duality, holography (AdS/CFT, gauge/gravity, etc.) [See also 83E05]

**81T40** Two-dimensional field theories, conformal field theories, etc. in quantum mechanics

**81T45** Topological field theories in quantum mechanics [See also 57R56, 58Dxx]

**81T50** Anomalies in quantum field theory

**81T55** Casimir effect in quantum field theory

**81T60** Supersymmetric field theories in quantum mechanics

**81T70** Quantization in field theory; cohomological methods [See also 58D29]

**81T75** Noncommutative geometry methods in quantum field theory [See also 46L85, 46L87, 58B34]

**81T99** None of the above, but in this section

## 81Uxx Quantum scattering theory [See also 34A55, 34L25, 34L40, 35P25, 47A40]

**81U05** 2-body potential quantum scattering theory {For WKB methods, see also 34E20}

**81U10** $n$-body potential quantum scattering theory

**81U15** Exactly and quasi-solvable systems arising in quantum theory

**81U20** $S$-matrix theory, etc. in quantum theory

**81U24** Resonances in quantum scattering theory

**81U26** Tunneling in quantum theory

**81U30** Dispersion theory, dispersion relations arising in quantum theory

**81U35** Inelastic and multichannel quantum scattering

**81U40** Inverse scattering problems in quantum theory

**81U90** Particle decays

**81U99** None of the above, but in this section

### 81Vxx Applications of quantum theory to specific physical systems

**81V05** Strong interaction, including quantum chromodynamics

**81V10** Electromagnetic interaction; quantum electrodynamics

**81V15** Weak interaction in quantum theory

**81V17** Gravitational interaction in quantum theory [See also 83Cxx, 83Exx]

**81V19** Other fundamental interactions in quantum theory

**81V22** Unified quantum theories

**81V25** Other elementary particle theory in quantum theory

**81V27** Anyons

**81V35** Nuclear physics

**81V45** Atomic physics

**81V55** Molecular physics [See also 92E10]

**81V60** Mono-, di- and multipole moments (EM and other), gyromagnetic relations

**81V65** Quantum dots as quasi particles [See also 82D20]

**81V70** Many-body theory; quantum Hall effect

**81V72** Particle exchange symmetries in quantum theory (general)

**81V73** Bosonic systems in quantum theory

**81V74** Fermionic systems in quantum theory

**81V80** Quantum optics

**81V99** None of the above, but in this section

# 82-XX Statistical mechanics, structure of matter

**82-00** General reference works (handbooks, dictionaries, bibliographies, etc.) pertaining to statistical mechanics

**82-01** Introductory exposition (textbooks, tutorial papers, etc.) pertaining to statistical mechanics

**82-02** Research exposition (monographs, survey articles) pertaining to statistical mechanics

**82-03** History of statistical mechanics [Consider also classification numbers from Section 01]

**82-04** Software, source code, etc. for problems pertaining to statistical mechanics

**82-05** Experimental work for problems pertaining to statistical mechanics

**82-06** Proceedings, conferences, collections, etc. pertaining to statistical mechanics

**82-10** Mathematical modeling or simulation for problems pertaining to statistical mechanics

**82-11** Research data for problems pertaining to statistical mechanics

## 82Bxx Equilibrium statistical mechanics

**82B03** Foundations of equilibrium statistical mechanics

**82B05** Classical equilibrium statistical mechanics (general)

**82B10** Quantum equilibrium statistical mechanics (general)

**82B20** Lattice systems (Ising, dimer, Potts, etc.) and systems on graphs arising in equilibrium statistical mechanics [See also 05Cxx]

**82B21** Continuum models (systems of particles, etc.) arising in equilibrium statistical mechanics

**82B23** Exactly solvable models; Bethe ansatz

**82B24** Interface problems; diffusion-limited aggregation arising in equilibrium statistical mechanics

**82B26** Phase transitions (general) in equilibrium statistical mechanics

**82B27** Critical phenomena in equilibrium statistical mechanics

**82B28** Renormalization group methods in equilibrium statistical mechanics [See also 81T17]

**82B30** Statistical thermodynamics [See also 80-XX]

**82B31** Stochastic methods applied to problems in equilibrium statistical mechanics

**82B35** Irreversible thermodynamics, including Onsager-Machlup theory [See also 92E20]

**82B40** Kinetic theory of gases in equilibrium statistical mechanics

**82B41** Random walks, random surfaces, lattice animals, etc. in equilibrium statistical mechanics [See also 60G50, 82C41]

**82B43** Percolation [See also 60K35]

**82B44** Disordered systems (random Ising models, random Schrödinger operators, etc.) in equilibrium statistical mechanics

**82B99** None of the above, but in this section

## 82Cxx Time-dependent statistical mechanics (dynamic and nonequilibrium)

**82C03** Foundations of time-dependent statistical mechanics

**82C05** Classical dynamic and nonequilibrium statistical mechanics (general)

**82C10** Quantum dynamics and nonequilibrium statistical mechanics (general)

**82C20** Dynamic lattice systems (kinetic Ising, etc.) and systems on graphs in time-dependent statistical mechanics [See also 05Cxx]

**82C21** Dynamic continuum models (systems of particles, etc.) in time-dependent statistical mechanics

**82C22** Interacting particle systems in time-dependent statistical mechanics [See also 60K35]

**82C23** Exactly solvable dynamic models in time-dependent statistical mechanics [See also 37K60]

**82C24** Interface problems; diffusion-limited aggregation in time-dependent statistical mechanics

**82C26** Dynamic and nonequilibrium phase transitions (general) in statistical mechanics

**82C27** Dynamic critical phenomena in statistical mechanics

**82C28** Dynamic renormalization group methods applied to problems in time-dependent statistical mechanics [See also 81T17]

**82C31** Stochastic methods (Fokker-Planck, Langevin, etc.) applied to problems in time-dependent statistical mechanics [See also 60H10]

**82C32** Neural nets applied to problems in time-dependent statistical mechanics [See also 68T05, 91E40, 92B20]

**82C35** Irreversible thermodynamics, including Onsager-Machlup theory

**82C40** Kinetic theory of gases in time-dependent statistical mechanics

**82C41** Dynamics of random walks, random surfaces, lattice animals, etc. in time-dependent statistical mechanics [See also 60G50]

**82C43** Time-dependent percolation in statistical mechanics [See also 60K35]

**82C44** Dynamics of disordered systems (random Ising systems, etc.) in time-dependent statistical mechanics

**82C70** Transport processes in time-dependent statistical mechanics

**82C99** None of the above, but in this section

## 82Dxx Applications of statistical mechanics to specific types of physical systems

**82D03** Statistical mechanics in condensed matter (general)

**82D05** Statistical mechanics of gases

**82D10** Statistical mechanics of plasmas

**82D15** Statistical mechanics of liquids

**82D20** Statistical mechanics of solids

**82D25** Statistical mechanics of crystals {For crystallographic group theory, see 20H15}

**82D30** Statistical mechanics of random media, disordered materials (including liquid crystals and spin glasses)

**82D35** Statistical mechanics of metals

**82D37** Statistical mechanics of semiconductors

**82D40** Statistical mechanics of magnetic materials

**82D45** Statistical mechanics of ferroelectrics

**82D50** Statistical mechanics of superfluids

**82D55** Statistical mechanics of superconductors

**82D60** Statistical mechanics of polymers

**82D75** Nuclear reactor theory; neutron transport

**82D77** Quantum waveguides, quantum wires [See also 78A50]

**82D80** Statistical mechanics of nanostructures and nanoparticles

**82D99** None of the above, but in this section

### 82Mxx Basic methods in statistical mechanics [See also 65-XX]

**82M10** Finite element, Galerkin and related methods applied to problems in statistical mechanics

**82M12** Finite volume methods applied to problems in statistical mechanics

**82M15** Boundary element methods applied to problems in statistical mechanics

**82M20** Finite difference methods applied to problems in statistical mechanics

**82M22** Spectral, collocation and related (meshless) methods applied to problems in statistical mechanics

**82M30** Variational methods applied to problems in statistical mechanics

**82M31** Monte Carlo methods applied to problems in statistical mechanics [See also 65C05]

**82M36** Computational density functional analysis in statistical mechanics

**82M37** Computational molecular dynamics in statistical mechanics

**82M60** Stochastic analysis in statistical mechanics [See also 65C35]

**82M99** None of the above, but in this section

# 83-XX Relativity and gravitational theory

**83-00** General reference works (handbooks, dictionaries, bibliographies, etc.) pertaining to relativity and gravitational theory

**83-01** Introductory exposition (textbooks, tutorial papers, etc.) pertaining to relativity and gravitational theory

**83-02** Research exposition (monographs, survey articles) pertaining to relativity and gravitational theory

**83-03** History of relativity and gravitational theory [Consider also classification numbers from Section 01]

**83-04** Software, source code, etc. for problems pertaining to relativity and gravitational theory

**83-05** Experimental work for problems pertaining to relativity and gravitational theory

**83-06** Proceedings, conferences, collections, etc. pertaining to relativity and gravitational theory

**83-08** Computational methods for problems pertaining to relativity and gravitational theory

**83-10** Mathematical modeling or simulation for problems pertaining to relativity and gravitational theory

**83-11** Research data for problems pertaining to relativity and gravitational theory

### 83Axx Special relativity

**83A05** Special relativity

**83A99** None of the above, but in this section

### 83Bxx Observational and experimental questions in relativity and gravitational theory

**83B05** Observational and experimental questions in relativity and gravitational theory

**83B99** None of the above, but in this section

## 83Cxx General relativity

**83C05** Einstein's equations (general structure, canonical formalism, Cauchy problems)

**83C10** Equations of motion in general relativity and gravitational theory

**83C15** Exact solutions to problems in general relativity and gravitational theory

**83C20** Classes of solutions; algebraically special solutions, metrics with symmetries for problems in general relativity and gravitational theory

**83C22** Einstein-Maxwell equations

**83C25** Approximation procedures, weak fields in general relativity and gravitational theory

**83C27** Lattice gravity, Regge calculus and other discrete methods in general relativity and gravitational theory

**83C30** Asymptotic procedures (radiation, news functions, $\mathcal{H}$-spaces, etc.) in general relativity and gravitational theory

**83C35** Gravitational waves

**83C40** Gravitational energy and conservation laws; groups of motions

**83C45** Quantization of the gravitational field

**83C47** Methods of quantum field theory in general relativity and gravitational theory [See also 81T20]

**83C50** Electromagnetic fields in general relativity and gravitational theory

**83C55** Macroscopic interaction of the gravitational field with matter (hydrodynamics, etc.)

**83C56** Dark matter and dark energy

**83C57** Black holes

**83C60** Spinor and twistor methods in general relativity and gravitational theory; Newman-Penrose formalism

**83C65** Methods of noncommutative geometry in general relativity [See also 58B34]

**83C75** Space-time singularities, cosmic censorship, etc.

**83C80** Analogues of general relativity in lower dimensions

**83C99** None of the above, but in this section

## 83Dxx Relativistic gravitational theories other than Einstein's, including asymmetric field theories

**83D05** Relativistic gravitational theories other than Einstein's, including asymmetric field theories

**83D99** None of the above, but in this section

## 83Exx Unified, higher-dimensional and super field theories

**83E05** Geometrodynamics and the holographic principle [See also 81T35]

**83E15** Kaluza-Klein and other higher-dimensional theories

**83E30** String and superstring theories in gravitational theory [See also 81T30]

**83E50** Supergravity

**83E99** None of the above, but in this section

### 83Fxx Relativistic cosmology {For astrophysical cosmology, see 85A40}

**83F05** Relativistic cosmology {For astrophysical cosmology, see 85A40}

**83F99** None of the above, but in this section

# 85-XX Astronomy and astrophysics {For celestial mechanics, see 70F15}

**85-00** General reference works (handbooks, dictionaries, bibliographies, etc.) pertaining to astronomy and astrophysics

**85-01** Introductory exposition (textbooks, tutorial papers, etc.) pertaining to astronomy and astrophysics

**85-02** Research exposition (monographs, survey articles) pertaining to astronomy and astrophysics

**85-03** History of astronomy and astrophysics [Consider also classification numbers from Section 01]

**85-04** Software, source code, etc. for problems pertaining to astronomy and astrophysics

**85-05** Experimental work for problems pertaining to astronomy and astrophysics

**85-06** Proceedings, conferences, collections, etc. pertaining to astronomy and astrophysics

**85-08** Computational methods for problems pertaining to astronomy and astrophysics

**85-10** Mathematical modeling or simulation for problems pertaining to astronomy and astrophysics

**85-11** Research data for problems pertaining to astronomy and astrophysics

### 85Axx Astronomy and astrophysics {For celestial mechanics, see 70F15}

**85A04** General questions in astronomy and astrophysics

**85A05** Galactic and stellar dynamics

**85A15** Galactic and stellar structure

**85A20** Planetary atmospheres

**85A25** Radiative transfer in astronomy and astrophysics

**85A30** Hydrodynamic and hydromagnetic problems in astronomy and astrophysics [See also 76Y05]

**85A35** Statistical astronomy

**85A40** Astrophysical cosmology {For relativistic cosmology, see 83F05}

**85A99** None of the above, but in this section

# 86-XX Geophysics [See also 76U05, 76V05]

**86-00** General reference works (handbooks, dictionaries, bibliographies, etc.) pertaining to geophysics

**86-01** Introductory exposition (textbooks, tutorial papers, etc.) pertaining to geophysics

**86-02** Research exposition (monographs, survey articles) pertaining to geophysics

**86-03** History of geophysics [Consider also classification numbers from Section 01]

**86-04** Software, source code, etc. for problems pertaining to geophysics

**86-05** Experimental work for problems pertaining to geophysics

**86-06** Proceedings, conferences, collections, etc. pertaining to geophysics

**86-08** Computational methods for problems pertaining to geophysics

**86-10** Mathematical modeling or simulation for problems pertaining to geophysics

**86-11** Research data for problems pertaining to geophysics

## 86Axx Geophysics [See also 76U05, 76V05]

**86A04** General questions in geophysics

**86A05** Hydrology, hydrography, oceanography [See also 76Bxx, 76E20, 76Q05, 76Rxx, 76Uxx]

**86A08** Climate science and climate modeling

**86A10** Meteorology and atmospheric physics [See also 76Bxx, 76E20, 76N15, 76Q05, 76Rxx, 76Uxx]

**86A15** Seismology (including tsunami modeling), earthquakes

**86A20** Potentials, prospecting

**86A22** Inverse problems in geophysics [See also 35R30]

**86A25** Geo-electricity and geomagnetism [See also 76W05, 78A25]

**86A30** Geodesy, mapping problems

**86A32** Geostatistics

**86A40** Glaciology

**86A60** Geological problems

**86A70** Vulcanology; magma and lava flow

**86A99** None of the above, but in this section

# 90-XX Operations research, mathematical programming

**90-00** General reference works (handbooks, dictionaries, bibliographies, etc.) pertaining to operations research and mathematical programming

**90-01** Introductory exposition (textbooks, tutorial papers, etc.) pertaining to operations research and mathematical programming

**90-02** Research exposition (monographs, survey articles) pertaining to operations research and mathematical programming

**90-03** History of operations research and mathematical programming [Consider also classification numbers from Section 01]

**90-04** Software, source code, etc. for problems pertaining to operations research and mathematical programming

**90-05** Experimental work for problems pertaining to operations research and mathematical programming

**90-06** Proceedings, conferences, collections, etc. pertaining to operations research and mathematical programming

**90-08** Computational methods for problems pertaining to operations research and mathematical programming

**90-10** Mathematical modeling or simulation for problems pertaining to operations research and mathematical programming

**90-11** Research data for problems pertaining to operations research and mathematical programming

## 90Bxx Operations research and management science

**90B05** Inventory, storage, reservoirs

**90B06** Transportation, logistics and supply chain management

**90B10** Deterministic network models in operations research {For network control, see 93B70}

**90B15** Stochastic network models in operations research {For network control, see 93B70}

**90B18** Communication networks in operations research [See also 68M10, 68M12, 68M18, 94A05] {For networks as computational models, see 68Q06}

**90B20** Traffic problems in operations research

**90B22** Queues and service in operations research [See also 60K25, 68M20]

**90B25** Reliability, availability, maintenance, inspection in operations research [See also 60K10, 62N05]

**90B30** Production models

**90B35** Deterministic scheduling theory in operations research [See also 68M20]

**90B36** Stochastic scheduling theory in operations research [See also 68M20]

**90B40** Search theory

**90B50** Management decision making, including multiple objectives [See also 90C29, 90C31, 91A35, 91B06]

**90B60** Marketing, advertising [See also 91B60]

**90B70** Theory of organizations, manpower planning in operations research [See also 91D35]

**90B80** Discrete location and assignment [See also 90C10]

**90B85** Continuous location

**90B90** Case-oriented studies in operations research

**90B99** None of the above, but in this section

## 90Cxx Mathematical programming {For numerical methods, see also 49Mxx, 65Kxx}

**90C05** Linear programming

**90C06** Large-scale problems in mathematical programming

**90C08** Special problems of linear programming (transportation, multi-index, data envelopment analysis, etc.)

**90C09** Boolean programming

**90C10** Integer programming

**90C11** Mixed integer programming

**90C15** Stochastic programming

**90C17** Robustness in mathematical programming

**90C20** Quadratic programming

**90C22** Semidefinite programming

**90C23** Polynomial optimization

**90C24** Tropical optimization (e.g., max-plus optimization)

**90C25** Convex programming

**90C26** Nonconvex programming, global optimization

**90C27** Combinatorial optimization

**90C29** Multi-objective and goal programming

**90C30** Nonlinear programming

**90C31** Sensitivity, stability, parametric optimization

**90C32** Fractional programming

**90C33** Complementarity and equilibrium problems and variational inequalities (finite dimensions) (aspects of mathematical programming)

**90C34** Semi-infinite programming

**90C35** Programming involving graphs or networks [See also 05C90, 90C27]

**90C39** Dynamic programming [See also 49L20]

**90C40** Markov and semi-Markov decision processes

**90C46** Optimality conditions and duality in mathematical programming [See also 49N15]

**90C47** Minimax problems in mathematical programming [See also 49K35]

**90C48** Programming in abstract spaces

**90C49** Extreme-point and pivoting methods

**90C51** Interior-point methods

**90C52** Methods of reduced gradient type

**90C53** Methods of quasi-Newton type

**90C55** Methods of successive quadratic programming type

**90C56** Derivative-free methods and methods using generalized derivatives [See also 49J52]

**90C57** Polyhedral combinatorics, branch-and-bound, branch-and-cut

**90C59** Approximation methods and heuristics in mathematical programming

**90C60** Abstract computational complexity for mathematical programming problems [See also 68Q25]

**90C70** Fuzzy and other nonstochastic uncertainty mathematical programming

**90C90** Applications of mathematical programming

**90C99** None of the above, but in this section

# 91-XX Game theory, economics, finance, and other social and behavioral sciences

**91-00** General reference works (handbooks, dictionaries, bibliographies, etc.) pertaining to game theory, economics, and finance

**91-01** Introductory exposition (textbooks, tutorial papers, etc.) pertaining to game theory, economics, and finance

**91-02** Research exposition (monographs, survey articles) pertaining to game theory, economics, and finance

**91-03** History of game theory, economics, and finance [Consider also classification numbers from Section 01]

**91-04** Software, source code, etc. for problems pertaining to game theory, economics, and finance

**91-05** Experimental work for problems pertaining to game theory, economics, and finance

**91-06** Proceedings, conferences, collections, etc. pertaining to game theory, economics, and finance

**91-08** Computational methods for problems pertaining to game theory, economics, and finance

**91-10** Mathematical modeling or simulation for problems pertaining to game theory, economics, and finance

**91-11** Research data for problems pertaining to game theory, economics, and finance

## 91Axx Game theory

**91A05** 2-person games

**91A06** $n$-person games, $n > 2$

**91A07** Games with infinitely many players

**91A10** Noncooperative games

**91A11** Equilibrium refinements

**91A12** Cooperative games

**91A14** Potential and congestion games

**91A15** Stochastic games, stochastic differential games

**91A16** Mean field games (aspects of game theory) {For partial differential equations, see 35Q89; for calculus of variations and optimal control, see 49N80}

**91A18** Games in extensive form

**91A20** Multistage and repeated games

**91A22** Evolutionary games

**91A23** Differential games (aspects of game theory) [See also 49N70]

**91A24** Positional games (pursuit and evasion, etc.) [See also 49N75]

**91A25** Dynamic games

**91A26** Rationality and learning in game theory

**91A27** Games with incomplete information, Bayesian games

**91A28** Signaling and communication in game theory

**91A30** Utility theory for games [See also 91B16]

**91A35** Decision theory for games [See also 62Cxx, 90B50, 91B06]

**91A40** Other game-theoretic models

**91A43** Games involving graphs {For games on graphs, see also 05C57}

**91A44** Games involving topology, set theory, or logic

**91A46** Combinatorial games

**91A50** Discrete-time games

**91A55** Games of timing

**91A60** Probabilistic games; gambling [See also 60G40]

**91A65** Hierarchical games (including Stackelberg games)

**91A68** Algorithmic game theory and complexity [See also 68Qxx, 68Wxx]

**91A70** Spaces of games

**91A80** Applications of game theory

**91A81** Quantum games

**91A86** Game theory and fuzziness

**91A90** Experimental studies

**91A99** None of the above, but in this section

## 91Bxx Mathematical economics {For econometrics, see 62P20}

**91B02** Fundamental topics (basic mathematics, methodology; applicable to economics in general)

**91B03** Mechanism design theory

**91B05** Risk models (general) {For actuarial and financial risk, see 91Gxx}

**91B06** Decision theory [See also 62Cxx, 90B50, 91A35]

**91B08** Individual preferences

**91B10** Group preferences

**91B12** Voting theory

**91B14** Social choice

**91B15** Welfare economics

**91B16** Utility theory [See also 91A30]

**91B18** Public goods

**91B24** Microeconomic theory (price theory and economic markets)

**91B26** Auctions, bargaining, bidding and selling, and other market models

**91B32** Resource and cost allocation (including fair division, apportionment, etc.)

**91B38** Production theory, theory of the firm

**91B39** Labor markets

**91B41** Contract theory (moral hazard, adverse selection)

**91B42** Consumer behavior, demand theory

**91B43** Principal-agent models

**91B44** Economics of information

**91B50** General equilibrium theory

**91B51** Dynamic stochastic general equilibrium theory

**91B52** Special types of economic equilibria

**91B54** Special types of economic markets (including Cournot, Bertrand)

**91B55** Economic dynamics

**91B60** Trade models

**91B62** Economic growth models

**91B64** Macroeconomic theory (monetary models, models of taxation)

**91B66** Multisectoral models in economics

**91B68** Matching models

**91B69** Heterogeneous agent models

**91B70** Stochastic models in economics

**91B72** Spatial models in economics [See also 91D25]

**91B74** Economic models of real-world systems (e.g., electricity markets, etc.)

**91B76** Environmental economics (natural resource models, harvesting, pollution, etc.)

**91B80** Applications of statistical and quantum mechanics to economics (econophysics)

**91B82** Statistical methods; economic indices and measures [See also 62P20]

**91B84** Economic time series analysis {For statistical theory of time series, see 62M10}

**91B86** Mathematical economics and fuzziness

**91B99** None of the above, but in this section

## 91Cxx Social and behavioral sciences: general topics {For statistics, see 62P25}

**91C05** Measurement theory in the social and behavioral sciences

**91C15** One- and multidimensional scaling in the social and behavioral sciences

**91C20** Clustering in the social and behavioral sciences [See also 62H30]

**91C99** None of the above, but in this section

## 91Dxx Mathematical sociology (including anthropology)

**91D10** Models of societies, social and urban evolution

**91D15** Social learning

**91D20** Mathematical geography and demography

**91D25** Spatial models in sociology [See also 91B72]

**91D30** Social networks; opinion dynamics

**91D35** Manpower systems in sociology [See also 90B70, 91B39]

**91D99** None of the above, but in this section

## 91Exx Mathematical psychology {For psychometrics, see 62P15}

**91E10** Cognitive psychology

**91E30** Psychophysics and psychophysiology; perception

**91E40** Memory and learning in psychology [See also 68T05]

**91E45** Measurement and performance in psychology

**91E99** None of the above, but in this section

## 91Fxx Other social and behavioral sciences (mathematical treatment)

**91F10** History, political science

**91F20** Linguistics [See also 03B65, 68T50]

**91F99** None of the above, but in this section

## 91Gxx Actuarial science and mathematical finance {For statistics, see 62P05}

**91G05** Actuarial mathematics

**91G10** Portfolio theory

**91G15** Financial markets

**91G20** Derivative securities (option pricing, hedging, etc.)

**91G30** Interest rates, asset pricing, etc. (stochastic models)

**91G40** Credit risk

**91G45** Financial networks (including contagion, systemic risk, regulation)

**91G50** Corporate finance (dividends, real options, etc.)

**91G60** Numerical methods (including Monte Carlo methods)

**91G70** Statistical methods; risk measures [See also 62P05, 62P20]

**91G80** Financial applications of other theories [See also 35Q91, 37N40, 49N90, 60J70, 60K10, 60H30, 93E20]

**91G99** None of the above, but in this section

# 92-XX Biology and other natural sciences

**92-00** General reference works (handbooks, dictionaries, bibliographies, etc.) pertaining to biology

**92-01** Introductory exposition (textbooks, tutorial papers, etc.) pertaining to biology

**92-02** Research exposition (monographs, survey articles) pertaining to biology

**92-03** History of biology [Consider also classification numbers from Section 01]

**92-04** Software, source code, etc. for problems pertaining to biology

**92-05** Experimental work for problems pertaining to biology

**92-06** Proceedings, conferences, collections, etc. pertaining to biology

**92-08** Computational methods for problems pertaining to biology

**92-10** Mathematical modeling or simulation for problems pertaining to biology

**92-11** Research data for problems pertaining to biology

## 92Bxx Mathematical biology in general

**92B05** General biology and biomathematics

**92B10** Taxonomy, cladistics, statistics in mathematical biology

**92B15** General biostatistics [See also 62P10]

**92B20** Neural networks for/in biological studies, artificial life and related topics [See also 68T05, 82C32, 94Cxx]

**92B25** Biological rhythms and synchronization

**92B99** None of the above, but in this section

## 92Cxx Physiological, cellular and medical topics

**92C05** Biophysics

**92C10** Biomechanics [See also 74L15]

**92C15** Developmental biology, pattern formation

**92C17** Cell movement (chemotaxis, etc.)

**92C20** Neural biology

**92C30** Physiology (general)

**92C32** Pathology, pathophysiology

**92C35** Physiological flow [See also 76Z05]

**92C37** Cell biology

**92C40** Biochemistry, molecular biology

**92C42** Systems biology, networks

**92C45** Kinetics in biochemical problems (pharmacokinetics, enzyme kinetics, etc.) [See also 80A30]

**92C47** Biosensors (not for medical applications)

**92C50** Medical applications (general)

**92C55** Biomedical imaging and signal processing [See also 44A12, 65R10, 94A08, 94A12]

**92C60** Medical epidemiology {For theoretical aspects, see 92D30}

**92C70** Microbiology

**92C75** Biotechnology

**92C80** Plant biology

**92C99** None of the above, but in this section

## 92Dxx Genetics and population dynamics

**92D10** Genetics and epigenetics {For genetic algebras, see 17D92}

**92D15** Problems related to evolution

**92D20** Protein sequences, DNA sequences

**92D25** Population dynamics (general)

**92D30** Epidemiology {For medical applications, see 92C60}

**92D40** Ecology

**92D45** Pest management

**92D50** Animal behavior

**92D99** None of the above, but in this section

## 92Exx Chemistry {For biochemistry, see 92C40}

**92E10** Molecular structure (graph-theoretic methods, methods of differential topology, etc.) [See also 05C92]

**92E20** Classical flows, reactions, etc. in chemistry [See also 80A30, 80A32]

**92E99** None of the above, but in this section

## 92Fxx Other natural sciences (mathematical treatment)

**92F05** Other natural sciences (mathematical treatment)

**92F99** None of the above, but in this section

# 93-XX Systems theory; control {For optimal control, see 49-XX}

**93-00** General reference works (handbooks, dictionaries, bibliographies, etc.) pertaining to systems and control theory

**93-01** Introductory exposition (textbooks, tutorial papers, etc.) pertaining to systems and control theory

**93-02** Research exposition (monographs, survey articles) pertaining to systems and control theory

**93-03** History of systems and control theory [Consider also classification numbers from Section 01]

**93-04** Software, source code, etc. for problems pertaining to systems and control theory

**93-05** Experimental work for problems pertaining to systems and control theory

**93-06** Proceedings, conferences, collections, etc. pertaining to systems and control theory

**93-08** Computational methods for problems pertaining to systems and control theory

**93-10** Mathematical modeling or simulation for problems pertaining to systems and control theory

**93-11** Research data for problems pertaining to systems and control theory

## 93Axx General systems theory

**93A05** Axiomatic systems theory

**93A10** General systems

**93A13** Hierarchical systems

**93A14** Decentralized systems

**93A15** Large-scale systems

**93A16** Multi-agent systems

**93A99** None of the above, but in this section

## 93Bxx Controllability, observability, and system structure

**93B03** Attainable sets, reachability

**93B05** Controllability

**93B07** Observability

**93B10** Canonical structure

**93B11** System structure simplification

**93B12** Variable structure systems

**93B15** Realizations from input-output data

**93B17** Transformations

**93B18** Linearizations

**93B20** Minimal systems representations

**93B24** Topological methods

**93B25** Algebraic methods

**93B27** Geometric methods

**93B28** Operator-theoretic methods [See also 47A48, 47A57, 47B35, 47N70]

**93B30** System identification

**93B35** Sensitivity (robustness)

**93B36** $H^\infty$-control

**93B45** Model predictive control

**93B47** Iterative learning control

**93B50** Synthesis problems

**93B51** Design techniques (robust design, computer-aided design, etc.)

**93B52** Feedback control

**93B53** Observers

**93B55** Pole and zero placement problems

**93B60** Eigenvalue problems

**93B70** Networked control

**93B99** None of the above, but in this section

## 93Cxx Model systems in control theory

**93C05** Linear systems in control theory

**93C10** Nonlinear systems in control theory

**93C15** Control/observation systems governed by ordinary differential equations [See also 34Hxx]

**93C20** Control/observation systems governed by partial differential equations

**93C23** Control/observation systems governed by functional-differential equations [See also 34K35]

**93C25** Control/observation systems in abstract spaces

**93C27** Impulsive control/observation systems

**93C28** Positive control/observation systems

**93C29** Boolean control/observation systems

**93C30** Control/observation systems governed by functional relations other than differential equations (such as hybrid and switching systems)

**93C35** Multivariable systems, multidimensional control systems

**93C40** Adaptive control/observation systems

**93C41** Control/observation systems with incomplete information

**93C42** Fuzzy control/observation systems

**93C43** Delay control/observation systems

**93C55** Discrete-time control/observation systems

**93C57** Sampled-data control/observation systems

**93C62** Digital control/observation systems

**93C65** Discrete event control/observation systems

**93C70** Time-scale analysis and singular perturbations in control/observation systems

**93C73** Perturbations in control/observation systems

**93C80** Frequency-response methods in control theory

**93C83** Control/observation systems involving computers (process control, etc.)

**93C85** Automated systems (robots, etc.) in control theory [See also 68T40, 70B15, 70Q05]

**93C95** Application models in control theory

**93C99** None of the above, but in this section

### 93Dxx Stability of control systems

**93D05** Lyapunov and other classical stabilities (Lagrange, Poisson, $L^p, l^p$, etc.) in control theory

**93D09** Robust stability

**93D10** Popov-type stability of feedback systems

**93D15** Stabilization of systems by feedback

**93D20** Asymptotic stability in control theory

**93D21** Adaptive or robust stabilization

**93D23** Exponential stability

**93D25** Input-output approaches in control theory

**93D30** Lyapunov and storage functions

**93D40** Finite-time stability

**93D50** Consensus

**93D99** None of the above, but in this section

### 93Exx Stochastic systems and control

**93E03** Stochastic systems in control theory (general)

**93E10** Estimation and detection in stochastic control theory [See also 60G35]

**93E11** Filtering in stochastic control theory [See also 60G35]

**93E12** Identification in stochastic control theory

**93E14** Data smoothing in stochastic control theory

**93E15** Stochastic stability in control theory

**93E20** Optimal stochastic control [See also 49J55, 49K45]

**93E24** Least squares and related methods for stochastic control systems

**93E35** Stochastic learning and adaptive control

**93E99** None of the above, but in this section

# 94-XX Information and communication theory, circuits

**94-00** General reference works (handbooks, dictionaries, bibliographies, etc.) pertaining to information and communication theory

**94-01** Introductory exposition (textbooks, tutorial papers, etc.) pertaining to information and communication theory

**94-02** Research exposition (monographs, survey articles) pertaining to information and communication theory

**94-03** History of information and communication theory [Consider also classification numbers from Section 01]

**94-04** Software, source code, etc. for problems pertaining to information and communication theory

**94-05** Experimental work for problems pertaining to information and communication theory

**94-06** Proceedings, conferences, collections, etc. pertaining to information and communication theory

**94-08** Computational methods for problems pertaining to information and communication theory

**94-10** Mathematical modeling or simulation for problems pertaining to information and communication theory

**94-11** Research data for problems pertaining to information and communication theory

## 94Axx Communication, information

**94A05** Communication theory [See also 60G35, 90B18]

**94A08** Image processing (compression, reconstruction, etc.) in information and communication theory [See also 68U10]

**94A11** Application of orthogonal and other special functions

**94A12** Signal theory (characterization, reconstruction, filtering, etc.)

**94A13** Detection theory in information and communication theory

**94A14** Modulation and demodulation in information and communication theory

**94A15** Information theory (general) [See also 62B10] {For quantum-theoretic aspects, see also 81P45}

**94A16** Informational aspects of data analysis and big data [See also 62R07, 68T09] {For homological aspects, see 55N31}

**94A17** Measures of information, entropy [See also 62B10]

**94A20** Sampling theory in information and communication theory

**94A24** Coding theorems (Shannon theory)

**94A29** Source coding [See also 68P30]

**94A34** Rate-distortion theory in information and communication theory

**94A40** Channel models (including quantum) in information and communication theory [See also 81P47]

**94A45** Prefix, length-variable, comma-free codes [See also 20M35, 68Q45]

**94A50** Theory of questionnaires

**94A55** Shift register sequences and sequences over finite alphabets in information and communication theory

**94A60** Cryptography [See also 11T71, 14G50, 68P25, 81P94]

**94A62** Authentication, digital signatures and secret sharing

**94A99** None of the above, but in this section

## 94Bxx Theory of error-correcting codes and error-detecting codes

**94B05** Linear codes (general theory)

**94B10** Convolutional codes

**94B12** Combined modulation schemes (including trellis codes) in coding theory

**94B15** Cyclic codes

**94B20** Burst-correcting codes

**94B25** Combinatorial codes

**94B27** Geometric methods (including applications of algebraic geometry) applied to coding theory [See also 11T71, 14G50]

**94B30** Majority codes

**94B35** Decoding

**94B40** Arithmetic codes [See also 11T71, 14G50]

**94B50** Synchronization error-correcting codes

**94B60** Other types of codes

**94B65** Bounds on codes

**94B70** Error probability in coding theory

**94B75** Applications of the theory of convex sets and geometry of numbers (covering radius, etc.) to coding theory [See also 11H31, 11H71]

**94B99** None of the above, but in this section

## 94Cxx Circuits, networks [See also 68Q06]

**94C05** Analytic circuit theory

**94C11** Switching theory, applications of Boolean algebras to circuits and networks

**94C12** Fault detection; testing in circuits and networks

**94C15** Applications of graph theory to circuits and networks [See also 05Cxx, 68R10]

**94C30** Applications of design theory to circuits and networks [See also 05Bxx]

**94C60** Circuits in qualitative investigation and simulation of models

**94C99** None of the above, but in this section

## 94Dxx Miscellaneous topics in information and communication theory

**94D05** Fuzzy sets and logic (in connection with information, communication, or circuits theory) [See also 03B52, 03E72, 28E10]

**94D10** Boolean functions [See also 06E30] {For connections with circuits and networks, see 94C11}

**94D99** None of the above, but in this section

# 97-XX Mathematics education

**97-00** General reference works (handbooks, dictionaries, bibliographies, etc.) pertaining to mathematics education

**97-01** Introductory exposition (textbooks, tutorial papers, etc.) pertaining to mathematics education

**97-02** Research exposition (monographs, survey articles) pertaining to mathematics education

**97-03** History of mathematics education [Consider also classification numbers from Section 01]

**97-06** Proceedings, conferences, collections, etc. pertaining to mathematics education

**97-11** Research data for problems pertaining to mathematics education

## 97Axx History and society (aspects of mathematics education)

**97A30** History in mathematics education {For mathematics history, see 01-XX; for biographies, see 01A70; for history of mathematics education, see 97-03}

**97A40** Mathematics education and society {For sociology (and profession) of mathematics, see 01A80}

**97A99** None of the above, but in this section

## 97Bxx Educational policy and systems

**97B10** Mathematics educational research and planning

**97B20** Educational policy for general education

**97B30** Educational policy for vocational education

**97B40** Educational policy for higher education

**97B50** Mathematics teacher education

**97B60** Educational policy for adult and further education

**97B70** Syllabuses, educational standards

**97B99** None of the above, but in this section

## 97Cxx Psychology of mathematics education, research in mathematics education

**97C10** Comprehensive works on psychology of mathematics education

**97C20** Affective behavior and mathematics education

**97C30** Cognitive processes, learning theories (aspects of mathematics education)

**97C40** Intelligence and aptitudes (aspects of mathematics education)

**97C50** Language and verbal communities (aspects of mathematics education)

**97C60** Sociological aspects of learning (aspects of mathematics education)

**97C70** Teaching-learning processes in mathematics education

**97C99** None of the above, but in this section

## 97Dxx Education and instruction in mathematics

**97D10** Comprehensive works and comparative studies on education and instruction in mathematics

**97D20** Philosophical and theoretical contributions (didactics of mathematics)

**97D30** Objectives and goals of mathematics teaching

**97D40** Mathematics teaching methods and classroom techniques

**97D50** Teaching mathematical problem solving and heuristic strategies

**97D60** Student assessment, achievement control, and rating (aspects of mathematics education)

**97D70** Learning difficulties and student errors (aspects of mathematics education)

**97D80** Mathematics teaching units and draft lessons

**97D99** None of the above, but in this section

## 97Exx Education of foundations of mathematics

**97E10** Comprehensive works on education of foundations of mathematics

**97E20** Philosophy and mathematics (educational aspects)

**97E30** Logic (educational aspects)

**97E40** Language of mathematics (educational aspects)

**97E50** Reasoning and proving in the mathematics classroom

**97E60** Sets, relations, set theory (educational aspects)

**97E99** None of the above, but in this section

## 97Fxx Education of arithmetic and number theory

**97F10** Comprehensive works on education of arithmetic and number theory

**97F20** Pre-numerical stage, concept of numbers

**97F30** Natural numbers (educational aspects)

**97F40** Integers, rational numbers (educational aspects)

**97F50** Real numbers, complex numbers (educational aspects)

**97F60** Number theory (educational aspects)

**97F70** Measures and units (educational aspects)

**97F80** Ratio and proportion, percentages (educational aspects)

**97F90** Real-life mathematics, practical arithmetic (educational aspects)

**97F99** None of the above, but in this section

## 97Gxx Geometry education

**97G10** Comprehensive works on geometry education

**97G20** Informal geometry (educational aspects)

**97G30** Area and volume (educational aspects)

**97G40** Plane and solid geometry (educational aspects)

**97G50** Transformation geometry (educational aspects)

**97G60** Plane and spherical trigonometry (educational aspects)

**97G70** Analytic geometry, vector algebra (educational aspects)

**97G80** Descriptive geometry (educational aspects)

**97G99** None of the above, but in this section

## 97Hxx Algebra education

**97H10** Comprehensive works on algebra education

**97H20** Elementary algebra (educational aspects)

**97H30** Equations and inequalities (educational aspects)

**97H40** Groups, rings, fields (educational aspects)

**97H50** Ordered algebraic structures (educational aspects)

**97H60** Linear algebra (educational aspects)

**97H99** None of the above, but in this section

## 97Ixx Analysis education

**97I10** Comprehensive works on analysis education

**97I20** Mappings and functions (educational aspects)

**97I30** Sequences and series (educational aspects)

**97I40** Differential calculus (educational aspects)

**97I50** Integral calculus (educational aspects)

**97I60** Functions of several variables (educational aspects)

**97I70** Functional equations (educational aspects)

**97I80** Complex analysis (educational aspects)

**97I99** None of the above, but in this section

## 97Kxx Education of combinatorics, graph theory, probability theory, and statistics

**97K10** Comprehensive works on combinatorics, graph theory, and probability (educational aspects)

**97K20** Combinatorics (educational aspects)

**97K30** Graph theory (educational aspects)

**97K40** Descriptive statistics (educational aspects)

**97K50** Probability theory (educational aspects)

**97K60** Distributions and stochastic processes (educational aspects)

**97K70** Foundations and methodology of statistics (educational aspects)

**97K80** Applied statistics (educational aspects)

**97K99** None of the above, but in this section

## 97Mxx Education of mathematical modeling and applications of mathematics

**97M10** Modeling and interdisciplinarity (aspects of mathematics education)

**97M20** Mathematics in vocational training and career education

**97M30** Financial and insurance mathematics (aspects of mathematics education)

**97M40** Operations research, economics (aspects of mathematics education)

**97M50** Physics, astronomy, technology, engineering (aspects of mathematics education)

**97M60** Biology, chemistry, medicine (aspects of mathematics education)

**97M70** Behavioral and social sciences (aspects of mathematics education)

**97M80** Arts, music, language, architecture (aspects of mathematics education)

**97M99** None of the above, but in this section

## 97Nxx Education of numerical mathematics

**97N10** Comprehensive works on education of numerical mathematics

**97N20** Rounding, estimation, theory of errors (educational aspects)

**97N30** Numerical algebra (educational aspects)

**97N40** Numerical analysis (educational aspects)

**97N50** Interpolation and approximation (educational aspects)

**97N60** Mathematical programming (educational aspects)

**97N70** Discrete mathematics (educational aspects)

**97N80** Mathematical software, computer programs (educational aspects)

**97N99** None of the above, but in this section

## 97Pxx Computer science (educational aspects)

**97P10** Comprehensive works on computer science (educational aspects)

**97P20** Theoretical computer science (educational aspects)

**97P30** Systems, databases (educational aspects)

**97P40** Programming languages (educational aspects)

**97P50** Programming techniques (educational aspects)

**97P80** Artificial intelligence (educational aspects)

**97P99** None of the above, but in this section

# 97Uxx Educational material and media and educational technology in mathematics education

**97U10** Comprehensive works on educational material and media and educational technology in mathematics education

**97U20** Textbooks, textbook research (aspects of mathematics education)

**97U30** Teachers' manuals and planning aids (aspects of mathematics education)

**97U40** Problem books, competitions, examinations (aspects of mathematics education)

**97U50** Computer-assisted instruction, e-learning (aspects of mathematics education)

**97U60** Manipulative materials (aspects of mathematics education)

**97U70** Technological tools, calculators (aspects of mathematics education)

**97U80** Audiovisual media (aspects of mathematics education)

**97U99** None of the above, but in this section