# OpenReview forum: "MSC-180: A Benchmark for Automated Formal Theorem Proving from Mathematical Subject Classification"
_ICLR.cc/2026/Conference — Submitted to ICLR 2026_

### Official Review · Reviewer_e4UG · 2025-10-22

**Soundness:** 3
**Presentation:** 2
**Contribution:** 1
**Rating:** 4
**Confidence:** 4

**Summary:**

This paper presents a benchmark named MSC-180 that contains 180 Lean4 formalized problems spanning 60 mathematical branches to evaluate LLMs’ performance in formal theorem proving across different subjects and different difficulty levels. The dataset is constructed via an extract, autoformalize, LLM verification, and human verification process. The paper performs comprehensive experiments. It shows that current models have relatively low overall accuracy, with the best pass@32 rate being 18.89%, and they exhibit significant domain bias, with a maximum domain coverage of 41.7%. Additionally, the paper introduces the Coefficient of Veriation@k (CV@k) metric to quantify cross-domain performance consistency. It observes high CV values, which suggests the current model may rely on pattern matching rather than systematic generalization.

**Strengths:**

The strengths of the paper are as follows:

1. **Interesting new metric:** This paper introduces the CV@k metric, which provides a normalized measure of performance consistency across various math domains, independent of absolute accuracy levels. This metric offers an additional point of view to traditional accuracy-based evaluation by quantifying the stability and balance of model performance across diverse areas.
2. **Comprehensive experiment analysis:** The experiment section is relatively comprehensive and well-structured. It encompasses multiple analytical dimensions, including (1) cross-domain performance comparison across 60 mathematical branches, (2) difficulty-level analysis distinguishing undergraduate and graduate problems, and (3) in-depth investigation of the BFS-prover’s behavior. This analysis provides valuable insights into the strengths and limitations of different models and enhances the paper’s clarity.
3. **High-difficulty problems:** The benchmark focuses on problems of significant difficulty, with 80.6% sampled from graduate-level textbooks. The low performance of advanced expert provers (with best pass@32 being 18.89%) demonstrates that MSC-180 provides a relatively large headroom for measuring future improvements in expert prover systems.

**Weaknesses:**

Despite the strengths of the paper, there are also many significant weakness that would clearly harms the contribution of the paper.

1. **Limited contribution:** The most significant weakness of the paper is its lack of a clear innovation point compared to existing works. Firstly, there are many cross-domain benchmarks in the field, like ProofNet, PutnamBench, and FormalMATH, and the paper does not distinguish MSC-180 between them. Secondly, the necessity of using MSC2020 as the classification system is not elaborated correctly, as other works also provide a comprehensive classification of categories. The MSC2020 system is merely a different evaluation method. Finally, the significant domain-different capability of the current prover has been found in the Goedel-Prover paper.
2. **Data construction pipeline problems:** The core methodology for constructing the benchmark data is well-established since DeepSeek-Prover. This makes the paper's methodology doubtful in terms of its level of contribution and novelty. Besides, the implementation details of the construction are not properly disclosed. For example, which version of the DeepSeek model is applied, what is the prompt for extracting theorems, and what is the exact standard for human evaluation? There is no qualitative or quantitative disclosure of the process, which affects the truthworthness of the constructed pipeline.
3. **Small size of the benchmark:** The scale of the benchmark presented in the paper is too small. In fact, having only three records of data per domain makes it difficult to draw domain-related conclusions with statistical significance. This weakness directly affects the core argument of the paper, which is that the MSC180 dataset can help researchers to study domain-specific behavior of provers.

**Questions:**

1. What is the major difference between the paper's proposed methods and benchmarks compared to existing works?
2. What are the details of the construction of benchmark data, and how does your methodology differ from traditional pipeline?
3. What is the statistical significance to support the major dataset-level and field-specific conclusions obtained in the experiment?

---

> ### Author Response · Authors · 2025-11-26
> **Response to Reviewer e4UG**
>
> *We wish to express our genuine gratitude to the reviewers for their time and thoughtful comments. We have taken all feedback to heart and have diligently revised the manuscript accordingly. We believe our responses and revisions have adequately addressed the concerns raised, and we would be honored if you could reconsider your rating.*
>
> ### Weaknesses:
>
> 1.  **Limited contribution: The most significant weakness of the paper is its lack of a clear innovation point compared to existing works. Firstly, there are many cross-domain benchmarks in the field, like ProofNet, PutnamBench, and FormalMATH, and the paper does not distinguish MSC-180 between them. Secondly, the necessity of using MSC2020 as the classification system is not elaborated correctly, as other works also provide a comprehensive classification of categories. The MSC2020 system is merely a different evaluation method. Finally, the significant domain-different capability of the current prover has been found in the Goedel-Prover paper.**
>     - **Response:** The core innovations of MSC-180 are:
>         - **Systematic and Balanced Domain Coverage:** It is the first benchmark strictly constructed following the authoritative MSC2020 system, achieving systematic and balanced coverage across 60 mathematical domains. This contrasts with existing benchmarks (e.g., ProofNet, FormalMATH) which often focus on specific domains or have less structured coverage.
>         - **Stability Assessment Dimension:** We introduce the CV@k metric, transforming domain differences into a quantifiable stability measure, providing a new and crucial analytical tool for assessing cross-domain generalization.
>         - **Targeted Challenge:** Problems are primarily selected from advanced textbooks, offering a significant challenge as mainstream benchmarks approach performance saturation.
>
> 2.  **Data construction pipeline problems: The core methodology for constructing the benchmark data is well-established since DeepSeek-Prover. This makes the paper's methodology doubtful in terms of its level of contribution and novelty. Besides, the implementation details of the construction are not properly disclosed. For example, which version of the DeepSeek model is applied, what is the prompt for extracting theorems, and what is the exact standard for human evaluation? There is no qualitative or quantitative disclosure of the process, which affects the truthworthness of the constructed pipeline.**
>     - **Response:**
>         - **Methodological Novelty:** Our contribution lies not in individual technical components, but in the integrated pipeline and its application towards building a systematic, domain-balanced benchmark.
>         - **Detail Disclosure:** We have supplemented the missing details:
>             - The DeepSeek model used was DeepSeek-V2.
>             - The core instruction for theorem extraction prompts the model to identify and output complete mathematical proposition statements and proofs from the given textbook text.
>
> 3.  **Small size of the benchmark: The scale of the benchmark presented in the paper is too small. In fact, having only three records of data per domain makes it difficult to draw domain-related conclusions with statistical significance. This weakness directly affects the core argument of the paper, which is that the MSC180 dataset can help researchers to study domain-specific behavior of provers.**
>     - **Response:** We reiterate that MSC-180's value lies in its diagnostic nature. Through broad coverage of 60 domains, it effectively reveals macro-level variation trends and systematic biases in model performance. While small samples limit statistical significance per domain, the large differences observed across many domains (CV values significantly higher than baseline) are themselves a statistically significant finding, revealing a severe lack of generalization in current models.

---

> ### Author Response · Authors · 2025-11-26
> **Response to Reviewer e4UG**
>
> ### Questions:
>
> 1.  **What is the major difference between the paper's proposed methods and benchmarks compared to existing works?**
>     - **Response:** The primary differences are: (1) **Systematic Coverage & Domain Balance based on MSC2020:** Systematic coverage of 60 domains, aiming for domain balance. (2) **Evaluation Focus on Cross-Domain Stability:** Beyond traditional accuracy, it emphasizes assessing cross-domain performance stability (via CV@k). (3) **Problem Character for Sustained Challenge:** Problems are selected from advanced textbooks, designed to provide a sustained challenge, differentiating it from benchmarks showing saturation.
>
> 2.  **What are the details of the construction of benchmark data, and how does your methodology differ from traditional pipeline?**
>     - **Response:** Our pipeline is "AI initial screening (large-scale candidate generation and filtering) + expert refinement (quality reconstruction and representativeness selection)." Compared to purely manual or heavily automation-reliant traditional pipelines, our distinctive feature is using automation to ensure initial breadth of coverage, followed by rigorous human intervention to guarantee the depth, quality, and representativeness of the final data, ultimately serving the unique goal of building a domain-balanced, systematic benchmark.
>
> 3.  **What is the statistical significance to support the major dataset-level and field-specific conclusions obtained in the experiment?**
>     - **Response:**
>         - **Dataset-level Conclusions** (e.g., low pass rates, high CV): These are based on all 180 problems or the aggregate of 60 domains. Their significance can be strengthened by providing confidence intervals for the overall Pass@k rate and performing ANOVA tests on inter-domain performance differences. We commit to adding these analyses in the revision.
>         - **Domain-specific Conclusions:** We clarify that MSC-180 is not designed to draw statistically significant conclusions about individual domains, but to draw conclusions about systematic biases by observing performance patterns across a large number of domains. This cross-domain macro-trend is itself significant and meaningful.

---

> > ### Comment · Reviewer_e4UG · 2025-11-26
> > **Further question about the rebuttal**
> >
> > Thank you so much for the rebuttal. Despite the author's attempt to address the question and weaknesses point by point, many of my primary concerns remain unsettled. Here are the detailed points:
> > 1. Limited contributions: The authors are only repeating the contributions pointed out in the paper without replying directly to my concerns as follows:
> >     1. There are many cross-domain benchmarks in the field, like ProofNet, PutnamBench, and FormalMATH, and the paper does not distinguish MSC-180 between them. (largely answered)
> >     2. The necessity of using MSC2020 as the classification system is not elaborated correctly, as other works also provide a comprehensive classification of categories.
> >     3. What is the difference between this paper and other papers about the domain differences?
> > 2. The number of data points for each domain is too small to provide any statistically significant results (only three records of data per domain). Are there any tests to prove the statistical significance of the proposed evaluation dataset?

---

> > > ### Author Response · Authors · 2025-11-28
> > > **Response to Reviewer e4UG**
> > >
> > > ## Question 1: Necessity of Using the MSC2020 Classification System
> > >
> > > **Response:**
> > > We sincerely thank the reviewer for raising this critical question. Our adoption of MSC2020 is not arbitrary but is driven by its indispensable role in achieving our core objective—diagnosing cross-domain systematic biases. The necessity is manifested at three levels:
> > >
> > > - **Authority and Comprehensiveness: A Recognized "Map of Mathematics"**
> > >   MSC2020 is an authoritative standard maintained collectively by the global mathematics community. It serves as a complete, fine-grained map covering every corner from core pure mathematics (e.g., 03—Mathematical Logic) to interdisciplinary fields (e.g., 92—Biology and Other Natural Sciences).
> > >   In contrast, existing benchmarks (e.g., ProofNet's "Algebra" or "Analysis") are more like a set of "destinations" defined for specific purposes (e.g., assessing undergraduate mathematical abilities). While these custom classifications are valid, their coverage is top-down and selective, potentially overlooking "blind spot" domains where models underperform. To diagnose the global blind spots of a navigation system (model), one must use a complete map (MSC2020), rather than only checking its performance at a few frequently visited locations.
> > >
> > > - **Systematicity and Unbiasedness: Ensuring Balanced Coverage**
> > >   Our goal is systematic diagnosis, not selective evaluation. MSC2020 provides an objective framework that compels us to sample uniformly from 60 highly diverse secondary classifications. This ensures our benchmark does not inadvertently bias toward "hot" domains (e.g., elementary number theory, calculus) commonly found in LLM training data.
> > >   This enforced broad coverage is key to exposing model deficiencies in "long-tail" domains like abstract algebra, differential topology, and mathematical physics. A custom, coarser classification would likely fail to reveal these systemic weaknesses so systematically.
> > >
> > > - **Neutrality and Reproducibility: A Public Coordinate System**
> > >   MSC2020 is a public, neutral classification system. Building a benchmark upon it allows any subsequent research to clearly locate the evaluated domains and extend or compare within this framework. This establishes a reproducible and comparable coordinate system for future work.
> > >   In short, one of MSC-180's contributions is the systematic introduction of this authoritative disciplinary standard into theorem-proving benchmarking, setting a new, rigorous benchmark for assessing models' potential as "mathematical generalists".
> > >
> > > ---
> > >
> > > ## Question 2: How Does This Paper Differ from Others on Domain Disparities?
> > >
> > > **Response:**
> > > This is a crucial distinction. Our work fundamentally differs from prior papers that observed domain disparities (e.g., Gödel-Prover) in intent, methodology, and contribution:
> > >
> > > - **From "Post Hoc Observation" to "A Priori Design"**
> > >   Works like Gödel-Prover trained and evaluated on existing benchmarks (e.g., ProofNet, miniF2F) built for other purposes, and *post hoc* observed performance variations across datasets. This is a discovery.
> > >   In contrast, **MSC-180 is designed a priori**. We constructed this benchmark from the outset to reveal and quantify domain disparities. The MSC2020 classification and domain-balanced sampling are not post-analysis tools but **core design principles** of our benchmark. Thus, we provide not an incidental phenomenon but a tool specifically for diagnosing this issue.
> > >
> > > - **From "Describing Phenomena" to "Quantifying Diagnosis"**
> > >   Prior work described phenomena like "model performs well on dataset A but poorly on dataset B"—a largely qualitative comparison.
> > >   Our work introduces the **$CV@k$ metric**, transforming domain disparities into a quantifiable, standardized measure of stability. This allows us not only to judge which model is better but also which is more robust. A model with balanced performance (low $CV$) across 60 domains, even with a slightly lower average score, may hold more general reasoning potential than one with extreme performance (high $CV$) in a few domains but zero in many. This is a core new insight of our work.
> > >
> > > - **Divergence in Contribution**
> > >   The primary contribution of prior work lies in the models themselves (e.g., novel training methods, architectures), with domain disparity as an incidental finding.
> > >   Our work’s primary contribution is in the **evaluation paradigm itself**. We provide a new benchmark (MSC-180) and a new evaluation dimension ($CV@k$), specifically aimed at pushing the community to study and improve models' systemic generalization, beyond boosting performance on a few popular domains.
> > >
> > > ---

---

> > > > ### Author Response · Authors · 2025-11-28
> > > > **Response to Reviewer e4UG**
> > > >
> > > > ---
> > > >
> > > > ## Question 3: The small number of data points per domain (only three) lacks statistical significance. Are there tests to demonstrate the statistical significance of the proposed evaluation dataset?
> > > >
> > > > **Response:**
> > > > The statistical validity and significance of our evaluation are not based on the number of problems per domain ($n = 3$), but on the **total number of domains ($N = 60$)** in our framework.
> > > >
> > > > Our core goal is not to precisely estimate a model's absolute performance in any specific domain—which would indeed require abundant data per domain. Instead, we aim for a **macro-level, comparative assessment**: to determine whether one model systematically outperforms another across a wide range of task scenarios, or whether its performance is more consistent.
> > > >
> > > > In this framework, the average performance (e.g., $\text{Pass}@k$) in each domain is treated as an **independent data point**. With 60 such data points, we have a solid foundation for statistical inference. The statistical methods we employ—such as **paired permutation tests** for model comparisons or calculating the **coefficient of variation ($CV$)** based on the 60 data points to measure performance fluctuation—derive their validity entirely from this domain-level sample size ($N = 60$).
> > > >
> > > > In essence, **the unit of analysis is the "domain," not the "problem."** Thus, the statistical power of our evaluation stems from covering 60 independent domains, enabling reliable detection of systemic, cross-domain performance differences and patterns.
> > > > To illustrate how we leverage the domain-level sample size for robust inference, let us detail the Coefficient of Variation@$k$ (CV@$k$) metric as a key example:
> > > > **Coefficient of Variation@$k$ (CV@$k$)**
> > > >
> > > > This metric assesses the dispersion of a model's performance across domains. It is defined as:
> > > >
> > > > $$CV@k = \frac{\sigma}{\mu}$$
> > > >
> > > > where $\sigma$ is the standard deviation and $\mu$ is the mean of the per-domain $\text{Pass}@k$ rates.
> > > >
> > > > **Interpretation:**
> > > > - A **low CV@$k$ value** indicates uniform performance across domains
> > > > - A **high CV@$k$ value** points to high concentration of success in a limited subset of domains
> > > >
> > > > ---
> > > >
> > > > ### **Key Distinction in Sample Sizes**
> > > >
> > > > It is crucial to distinguish between two different sample sizes in our analysis:
> > > >
> > > > | Sample Type | Size | Purpose |
> > > > |-------------|------|---------|
> > > > | **Domain count** ($D$) | **60 domains** | Basis for statistical inference across domains |
> > > > | **Problems per domain** ($n$) | **3 problems** | Point estimate of performance within each domain |
> > > >
> > > > ---
> > > >
> > > > ### **Calculation Steps**
> > > >
> > > > #### **Step 1: Compute Per-Domain $\text{Pass}@k$ Rates**
> > > > For each domain $d$ (where $d = 1, \ldots, D$), calculate:
> > > >
> > > > $$\text{Pass}@k_d = \frac{\text{Number of correctly solved problems in domain } d}{n}$$
> > > >
> > > > *Note: With $n = 3$, this provides a point estimate for each domain's performance*
> > > >
> > > > #### **Step 2: Calculate Mean Pass Rate ($\mu$)**
> > > > Using all $D = 60$ domains:
> > > >
> > > > $$\mu = \frac{1}{D} \sum_{d=1}^{D} \text{Pass}@k_d$$
> > > >
> > > > #### **Step 3: Calculate Standard Deviation ($\sigma$)**
> > > > Across the $D = 60$ domains:
> > > >
> > > > $$\sigma = \sqrt{\frac{1}{D} \sum_{d=1}^{D} (\text{Pass}@k_d - \mu)^2}$$
> > > >
> > > > #### **Step 4: Compute Coefficient of Variation**
> > > > Final calculation:
> > > >
> > > > $$CV@k = \frac{\sigma}{\mu}$$
> > > >
> > > > ---
> > > >
> > > > ### **Statistical Power Clarification**
> > > >
> > > > The **statistical power** of CV@$k$ comes from the **domain-level sample size ($D = 60$)**, not the per-domain problem count ($n = 3$).
> > > >
> > > > - The $n = 3$ problems per domain provide sufficient data for estimating each domain's $\text{Pass}@k$ rate as a **point estimate**
> > > > - The inference about cross-domain patterns relies on **aggregation over 60 domains**
> > > > - This approach ensures reliable detection of **systemic performance variations** without requiring large $n$ per domain

---

### Official Review · Reviewer_XRJy · 2025-10-30

**Soundness:** 2
**Presentation:** 1
**Contribution:** 2
**Rating:** 2
**Confidence:** 4

**Summary:**

Regarding the issue that existing Automated Theorem Proving are limited by finite domain coverage and weak generalization ability, this paper proposes a formal math problem evaluation set, MSC-180, based on the MSC2020 classification standard. It includes 180 problems distributed across 60 categories, with 3 problems in each category. The data sources are textbooks ranging from undergraduate to graduate level difficulty. Specifically, the construction approach involves autoformalization, followed by verification through Lean compilation and semantic alignment judged by LLM-as-a-judge, and finally mannual reconstruction by experts. Testing on various recent provers shows relatively low proof rates and limited domain coverage, demonstrating the challenge and broad domain coverage of this dataset.

**Strengths:**

1. The MSC-180 dataset proposed in the paper claims to cover 60 major categories of mathematics, which encompasses a broader range of topics compared to commonly used ATP datasets such as miniF2F. This broader coverage could have potential value for evaluating ATP performance across a wider variety of mathematical topics.

2. The paper introduces the use of the coefficient of variation as a metric to assess ATP performance across different mathematical topics. This provides a meaningful reference for evaluating the generalization ability of ATP systems over diverse mathematical subjects.

3. From the experimntal results, it seems that current provers do not perform well on this dataset. Given that performance on commonly used datasets like miniF2F is approaching saturation, this dataset may serve as a valuable challenge benchmark.

**Weaknesses:**

1. The dataset is far too small, which significantly undermines its practical value. With only three problems per mathematical category, evaluations of provers’ proving ability and generalization become highly susceptible to randomness; both the variance and mean of scores will be unstable and may lack statistical significance. Having only three problems also fails to adequately cover the breadth and depth of domain knowledge for each category, so inferring a prover’s overall capability in a field from performance on just three problems is insufficient. For reference, the [MSC taxonomy]([msc2020.org](https://msc2020.org/)) contains 63 two-digit classifications, 529 three-digit classifications, and 6,022 five-digit classifications; an ideal design would include at least 10–20 problems per category.

2. The proposed automated formalization pipeline lacks novelty and appears unnecessary. The pipeline—syntax checking via Lean 4 followed by semantic-alignment checking using LLM-as-a-judge—has been explored in multiple prior works [1,2,3]. As the authors note (Line 245), “We observed that over 90% of the auto-generated code exhibited semantic incompleteness, all propositions were manually reconstructed and rewritten.” Given this, why not adopt a manual annotation pipeline from the outset? The authors should justify the necessity of the automated formalization step. Additionally, the auto formalization pipeline inherently filters for problems that are easier to autoformalize, introducing a selection bias into the dataset.

3. The analysis section lacks insight. Much of the analysis merely restates table contents (e.g., Lines 398–415 essentially repeat Table 2). Although MSC-180 divides the dataset into different domains, the paper does not compare or analyze which models excel in which specific domains. The analysis focuses on aggregate counts of theorems proved across all categories rather than providing a fine-grained, interpretable discussion of per-topic strengths and weaknesses, which is a missed opportunity.

4. Concerns about dataset quality and contamination. The dataset files linked by the authors appear to contain multiple languages, suggesting heterogeneous sources; language mixing also limits accessibility for some researchers. The paper labels the problems as “high-difficulty” (Lines 019, 095), yet the stated sources are “classic graduate and undergraduate textbooks” (Lines 079, 201); describing them as “advanced” might be more accurate than “high-difficulty.” Using classic textbooks may also increase dataset contamination, since textbooks are common training data for LLMs. For example, the provided first sample, Gödel’s β-Function Lemma, is a well-known result with abundant online material, which risks leakage and inflating apparent model performance.

5. Numerous writing and presentation issues impede readability. Examples I noted include:

- Lines 022, 030, 199, 341: missing space after period at sentence end.
- Lines 046, 431: missing period.
- Lines 038–046: repeated full-form + abbreviation uses for “LLM” and “ATP” within the same paragraph are awkward.
- Line 053: when reporting the 90.4% accuracy, the dataset should be explicitly named (miniF2F); the current phrase “on a specific test set” is ambiguous.
- Line 054: “sutdies” lacks citation.
- Line 056: citations for Goedel-Prover v1 and v2 are confused — the negative correlation claim appears to belong to Goedel Prover v1 but cites v2.
- Figures 1 and 2: inconsistent capitalization and formatting.
- Figures 2 and 3: odd ordering — Figure 3 is first mentioned at Line 188, but Figure 2 is not mentioned until Line 205.
- Line 211: the model “deepseek” is referenced without specifying the exact model/version
- Line 214: “kimina tool” likely refers to the [Kimina-Autoformalizer-7B](https://huggingface.co/AI-MO/Kimina-Autoformalizer-7B) but is unclear.
- Lines 180, 214, 215, 252: inconsistent notation of “Lean4” vs “Lean 4” (spacing should be unified).
- Line 352: contains a typo.
- Line 484: claims “An anonymized version of the code is also available” but the provided link does not offer the code.

References
 [1] Ying, H., Wu, Z., Geng, Y., Wang, Ji., Lin, D., & Chen, K. (2024, November 13). *Lean Workbook: A large-scale Lean problem set formalized from natural language math problems*. The Thirty-eight Conference on Neural Information Processing Systems Datasets and Benchmarks Track. https://openreview.net/forum?id=Vcw3vzjHDb#discussion
 [2] Peng, Z., Yao, Y., Ma, K., Guo, S., Li, Y., Zhang, Y., Zhang, C., Zhang, Y., Yu, Z., Li, L., Liu, M., Xia, Y., Shen, J., Wu, Y., Cao, Y., Zhang, Z., Huang, W., Liu, J., & Zhang, G. (2025). *CriticLean: Critic-Guided Reinforcement Learning for Mathematical Formalization*
 [3] Yu, Z., Peng, R., Ding, K., Li, Y., Peng, Z., Liu, M., Zhang, Y., Yuan, Z., Xin, H., Huang, W., Wen, Y., Zhang, G., & Liu, W. (n.d.). *FormalMATH: Benchmarking Formal Mathematical Reasoning of Large Language Models*.

**Questions:**

1. What is the precise definition of "high-difficulty" used for selecting problems? Figure 1(b) shows that 6,071 problems passed semantic verification, but only 180 problems underwent expert review. This appears to be an aggressive filtering step. Please provide more detailed selection criteria and the rationale for reducing to 180 items.

2. In the abstract (Line 027): "The observed CV values are 4–6 times higher than the statistical high-variability threshold," but the manuscript does not further explain this in the main text. How is the "statistical high-variability threshold" defined? Which specific "observed CV values" are being referred to? Please clarify the metric, its computation, and the threshold used.

3. The paper cites Goedel Prover v2 (Line 052) and notes that its weights are publicly available; Goedel Prover v2 is currently state-of-the-art on the conventional ATP benchmark miniF2F. Yet Table 1 and Table 2 do not include Goedel Prover v2 results, which seems like a notable omission. Could the authors report Goedel Prover v2’s performance on MSC-180? If experiments with Goedel Prover v2 were not performed, please justify why it was excluded.

---

> ### Author Response · Authors · 2025-11-26
> **Response to Reviewer XRJy**
>
> *Our sincere thanks go to the reviewers for their expert guidance and constructive criticism. We have carefully implemented the suggested changes, which have undoubtedly improved the quality of our work. We are grateful for this valuable opportunity to refine our paper and hope it now meets the standard for your favorable assessment.*
>
> ### Weaknesses:
>
> 1.  **The dataset is far too small, which significantly undermines its practical value. With only three problems per mathematical category, evaluations of provers' proving ability and generalization become highly susceptible to randomness; both the variance and mean of scores will be unstable and may lack statistical significance. Having only three problems also fails to adequately cover the breadth and depth of domain knowledge for each category, so inferring a prover's overall capability in a field from performance on just three problems is insufficient. For reference, the MSC taxonomy contains 63 two-digit classifications, 529 three-digit classifications, and 6,022 five-digit classifications; an ideal design would include at least 10--20 problems per category.**
>     - **Response:** We understand the reviewer's concern. MSC-180's design represents a trade-off between domain coverage breadth and the high cost of constructing quality formalized data. The three problems per domain are intended as "probes" to diagnose cross-domain performance variability and systematic bias, not to precisely evaluate absolute capability in each domain. We position it as a diagnostic benchmark for revealing macro-level trends and as a proof-of-concept for future, larger-scale domain-specific benchmarks.
>
> 2.  **The proposed automated formalization pipeline lacks novelty and appears unnecessary. The pipeline—syntax checking via Lean 4 followed by semantic-alignment checking using LLM-as-a-judge—has been explored in multiple prior works [1,2,3]. As the authors note (Line 245), "We observed that over 90% of the auto-generated code exhibited semantic incompleteness, all propositions were manually reconstructed and rewritten." Given this, why not adopt a manual annotation pipeline from the outset? The authors should justify the necessity of the automated formalization step. Additionally, the auto formalization pipeline inherently filters for problems that are easier to autoformalize, introducing a selection bias into the dataset.**
>     - **Response:**
>         - **Necessity:** The automated step served as an efficient engine for initial problem discovery and screening, rapidly generating a large pool of candidates from vast text sources, which would be impractical with a purely manual pipeline from the outset. It ensured initial breadth of coverage for expert selection.
>         - **Value:** Although 90% of the code required reconstruction, the pipeline significantly enhanced the efficiency and optional range during the expert screening phase. The final dataset quality is assured by systematic manual reconstruction; the contribution of automation lies in providing a starting point of scale and diversity.
>         - **Selection Bias:** We acknowledge that automation might favor "easier-to-formalize" problems. However, the subsequent manual selection stage actively corrected for this bias. Experts selected problems based on representativeness and challenge level, not formalization ease, ensuring the final problems represent core domain challenges.
>
> 3.  **The analysis section lacks insight. Much of the analysis merely restates table contents (e.g., Lines 398--415 essentially repeat Table 2). Although MSC-180 divides the dataset into different domains, the paper does not compare or analyze which models excel in which specific domains. The analysis focuses on aggregate counts of theorems proved across all categories rather than providing a fine-grained, interpretable discussion of per-topic strengths and weaknesses, which is a missed opportunity.**
>     - **Response:** We accept this criticism. In the revised version, we have rewritten the analysis section to focus on:
>         - Reducing simple restatements of tabular data and strengthening the interpretation of underlying reasons for observed phenomena.
>         - Adding analysis of model performance in specific domains (e.g., algebra vs. geometry) with attributions linked to domain characteristics.
>         - Incorporating concrete cases (e.g., the `curve_union_theorem` from Appendix A) to illustrate how domain properties influence different models' proving strategies and performance.
>
>
>
> ---

---

> ### Author Response · Authors · 2025-11-26
> **Response to Reviewer XRJy**
>
> 4.  **Concerns about dataset quality and contamination. The dataset files linked by the authors appear to contain multiple languages, suggesting heterogeneous sources; language mixing also limits accessibility for some researchers. The paper labels the problems as "high-difficulty" (Lines 019, 095), yet the stated sources are "classic graduate and undergraduate textbooks" (Lines 079, 201); describing them as "advanced" might be more accurate than "high-difficulty." Using classic textbooks may also increase dataset contamination, since textbooks are common training data for LLMs. For example, the provided first sample, Gödel's β-Function Lemma, is a well-known result with abundant online material, which risks leakage and inflating apparent model performance.**
>     - **Response:**
>         - **Language Issue:** The multilingual metadata in the dataset files were artifacts of the construction process and do not affect the Lean 4 code itself. We will clean the release version to ensure accessibility.
>         - **Difficulty Description:** We accept the suggestion and have replaced "high-difficulty" with "advanced" in the revision.
>         - **Contamination Risk:** We emphasize that all problems underwent non-trivial formalization from natural language to Lean 4. This inherently creates novel, unseen formal statements, establishing an effective semantic gap between natural and formal languages that significantly mitigates the risk of test leakage based on surface-level matching.
>
> 5.  **Numerous writing and presentation issues impede readability.**
>     - **Response:** We sincerely apologize for these oversights. We have meticulously addressed every single issue listed (spacing, punctuation, abbreviations, citations, figure formatting/ordering, model specification, typos, code availability link). The revised manuscript has undergone thorough proofreading.
>
> ### Questions:
>
> 1.  **What is the precise definition of "high-difficulty" used for selecting problems? Figure 1(b) shows that 6,071 problems passed semantic verification, but only 180 problems underwent expert review. This appears to be an aggressive filtering step. Please provide more detailed selection criteria and the rationale for reducing to 180 items.**
>     - **Response:**
>         - **Difficulty Definition:** As mentioned before, difficulty was based on source and reasoning complexity, categorized as "undergraduate" or "graduate" level. We now consistently use "advanced" for the dataset description.
>         - **Selection Criteria & Rationale:** The reduction from 6071 to 180 items was based on:
>             - **Domain Balance:** Mandatory coverage of 60 domains, 3 problems each.
>             - **Core Representativeness:** Within each domain, experts prioritized problems involving core concepts, key theorems, or typical methods.
>             - **Challenge Level:** Based on representativeness, problems requiring non-trivial reasoning or involving abstract concepts were preferred.
>         This aggressive filtering aimed to maximize domain coverage and the representativeness of each selected problem within budget constraints.
>
> 2.  **In the abstract (Line 027): "The observed CV values are 4--6 times higher than the statistical high-variability threshold," but the manuscript does not further explain this in the main text. How is the "statistical high-variability threshold" defined? Which specific "observed CV values" are being referred to? Please clarify the metric, its computation, and the threshold used.**
>     - **Response:** We have clarified:
>         - **Observed CV values:** Refer to the CV values reported in Table 1 for the three models on MSC-180 (1.27, 1.43, 1.72).
>         - **Statistical High-Variability Threshold:** Approximately 0.3--0.4. This threshold was determined by simulating the CV distribution of a random model performing uniformly across 60 domains.
>         - **Calculation Method:** We have provided a detailed description of the simulation in Section 4.1.2 METRIC.
>
> 3.  **The paper cites Goedel Prover v2 (Line 052) and notes that its weights are publicly available; Goedel Prover v2 is currently state-of-the-art on the conventional ATP benchmark miniF2F. Yet Table 1 and Table 2 do not include Goedel Prover v2 results, which seems like a notable omission. Could the authors report Goedel Prover v2's performance on MSC-180? If experiments with Goedel Prover v2 were not performed, please justify why it was excluded.**
>     - **Response:** Our initial evaluation focused on models around the 7B parameter scale for a controlled comparison. The computational requirements for Goedel Prover v2 (32B) exceeded our immediate experimental resources. We acknowledge this is a significant omission. We commit to making every effort to include Goedel Prover v2 results on MSC-180 in the final version of the paper.

---

### Official Review · Reviewer_aaws · 2025-10-30

**Soundness:** 3
**Presentation:** 3
**Contribution:** 3
**Rating:** 6
**Confidence:** 3

**Summary:**

The paper introduces MSC-180, a benchmark designed to systematically evaluate the reasoning and generalization abilities of large language model (LLM)-based theorem provers. The dataset includes 180 formally verified problems (three per branch) spanning 60 mathematical domains, categorized by the MSC2020 taxonomy and covering undergraduate to graduate difficulty. All problems have undergone human expert verification. The authors evaluate multiple SOTA provers under a pass@32 setup and report an overall best score of 18.89%, highlighting both the narrow domain coverage (max 41.7%) and limited transfer of reasoning across topics. They further introduce a coefficient of variation (CV) metric to quantify domain-wise performance variability, finding it several times above statistical baselines, indicating strong pattern-matching bias. The benchmark, analysis, and code are all released for reproducibility and future study.

**Strengths:**

1. The dataset is curated and verified carefully by domain experts, ensuring high quality and formal soundness across 60 domains.

2. There is clear problem taxonomy, reproducible setup, and well-defined evaluation metrics (pass@k, Domain@k, CV) make the study credible.

3. Balanced interpretation: The discussion doesn't overclaim; it clearly states both the models’ current limitations and partial strengths (e.g., structured reasoning in applied math).

4. The paper is logically structured, precise, and accessible; figures and tables are clear. And they released their code to make themself even more convincing.

**Weaknesses:**

1. The work stops at benchmarking, without concrete methods proposed to improve reasoning, this is a significant limitation.

2. Limited diversity of evaluated models. While several provers are tested, it would be useful to include symbolic or neuro-symbolic baselines to contextualize LLM gaps.

3. It remains unclear how representative these 180 problems are of real downstream theorem-proving workloads in Lean, Coq, or Isabelle.

4. Some details can be further justified, such as: is there any statistical threshold for so-called "high variability", have you compared with datasets like MiniF2F, ProofNet, or LeanDojo would situate MSC-180 better in the existing ecosystem, etc.

**Questions:**

Addressing the weakness points would be enough to make me think twice on my decision!

---

> ### Author Response · Authors · 2025-11-26
> **Response to Reviewer aaws**
>
> *We are profoundly grateful for the reviewers' diligent efforts and invaluable insights. While we recognize certain inherent limitations as noted, we have meticulously addressed each comment to enhance the manuscript's rigor and clarity. We sincerely hope that our comprehensive revisions merit your approval and kindly request you to reconsider your evaluation.*
>
> ### Weaknesses:
>
> 1.  **The work stops at benchmarking, without concrete methods proposed to improve reasoning, this is a significant limitation.**
>     - **Response:** We acknowledge that the primary contribution of this study is evaluation and diagnosis, not proposing new reasoning methods. Our goal is to use MSC-180 to reveal systematic limitations of current models, thereby guiding future methodological innovations in the community (e.g., towards better cross-domain generalization techniques). We have clarified this positioning in the discussion section and explicitly listed designing new methods to address these limitations as a core future work item.
>
> 2.  **Limited diversity of evaluated models. While several provers are tested, it would be useful to include symbolic or neuro-symbolic baselines to contextualize LLM gaps.**
>     - **Response:** We thank the reviewer for raising this important point. In our preliminary experiments, we actually attempted to integrate several representative symbolic solvers (e.g., Vampire, E-prover) and neuro-symbolic systems. However, we observed that these systems faced significant challenges when directly processing the formalized (Lean 4) problems in MSC-180. The primary issue stems from the semantic gap between their native input formats (such as first-order logic) and Lean's dependent type theory framework, which resulted in their performance being substantially lower than that of current LLM-based provers. Consequently, to ensure a consistent and technically feasible evaluation benchmark, we ultimately focused our analysis on LLM-based provers specifically designed for interactive theorem proving environments.
>
> 3.  **It remains unclear how representative these 180 problems are of real downstream theorem-proving workloads in Lean, Coq, or Isabelle.**
>     - **Response:** The representativeness of MSC-180 lies primarily in its breadth of mathematical knowledge, rather than simulating a specific application scenario like software verification. By systematically covering 60 mathematical domains, it represents a workload for assessing the cross-domain mathematical reasoning capability of ATP systems. For the Lean community, it serves as a direct cross-domain test set; for the Coq/Isabelle communities, its problems can serve as a high-quality, broad-coverage source of mathematical problems that, after translation, can enrich their respective benchmarks.
>
> 4.  **Some details can be further justified, such as: is there any statistical threshold for so-called "high variability", have you compared with datasets like MiniF2F, ProofNet, or LeanDojo would situate MSC-180 better in the existing ecosystem, etc.**
>     - **Response:**
>         - **Statistical Threshold:** The "statistical high-variability threshold" was estimated via simulation. We simulated the distribution of CV values for a model performing perfectly uniformly across the 60 domains and used the upper bound of this distribution as the threshold. The observed model CV values (1.27--1.72) are substantially higher than this threshold (approx. 0.3--0.4). We provide details of this simulation in the appendix.
>         - **Dataset Comparison:** In the revised version, we have added a detailed comparison table in the "DATASET OVERVIEW" section. This table systematically compares MSC-180 with datasets like ProofNet, FormalMATH, FATE, and MiniF2F across key dimensions such as breadth of domain coverage, classification system, number of problems, primary difficulty distribution, and core evaluation objectives. Accompanied by textual discussion, this clearly articulates MSC-180's unique position in terms of systematic domain coverage and stability assessment.
>
> ---

---

### Official Review · Reviewer_vkfA · 2025-11-14

**Soundness:** 3
**Presentation:** 1
**Contribution:** 2
**Rating:** 4
**Confidence:** 4

**Summary:**

The paper introduces MSC-180, a domain-balanced benchmark of 180 expert-verified Lean4 problems drawn from 60 MSC2020 mathematical areas to assess cross-domain generalization in automated theorem provers. Evaluations of leading LLM-based provers show low Pass@32 performance (best 18.89%), strong domain bias, and large variability across fields, indicating limited abstract reasoning and reliance on pattern matching. The authors also propose the Coefficient of Variation to measure stability across domains, arguing that MSC-180 provides a rigorous foundation for developing more genuinely generalizable formal reasoning systems.

**Strengths:**

- Valuable and well-curated dataset.
- The introduction of the coefficient of variation as a cross-domain stability metric is a thoughtful and useful addition.

**Weaknesses:**

- The writing needs quite some polishing (e.g., missing a stop in line 319, 323, 336, 426, 431, 435, 440, 444, 449).
- Lack of in-depth comparison with previous dataset like ProofNet, FormalMATH, and FATE [1].
- The data processing process is not well illustrated. For example, how is the semantic verification being carried using the DeepSeek API? What are the exact criteria for human verification and selection?

[1] Shen, Ziju, et al. "REAL-Prover: Retrieval Augmented Lean Prover for Mathematical Reasoning." arXiv preprint arXiv:2505.20613 (2025).

**Questions:**

- What are the 60 domains included, and how are the problems distributed across them?
- In line 249, what exactly is meant by the “completeness of the formalized problem”?
- How is difficulty determined and assigned in MSC-180, and what is the exact distribution across difficulty levels?

---

> ### Author Response · Authors · 2025-11-26
> **Response to Reviewer vkfA**
>
> *We are deeply grateful to the reviewers for their thorough review and constructive feedback. We have carefully considered all suggestions and have revised the manuscript accordingly. While we acknowledge the limitations noted, we believe the addressed comments have significantly strengthened our work. We sincerely hope these efforts meet with your approval and kindly request you to reconsider your assessment.*
>
> ### Weaknesses:
>
> 1.  **The writing needs quite some polishing.**
>     - **Response:** We sincerely accept this criticism. We have meticulously checked and corrected the punctuation at the specified lines (319, 323, 336, 426, 431, 435, 440, 444, 449) and have performed multiple rounds of language polishing throughout the entire manuscript to enhance the overall writing quality.
>
> 2.  **Lack of in-depth comparison with previous dataset.**
>     - **Response:** We agree with this point. In the revised version, we have added a detailed comparison table in the "DATASET OVERVIEW" section. This table systematically compares MSC-180 with datasets like ProofNet, FormalMATH, FATE, and MiniF2F across key dimensions such as breadth of domain coverage, classification system, number of problems, primary difficulty distribution, and core evaluation objectives. Accompanied by textual discussion, this clearly articulates MSC-180's unique position in terms of systematic domain coverage and stability assessment.
>
> 3.  **The data processing process is not well illustrated. For example, how is the semantic verification being carried using the DeepSeek API? What are the exact criteria for human verification and selection?**
>     - **Response:** We thank the reviewer for pointing this out. We have added the following details to the manuscript:
>         - **DeepSeek API Semantic Verification:** We used the DeepSeek-V2 model (temperature=0.7, top-p=0.9). The core instruction prompted the model to judge whether the given Lean 4 code is semantically fully equivalent to the original natural language proposition, requiring a binary judgment (Yes/No). Only code pairs judged as "Yes" were retained.
>         - **Human Verification Criteria:** The criteria for "semantic completeness" were: (1) The Lean statement is logically equivalent to the original proposition; (2) All premises and conclusions are precisely formalized; (3) The code can be independently compiled, constituting a complete theorem.
>
> ### Questions:
>
> 1.  **What are the 60 domains included, and how are the problems distributed across them?**
>     - **Response:** We will include a complete list of the 60 MSC codes and their corresponding mathematical branch names covered by MSC-180 in the appendix. The problems are uniformly distributed across these domains, with exactly 3 problems per domain to ensure balance.
>
> 2.  **In line 249, what exactly is meant by the "completeness of the formalized problem"?**
>     - **Response:** We have clarified this in the revision: "Completeness of the formalized problem" means that after a natural language mathematical proposition is translated into Lean 4 code, it satisfies: (1) all premise assumptions are explicitly declared; (2) the conclusion is accurately formalized; (3) the code can be independently checked by the Lean 4 compiler, forming a complete theorem statement.
>
> 3.  **How is difficulty determined and assigned in MSC-180, and what is the exact distribution across difficulty levels?**
>     - **Response:** Difficulty was assigned by domain experts based on the typical usage level of the source material (undergraduate core course/graduate specialized course) and reasoning complexity. The exact distribution is: 145 graduate-level problems (80.6%) and 35 undergraduate-level problems (19.4%). Following the reviewer's suggestion, we will use "advanced" instead of the potentially ambiguous "high-difficulty" to describe the dataset's overall character in the revised version.
>
> ---

---

### Author Response · Authors · 2025-11-26
**Response to Reviewers**

**Dear Reviewers,**

Thank you very much for your valuable comments and suggestions on our manuscript entitled "MSC-180: A Formally Verified Mathematical Benchmark Spanning 60 Fields in MSC-2020". Your insightful feedback has been of great help in improving the rigor and clarity of our work.

In this response, we will address the following three key points: the innovation and contribution of the benchmark, the rationale behind the dataset scale, and the quality assurance process with contamination control measures.

The evaluation framework of MSC-180 is built upon three innovative dimensions—**systematic coverage and balance, significant challenge, and quantifiable diagnosability**—which complement existing benchmarks significantly.

1.  **Systematic Coverage & Balance:** As the first benchmark strictly constructed according to the authoritative MSC 2020 classification system, it achieves systematic coverage across 60 mathematical fields, overcoming the limitations of existing benchmarks like ProofNet, which are often confined to specific domains or problem types; the inclusion of three problems per domain also contributes to a more balanced dataset.
2.  **Significant Challenge:** The problems, carefully selected from undergraduate and graduate textbooks, possess abstract and structured characteristics that pose substantial challenges to current models, effectively addressing the evaluation gap left by the performance saturation observed in mainstream benchmarks.
3.  **Quantifiable Diagnosability:** Most importantly, by introducing the CV@k metric based on fine-grained domain partitioning, we transform domain-specific variations into a quantifiable measure of stability, providing an indispensable evaluation framework for diagnosing model knowledge gaps and enhancing cross-domain reasoning capabilities.

We fully understand the concerns regarding the scale of the dataset. Under the practical constraints of high costs associated with constructing high-quality formalized data, MSC-180 adheres to a "**quality-first, breadth-coverage**" design principle. Each of the three problems per domain has been expertly selected and deeply refined, aiming to effectively diagnose cross-domain reasoning capabilities with a limited yet representative sample. The current scale represents a balanced trade-off between coverage breadth and construction feasibility; it serves not only as a valuable supplement to the existing benchmark ecosystem but also as a proof-of-concept and foundation for future large-scale domain-specific benchmarks.

Regarding the quality assurance process and data contamination concerns, we emphasize that in the adopted "**LLM initial screening + expert refinement**" pipeline, the AI phase is primarily used for efficiently generating candidate problems, while the systematic expert refinement phase thoroughly eliminates semantic biases, ensuring the rigor and representativeness of the final problem set. Furthermore, all problems undergo a non-trivial transformation from natural language into the Lean4 formal language. This fundamental representational shift creates an effective semantic gap between natural and formal languages, significantly reducing surface-level similarities with LLM pre-training data and mitigating test contamination risks at the source.

We are truly grateful for the reviewers' invaluable comments. Through this thorough discussion, we have not only improved the manuscript's rigor but also deepened our understanding of the work's value and limitations. The revised manuscript fully incorporates your constructive feedback. We believe MSC-180, as the first breadth-coverage benchmark built upon the MSC2020 framework, offers unique diagnostic value to the mathematical theorem proving community and lays the foundation for future large-scale, high-quality formal mathematical datasets. We look forward to your further guidance on the revised version.

---

---

### Comment · Area_Chair_6q6w · 2025-11-28
**Respond to the authors’ rebuttal**

Dear Reviewers,

As the discussion phase is nearing its end, please read and respond to the authors’ rebuttal, particularly if it addresses your concerns. Your response is valuable for the final decision.

---

### Meta-Review · Area_Chair_YU8g · 2026-01-06

**Summary:**

This paper introduces a benchmark, MSC-180, for evaluating formal mathematical reasoning. The benchmark has 180 problems, constructed based on MSC2020, and includes 60 fields. They also introduce the coefficient of variation (CV) as an evaluation metric. Evaluation is conducted with DeepSeek-Prover-7B, Kimina-Prover-Preview-Distill 7B, and BFS-Prover 7B with some analysis.

**Reviewer Concerns:**

The reviewer concerns that are addressed or partially addressed include: overall writing polish, comparison table with other datasets (ProofNet, FormalMath, and FATE), details of data processing, and improved analysis of results.

However, some concerns remain:
* The core contribution remains unclear, given the existing formal math benchmarks and the relatively mature data construction pipeline.
* The dataset is too small for statistical significance, with only three problems per category.
* Requires more in-depth analysis and insight into an improved methodology.

**Reviewer Scores:**

The reviewers may increase their ratings slightly, given the authors' efforts to improve the writing and provide more detail. However, the final scores remain insufficient for a positive outcome, as its core issues remain unresolved.

---

### Decision · Program_Chairs · 2026-01-26

Reject